Journal of Data-centric Machine Learning Research (2024)        Submitted 12/23; Revised 7/24; Published 8/24

# Deep Neural Network Benchmarks
# for Selective Classification

**Andrea Pugnana**                                                                 ANDREA.PUGNANA@DI.UNIPI.IT
*Scuola Normale Superiore, University of Pisa, ISTI-CNR*
*Pisa, Italy*

**Lorenzo Perini**                                                                 LORENZO.PERINI@KULEUVEN.BE
*KU Leuven*
*Leuven, Belgium*

**Jesse Davis**                                                                     JESSE.DAVIS@KULEUVEN.BE
*KU Leuven*
*Leuven, Belgium*

**Salvatore Ruggieri**                                                             SALVATORE.RUGGIERI@UNIPI.IT
*University of Pisa*
*Pisa, Italy*

*https://openreview.net/forum?id=xDPzHbtAEs*

**Editor:** Mykola Pechenizkiy

## Abstract

With the increasing deployment of machine learning models in many socially-sensitive tasks, there is a growing demand for reliable and trustworthy predictions. One way to accomplish these requirements is to allow a model to abstain from making a prediction when there is a high risk of making an error. This requires adding a selection mechanism to the model, which *selects* those examples for which the model will provide a prediction. The selective classification framework aims to design a mechanism that balances the fraction of rejected predictions (i.e., the proportion of examples for which the model does not make a prediction) versus the improvement in predictive performance on the selected predictions. Multiple selective classification frameworks exist, most of which rely on deep neural network architectures. However, the empirical evaluation of the existing approaches is still limited to partial comparisons among methods and settings, providing practitioners with little insight into their relative merits. We fill this gap by benchmarking 18 baselines on a diverse set of 44 datasets that includes both image and tabular data. Moreover, there is a mix of binary and multiclass tasks. We evaluate these approaches using several criteria, including selective error rate, empirical coverage, distribution of rejected instance's classes, and performance on out-of-distribution instances. The results indicate that there is not a single clear winner among the surveyed baselines, and the best method depends on the users' objectives.

## 1 Introduction

Artificial Intelligence (AI) systems are increasingly being deployed to support or even automate decision-making. Ensuring the trustworthiness of AI systems is crucial in many applications (Kaur et al., 2023), and is one of the main goals of the recent European AI Act (European Commission, 2021). More precisely, "[h]igh-risk AI systems shall be designed and developed in such a way that they achieve, in the light of their intended purpose, an appropriate level of accuracy [and] robustness".

High-risk AI systems pertain to socially sensitive domains, such as: healthcare, where predictions might be used to determine treatments (Craig et al., 2023); justice, where predictions can evaluate the risk of recidivism (Berk et al., 2021); hiring, where predictions can determine rankings of candidates or explain their turnover intention (Fabris et al., 2023; Lazzari et al., 2022); and credit scoring, where predictions can be used to estimate the probability of repaying a debt (Dastile et al., 2020).

In all such high-risk contexts, we aim to reduce the number of mistakes made by AI systems because their mistakes can have critical consequences. For example, consider a bank that uses a Machine Learning (ML) model to score the credit risk of loan applications. In such a setting, a misprediction could either translate into a money loss for the bank or an unjust denial of credit to the applicant.

One potential way to improve the trustworthiness of a model is to allow it to abstain from making a prediction when there is a high chance of making an error (Chow, 1970). Such a strategy is inherent in human reasoning when facing an unknown phenomenon. For example, human bankers who are unsure about a specific loan application do not (have to) provide an answer as soon as they are asked. Indeed, they may require additional financial documents to verify the loan's feasibility or ask for an external expert consultation. This approach aims to minimize the risk of an incorrect evaluation.

Likewise, allowing ML models to predict only when confident enough helps mitigate the risk of incorrect predictions (Pugnana, 2023). On the one hand, including a reject option results in the ML model having better performance when it does provide a prediction because it is only offering predictions in those cases where it is highly likely to be correct. On the other hand, rejected instances can be dealt with in other ways. For example, human experts can be involved in the loop to oversee difficult instances, e.g., a banker can oversee difficult-to-evaluate loan applications. Alternatively, the prediction task can be deferred to more complex ML models, possibly using additional and costly-to-compute features.

Selective Classification (SC) (El-Yaniv and Wiener, 2010) is one well-known framework that allows a model not always to offer a prediction. Intuitively, this framework imbues a model with a mechanism that *selects* whether a prediction is made on a per-example basis. The goal is to navigate the tradeoff between the proportion of examples for which a prediction is made (i.e., the model's *coverage*) and the performance improvement on the selected examples (i.e., the ones for which a prediction is made) that arises from focusing only on those cases where the model has a small chance of making a misprediction. Typically, this is done by maximizing the performance on the selected examples given a target coverage. Given the appeal of SC, there are wide range of approaches for this problem setting (Geifman and El-Yaniv, 2017, 2019; Liu et al., 2019; Huang et al., 2020; Gangrade et al., 2021; Pugnana

and Ruggieri, 2023a,b; Feng et al., 2023). The primary emphasis is on implementing SC in the context Deep Neural Networks (DNN) models.

Unfortunately, we lack insights into the relative merits of existing SC approaches for DNNs because existing empirical evaluations in the literature suffer from several shortcomings. First, they always involve $\leq 10$ datasets, and primarily consider only image data. Second, only a handful of approaches (never more than seven) are compared. Third, most studies mainly focus on comparing approaches based on single metric: their predictive accuracy on the selected examples. However, there are other relevant performance characteristics of SC methods such as whether their coverage constraint holds, whether they disproportionately reject instances from one class, or how they behave on unseen data.

Our goal is to fill this gap by performing the first comprehensive benchmarking of SC methods for DNN architectures. Specifically, our evaluation goes substantially beyond existing studies by:

1. Including 18 SC methods;

2. Evaluating the considered methods on 44 datasets that include both image and tabular data; and

3. Considering five different aspects of SC models' performance.

Our results suggest that the choice of the baseline depends on the performance criterion to be prioritized. In fact, most methods perform with no statistically significant difference across the different tasks. To summarize, the main contributions of this paper are that we:

*(i)* briefly survey the state-of-the-art methods in SC;

*(ii)* provide the widest experimental evaluation of SC methods in terms of baselines, datasets and tasks;

*(iii)* point out the limitations of compared methods, which highlights potential avenues for future research directions; and

*(iv)* release a public repository with all software code and datasets for reproducing the baseline algorithms and the experiments.[1]

## 2 Background

Let $\mathcal{X}$ be an $d$-dimensional input space, $\mathcal{Y} = \{1, \ldots, m\}$ be the target space and $P(\mathbf{X}, Y)$ be the probability distribution over $\mathcal{X} \times \mathcal{Y}$. Given a hypothesis space $\mathcal{H}$ of functions that map $\mathcal{X}$ to $\mathcal{Y}$ (called models or classifiers), the goal of a learning algorithm is to find the hypothesis $h \in \mathcal{H}$ that minimizes the *risk*:

$$R(h) = \mathbb{E}[l(h(\mathbf{X}), Y)] \tag{1}$$

where $l : \mathcal{Y} \times \mathcal{Y} \to \mathbb{R}$ is a user-specified loss function. Because $P(\mathbf{X}, Y)$ is generally unknown, it is typically assumed that we have access to an i.i.d. sample $\mathcal{T}_n = \{(\mathbf{x}_i, y_i)\}_{i=1}^n$ that can

---

1. The code is available at `github.com/andrepugni/ESC/`.

be used to learn a classifier $\hat{h}(\cdot)$, such that:

$$\hat{h} \in \underset{h \in \mathcal{H}}{\arg\min} \, \hat{R}(h, \mathcal{T}_n) \tag{2}$$

where $\hat{R}(h, \mathcal{T}_n) = 1/|\mathcal{T}_n| \sum_{(\mathbf{x},y) \in \mathcal{T}_n} l(h(\mathbf{x}), y)$ is the *empirical risk* over the sample $\mathcal{T}_n$.

Because the learned model is prone to making mistakes, one can extend the canonical setting to include a selection mechanism that allows the model to refrain from offering a prediction for those instances likely to be misclassified.

Formally, a *selective classifier* is a pair $(h, g)$ where $h$ is a standard classifier and $g : \mathcal{X} \to \{0, 1\}$ is a *selection function* that determines whether $h$'s prediction is provided or the model abstains (or rejects):

$$(h, g)(\mathbf{x}) = \begin{cases} h(\mathbf{x}) & \text{if } g(\mathbf{x}) = 1 \\ \text{abstain} & \text{otherwise} \end{cases} \tag{3}$$

In practice, rather than directly learning the selection function in Eq. 3, one approximates it by (1) learning a *confidence function*[2] $k_h : \mathcal{X} \to [0, 1]$ (sometimes called soft selection (Geifman and El-Yaniv, 2017)) that measures how likely it is that the predictor $h$ is correct, and (2) setting a threshold $\tau \in [0, 1]$ that defines the minimum confidence for providing a prediction. A low confidence value indicates that the model is likely to make a misprediction for the instance and therefore it should abstain, which yields the following selection function:

$$g(\mathbf{x}) = \mathbb{1}(k_h(\mathbf{x}) > \tau) \tag{4}$$

To prevent the selective classifier from abstaining on too many (test) instances, SC methods also consider the *coverage* metric, which is defined as

$$\phi(g) = \mathbb{E}[g(\mathbf{X})]. \tag{5}$$

The coverage computes the expected proportion of instances for which the model would make a prediction. These non-rejected instances are commonly referred to as either *accepted* or *selected*, and we will use these terms interchangeably. We will refer to the *rejection rate* as the complement of the coverage, i.e., $1 - \phi(g)$ (Perini and Davis, 2023). Another core measure of the SC framework is the risk over the accepted region, commonly called the *selective risk* which is defined as:

$$R(h, g) = \frac{\mathbb{E}[l(h(\mathbf{X}), Y)g(\mathbf{X})]}{\phi(g)} = \mathbb{E}[l(h(\mathbf{X}), Y)|g(\mathbf{X}) = 1] \tag{6}$$

A widely adopted instance of the selective risk is the *selective error rate*, which corresponds to the selective risk for the 0-1 loss $l(h(\mathbf{X}), Y) = \mathbb{1}\{h(\mathbf{X}) \neq Y\}$.

Coverage and risk are estimated over a given test set $\mathcal{T}_{test}$ as follows. The *empirical risk* over the set of accepted instances is defined as:

$$\hat{R}(h, g, \mathcal{T}_{test}) = \frac{1}{|\mathcal{T}_{test}| \cdot \hat{\phi}(g, \mathcal{T}_{test})} \sum_{(\mathbf{x},y) \in \mathcal{T}_{test}} l(h(\mathbf{x}), y) \cdot g(\mathbf{x}) \tag{7}$$

---

2. A good confidence function $k_h$ should rank instances based on descending loss, i.e., if $k_h(\mathbf{x}_i) \leq k_h(\mathbf{x}_j)$ then $l(h(\mathbf{x}_i), y_i) \geq l(h(\mathbf{x}_j), y_j)$.

where $\hat{\phi}(g, \mathcal{T}_{test}) = 1/|\mathcal{T}_{test}| \sum_{(\mathbf{x},y) \in \mathcal{T}_{test}} g(\mathbf{x})$ is the *empirical coverage* over the test set. Observe that $\hat{R}(h, g, \mathcal{T}_{test}) = \hat{R}(h, \mathcal{T}_{test}^g)$, where $\mathcal{T}_{test}^g = \{(\mathbf{x}, y) \in \mathcal{T}_{test} \mid g(\mathbf{x}) = 1\}$, i.e., the empirical risk of a selective classifier boils down to the empirical risk of the classifier over the set of accepted instances. The inherent trade-off between coverage and risk can be summarized by a *risk-coverage curve* (El-Yaniv and Wiener, 2010). Moreover, this trade-off allows framing the SC task according to two different formulations: the bounded improvement model and the bounded abstention model (Franc et al., 2023). In the bounded improvement model, the problem is formulated by fixing an upper bound $r$ - the *target risk* - for the selective risk and then looking for a selective classifier that maximizes coverage (Geifman and El-Yaniv, 2017).

**Problem 1 (Bounded-improvement model)** *Given a target risk $r$, an optimal selective classifier $(h, g)$ is a solution to:*

$$\max_{\theta, \psi} \phi(g_\psi) \quad s.t. \quad R(h_\theta, g_\psi) \leq r \tag{8}$$

Conversely, in the bounded-abstention model, we fix a lower bound $c$ for coverage (called *target coverage*) and then look for a selective classifier that minimizes the selective risk (Geifman and El-Yaniv, 2019).

**Problem 2 (Bounded-abstention model)** *Given a target coverage $c$, an optimal selective classifier $(h, g)$ is a solution to:*

$$\min_{\theta, \psi} R(h_\theta, g_\psi) \quad s.t. \quad \phi(g_\psi) \geq c \tag{9}$$

We call *coverage-calibration* the post-training procedure of estimating the threshold $\tau$ in (4) for the target coverage $c$ specified in Problem 2. This is generally done by estimating the $(1-c) \cdot 100$-th percentile of the confidence function $k_h$ over a held-out calibration set $\mathcal{T}_{cal}$.

## 3 Baselines

There are multiple ways to devise abstaining classifiers. We restrict our attention to DNN approaches aiming to solve the bounded-abstention problem (Eq. 9). We present and categorize a few baselines according to their definition of the confidence function, extending the work of Feng et al. (2023). We distinguish among three categories of methods: **Learn-to-Abstain**, **Learn-to-Select** and **Score-based**.

### 3.1 Learn-to-Abstain Methods

Learn-to-Abstain methods tackle the selective classification task by adding a new class label $(m + 1)$ representing abstention to the classification problem. While there are no actual instances belonging to this class, these approaches design loss functions to enable the classifier to assign a positive score $s_{m+1}(\mathbf{x})$ to ambiguous instances. This score serves as a confidence function, i.e., $k_h(\mathbf{x}) = 1 - s_{m+1}(\mathbf{x})$ (Feng et al., 2023). In Figure 1, we provide an example of a canonical Learn-to-Abstain architecture.

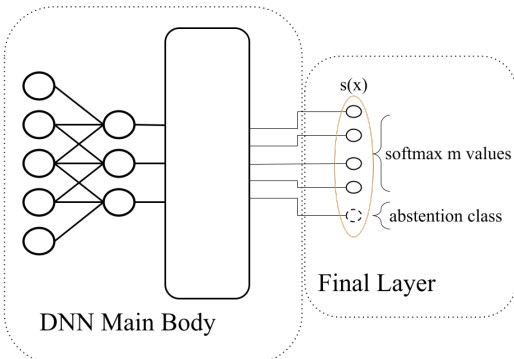

Figure 1: A generic Learn-to-Abstain architecture

The first method to take this approach was **DG** (Liu et al., 2019). It uses a reward hyperparameter $o$ for the class $m+1$ to set how often the classifier should abstain. Formally, DG trains a neural network minimizing the following loss:

$$\mathcal{L}_{\text{DG}} = \mathbb{E}_{P(\mathbf{X},Y)} \left[ \log(s_y(\mathbf{x}) + \frac{1}{o} s_{m+1}(\mathbf{x})) \right], \tag{10}$$

where $s_y(\mathbf{x})$ and $s_{m+1}(\mathbf{x})$ are the neural network softmax values, respectively, over the true class $Y = y$ and $m + 1$ (abstention). Intuitively, a higher $o$ encourages the network to be confident in its prediction, and a low $o$ makes it less confident and more likely to abstain. However, DG does not have any explicit way to guide abstention towards more difficult examples during training, as the reward $o$ remains fixed for the whole training procedure.

To overcome this limitation, Self-Adaptive Training (**SAT**) (Huang et al., 2020) trains the selective classifier through a convex combination of predictions and true labels. This combination is dynamically adapted during the training process to identify those instances that are more difficult to correctly classify and, hence are good candidates for abstention. More precisely, for the first $E_s$ (user-defined) epochs, the training target - $\mathbf{t} \in [0, 1]^m$ - is equal to the one-hot encoded true label vector $\mathbf{y}$. Afterwards, it becomes the convex combination of (probabilistic) predictions and true labels, namely $\mathbf{t} = \gamma \mathbf{t} + (1 - \gamma)\mathbf{s}(\mathbf{x})$, with $\mathbf{s}(\mathbf{x})$ representing the neural network softmax values and $\gamma$ the weight of the convex combination. The final selective classifier is then optimized by minimizing the loss function:

$$\mathcal{L}_{\text{SAT}} = -\mathbb{E}_{P(\mathbf{X},Y)} \left[ \mathbf{t}' \log(\mathbf{s}(\mathbf{x})) + (1 - t_y) \log s_{m+1}(\mathbf{x}) \right], \tag{11}$$

where $t_y$ is the value of vector $\mathbf{t}$ corresponding to the index of true value $y$ and $s_{m+1}(\mathbf{x})$ represents the softmax value for the abstention class.[3] Both DG and SAT add an extra softmax value to the neural network output to identify difficult-to-predict instances. However,

---

3. For instance, if $y = 1$ and $m = 2$, then $\mathbf{t}' = [0, 1]$ and $t_y = 1$ when epoch is below $E_s$. Intuitively, the first term is the cross-entropy loss between the classifier and the adaptive training target, which allows learning a good multi-class classifier. The second term serves as a confidence function, identifying uncertain samples in the dataset. The balance between these terms is controlled by the value of $t_y$, which determines whether the classifier learns to abstain or make accurate predictions.

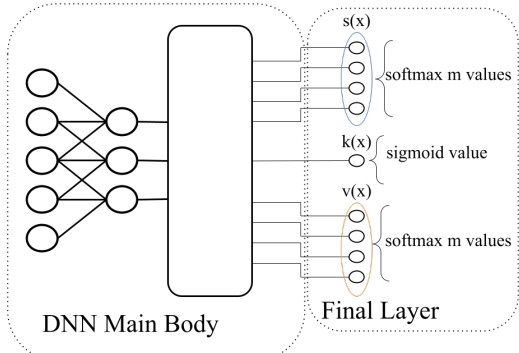

Figure 2: An example of SELNET, a Learn-to-Select architecture.

Feng et al. (2023) argue that incorporating this extra class in the training loss could result in overfitting on examples that are easier to classify. To mitigate this, **SAT+EM** (Feng et al., 2023) adds an average entropy term $\mathcal{E}(\mathbf{s}(\mathbf{x}))$ to SAT's loss :

$$\mathcal{L}_{\text{SAT+EM}} = \mathcal{L}_{\text{SAT}} + \beta\mathcal{E}(\mathbf{s}(\mathbf{x})) \tag{12}$$

where $\mathbf{s}(\mathbf{x})$ represents the neural network of $m$ softmax values, and $\beta$ is a hyperparameter that measures the impact of the entropy term. All the learn-to-abstain methods are calibrated for the target coverage $c$ using a calibration set (as discussed in Section 2).

### 3.2 Learn-to-Select Methods

Like Learn-to-Abstain methods, Learn-to-Select methods simultaneously learn the classifier and its specific confidence function. However, in this setting, the confidence function does not rely on an additional abstention class but aims at achieving a specific target coverage $c$. This procedure ensures that the classifier's parameters are optimized to correctly predict instances less likely to be rejected.

The main architecture belonging to this class is SelectiveNet (**SELNET**) (Geifman and El-Yaniv, 2019). Given a target coverage $c$, SELNET jointly trains the final classifier and the confidence function to maximize the performance over the $100 \cdot c\%$ most confident instances. SELNET's architecture has four main components, each with a different purpose, as depicted in Figure 2: the main body, the predictive head $s$, the selective head $k$, and the auxiliary head $v$. The main body consists of deep layers shared by all three heads: any deep-learning architecture can be used in this part (e.g., convolutional layers, linear layers, recurrent layers, etc.). The predictive head $s(\mathbf{x})$, consisting of a final linear layer with softmax, is used to make the classifier prediction. The selective head $k(\mathbf{x})$ outputs a confidence function using a linear layer with a final sigmoid activation. The auxiliary head $v(\mathbf{x})$ replicates the structure of the predictive head and mitigates the risk of overfitting on the accepted instances. Given the target coverage $c$, SELNET is trained using the following

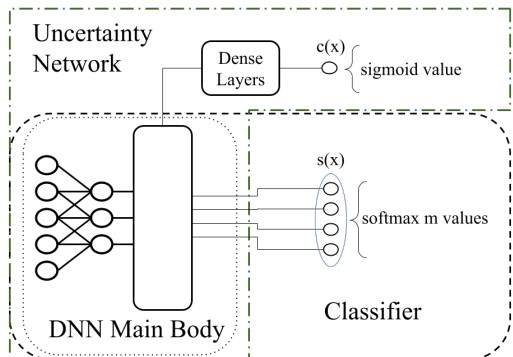

Figure 3: An example of CONFIDNET, a score-based architecture.

loss function:

$$\mathcal{L}_{\text{SELNET}} = \alpha \left( \frac{\mathbb{E}_{P(\mathbf{X},Y)}\left[l(s(\mathbf{x}),y)k(\mathbf{x})\right]}{\phi(k)} + \lambda(\max(0, c - \phi(k)))^2 \right) + (1-\alpha)\mathbb{E}_{P(\mathbf{X},Y)}\left[l(v(\mathbf{x}),y)\right] \tag{13}$$

where $l(s(\mathbf{x}), y)$ is the cross-entropy loss for the predictive head $s(\mathbf{x})$; $\phi(k)$ is the coverage obtained by selective head $k$; $l(v(\mathbf{x}), y)$ is the cross entropy loss for auxiliary head prediction $v(\mathbf{x})$; $\alpha$ is a hyperparameter to control the relative importance between the losses for the predictive and the auxiliary head; and $\lambda$ is a penalization term for coverage violations.

For the sake of completeness, following the same reasoning as for SAT+EM (Feng et al., 2023), we include **SELNET+EM** in the comparison. This approach adapts the SELNET objective function to contain an additional entropy term.

### 3.3 Score-based Methods

Score-based methods compute and set a threshold on a confidence function - as formalized in Eq. 4 - that is based on the classifier's output. Conceptually, this means that predictions are only made for the test examples for which the model is most confident. Because this can be viewed as a post-hoc approach, this confers the advantage of being applicable to already trained models.

The most popular technique is the Softmax Response **SR** (Geifman and El-Yaniv, 2017), which defines the confidence function as the maximum value of a final softmax layer, i.e., $k_{\text{SR}}(x) = \max_{y \in \mathcal{Y}} s_y(\mathbf{x})$. Given a coverage $c$, SR sets the selection threshold $\tau$ using the calibration procedure explained in Section 2.

Since the SR principle is very general, it can be applied to any method that provides scores for the classes (Franc et al., 2023). For example, Feng et al. (2023) propose to improve learn-to-abstain and learn-to-select methods by replacing their confidence function with the SR confidence. In particular, three novel methods are presented, i.e., **SAT+SR**, **SAT+EM+SR**, **SELNET+SR**, which are trained using $\mathcal{L}_{\text{SAT}}$, $\mathcal{L}_{\text{SAT+EM}}$ and $\mathcal{L}_{\text{SELNET}}$ respectively. For the sake of completeness, we include also **SELNET+EM+SR** in the comparison, i.e., a network trained with the SELNET+EM's loss and using the SR selection strategy.

Another score-based popular option is using ensembles of neural networks. For example, Lakshminarayanan et al. (2017) train multiple networks with different initialization and build a selective classifier by computing the entropy of the (multiple) network outputs (**ENS**). The intuition is that more disagreement among the outputs indicates that the ensemble is uncertain about its prediction, and hence rejection is appropriate. However, despite the advantages of using ensembles in terms of performance, relating a dispersion measure to the correctness of predictions is not straightforward. Hence, the authors also propose using the average softmax response (i.e., $k_{\text{ENS}}(\mathbf{x}) = {}^1\!/\!_J \sum_{j=1}^{J} k_{\text{SR},j}(\mathbf{x})$, where $J$ is the number of networks in the ensemble) as a confidence measure. We will refer to this baseline as **ENS+SR**. A theoretical analysis of the advantages of using ENS+SR can be found in Ding et al. (2023).

The main concern with using $k_{\text{SR}}(\mathbf{x})$ as a confidence measure is that may provide high values both for mistakes and correct predictions, making them indistinguishable. On the other hand, when the model misclassifies an example, the score $s_y(\mathbf{x})$ associated with the true class probability $P(Y = y | \mathbf{X} = \mathbf{x})$ should be low, making it a viable option to perform selective classification. However, one cannot access true labels at test time, making it impossible to use $s_y(\mathbf{x})$ directly. Corbière et al. (2019) address this concern by estimating $s_y(\mathbf{x})$ with a two-step procedure called **CONFIDNET** (as depicted in Figure 3). First, they estimate $s_y(\mathbf{x})$ by training a neural network classifier. Next, they build a second (uncertainty) network on top of the classifier: the main body is kept unchanged, while the final part of the original classifier is replaced with a series of dense layers. This uncertainty network is then trained considering the following loss function:

$$\mathcal{L}_{\text{CONFIDNET}} = \mathbb{E}_{P(\mathbf{X},Y)}[(c(\mathbf{x}) - s_y(\mathbf{x}))^2] \tag{14}$$

with $c(\mathbf{x})$ referring to the final output of the uncertainty network. Intuitively, $c(\mathbf{x})$ should mimic $s_y(\mathbf{x})$ and can be used as a confidence function: the higher $c(\mathbf{x})$, the higher the chance the classifier is right.

Franc et al. (2023) also use a classifier and an uncertainty estimator. They propose two different approaches, named **REG** and **SELE**. Both of them learn the classifier on half of the training data and use the other half to directly estimate where the classifier is more likely to make mistakes. In particular, these two methods focus on learning an uncertainty score $f$, which mirrors the confidence function $k$: the higher $f$ is, the higher the likelihood of making mistakes (thus, abstention is preferable in the latter). Neither SELE nor REG are tied to specific neural network architectures, i.e., they are model-agnostic and can be adapted to other learning models. REG poses the problem of learning the uncertainty score as a regression problem, where given a set of hypotheses $\mathcal{F}$, the uncertainty score $f \in \mathcal{F}$ minimizes the following:

$$\mathcal{L}_{\text{REG}} = \mathbb{E}_{P(\mathbf{X},Y)}[(l(h(\mathbf{x}, y) - f(\mathbf{x}))^2] \tag{15}$$

Intuitively, the higher the value of $f$, the higher the loss. Hence, abstention should be preferred. On the other hand, given a hypothesis space $\mathcal{F}$, SELE considers a surrogate loss of the risk coverage curve, i.e.,

$$\mathcal{L}_{\text{SELE}} = \mathbb{E}_{(\mathbf{x}_1,y_1),(\mathbf{x}_2,y_2)\sim P(X,Y)}[l(h(\mathbf{x}_1, y_1)) \log(1 + \exp(f(\mathbf{x}_2) - f(\mathbf{x}_1)))] \tag{16}$$

and then learns the uncertainty score $f \in \mathcal{F}$ by minimizing $\mathcal{L}_{\mathrm{SELE}}$.

The approaches presented so far require a held-out calibration dataset. Unfortunately, for problems where only little data is available, reducing the amount of training data may deteriorate the classifier's performance. Moreover, splitting the data into a dataset for training and a dataset for calibration may introduce randomness effects on both the classifier and the selection function. **SCROSS** (Pugnana and Ruggieri, 2023a) is a model-agnostic approach that overcomes the need for a calibration set by employing a cross-validation strategy that follows three steps. First, it splits the available data into $K$ folds. Second, it trains a classifier over $K-1$ folds and predicts the SR confidence values over the remaining $K$-th fold. Finally, it stacks the predicted confidence values altogether. This approach approximates the confidence over the full dataset. Then, SCROSS uses SR's approach to threshold such confidence values.

Moreover, in high-risk scenarios where SC is sought, such as healthcare and finance, we often deal with imbalanced (binary) classes (He and Garcia, 2009). A common metric to evaluate the performance of classifiers in such contexts is the Area Under the ROC Curve (AUC) (Yang and Ying, 2023), which measures the classifier's ability to rank instances from minority and majority classes correctly. Pugnana and Ruggieri (2023b) provide a theoretical condition - two bounds over the minority class score - that guarantees not to worsen AUC once we allow for abstention. The selection function is implemented by (1) estimating these lower and upper bounds for the minority class score, and (2) rejecting instances with minority class scores between the two (estimated) bounds. To implement such a strategy, the authors devise two algorithms, i.e., **PLUGINAUC** and **AUCROSS**. The difference between the two methods lies in how their selection strategy is calibrated: PLUGINAUC adopts a held-out approach to calibrate the bounds, while AUCROSS uses a cross-fitting approach similar to SCROSS.

## 4 Research Questions

This paper intends to evaluate the relative strength of the baselines introduced in Section 3 with respect to the following research questions:

**Q1**: Are there significant differences across baselines and scenarios regarding selective error rate?

**Q2**: Are there significant differences across baselines and scenarios regarding violations of the target coverage?

**Q3**: How are the methods' rejection rates distributed among the classes?

**Q4**: How do the methods behave when flipping the learning task to maximise the coverage under constraints on the error rate?

**Q5**: How do the methods react to out-of-distribution test examples?

We differentiate from previous works in several respects:

- Regarding **Q1**, our study goes beyond existing ones in two important ways. First, prior evaluations involving SC methods were performed using less than seven baselines

and less than ten datasets whereas we consider 18 methods and 44 datasets. Second, prior benchmarks largely focused on image data whereas our benchmark also include tabular data.

- Concerning **Q2**, only a few works investigate coverage violations, i.e., Geifman and El-Yaniv (2019); Pugnana and Ruggieri (2023a,b). As in **Q1**, this was done on a much smaller scale: for example, Geifman and El-Yaniv (2019) considered only a single image dataset, while Pugnana and Ruggieri (2023a) and Pugnana and Ruggieri (2023b) considered eight and nine binary datasets respectively;

- Only the work by Pugnana and Ruggieri (2023b) addresses **Q3** and highlights that the rejection rate is biased against the minority class. However, they considered nine binary datasets and only six baselines;

- We are the first to empirically evaluate **Q4** and assess performances when switching from minimizing selective risk to maximizing coverage on a large and diverse set of data and settings;

- We are the first to evaluate **Q5** and evaluate how SC methods perform when dealing with shifts in the feature space.

## 5 Experimental Evaluation

### 5.1 Experimental Setting

**Datasets and Baselines.** We run experiments on 44 benchmark datasets from real-life scenarios, such as finance and healthcare (Yang et al., 2023). Among these, 20 are image data and 24 are tabular data. Moreover, 13 of these datasets were previously used in testing (at least one) the baselines in their original paper. Details are provided in Tables A1-A2 of the Appendix A.1. We compare a total of 18 baseline methods (presented in Section 3) representing the state-of-the-art SC methods: DG, SAT, SAT+EM (*learn-to-abstain*); SEL-NET, SELNET+EM (*learn-to-select*); SR, SAT+SR, SAT+EM+SR, SELNET+SR, SELNET+EM+SR, ENS, ENS+SR, CONFIDNET, REG, SELE, SCROSS, PLUG-INAUC, AUCROSS (*score-based*).

**Hyperparameters.** The baselines share the same neural-network architecture. For image data, we use either a Resnet34 architecture (He et al., 2016) or the one specified in the original paper. For tabular data, since neural networks are not state-of-the-art methods, we use the architectures proposed by Gorishniy et al. (2021); Grinsztajn et al. (2022), which revised DNN models for tabular data. Overall, we consider two sets of hyperparameters: network-specific (e.g., hidden layers, learning rate), and loss-specific (e.g., $\beta$ for SAT+EM). All networks are trained for 300 epochs. We optimize the hyperparameters using `optuna` (Akiba et al., 2019), a framework for multi-objective Bayesian optimization, with the following inputs: coverage violation and cross-entropy loss as target metrics, `BoTorch` as sampler (Balandat et al., 2020), 10 initial independent trials out of 20 total trials. Among the 20 trials, we select the configuration that (1) has the highest accuracy on the validation set and (2) reaches the target coverage ($\pm 0.05$). Moreover, some baselines require the target coverage $c$ to be known at training time (e.g., SELNET). For the

sake of reducing the computational cost[4], we optimize their hyperparameters using only three values $c \in \{.99, .85, .70\}$ and fix the best-performing architecture for all target coverages. Moreover, SCROSS, AUCROSS, ENS, ENS+SR and PLUGINAUC use the same optimal hyperparameters found for SR as they share the same training loss. Similarly, SEL-NET+SR, SELNET+EM+SR, SAT+SR and SAT+EM+SR employ the same optimal configuration as, respectively, SELNET, SELNET+EM, SAT and SAT+EM. We detail the parameter choices in Appendix A.2.

**Experimental setup.**  For each combination of datasets and baselines, we run the following experiment: (i) we randomly split the available data into training, calibration, validation, and test sets using the proportion 60/10/10/20%, (ii) we consider the following 7 target coverages $c \in \{.7, .75, .8, .85, .9, .95, .99\}$, (iii) we tune the baseline's hyperparameters using training, calibration, and validation sets as described in the previous paragraph, (iv) we use such optimal hyperparameters to train the baseline on the training set and calibrate the confidence function on the calibration set, (v) we draw 100 bootstraps datasets from the test set (see (Rajkomar et al., 2018)) with the same size at the test set, and, finally, (vi) we compute the empirical selective error rate[5] $\widehat{Err}(h, g, \mathcal{T}_{test})$, the empirical coverage $\hat{\phi}(g, \mathcal{T}_{test})$, and, for binary datasets, the class distribution over the accepted instances for each of the 100 bootstrapped datasets $\mathcal{T}_{test}$. For each evaluation metric, we compute its mean and standard deviation over the 100 bootstrap runs. In reporting results, we distinguish between binary and multi-class (i.e., $> 2$ classes) problems because PLUGINAUC and AUCROSS are specific for binary classification.

Regarding computational resources, we split the workload over three machines: (1) a 25 nodes cluster equipped with $2 \times 16$-core @ 2.7 GHz (3.3 GHz Turbo) POWER9 Processor and 4 NVIDIA Tesla V100 each, OS RedHatEnterprise Linux release 8.4; (2) a 96 cores machine with Intel(R) Xeon(R) Gold 6342 CPU @ 2.80GHz and two NVIDIA RTX A6000, OS Ubuntu 20.04.4; (3) a 128 cores machine with AMD EPYC 7502 32-Core Processor and four NVIDIA RTX A5000, OS Ubuntu 20.04.6.

## 5.2 Experimental Results

We report here the main experimental results w.r.t. the research questions Q1–Q5. Additional results are reported in the Appendix B.

**Q1. Comparing the error rates.**  We introduce a normalized version of the empirical selective error rate, called *relative error* rate:

$$RelErr(h, g, \mathcal{T}_{test}) = \frac{\widehat{Err}(h, g, \mathcal{T}_{test})}{\widehat{Err}(h_{maj}, g, \mathcal{T}_{test})}, \tag{17}$$

where $\widehat{Err}(h_{maj}, g, \mathcal{T}_{test})$ is the empirical selective error rate obtained by always predicting the majority class in the training set. This normalization accounts for variability in task

---

4. Tuning the networks is computationally expensive, requiring more than 15 days on some large datasets, such as `food101`.
5. The empirical selective error rate is the empirical risk (7) w.r.t. the 0-1 loss. Almost all of the baselines are optimized for such a metric, except PLUGINAUC and AUCROSS that are designed for increasing AUC.

prediction difficulty. Intuitively, the closer the relative error rate to 0 the better. Values close to 1 denote selective error rates similar to the ones of a majority classifier.

Figure 4 reports the mean relative error rates for the top two and the worst two[6] baselines. We limit the number of reported baselines for clarity. Tables with detailed results at the dataset level are reported in the Appendix B.3.

For binary data, the best-performing methods are ENS+SR and SR. ENS+SR's relative error rate is $\approx .485$ at $c = .99$, decreasing to $\approx .365$ at $c = .70$. SR ranges from $\approx .488$ at $c = .99$ to $\approx .363$ at $c = .70$. The worst baselines are DG and REG, with relative error rates of $\approx .615$ and $.544$ at $c = .99$ respectively, up to $\approx .564$ and $\approx .529$ at $c = .70$.

Also for multiclass data, ENS+SR and SR achieve the best results. The relative error rate ENS+SR ranges from $\approx .182$ at $c = .99$ to $\approx .117$ at $c = .70$, while SR starts from $\approx ..197$ at $c = .99$, and decreases down to $\approx .127$ for $c = .70$. SELE and REG are the worst methods. The former passes from $\approx .252$ at $c = .99$ to $\approx .217$ at $c = .70$. The latter achieves $\approx .256$ at $c = .99$ and $c = .70$, with no improvement for small target coverages.

Next, we check the statistical significance of these results. For each target coverage and bootstrapped dataset, we rank the compared methods from 1 (the best) to 18 (the worst) w.r.t. the relative error rate. These rankings are then used in the Friedman's omnibus test of equality of means and in its post-hoc Nemenyi test, following the steps described in Demsar (2006). Figure 5 shows Critical Difference (CD) plots, which provide a graphical representation of the output of the Nemenyi test. In each plot, the horizontal axis reports the average rank of each method – where being closer to one (farther to the right) implies better performances. A bold line connects methods whose differences are not statistically significant at 0.05 significance level. The plots show that there is no clear winner regardless of the coverage and of the binary/multiclass classification task. The group of not-statistically-different top methods contains between 8 and 14 baselines. However, ENS+SR is always the top ranked baseline, which makes it a good choice in general.

**Q2. Comparing the empirical coverages.** The constraint on the target coverage $c$ in (9) is essential in many scenarios. Nevertheless, most papers do not sufficiently investigate the actual coverage achieved by the baselines. We assess how much the empirical coverage deviates from the target coverage $c$ on the bootstrap dataset $\mathcal{T}_{test}$. To account for small coverage violations, we introduce a user-defined tolerance $\varepsilon$, and define the $\varepsilon$-*coverage violation*:

$$CovViol_\varepsilon(g, \mathcal{T}_{test}) = \min(0, \ \hat{\phi}(g, \mathcal{T}_{test}) - c + \varepsilon),$$

where $\hat{\phi}(g, \mathcal{T}_{test})$ is the empirical coverage on $\mathcal{T}_{test}$. Intuitively, $CovViol_\varepsilon$ is zero when the empirical coverage is greater or equal than the target coverage minus the tolerance; and it is greater than zero when the empirical coverage is smaller than $c - \varepsilon$. By looking at different tolerances $\varepsilon$, one can evaluate how the baselines perform w.r.t. small, medium or large coverage violations. We define the satisfaction of the constraint as:

$$ConSat(\varepsilon) = \mathbb{1}(CovViol_\varepsilon = 0)$$

and report in Figure 6 the mean and standard deviation of *ConSat* for $\varepsilon$ in $\{0, .01, .02, .05, .10\}$. As for Figure 4, we limit the number of baselines to the two best and worst ones w.r.t. *ConSat*.

---

6. We rank baselines based on the mean value of relative error rate over all the target coverages.

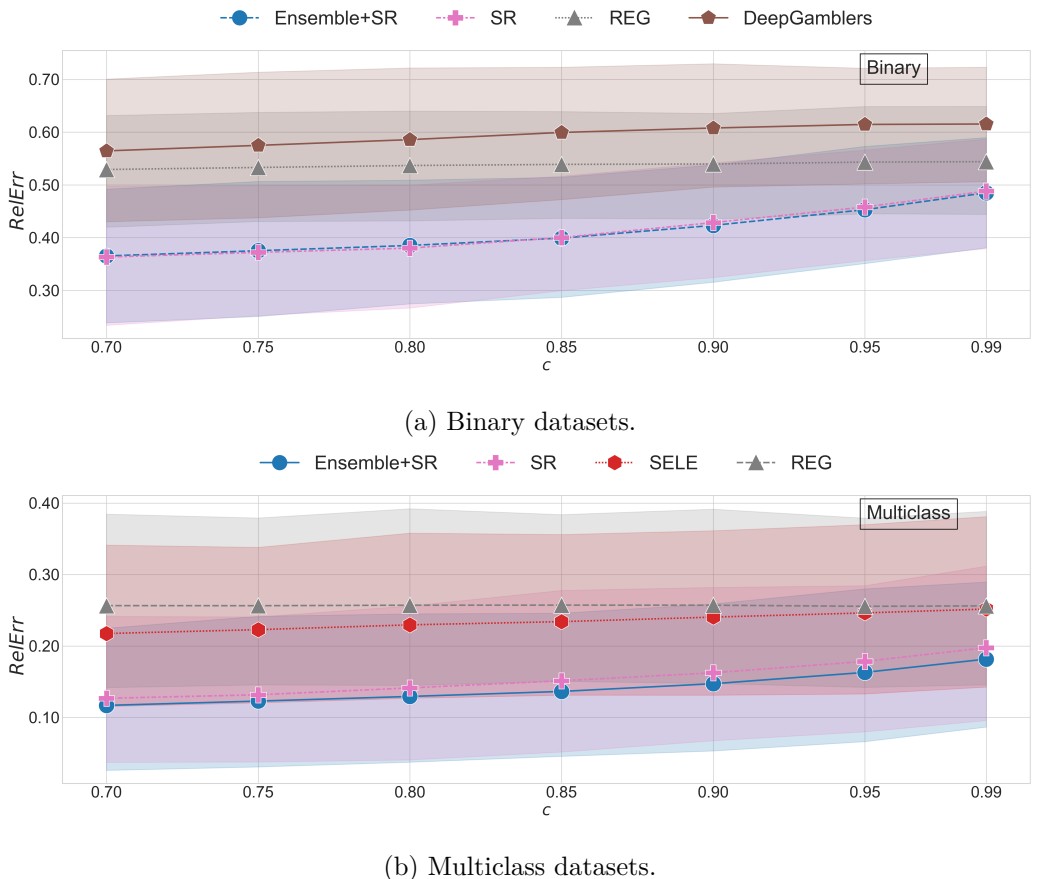

(a) Binary datasets.

(b) Multiclass datasets.

Figure 4: Q1: *RelErr* as a function of target coverage $c$ for two best and worst approaches on (a) binary and (b) multiclass problems. On each subplot, only the two best and worst approaches are shown for readability.

As one would expect, the overall performances gradually improve for all baselines when increasing $\varepsilon$, and the gap among the baselines decreases. For binary data, the best methods are ENS and PLUGINAUC, and the worst methods are AUCROSS and SCROSS. When considering that no violation is allowed, i.e., $\varepsilon = 0$, the baselines satisfy the constraint between $\approx 39.9\%$ (CONFIDNET) and $\approx 56.5\%$ (SCROSS) of the times. For $\varepsilon = .01$ ENS has the highest value of *ConSat* ($\approx .887$); for $\varepsilon = .02$ SAT is the best method ($\approx 0.976$); for $\varepsilon = .05$ both PLUGINAUC and SAT satisfy the constraint all the times.

For multiclass data, the top performers are SCROSS and SAT+EM, while the worst methods are REG and DG. At $\varepsilon = 0$, SCROSS has no coverage violations $\approx 75\%$ of the times, which is 25 percentage points more than the worst performing methods (SELE and REG). Interestingly, already at $\varepsilon = .02$, four methods (i.e., SCROSS, SAT+EM, SR, SAT+SR) always reach zero violations. For $\varepsilon = .05$, only CONFIDNET and SEL-NET+SR do not reach zero violations. In summary, coverage violations are generally limited, and noticeable differences among the baselines only occur at very small tolerances.

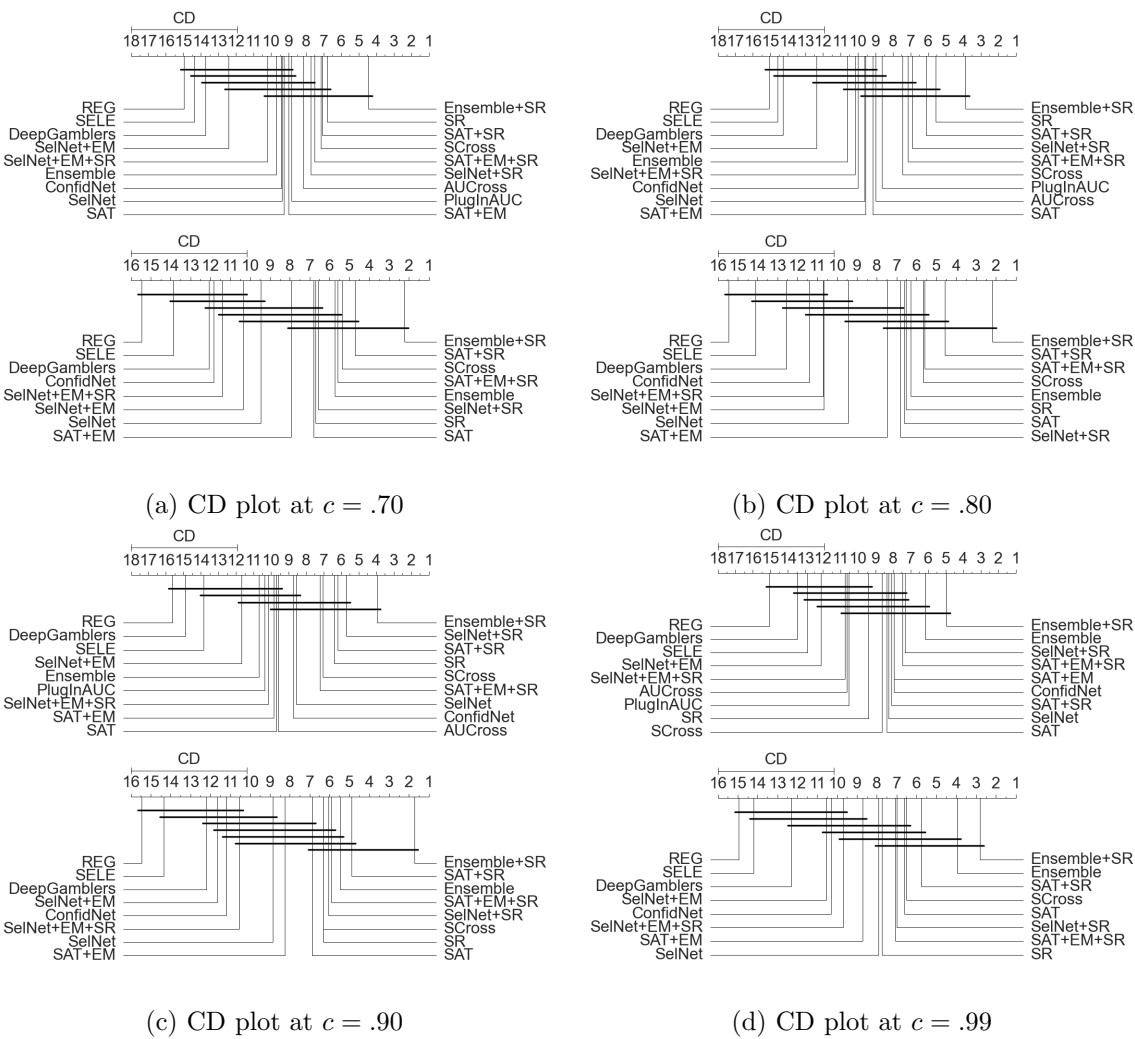

(a) CD plot at $c = .70$

(b) CD plot at $c = .80$

(c) CD plot at $c = .90$

(d) CD plot at $c = .99$

Figure 5: Q1: CD plots of relative error rate *RelErr* for different target coverages. Top plots for binary datasets. Bottom plots for multiclass datasets.

**Q3. Rejection rate over classes.** Pugnana and Ruggieri (2023b) observed that, in imbalanced classification tasks, selective classification methods reject proportionally more instances from the minority class. In this paragraph, we analyze this behavior on 7 binary class datasets of our collection with a minority class prior estimate $p \leq 0.25$ (Perini et al., 2020). Detailed results for the other binary datasets are reported in the Appendix B.3. First, let us introduce the *minority coefficient*:

$$MinCoeff = \frac{p_a}{p}, \tag{18}$$

defined as the ratio of the minority class proportion $p_a$ in the accepted instances over the minority class prior $p$. Ideally, the minority coefficient should be $\approx 1$. Lower values indicate that the selective function introduces a bias against the minority class.

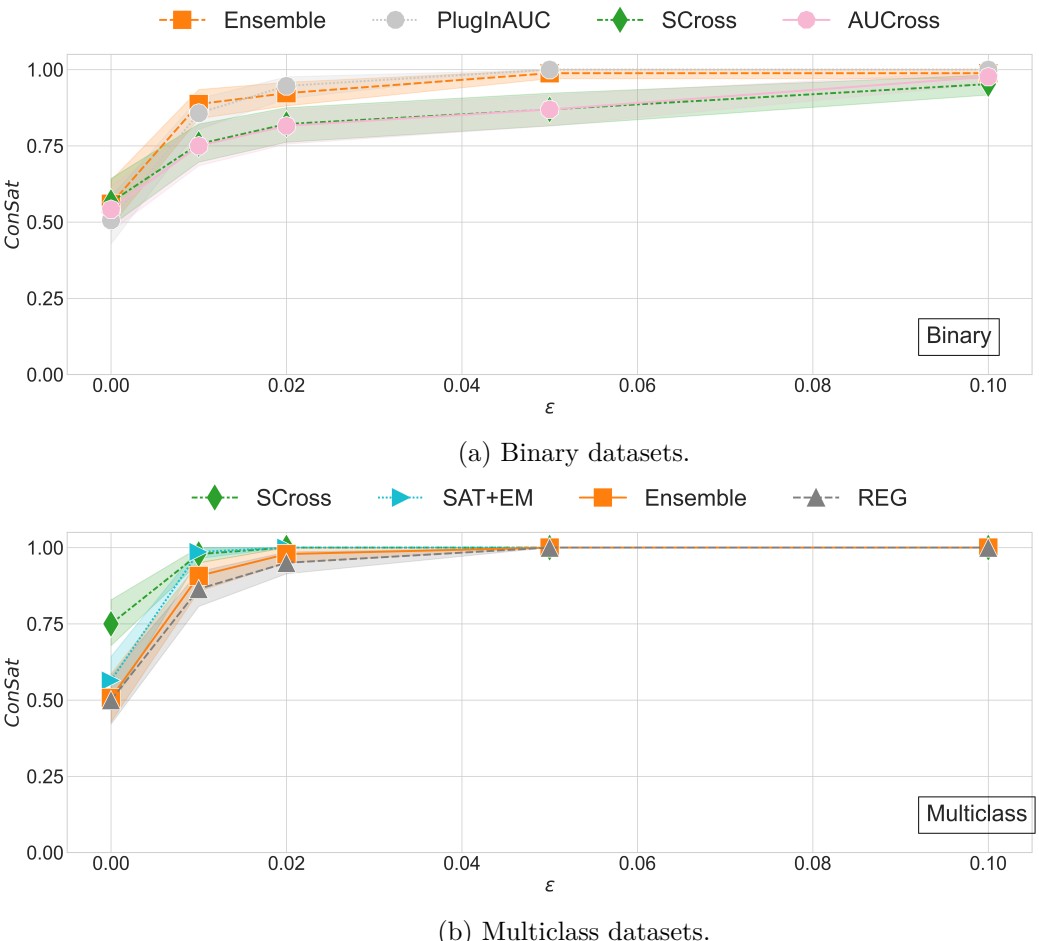

(a) Binary datasets.

(b) Multiclass datasets.

Figure 6: Q2: *ConSat* as a function of tolerance $\varepsilon$ for two best and worst approaches on (a) binary and (b) multiclass problems. On each subplot, only the two best and worst approaches are shown for readability.

Figure 7 shows the mean minority coefficient for the best two and worst two baselines at the variation of the target coverage $c$. The best methods are AUCROSS and PLUGIN-AUC. Their minority coefficient is $\approx 1.00$ and $\approx 1.01$ respectively at $c = .99$, and it remains steady for lower coverages. At $c = .70$, PLUGINAUC reaches a mean *MinCoeff* $\approx 1.05$, and AUCROSS achieves *MinCoeff* $\approx 0.997$. For all the other baselines, there is a clear trend: the smaller the target coverage, the smaller the minority coefficient. For 11 out of 18 baselines, *MinCoeff* drops below .50 at $c = .70$. The worst methods are SELNET and DG. For the former, the mean *MinCoeff* ranges from $\approx .946$ at $c = .99$ to $\approx .375$ at $c = .70$. For the latter, the mean *MinCoeff* ranges from $\approx .966$ at $c = .99$ to $\approx .413$ for $c = .70$.

These results support the findings by Pugnana and Ruggieri (2023b), highlighting that the current approaches to SC, with the exception of AUCROSS and PLUGINAUC, do not take into account the issue of class balancing in the selected instances.

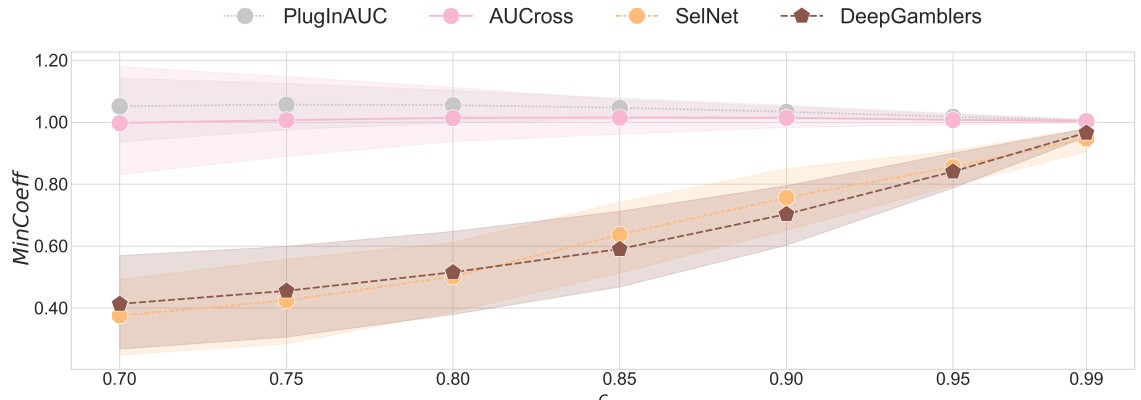

Figure 7: Q3: *MinCoeff* as a function of target coverage *c* for two best and worst approaches. Only the two best and worst approaches are shown for readability.

**Q4. Flipping the learning task to maximize the model coverage under error constraints.** The vast majority of methods focus on the bounded-abstention model of Problem 2. To the best of our knowledge, the only method explicitly addressing the bounded-improvement model of Problem 1 is due to Gangrade et al. (2021), whose code has not been fully released. However, tackling the bounded-improvement model is useful in some application scenarios. Consider the bank example again. SC here can be used in two ways: on the one hand, the bank can set a target coverage *c* and calibrate a selective classifier so that *c*% of the cases are directly handled by the ML model, while the remaining - most difficult - ones are deferred to human experts. Here *c* is chosen on the basis of the personnel capacity of the bank. On the other hand, the bank can also be interested in maximizing the model coverage without incurring too many (costly) mistakes. Measuring such maximal coverage allows for planning the amount of human effort needed for the difficult cases.

In this subsection, we evaluate the performances of the bounded-abstention baselines when flipping the task to the bounded-abstention problem through the *Selection with Guaranteed Risk* (SGR) algorithm proposed by Geifman and El-Yaniv (2017). SGR is a [classifier $h$, confidence $k_h$]-agnostic approach that optimizes the selection threshold $\tau$ (see (4) such that the selective error rate at test time is guaranteed to be bounded ($\leq r$) with probability $> 1 - \delta$ and the coverage is maximized. We apply SGR on all the baselines but AUCROSS and PLUGINAUC, as their hard selection function is not compatible with SGR. Moreover, since SELNET needs specific coverage for training, we use all $c$'s one at a time, and compute the average results after applying SGR. We run experiments for four target error rates $r \in \{e/10, e/5, e/2, e\}$, where $e$ is the dataset-specific error of the majority-class classifier $h_{maj}$ on the whole test set, and set $\delta = 0.001$.

Figure 8 reports the results for the best two and the worst two baselines. The top plot shows the mean empirical coverage over the test sets of all baseline datasets (the higher, the better). The bottom plot shows the mean *error ratio* (the smaller, the better):

$$ErrCoeff = \hat{r}/r,$$

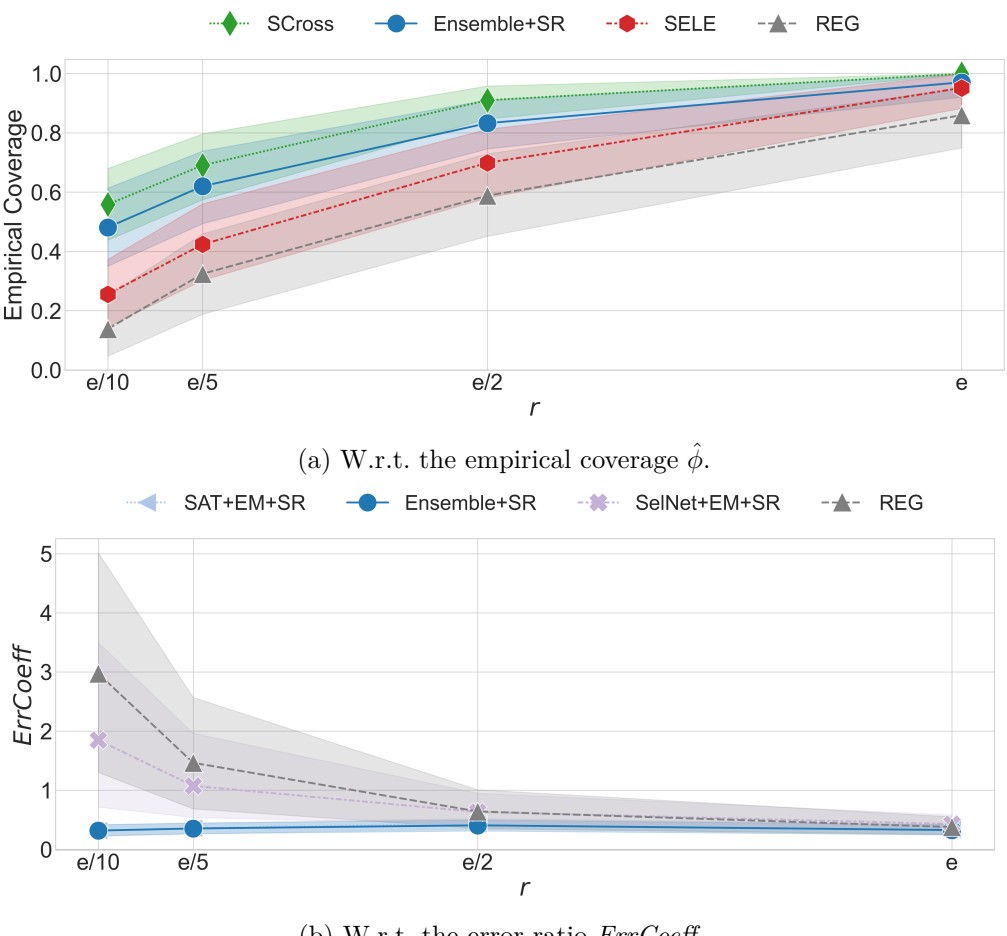

(a) W.r.t. the empirical coverage $\hat{\phi}$.

(b) W.r.t. the error ratio *ErrCoeff*.

Figure 8: Q4: SGR performance as a function of target error rate $r$ for the two best and worst approaches in terms of (a) coverage $\hat{\phi}$, and (b) *ErrCoeff*. Only the two best and worst approaches are shown for readability.

between the empirical selective error rate $\hat{r}$ and the target error rate $r$. When looking at the empirical coverage, the best performing baseline is SCROSS, with coverage ranging from .999 for $r = e$ to .558 for $r = e/10$. This is 40 percentage points higher than the worst method, namely REG. The second-best method is ENS+SR, with a mean coverage of $\approx .970$ at $r = e$ and $\approx .481$ at $r = e/10$.

Concerning *ErrCoeff*, we observe that for less strict target errors (i.e., $e$ and $e/2$), all the baselines have error ratios close to 0. For more restrictive target errors, there is a gradual increase in the mean value of *ErrCoeff*. The methods with the smallest error ratios are ENS+SR and SAT+EM+SR, reaching *ErrCoeff* $\approx .316$ and *ErrCoeff* $\approx .317$ respectively at $r = e/10$. The worst methods are REG and SELNET+EM+SR, with a mean error ratio of $\approx 2.97$ and $\approx 1.84$ at $r = e/10$.

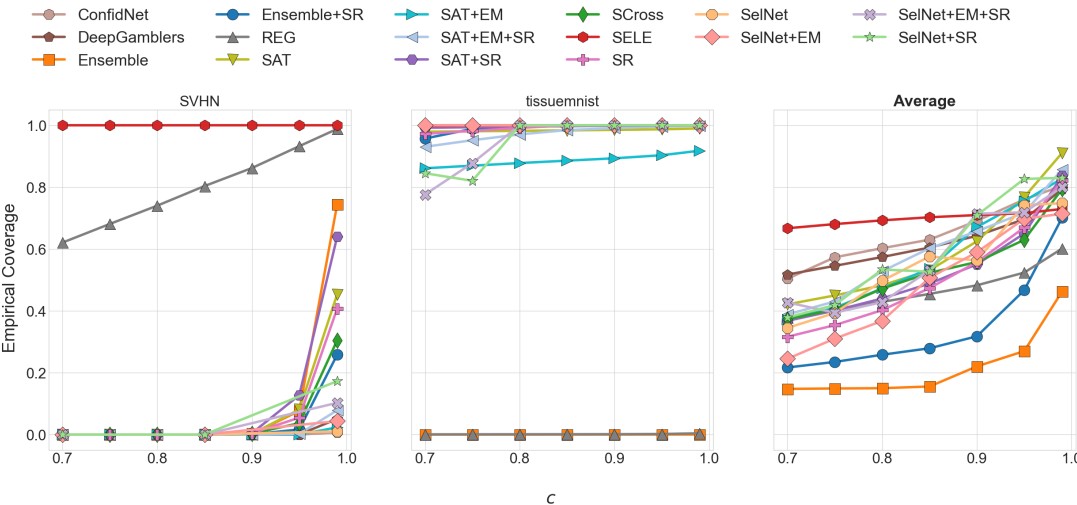

Figure 9: Q5: Empirical coverage $\hat{\phi}$ for out-of-distribution test sets (two selected image datasets and average results over the 20 image datasets) when varying coverages $c$.

**Q5. Testing the methods on out-of-distribution examples.** Although SC methods are not necessarily designed for working in out-of-distribution (o.o.d.) settings[7], robustness of selective classifiers w.r.t. data shifts is highly sought. We investigate this property here by generating o.o.d. instances at test time for image datasets. We take an extreme approach by creating a test set with images made of uniformly random pixel values.[8] An ideal selection function should reject the whole test set, since the images have close to zero probability of being drawn from the same distribution generating the training set.

For each of the 16 baselines that tackle multi-class classification, Figure 9 shows the empirical coverage obtained over the o.o.d. test set over the two selected image datasets and the average results over the 20 image datasets. Detailed results for all 20 datasets are provided in Appendix B.2.

First, we observe that there is no single method that manages to reject all the test instances across all datasets. For example, most of the baselines obtain empirical coverage close to 0 (best) for the 5 lowest coverage values on `SVHN`. On the other hand, on `tissuemnist`, the majority of baselines have always nearly maximum empirical coverage, with the sole exception of REG and ENS that manage to reject all the o.o.d. images.

Ensemble methods are generally better than other methods: ENS reaches the lowest mean empirical coverage on all datasets of $\approx .221$, ranging from $\approx .462$ at $c = .99$ to $\approx .148$ at $c = .70$, and ENS+SR is the second best method with a mean coverage $\approx .353$, ranging from $\approx .701$ at $c = .99$ to $\approx .217$ at $c = .70$.

### 5.3 Discussion

We conclude the experimental section by briefly summarizing the main findings.

---

7. See the novelty rejection approaches in the related work Section 6.

8. We provide additional results with less extreme shifts in the Appendix.

Regarding **Q1**, our results do not contradict the folk wisdom that an ensemble strategy (paired with the softmax response) overcome other baseline methods: ENS+SR always ranks first in terms of relative error rate. However, we stress that, depending on the target coverage, there are always at least nine baselines whose performance can't be distinguished from ENS+SR's one in a statistically significant sense. Conversely, some methods, i.e., REG, SELE, DG, SELNETSELNET+EM,SELNET+EM+SR, CONFIDNET, ENS, and PLUGINAUC, perform worse at least once (in a statistically significant sense) than top-performing baselines. Hence, our findings suggest that the claimed "superiority" of the considered state-of-the-art methods should be treated with caution: when we increase the number of experimental datasets, most methods perform equally well.

Our results on **Q2** show that coverage violations are generally small with a few exceptions of coverage violations above 10%. This confirms that the employed calibration strategies are well suited to achieving an empirical coverage that is fairly close to the target one.

For **Q3**, a significant difference arises among the methods. Only PLUGINAUC and AUCROSS reject equally across classes, while all the other methods abstain more relatively more frequently on the minority class. This behavior can have unforeseen consequences such as inducing cognitive bias in the human decision-maker that must make the decision on the rejected instances (Rastogi et al., 2022). For example, by abstaining more often on bad loan applicants, humans could be prone to associate the model's rejections with bad applicants, even if this might not be necessarily true (Bondi et al., 2022).

The experiments for **Q4** show that SGR can effectively switch from the bounded abstention to the bounded improvement model assuming that target error rate is not too strict. In highly sensitive scenarios, where stronger guarantees are required, SGR often fails, thus suggesting a potential direction for future research towards methods specifically designed for the bounded improvement model.

For **Q5**, the results indicate that the current state-of-the-art baselines fail to reject consistently under distribution shifts. Consequently, practitioners should be cautious about applying SC techniques in the wild without considering potential issues deriving from data shifts. From a research perspective, this opens an intriguing future direction for shift-aware selective classification methods.

We point out that the methods which require training several neural networks might not be a feasible option for very large datasets, due to the huge computational power required. Such methods include the baselines ENS, ENS+SR, SCROSS, AUCROSS, CONFIDNET as well as the learn-to-select methods that require training a separate model for every target coverage.

## 6 Related Work

We present here a few related approaches and discuss in which respect they differ from SC.

**Ambiguity Rejection.** Ambiguity rejection focuses on abstaining on instances close to the decision boundary of the classifier (Hendrickx et al., 2024). SC is one of the main ways to perform ambiguity rejection. In particular, SC methods rely on confidence functions, which identify those instances where the classifier is more prone to make mistakes. Confidence values allow one to trade off coverage for selective risk.

The other main framework to perform ambiguity rejection is generally referred to as Learning to Reject (LtR) and is based on the seminal work by Chow (1970). Similarly to SC, LtR aims at learning a pair (classifier, rejector) such that the rejector determines when the classifier makes a prediction, limiting the predictions to the region where the classifier is likely correct (Cortes et al., 2023). However, LtR deviates from SC in two major aspects. First, the LtR methods learn the trade-off between abstention and prediction not by using confidence functions, but through a parameter $a$, representing the cost of rejection (Herbei and Wegkamp, 2006; Cortes et al., 2016; Tortorella, 2005; Condessa et al., 2013). However, setting the value of this hyperparameter is not straightforward, and it is context-dependant (Denis and Hebiri, 2020). Second, LtR methods are not meant to tackle the problem of minimizing a risk given a target coverage $c$. A more in-depth theoretical analysis for both LtR and SC can be found in (Franc et al., 2023), where the authors show that both frameworks share similar optimal strategies.

**Novelty Rejection.** A strategy orthogonal to ambiguity rejection consists of abstaining on instances that are unlikely to be seen according to the distribution of the training set. This approach is commonly referred to as *novelty rejection* (Dubuisson and Masson, 1993; Cordella et al., 1995), and is highly sought whenever there is a shift between the training and the test set distributions (Hendrickx et al., 2024; Van der Plas et al., 2023). Several techniques have been proposed for building novelty rejectors. As a first approach, one can estimate the marginal density and reject an instance if its probability is below a certain threshold (Nalisnick et al., 2019; Wang and Yiu, 2020). Another option is to employ a one-class classification model that predicts as novel the instances falling out of the region learnt from the training set (Coenen et al., 2020). Further approaches assign a score representing the novelty of an instance and abstain when such a score is above a certain level (Liang et al., 2018; Kühne et al., 2021; Perini and Davis, 2023; Van der Plas et al., 2023). To conclude, we highlight that the goal of novelty rejection differs from the SC goal, i.e. trading off risk and coverage, and linking the two problems is not straightforward (Hendrickx et al., 2024).

**Conformal Prediction.** Conformal prediction (Shafer and Vovk, 2008) augments the prediction of a M model by providing a set of target labels that comprise the true value with a specified (desired) level of confidence (Papadopoulos et al., 2002; Vovk, 2012; Kim et al., 2020; Abad et al., 2022; Angelopoulos et al., 2021). Differently from SC, conformal prediction focuses on quantifying the uncertainty associated with predictions rather than minimizing a specific type of error (Gangrade et al., 2021). Some works try to merge these two frameworks: for instance, in (Angelopoulos and Bates, 2021), conformal prediction is used to give guarantees over the selective error rate in an SC scenario by: (1) training a conformal predictor (e.g., SVC (Romano et al., 2020)), (2) calibrating its confidence levels, (3) setting a selection threshold over the confidence or p-values generated by the conformal predictor.

**Learning to Defer.** Learning to defer (Madras et al., 2018) is a generalization of LtR, where rather than incurring a rejection cost, the AI system can defer instances to human expert(s). One of the main differences in comparison to LtR and SC, is that the expert's predictions might be wrong under the learning to defer framework. This is generally modelled using a cost function (Mozannar and Sontag, 2020). Thus, common methods include

the expert in the loop and aim to find an optimal assignment strategy for the whole human-AI system. Roughly speaking, such a strategy decides whether or not to make the model predict, which results in a cost equal to the model loss, or defer the prediction to the user, which incurs the user cost (Okati et al., 2021; De et al., 2020; Mozannar et al., 2023; Verma et al., 2023; Straitouri et al., 2022).

**Real-world Applications.** In recent years, abstaining AI systems have been deployed to foster human decision-making in increasingly many domains. For example, Van der Plas et al. (2023) describe a novelty rejector for sleep stage scoring. Cianci et al. (2023) exploit the SC strategy by Pugnana and Ruggieri (2023b) to augment a credit scoring ML model with an uncertainty self-assessment. Coenen et al. (2020) use unlabeled data on unaccepted loan applications to build a credit scoring model that can abstain from predicting. Hendrickx et al. (2021) propose a novelty rejector to find unexpected vehicle usage from sensor data and refrain from providing a prediction for such cases. Van Roy and Davis (2023) flag annotation errors in soccer data considering a specific confidence function for tree-based methods (Devos et al., 2023). Bondi et al. (2022) study a selective classifier deferring to humans to evaluate the presence of animals in photo traps. For other applications of abstaining classifiers, we refer to Hendrickx et al. (2024), while we refer to Punzi et al. (2024) for applications of hybrid-decision-making systems.

## 7 Conclusions

**Limitations.** For the sake of a fair comparison, our study focuses on neural network classifiers, as some of the methods assume a deep learning architecture for the classifier.

Due to the large computational costs of the experiments, for each dataset, we consider only a single deep-learning architecture chosen among the ones at the state-of-the-art. E.g., for `cifar10`, we implemented all the baselines using a VGG16 architecture. This might reduce the generalizability of our results to other deep-learning architectures.

We also acknowledge that a few studies, e.g., Gorishniy et al. (2021); Grinsztajn et al. (2022), point out that for tabular datasets, the usage of tree-based models is the current state of the art. In this sense, model-agnostic methods could benefit from using other base classifiers, as shown in Pugnana and Ruggieri (2023a,b).

Another limitation of our benchmark is that we consider only images and tabular data, since they are the main data type over which SC methods have been tested so far. This choice is in line with the goal of this paper, which aims to compare existing approaches fairly. However, our results do not necessarily extend to other kinds of data such as text, audio or time series.

A possible concern could also regard the size of the datasets in our benchmark, which never exceeds $\approx 300k$ instances. This aspect could impact the external validity of our discussion. However, there are reasons for this choice. First, the considered datasets are used either in popular benchmarks (Yang et al., 2023; Gorishniy et al., 2021; Grinsztajn et al., 2022), or by selective classification works, e.g., (Geifman and El-Yaniv, 2019; Franc et al., 2023; Pugnana and Ruggieri, 2023b). Second, since we trained and fine-tuned all the models from scratch, with considerable computational costs, we decided to prioritize the variety of data over the size of a single dataset.

Moreover, our bootstrap procedure quantifies variability only in the test set. According to several works, such as Kohavi (1995), the best resampling method is stratified k-fold cross-validation with $K = 10$. Unfortunately, these strategies are not computationally feasible when employing large neural networks as in our study. Hence, we had to opt for a single train-test-split and bootstrap only the test set, as done for instance by Rajkomar et al. (2018).

Finally, our study does not report on the running times of the baselines, since, due to load balancing issues, we had to distribute the experiments over several machines with different hardware settings. This made it impossible to compare the running times of runs over different machines. However, we point out that the running times are proportional to the number of training tasks required by each method. E.g., ENS requires to train ten neural networks (see Appendix A.2), leading to a running time of about ten times the one of PLUGINAUC, which requires to train a single neural network.

**Conclusions.** We extensively evaluated 18 SC baselines over 44 datasets, taking into account both images and tabular data as well as both binary and multiclass classification tasks. Regarding previously investigated tasks, our extended analysis shows that: *(i)* there are no statistically significant differences among most of the methods in terms of selective error rate, even though ENS+SR always ranks first across all the baselines; *(ii)* large coverage violations are rare for all the methods with no significant difference among baselines for our data; *(iii)* on binary classification tasks, we observed different patterns between imbalanced and balanced domains regarding rejection rates across classes: in the former case, only AUCROSS and PLUGINAUC succeeded in not primarily rejecting the minority class. Moreover, we also emphasize novel findings: *(iv)* we tested empirically the effectiveness of SGR to switch from the bounded-abstention setting to the bounded-improvement one, noticing room for improvement when a very small target error rate is required; *(v)* we show how current methods fail in correctly rejecting instances when extreme feature shifts occur, pointing to a highly relevant open problem in the area.

## Broader Impact Statement

Because Machine Learning models can make errors in their predictions, adding a reject option is a means for improving their trustworthiness. The selective classification framework is one of the most popular ways to achieve such a goal by coupling a classifier with a selective function that decides whether to accept or reject making a prediction. However, existing selective classification methods have never been evaluated on a large scale. Our work is the first to fill this gap, providing the first extensive benchmark for testing selective classification methods. The experimental evaluation sheds light on the strengths and weaknesses of selective classification methods for what concerns their error rate, acceptance rate (called coverage), distribution of rejection over classes, and robustness to data shifts.

## Acknowledgments and Disclosure of Funding

The work of A. Pugnana and S. Ruggieri has been partly funded by PNRR - M4C2 - Investimento 1.3, Partenariato Esteso PE00000013 - "FAIR - Future Artificial Intelligence

Research" - Spoke 1 "Human-centered AI", funded by the European Commission under the NextGeneration EU programme, and by the project FINDHR funded by the EU's Horizon Europe research and innovation program under g.a. No 101070212. Views and opinions expressed are however those of the authors only and do not necessarily reflect those of the EU. Neither the EU nor the granting authority can be held responsible for them.

L. Perini received funding from FWOVlaanderen (aspirant grant 1166222N). Moreover, L. Perini and J. Davis received funding from the Flemish Government under the "Onderzoeksprogramma Artificiële Intelligentie (AI) Vlaanderen" programme.

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

## Appendix A. Experimental Details

We provide all the additional information on datasets, settings, and code for replicating the experiments of the paper.

### A.1 Datasets

Table A1 reports the datasets used in our benchmarking and a link to retrieve the original data. We also include whether the dataset was considered in a previous SC evaluation.

Table A2 reports some experimental details, including training size; batch size used for training; feature space in terms of features for tabular data and image size for image data; target space number of classes $m$; the percentage of the minority class in each dataset. We also report the Deep Neural Network (DNN) architectures we employed for each dataset. Such a choice was made according to the following criteria:

*(i)* a former paper in the literature used this dataset and employed a specific architecture;

*(ii)* if point *(i)* does not apply, we did the following:

  – for image data, we employed a ResNet34 architecture;
  – for tabular data, if a dataset was tested in Gorishniy et al. (2021), we applied the best-performing architecture on that specific dataset. Otherwise, we employed the FTTransformer architecture following the suggestion by Grinsztajn et al. (2022).

All the data were re-shuffled, normalized and split into training, test, calibration and validation sets, according to a 60%, 20%, 10%, and 10% proportion (respectively). In the code repository, we provide the `Python` scripts to recreate the data employed in this analysis.

### A.2 Hyperparameter Settings

We optimize the hyperparameters using `Optuna` (Akiba et al., 2019), a framework for multi-objective Bayesian optimization, with the following inputs: coverage violation and cross-entropy loss as target metrics, `BoTorch` as sampler (Balandat et al., 2020), 10 initial independent trials out of 20 total trials. We report in Table A3 the parameter space we used during the tuning procedure. Some hyperparameters are loss-specific, as they refer to a specific baseline loss, while others are network-specific, as they refer to a specific deep neural network architecture. We report the search space in the last column, where the notation $[a; b]_X$ stands for all values within $[a; b]$ that are linearly spaced with a gap equal to $X$. For example, $[0, 60]_{15}$ indicates the set of values $\{0, 15, 30, 45, 60\}$. For small sets of values, we directly indicate all the possibilities using the notation $\{a_1, a_2, \dots\}$. We discretize the search space of most hyperparameters because Bayesian Optimization suffers from high-dimensional spaces. We specify that if `time decay` is set to `True`, we halve the learning rate every 25 epochs as done in Geifman and El-Yaniv (2019). Moreover, for image data, which we found to be more unstable during the training, we start the `Optuna` optimization procedure using default values if these were suggested in some SC paper. Due to the huge computational cost of the tuning procedure, SELNET+SR, SELNET+EM+SR, SAT+SR and SAT+EM+SR use the same optimal hyperparameters as,

Table A1: Dataset sources.

| Dataset | Data Type | Link | Previous SC paper |
|---|---|---|---|
| adult | Tabular | uci/adult | Pugnana and Ruggieri (2023a,b) |
| aloi | Tabular | openml/id=1592 | – |
| bank | Tabular | uci/bank+marketing | Franc et al. (2023) |
| bloodmnist | Image | zenodo/6496656/files/bloodmnist | – |
| breastmnist | Image | zenodo/6496656/files/breastmnist | – |
| catsdogs | Image | kaggle/dogs-vs-cats | Geifman and El-Yaniv (2019); Liu et al. (2019); Huang et al. (2020); Pugnana and Ruggieri (2023a,b) |
| chestmnist | Image | zenodo/6496656/files/chestmnist | – |
| cifar10 | Image | pytorch/vision/CIFAR10 | Geifman and El-Yaniv (2019); Corbière et al. (2019); Huang et al. (2020); Feng et al. (2023); Pugnana and Ruggieri (2023b) |
| compass | Tabular | openml/id=44162 | – |
| covtype | Tabular | openml/id=1596 | Franc et al. (2023) |
| dermamnist | Image | zenodo/6496656/files/dermamnist | – |
| electricity | Tabular | openml/id=44120 | – |
| eye | Tabular | openml/id=44157 | – |
| food101 | Image | pytorch/vision/Food101 | Feng et al. (2023) |
| giveme | Tabular | kaggle/GiveMeSomeCredit | Pugnana and Ruggieri (2023a,b) |
| helena | Tabular | openml/id=41169 | – |
| heloc | Tabular | openml/id=45023 | – |
| higgs | Tabular | openml/id=23512 | – |
| house | Tabular | openml/id=43957 | – |
| indian | Tabular | openml/id=41972 | – |
| jannis | Tabular | openml/id=44079 | – |
| kddipums97 | Tabular | openml/id=44124 | – |
| letter | Tabular | openml/id=6 | Franc et al. (2023) |
| magic | Tabular | openml/id=44125 | – |
| miniboone | Tabular | openml/id=44119 | – |
| MNIST | Image | pytorch/vision/MNIST | Lakshminarayanan et al. (2017); Liu et al. (2019); Corbière et al. (2019) |
| octmnist | Image | zenodo/6496656/files/octmnist | – |
| online | Tabular | openml/id=45060 | – |
| organamnist | Image | zenodo/6496656/files/organamnist | – |
| organcmnist | Image | zenodo/6496656/files/organcmnist | – |
| organsmnist | Image | zenodo/6496656/files/organsmnist | – |
| oxfordpets | Image | pytorch/vision/oxfordpets | – |
| pathmnist | Image | zenodo/6496656/files/pathmnist | – |
| phoneme | Tabular | openml/id=44127 | – |
| pneumoniamnist | Image | zenodo/6496656/files/pneumoniamnist | – |
| pol | Tabular | openml/id=43991 | – |
| retinamnist | Image | zenodo/6496656/files/retinamnist | – |
| rl | Tabular | openml/id=44160 | – |
| stanfordcars | Image | pytorch/vision/StanfordCars | Feng et al. (2023) |
| SVHN | Image | pytorch/vision/SVHN | Geifman and El-Yaniv (2017, 2019); Liu et al. (2019); Corbière et al. (2019) |
| tissuemnist | Image | zenodo/6496656/files/tissuemnist | – |
| ucicredit | Tabular | openml/id=42477 | Pugnana and Ruggieri (2023a,b) |
| upselling | Tabular | openml/id=44158 | – |
| waterbirds | Image | stanford.edu/dro/waterbird | Jones et al. (2021) |

Table A2: Dataset details.

| Dataset | Training Size | Batch Size | # Features | # Classes | Minority Ratio | DNN Architecture |
|---|---|---|---|---|---|---|
| adult | 29,303 | 256 | 13 | 2 | 23.9% | FTTransformer |
| aloi | 64,800 | 512 | 128 | 1000 | 0.1% | TabResnet |
| bank | 27,126 | 128 | 16 | 2 | 11.7% | TabResnet |
| bloodmnist | 10,253 | 128 | $28 \times 28$ | 8 | 7.1% | Resnet18 |
| breastmnist | 468 | 64 | $28 \times 28$ | 2 | 26.9% | Resnet18 |
| catsdogs | 15,000 | 128 | $64 \times 64$ | 2 | 50.0% | VGG |
| chestmnist | 67,272 | 512 | $28 \times 28$ | 2 | 10.3% | Resnet18 |
| cifar10 | 36,000 | 128 | $32 \times 32$ | 1 | 100.0% | VGG |
| compass | 9,985 | 128 | 17 | 2 | 50.0% | FTTransformer |
| covtype | 348,605 | 1024 | 54 | 7 | 0.5% | FTTransformer |
| dermamnist | 6,008 | 128 | $28 \times 28$ | 7 | 1.1% | Resnet18 |
| electricity | 23,083 | 128 | 7 | 2 | 50.0% | FTTransformer |
| eye | 4,564 | 128 | 23 | 2 | 50.0% | FTTransformer |
| food101 | 60,600 | 256 | $224 \times 224$ | 101 | 1.0% | Resnet34 |
| giveme | 90,000 | 512 | 8 | 2 | 6.7% | TabResnet |
| helena | 39,116 | 512 | 27 | 100 | 0.2% | TabResnet |
| heloc | 6,000 | 128 | 22 | 2 | 50.0% | FTTransformer |
| higgs | 58,829 | 512 | 28 | 2 | 47.1% | FTTransformer |
| house | 8,092 | 128 | 16 | 2 | 50.0% | FTTransformer |
| indian | 5,485 | 128 | 220 | 8 | 0.2% | TabResnet |
| jannis | 34,548 | 512 | 54 | 2 | 50.0% | FTTransformer |
| kddipums97 | 3,112 | 128 | 20 | 2 | 50.0% | FTTransformer |
| letter | 12,000 | 128 | 16 | 26 | 3.7% | FTTransformer |
| magic | 8,024 | 128 | 10 | 2 | 50.0% | FTTransformer |
| miniboone | 43,798 | 256 | 50 | 2 | 50.0% | TabResnet |
| MNIST | 42,000 | 128 | $28 \times 28$ | 10 | 9.0% | Resnet34 |
| octmnist | 65,585 | 512 | $28 \times 28$ | 4 | 8.1% | Resnet18 |
| online | 7,398 | 128 | 17 | 2 | 15.5% | FTTransformer |
| organamnist | 35,310 | 256 | $28 \times 28$ | 11 | 4.0% | Resnet18 |
| organcmnist | 14,196 | 128 | $28 \times 28$ | 11 | 4.8% | Resnet18 |
| organsmnist | 15,132 | 128 | $28 \times 28$ | 11 | 4.6% | Resnet18 |
| oxfordpets | 4,409 | 128 | $224 \times 224$ | 2 | 32.3% | Resnet34 |
| pathmnist | 64,308 | 512 | $28 \times 28$ | 9 | 8.9% | Resnet18 |
| phoneme | 1,901 | 128 | 5 | 2 | 50.0% | FTTransformer |
| pneumoniamnist | 3,512 | 128 | $28 \times 28$ | 2 | 27.1% | Resnet18 |
| pol | 9,000 | 128 | 26 | 11 | 1.7% | FTTransformer |
| retinamnist | 960 | 128 | $28 \times 28$ | 5 | 5.8% | Resnet18 |
| rl | 2,982 | 128 | 12 | 2 | 50.0% | FTTransformer |
| stanfordcars | 9,710 | 128 | $224 \times 224$ | 196 | 0.3% | Resnet34 |
| SVHN | 59,573 | 128 | $32 \times 32$ | 10 | 6.3% | VGG |
| tissuemnist | 141,830 | 1024 | $28 \times 28$ | 8 | 3.5% | Resnet18 |
| ucicredit | 18,000 | 128 | 23 | 2 | 22.1% | TabResnet |
| upselling | 3,017 | 128 | 45 | 2 | 50.0% | FTTransformer |
| waterbirds | 7,072 | 128 | $224 \times 224$ | 2 | 22.6% | Resnet50 |

respectively, SELNET, SELNET+EM, SAT and SAT+EM. Similarly, SCROSS, ENS, ENS+SR, AUCROSS, and PLUGINAUC employ the best configuration found for SR as they share the same training loss, i.e., cross-entropy. For both SCROSS and AUCROSS we set $K = 5$, following the suggestions in Pugnana and Ruggieri (2023a,b). For both ENS and ENS+SR we used the default value of $K = 10$, following the suggestions in Lakshminarayanan et al. (2017). For the uncertainty network of CONFIDNET, we employed the same choice architecture detailed in the original paper (Corbière et al., 2019), i.e., the same main body as the network classifier followed by 4 dense layers in a single node with sigmoid activation. We used such a structure also for building SELE and REG uncertainty

Table A3: Hyperparameter spaces.

| Parameter | Loss-Specific | Network-Specific | Search Space |
|---|---|---|---|
| $o$ | DG | No | $[1; m]_{.02}$ |
| $\gamma$ | SAT, SAT+EM | No | $[0.9; 0.99]_{.01}$ |
| $E_s$ | SAT, SAT+EM | No | $[0; 60]_{15}$ |
| $\beta$ | SAT+EM, SELNET+EM | No | $\{1e{-}4, 1e{-}3, , 1e{-}2, 1e{-}1\}$ |
| $\alpha$ | SELNET, SELNET+EM | No | $\{.25; .75\}_{.05}$ |
| $\lambda$ | SELNET, SELNET+EM | No | $\{8, 16, 32, 64\}$ |
| optimizer | No | No | $\{\texttt{SGD}, \texttt{Adam}, \texttt{AdamW}\}$ |
| learning rate | No | No | $\{1e-5, 1e-4, 1e-3, 1e-2\}$ |
| optimizer unc. | CONFIDNET, SELE, REG | Yes | $\{\texttt{SGD}, \texttt{Adam}, \texttt{AdamW}\}$ |
| learning rate unc. | CONFIDNET, SELE, REG | Yes | $\{1e-8, 1e-7, 1e-6, 1e-5, 1e-4, 1e-3, 1e-2\}$ |
| time decay | No | No | $\{\texttt{True}, \texttt{False}\}$ |
| nesterov | No | No | $\{\texttt{True}, \texttt{False}\}$ |
| nesterov unc. | CONFIDNET, SELE, REG | Yes | $\{\texttt{True}, \texttt{False}\}$ |
| weight decay | No | No | $\{1e{-}6, 1e{-}5, 1e{-}4, 1e{-}3\}$ |
| d_token | No | FTTransformer, TabResNet | $[64, 512]_{64}$ |
| n_blocks | No | FTTransformer, TabResNet | $\{1, 2, 3, 4\}$ |
| d_hidden_factor | No | FTTransformer, TabResNet | $[2/3; 8/3]_{1/3}$ |
| attention_dropout | No | FTTransformer | $\{0; .5\}_{.05}$ |
| residual_dropout | No | FTTransformer | $\{0; .2\}_{.05}$ |
| ffn_dropout | No | FTTransformer | $\{0; .5\}_{.05}$ |
| d_main | No | TabResNet | $[64, 512]_{64}$ |
| d_dropout_first | No | TabResNet | $\{0; .5\}_{.05}$ |
| d_dropout_second | No | TabResNet | $\{0; .5\}_{.05}$ |
| batch_norm | No | VGG | $\{\texttt{True}, \texttt{False}\}$ |
| zero_init_residual | No | ResNet34, ResNet50 | $\{\texttt{True}, \texttt{False}\}$ |

networks. Following the empirical evaluation in (Franc et al., 2023), we split the training data in half to train SELE and REG networks: on the one half we train the classifier, on the other half, the uncertainty network. We provide the best configurations we employed in the final analysis in Tables A4-A12.

Table A4: Best configurations for DG, divided by architectures.

**FTTransformer**

| Dataset | optim. | l. rate | w. decay | t. decay | mom. | SGD nest. | n_blocks | d_token | att._drop. | res._drop. | d_ffn_factor | ffn_drop. |
|---|---|---|---|---|---|---|---|---|---|---|---|---|
| adult | SGD | 1e−03 | 1e−04 | False | .92 | False | 1 | 320 | .30 | | 2.67 | .45 |
| compass | Adam | 1e−04 | 1e−04 | False | | | 1 | 64 | .20 | | 1.33 | .10 |
| covtype | Adam | 1e−04 | 1e−04 | False | | | 2 | 512 | .15 | | 1.67 | .10 |
| electricity | Adam | 1e−04 | 1e−03 | False | | | 1 | 192 | .05 | .20 | 1.00 | .50 |
| eye | Adam | 1e−05 | 1e−04 | False | | | 1 | 64 | .15 | | 1.33 | .45 |
| heloc | AdamW | 1e−05 | 1e−04 | True | | | 3 | 384 | .25 | .10 | 1.33 | .45 |
| higgs | Adam | 1e−05 | 1e−05 | False | | | 1 | 64 | .15 | | 1.33 | .45 |
| house | Adam | 1e−05 | 1e−04 | False | | | 1 | 64 | .15 | | 1.33 | .45 |
| jannis | Adam | 1e−05 | 1e−04 | False | | | 1 | 64 | .15 | | 1.33 | .45 |
| kddipums97 | Adam | 1e−03 | 1e−04 | False | | | 1 | 64 | .15 | | 1.33 | .45 |
| letter | Adam | 1e−03 | 1e−03 | False | | | 4 | 64 | .05 | | 1.00 | .40 |
| magic | Adam | 1e−05 | 1e−04 | False | | | 4 | 64 | .15 | | 1.33 | .45 |
| online | Adam | 1e−03 | 1e−04 | False | | | 1 | 64 | .20 | | 1.67 | .50 |
| phoneme | Adam | 1e−05 | 1e−04 | False | | | 1 | 64 | .15 | | 1.33 | .45 |
| pol | AdamW | 1e−03 | 1e−05 | True | | | 4 | 256 | .05 | .10 | 1.67 | .40 |
| rl | Adam | 1e−04 | 1e−03 | True | | | 2 | 320 | .30 | | 1.67 | .15 |
| upselling | Adam | 1e−05 | 1e−04 | False | | | 1 | 64 | .15 | | 1.33 | .45 |

**TabResnet**

| Dataset | optim. | l. rate | w. decay | t. decay | mom. | SGD nest. | o | n_blocks | d_token / d_main | d_hidden_f | d_drop_first | d_drop_sec. | zero_init_resid |
|---|---|---|---|---|---|---|---|---|---|---|---|---|---|
| aloi | SGD | 1e−02 | 1e−04 | False | .96 | True | 748.4 | 4 | 128 | 192 | 2.00 | .20 | True |
| bank | SGD | 1e−01 | 1e−03 | True | .9 | False | 1.2 | 4 | 64 | 192 | 1.00 | .45 | False |
| gisette | AdamW | 1e−05 | 1e−04 | True | | | 2.0 | 1 | 384 | 512 | 4.00 | .30 | False |
| helena | AdamW | 1e−04 | 1e−04 | True | | | 89.0 | 1 | 384 | 448 | 3.00 | .40 | True |
| indian | SGD | 1e−01 | 1e−03 | True | .9 | False | 3.2 | 4 | 64 | 192 | 1.00 | .40 | True |
| miniboone | AdamW | 1e−04 | 1e−04 | True | .9 | | 2.0 | 1 | 384 | 448 | 3.00 | .30 | False |
| ucicredit | AdamW | 1e−05 | 1e−04 | True | | | 2.0 | 1 | 448 | 448 | 3.00 | .30 | False |

**Resnet18-50**

| Dataset | optim. | l. rate | w. decay | t. decay | mom. | SGD nest. | o | b_norm |
|---|---|---|---|---|---|---|---|---|
| MMIST | Adam | 1e−04 | 1e−04 | True | | | 2.8 | True |
| bloodmnist | Adam | 1e−04 | 1e−04 | True | | | 4.0 | False |
| breastmnist | SGD | 1e−01 | 1e−04 | True | .9 | False | 5.0 | False |
| chestmnist | SGD | 1e−01 | 1e−04 | False | | False | 5.0 | True |
| dermamnist | SGD | 1e−01 | 1e−03 | True | .9 | False | 3.4 | True |
| food101 | AdamW | 1e−04 | 1e−03 | True | | | 31.4 | True |
| octmnist | Adam | 1e−03 | 1e−06 | True | | | 5.0 | False |
| organamnist | SGD | 1e−01 | 1e−04 | False | .9 | False | 7.2 | True |
| organcmnist | SGD | 1e−02 | 1e−03 | False | .93 | True | 6.8 | True |
| organsmnist | SGD | 1e−01 | 1e−04 | False | .94 | True | 5.8 | True |
| oxfordpets | AdamW | 1e−04 | 1e−03 | True | | | 5.8 | True |
| pathmnist | SGD | 1e−01 | 1e−05 | False | .9 | False | 1.2 | True |
| pneumoniamnist | SGD | 1e−01 | 1e−04 | False | | | 3.4 | True |
| retinamnist | AdamW | 1e−04 | 1e−03 | True | | | 5.0 | False |
| stanfordcars | Adam | 1e−04 | 1e−06 | False | | | 2.8 | True |
| tissuemnist | SGD | 1e−03 | 1e−03 | False | .96 | True | 161.0 | False |
| waterbirds | AdamW | 1e−04 | 1e−06 | False | | | 2.0 | False |

**VGG**

| Dataset | optim. | l. rate | w. decay | t. decay | mom. | SGD nest. | o | b_norm |
|---|---|---|---|---|---|---|---|---|
| SVHN | SGD | 1e−04 | 1e−05 | False | .99 | True | 7.8 | True |
| catsdogs | SGD | 1e−02 | 1e−03 | False | .91 | True | 1.4 | True |
| cifar10 | SGD | 1e−02 | 1e−03 | False | .91 | True | 5.2 | True |

Table A5: Best configurations for SAT, divided by architectures.

| Dataset | optim. | l. rate w. | decay t. | decay mom. | SGD nest. | $E_s$ | $\gamma$ | n_blocks | d_token | att._drop. | res._drop. | d_ffn_factor | ffn_drop. | Arch. |
|---|---|---|---|---|---|---|---|---|---|---|---|---|---|---|
| adult | AdamW | 1e−05 | 1e−05 | True | | 45 | .97 | 3 | 128 | .40 | | 1.67 | .45 | FTTransformer |
| compass | AdamW | 1e−05 | 1e−05 | False | | 45 | .92 | 2 | 128 | .50 | | 2.33 | .05 | |
| covtype | Adam | 1e−04 | 1e−06 | True | | 15 | .95 | 4 | 320 | .15 | | .67 | .10 | |
| electricity | Adam | 1e−04 | 1e−06 | False | | 15 | .98 | 3 | 128 | .45 | | 1.00 | | |
| eye | AdamW | 1e−05 | 1e−06 | True | | 45 | .97 | 3 | 256 | .20 | | 2.33 | | |
| heloc | AdamW | 1e−05 | 1e−05 | False | | 45 | .92 | 2 | 192 | .45 | | 2.00 | .45 | |
| higgs | Adam | 1e−04 | 1e−05 | True | | 60 | .99 | 2 | 192 | .15 | .10 | 1.00 | | |
| house | AdamW | 1e−03 | 1e−06 | False | | 15 | .97 | 1 | 384 | .25 | .20 | 1.00 | .35 | |
| jannis | Adam | 1e−04 | 1e−06 | True | | 15 | .94 | 3 | 128 | .25 | | 1.00 | .30 | |
| kddipums97 | Adam | 1e−04 | 1e−06 | True | | 15 | .93 | 4 | 128 | .25 | | 1.00 | .30 | |
| letter | Adam | 1e−04 | 1e−06 | True | | 15 | .92 | 4 | 256 | .25 | | 1.00 | .30 | |
| magic | Adam | 1e−03 | 1e−06 | True | | 15 | .94 | 4 | 128 | .25 | | 1.00 | .30 | |
| online | Adam | 1e−04 | 1e−05 | True | | 60 | .99 | 2 | 192 | .15 | .10 | 1.00 | | |
| phoneme | Adam | 1e−03 | 1e−06 | True | | 15 | .93 | 4 | 64 | .30 | | 1.00 | .30 | |
| pol | Adam | 1e−03 | 1e−06 | True | | 15 | .93 | 4 | 192 | .15 | | 1.00 | .30 | |
| rl | Adam | 1e−04 | 1e−06 | False | | 0 | .98 | 3 | 64 | .45 | | 1.00 | .10 | |
| upselling | Adam | 1e−04 | 1e−06 | True | | 15 | .93 | 4 | 128 | .25 | | 1.00 | .30 | |

| Dataset | optim. | l. rate w. | decay t. | decay mom. | SGD nest. | $E_s$ | $\gamma$ | n_blocks | d_token | d_main | d_hidden_f | d_drop_first | d_drop_sec. | Arch. |
|---|---|---|---|---|---|---|---|---|---|---|---|---|---|---|
| aloi | Adam | 1e−04 | 1e−05 | False | | 0 | .93 | 2 | 320 | 384 | 3.00 | .50 | .50 | TabResnet |
| bank | Adam | 1e−02 | 1e−04 | True | | 15 | .93 | 2 | 320 | 320 | 2.00 | .45 | .15 | |
| giveme | Adam | 1e−04 | 1e−04 | False | | 30 | .91 | 1 | 512 | 192 | 3.00 | .15 | .25 | |
| helena | Adam | 1e−04 | 1e−04 | False | | 30 | .9 | 4 | 192 | 128 | 2.00 | .45 | .15 | |
| indian | Adam | 1e−03 | 1e−03 | True | | 0 | .98 | 4 | 512 | 192 | 1.00 | .10 | .20 | |
| miniboone | Adam | 1e−03 | 1e−03 | True | | 0 | .98 | 4 | 512 | 192 | 1.00 | .10 | .20 | |
| ucicredit | Adam | 1e−03 | 1e−03 | True | | 0 | .98 | 4 | 512 | 192 | 1.00 | .10 | .20 | |

| Dataset | optim. | l. rate w. | decay t. | decay mom. | SGD nest. | $E_s$ | $\gamma$ | zero_init_resid | Arch. |
|---|---|---|---|---|---|---|---|---|---|
| MNIST | SGD | 1e−02 | 1e−05 | False | .92 | 0 | .99 | True | Resnet18-50 |
| bloodmnist | Adam | 1e−03 | 1e−06 | True | | 60 | .9 | True | |
| breastmnist | AdamW | 1e−05 | 1e−03 | False | | 0 | .91 | False | |
| chestmnist | Adam | 1e−02 | 1e−06 | True | | 60 | .9 | True | |
| dermamnist | AdamW | 1e−04 | 1e−05 | True | | 30 | .94 | True | |
| food101 | Adam | 1e−02 | 1e−06 | True | | 45 | .9 | False | |
| octmnist | AdamW | 1e−03 | 1e−05 | True | | 60 | .92 | True | |
| organamnist | Adam | 1e−03 | 1e−06 | False | | 0 | .98 | True | |
| organcmnist | Adam | 1e−03 | 1e−06 | True | | 45 | .91 | True | |
| organsmnist | Adam | 1e−03 | 1e−06 | True | | 45 | .93 | True | |
| oxfordpets | AdamW | 1e−03 | 1e−06 | False | | 45 | .91 | False | |
| pathmnist | Adam | 1e−02 | 1e−06 | True | .9 | 15 | .91 | True | |
| pneumoniamnist | SGD | 1e−01 | 1e−04 | False | | 60 | .9 | False | |
| retinamnist | AdamW | 1e−04 | 1e−06 | True | | 0 | .95 | True | |
| stanfordcars | Adam | 1e−03 | 1e−05 | False | | 15 | .93 | True | |
| tissuemnist | Adam | 1e−02 | 1e−06 | True | | 15 | .91 | True | |
| waterbirds | SGD | 1e−02 | 1e−03 | False | .9 | 30 | .92 | False | |

| Dataset | optim. | l. rate w. | decay t. | decay mom. | SGD nest. | $E_s$ | $\gamma$ | b_norm | Arch. |
|---|---|---|---|---|---|---|---|---|---|
| SVHN | Adam | 1e−02 | 1e−06 | True | | 15 | .9 | True | VGG |
| catsdogs | Adam | 1e−02 | 1e−06 | True | | 45 | .9 | True | |
| cifar10 | Adam | 1e−02 | 1e−06 | True | | 45 | .93 | True | |

Table A6: Best configurations for SAT+EM, divided by architectures.

**FTTransformer**

| Dataset | optim. | l. rate | w. decay | t. decay | mom. | SGD nest. | $E_s$ | $\gamma$ | $\beta$ | n_blocks | d_token | att._drop. | res._drop. | d_ffn_factor | ffn_drop. | Arch. |
|---|---|---|---|---|---|---|---|---|---|---|---|---|---|---|---|---|
| adult | SGD | 1e−04 | 1e−06 | False | .95 | True | 45 | .9 | .01 | 3 | 256 | | | 2.67 | .50 | |
| compass | Adam | 1e−03 | 1e−06 | False | | | 60 | .96 | .0001 | 3 | 512 | | | 1.33 | .10 | |
| covtype | AdamW | 1e−05 | 1e−05 | False | | | 30 | .98 | .001 | 1 | 320 | | | 2.33 | .10 | |
| electricity | AdamW | 1e−05 | 1e−05 | False | | | 30 | .96 | .001 | 3 | 320 | | | 2.33 | .10 | |
| eye | AdamW | 1e−05 | 1e−04 | True | | | 30 | .91 | .01 | 2 | 448 | | | 1.33 | .15 | |
| heloc | AdamW | 1e−05 | 1e−04 | True | | | 30 | .92 | .0001 | 2 | 384 | | .15 | 2.33 | .50 | |
| higgs | AdamW | 1e−03 | 1e−05 | True | | | 30 | .98 | .001 | 1 | 448 | | | 1.67 | .30 | |
| house | AdamW | 1e−03 | 1e−06 | False | | | 60 | .96 | .0001 | 1 | 512 | | | 1.33 | .10 | |
| jannis | AdamW | 1e−03 | 1e−06 | True | | | 30 | .97 | .0001 | 3 | 448 | | | 2.00 | .25 | |
| kddipums97 | AdamW | 1e−03 | 1e−06 | False | | | 60 | .95 | .0001 | 1 | 512 | | | 1.33 | .10 | |
| letter | Adam | 1e−03 | 1e−06 | False | | | 30 | .96 | .001 | 3 | 384 | | | 2.67 | .10 | |
| magic | Adam | 1e−04 | 1e−05 | False | | | 60 | .94 | .0001 | 1 | 512 | | | 1.33 | .10 | |
| online | Adam | 1e−04 | 1e−06 | False | | | 60 | .96 | .0001 | 1 | 512 | | | 1.33 | .10 | |
| phoneme | AdamW | 1e−04 | 1e−06 | True | | | 15 | .99 | .01 | 3 | 320 | | | 2.67 | .40 | |
| pol | AdamW | 1e−04 | 1e−06 | True | | | 30 | .98 | .0001 | 3 | 448 | | | 2.00 | .30 | |
| r1 | Adam | 1e−04 | 1e−06 | False | | | 60 | .97 | .0001 | 1 | 512 | | | 1.33 | .15 | |
| upselling | AdamW | 1e−05 | 1e−06 | True | | | 30 | .99 | .0001 | 3 | 448 | | | 2.00 | .30 | |

**TabResnet**

| Dataset | optim. | l. rate | w. decay | t. decay | mom. | SGD nest. | $E_s$ | $\gamma$ | $\beta$ | n_blocks | d_token | d_main | d_hidden_f | d_drop_first | d_drop_sec. | Arch. |
|---|---|---|---|---|---|---|---|---|---|---|---|---|---|---|---|---|
| aloi | SGD | 1e−01 | 1e−04 | False | .93 | False | 15 | .91 | .001 | 3 | 192 | 512 | 4.00 | .10 | .25 | |
| bank | Adam | 1e−02 | 1e−06 | True | | | 15 | .94 | .001 | 1 | 384 | 256 | 1.00 | .30 | .05 | |
| givene | Adam | 1e−04 | 1e−06 | True | | | 0 | .93 | .0001 | 2 | 384 | 256 | 2.00 | .25 | .05 | |
| helena | Adam | 1e−04 | 1e−06 | True | | | 15 | .95 | .0001 | 2 | 384 | 256 | 2.00 | .25 | .05 | |
| indian | Adam | 1e−03 | 1e−06 | True | | | 0 | .94 | .001 | 2 | 384 | 256 | 1.00 | .30 | .05 | |
| miniboone | Adam | 1e−02 | 1e−06 | True | | | 15 | .94 | .001 | 1 | 384 | 256 | 1.00 | .30 | .05 | |
| ucicredit | AdamW | 1e−05 | 1e−06 | False | | | 60 | .92 | .001 | 2 | 512 | 128 | 3.00 | .40 | .10 | |

**Resnet18-50**

| Dataset | optim. | l. rate | w. decay | t. decay | mom. | SGD nest. | $E_s$ | $\gamma$ | $\beta$ | zero_init_resid | Arch. |
|---|---|---|---|---|---|---|---|---|---|---|---|
| MNIST | SGD | 1e−01 | 1e−05 | True | | True | 60 | .98 | .0001 | True | |
| bloodmnist | Adam | 1e−03 | 1e−06 | False | | | 60 | .96 | .0001 | False | |
| breastmnist | SGD | 1e−02 | 1e−04 | True | .9 | False | 45 | .9 | .01 | False | |
| chestmnist | Adam | 1e−03 | 1e−06 | False | | | 45 | .93 | .0001 | False | |
| dermamnist | SGD | 1e−04 | 1e−05 | False | .99 | True | 60 | .98 | .0001 | True | |
| food101 | Adam | 1e−02 | 1e−06 | True | | | 0 | .9 | .01 | True | |
| octmnist | Adam | 1e−03 | 1e−06 | True | | | 0 | .93 | .0001 | False | |
| organamnist | Adam | 1e−04 | 1e−06 | False | | | 0 | .96 | .0001 | False | |
| organcmnist | Adam | 1e−03 | 1e−06 | False | | | 60 | .96 | .0001 | False | |
| organsmnist | SGD | 1e−02 | 1e−06 | False | .95 | | 60 | .98 | .0001 | True | |
| oxfordpets | SGD | 1e−02 | 1e−04 | True | .93 | True | 0 | .98 | .0001 | False | |
| pathmnist | Adam | 1e−03 | 1e−06 | False | | | 0 | .95 | .0001 | False | |
| pneumoniamnist | Adam | 1e−05 | 1e−06 | False | | | 0 | .94 | .0001 | False | |
| retinamnist | Adam | 1e−04 | 1e−06 | True | | | 0 | .93 | .0001 | True | |
| stanfordcars | Adam | 1e−03 | 1e−04 | False | | | 15 | .98 | .001 | False | |
| tissuemnist | Adam | 1e−02 | 1e−06 | True | | | 0 | .93 | .0001 | False | |
| waterbirds | Adam | 1e−03 | 1e−06 | False | | | 0 | .96 | .0001 | False | |

**VGG**

| Dataset | optim. | l. rate | w. decay | t. decay | mom. | SGD nest. | $E_s$ | $\gamma$ | $\beta$ | b_norm | Arch. |
|---|---|---|---|---|---|---|---|---|---|---|---|
| SVHN | Adam | 1e−04 | 1e−06 | True | | | 0 | .93 | .0001 | False | |
| catsdogs | SGD | 1e−01 | 1e−05 | False | .96 | True | 60 | .98 | .0001 | True | |
| cifar10 | SGD | 1e−01 | 1e−04 | False | .93 | True | 60 | .98 | .0001 | True | |

Table A7: Best configurations for SELNET, divided by architectures.

| Dataset | optim. | l. rate w. | decay w. | decay t. | decay mom. | SGD nest. | λ | α | n_blocks | d_token | att._drop | res._drop | d_ffn_factor | ffn_drop | Arch. |
|---|---|---|---|---|---|---|---|---|---|---|---|---|---|---|---|
| adult | AdamW | 1e − 05 | 1e − 06 | True | | | 32 | .65 | 3 | 192 | .40 | | 2.00 | | |
| compass | Adam | 1e − 04 | 1e − 05 | True | | | 64 | .75 | 2 | 192 | .15 | .10 | 1.00 | | |
| covtype | Adam | 1e − 04 | 1e − 06 | True | | | 16 | .25 | 3 | 384 | .10 | | .67 | | |
| electricity | Adam | 1e − 04 | 1e − 04 | True | | | 64 | .75 | 2 | 192 | .05 | .05 | 1.00 | .05 | |
| eye | AdamW | 1e − 05 | 1e − 06 | True | | | 32 | .70 | 2 | 320 | .35 | | 2.67 | .10 | |
| heloc | Adam | 1e − 05 | 1e − 06 | False | | | 8 | .70 | 3 | 64 | .50 | | 1.33 | | |
| higgs | AdamW | 1e − 04 | 1e − 05 | True | | | 64 | .65 | 3 | 192 | .25 | | 2.67 | | |
| house | Adam | 1e − 04 | 1e − 05 | True | | | 64 | .75 | 2 | 192 | .15 | .10 | 1.00 | | |
| jannis | AdamW | 1e − 04 | 1e − 05 | False | | | 64 | .35 | 2 | 256 | .50 | | 2.33 | .50 | |
| kddipums97 | Adam | 1e − 04 | 1e − 06 | False | | | 8 | .70 | 3 | 64 | .50 | | 1.33 | .10 | |
| letter | Adam | 1e − 04 | 1e − 06 | True | | | 16 | .40 | 4 | 320 | .10 | | .67 | .20 | |
| magic | Adam | 1e − 04 | 1e − 05 | True | | | 32 | .55 | 3 | 256 | .15 | | .67 | .10 | |
| online | Adam | 1e − 03 | 1e − 05 | True | | | 64 | .70 | 2 | 192 | .15 | .20 | 1.00 | | |
| phoneme | Adam | 1e − 03 | 1e − 06 | True | | | 8 | .35 | 3 | 320 | .15 | | 1.33 | .30 | |
| pol | Adam | 1e − 04 | 1e − 05 | True | | | 64 | .75 | 2 | 192 | .15 | .10 | 1.00 | | |
| rl | Adam | 1e − 03 | 1e − 06 | True | | | 8 | .70 | 3 | 64 | .50 | | 1.00 | .10 | |
| upselling | Adam | 1e − 04 | 1e − 06 | True | | | 16 | .45 | 4 | 128 | .25 | | 1.00 | .30 | |
| | | | | | | | | | | | | | | | FTTransformer |

| Dataset | optim. | l. rate w. | decay w. | decay t. | decay mom. | SGD nest. | λ | α | n_blocks | d_token | d_main | d_hidden_f | d_drop_first | d_drop_sec. | Arch. |
|---|---|---|---|---|---|---|---|---|---|---|---|---|---|---|---|
| aloi | Adam | 1e − 04 | 1e − 05 | False | | | 32 | .25 | 4 | 128 | 128 | 2.00 | .50 | .05 | |
| bank | Adam | 1e − 03 | 1e − 03 | True | | | 8 | .70 | 4 | 512 | 192 | 1.00 | .10 | .20 | |
| giveme | Adam | 1e − 03 | 1e − 03 | True | | | 8 | .70 | 4 | 512 | 192 | 1.00 | .10 | .20 | |
| helena | Adam | 1e − 04 | 1e − 04 | False | | | 32 | .25 | 4 | 128 | 128 | 2.00 | .50 | .10 | |
| indian | AdamW | 1e − 02 | 1e − 04 | True | | | 32 | .65 | 2 | 256 | 320 | 2.00 | .40 | .15 | |
| miniboone | Adam | 1e − 03 | 1e − 03 | True | | | 8 | .70 | 4 | 512 | 192 | 1.00 | .10 | .20 | |
| ucicredit | Adam | 1e − 04 | 1e − 06 | False | | | 32 | .25 | 4 | 192 | 64 | 2.00 | .45 | .10 | |
| | | | | | | | | | | | | | | | TabResnet |

| Dataset | optim. | l. rate w. | decay w. | decay t. | decay mom. | SGD nest. | λ | α | zero_init_resid | Arch. |
|---|---|---|---|---|---|---|---|---|---|---|
| MNIST | SGD | 1e − 01 | 1e − 04 | True | .94 | True | 32 | .55 | False | |
| bloodmnist | SGD | 1e − 01 | 1e − 04 | True | .99 | True | 32 | .45 | False | |
| breastmnist | AdamW | 1e − 03 | 1e − 05 | True | | | 32 | .45 | True | |
| chestmnist | Adam | 1e − 02 | 1e − 06 | True | | | 64 | .25 | True | |
| dermamnist | AdamW | 1e − 05 | 1e − 03 | False | | | 16 | .30 | True | |
| food101 | SGD | 1e − 01 | 1e − 04 | True | .9 | False | 32 | .50 | False | |
| octmnist | Adam | 1e − 02 | 1e − 06 | True | | | 16 | .30 | True | |
| organamnist | SGD | 1e − 01 | 1e − 04 | True | .94 | True | 32 | .50 | False | |
| organcmnist | Adam | 1e − 03 | 1e − 06 | True | | | 32 | .30 | True | |
| organsmnist | Adam | 1e − 03 | 1e − 06 | True | | | 16 | .25 | True | |
| oxfordpets | AdamW | 1e − 03 | 1e − 06 | True | | | 16 | .25 | False | |
| pathmnist | SGD | 1e − 01 | 1e − 04 | True | .93 | True | 64 | .60 | False | |
| pneumoniamnist | Adam | 1e − 02 | 1e − 06 | True | | | 16 | .30 | True | |
| retinamnist | Adam | 1e − 02 | 1e − 06 | True | | | 16 | .30 | True | |
| stanfordcars | Adam | 1e − 03 | 1e − 06 | True | | | 64 | .25 | True | |
| tissuemnist | Adam | 1e − 02 | 1e − 06 | True | | | 16 | .30 | True | |
| waterbirds | Adam | 1e − 03 | 1e − 06 | False | | | 8 | .45 | True | |
| | | | | | | | | | | Resnet18-50 |

| Dataset | optim. | l. rate w. | decay w. | decay t. | decay mom. | SGD nest. | λ | α | b_norm | Arch. |
|---|---|---|---|---|---|---|---|---|---|---|
| SVHN | Adam | 1e − 02 | 1e − 06 | True | | | 64 | .25 | True | |
| catsdogs | Adam | 1e − 02 | 1e − 06 | True | | | 32 | .25 | True | |
| cifar10 | Adam | 1e − 03 | 1e − 06 | False | | | 8 | .70 | True | |
| | | | | | | | | | | VGG |

Table A8: Best configurations for SELNET+EM, divided by architectures.

| Dataset | optim. | l. rate | w. decay | t. decay mom. | SGD nest. | λ | α | β | n_blocks | d_token | att._drop. | res._drop. | d_ffn_factor | ffn_drop. | Arch. |
|---|---|---|---|---|---|---|---|---|---|---|---|---|---|---|---|
| adult | AdamW | 1e − 02 | 1e − 05 | True | | | 16 | .25 | .0001 | 2 | 384 | .15 | | 2.33 | .50 | |
| compass | AdamW | 1e − 03 | 1e − 06 | True | | | 16 | .75 | .0001 | 3 | 448 | .15 | | 2.33 | .30 | |
| cortype | AdamW | 1e − 05 | 1e − 05 | False | | | 16 | .60 | .001 | 3 | 320 | .05 | | 2.33 | .10 | |
| electricity | AdamW | 1e − 05 | 1e − 06 | False | | | 16 | .45 | .0001 | 4 | 256 | .10 | | 1.00 | .35 | |
| eye | AdamW | 1e − 04 | 1e − 04 | True | | | 64 | .25 | .01 | 1 | 448 | .25 | | 1.67 | .20 | |
| heloc | AdamW | 1e − 04 | 1e − 04 | True | | | 64 | .25 | .01 | 1 | 448 | .35 | | 1.33 | .15 | |
| higgs | AdamW | 1e − 03 | 1e − 04 | True | | | 64 | .35 | .01 | 1 | 384 | .20 | | 1.33 | .10 | |
| house | AdamW | 1e − 05 | 1e − 05 | True | | | 8 | .50 | .0001 | 3 | 384 | .20 | | 1.33 | | |
| jannis | Adam | 1e − 04 | 1e − 06 | False | | | 64 | .60 | .0001 | 3 | 512 | .50 | | 1.67 | .15 | |
| kddipums97 | AdamW | 1e − 03 | 1e − 06 | True | | | 16 | .70 | .0001 | 3 | 448 | .10 | | 2.00 | .30 | |
| letter | AdamW | 1e − 04 | 1e − 06 | True | | | 16 | .70 | .0001 | 3 | 448 | .10 | | 2.00 | .35 | |
| magic | AdamW | 1e − 03 | 1e − 06 | True | | | 16 | .70 | .0001 | 3 | 448 | .15 | | 2.00 | .25 | |
| online | AdamW | 1e − 03 | 1e − 06 | True | | | 16 | .75 | .0001 | 3 | 448 | .15 | | 2.00 | .35 | |
| phoneme | AdamW | 1e − 05 | 1e − 06 | False | | | 16 | .75 | .0001 | 4 | 192 | .15 | | 2.00 | .30 | |
| pol | AdamW | 1e − 05 | 1e − 06 | False | | | 16 | .50 | .0001 | 3 | 320 | | | 2.67 | .35 | |
| r1 | AdamW | 1e − 04 | 1e − 05 | False | | | 64 | .65 | .001 | 1 | 512 | | .45 | 1.33 | .10 | |
| upselling | AdamW | 1e − 05 | 1e − 06 | True | | | 16 | .70 | .0001 | 3 | 448 | .10 | | 2.00 | .30 | FTTransformer |

| Dataset | optim. | l. rate | w. decay | t. decay mom. | SGD nest. | λ | α | β | n_blocks | d_main | d_hidden_f | d_drop_first | d_drop_sec. | Arch. |
|---|---|---|---|---|---|---|---|---|---|---|---|---|---|---|
| aloi | SGD | 1e − 01 | 1e − 05 | False | .99 | False | 64 | .65 | .0001 | 4 | 320 | 384 | 3.00 | .15 | .05 |
| bank | Adam | 1e − 03 | 1e − 06 | True | | | 16 | .55 | .0001 | 1 | 320 | 320 | 1.00 | .30 | .05 |
| giveme | Adam | 1e − 03 | 1e − 06 | True | | | 16 | .45 | .0001 | 1 | 384 | 256 | 1.00 | .25 | .05 |
| helena | Adam | 1e − 03 | 1e − 06 | True | | | 8 | .50 | .0001 | 2 | 448 | 192 | 1.00 | .25 | .05 |
| indian | AdamW | 1e − 05 | 1e − 06 | False | | | 64 | .35 | .0001 | 2 | 512 | 128 | 3.00 | .40 | .05 |
| miniboone | Adam | 1e − 04 | 1e − 06 | True | | | 8 | .45 | .0001 | 2 | 384 | 256 | 2.00 | .25 | .05 |
| ucicredit | Adam | 1e − 03 | 1e − 06 | True | | | 16 | .45 | .001 | 1 | 384 | 256 | 1.00 | .30 | .05 TabResNet |

| Dataset | optim. | l. rate | w. decay | t. decay mom. | SGD nest. | λ | α | β | zero_init_resid | Arch. |
|---|---|---|---|---|---|---|---|---|---|---|
| MNIST | SGD | 1e − 03 | 1e − 05 | False | .93 | False | 64 | .35 | .0001 | False | |
| bloodmnist | Adam | 1e − 03 | 1e − 06 | False | | | 64 | .55 | .0001 | False | |
| breastmnist | Adam | 1e − 04 | 1e − 06 | True | | | 8 | .40 | .0001 | False | |
| chestmnist | SGD | 1e − 02 | 1e − 05 | False | .93 | True | 64 | .30 | .0001 | False | |
| dermamnist | Adam | 1e − 05 | 1e − 06 | False | | | 16 | .60 | .0001 | False | |
| food101 | Adam | 1e − 04 | 1e − 06 | True | | | 8 | .40 | .0001 | False | |
| octmnist | Adam | 1e − 04 | 1e − 06 | False | | | 16 | .60 | .0001 | False | |
| organamnist | Adam | 1e − 03 | 1e − 06 | True | | | 8 | .55 | .0001 | False | |
| organcmnist | Adam | 1e − 03 | 1e − 06 | False | | | 64 | .55 | .0001 | False | |
| organsmnist | Adam | 1e − 04 | 1e − 06 | False | | | 64 | .60 | .0001 | False | |
| oxfordpets | SGD | 1e − 02 | 1e − 04 | True | .95 | False | 32 | .50 | .0001 | False | |
| pathmnist | Adam | 1e − 03 | 1e − 06 | False | | | 64 | .55 | .0001 | False | |
| pneumoniamnist | Adam | 1e − 03 | 1e − 06 | False | | | 8 | .60 | .0001 | False | |
| retinamnist | Adam | 1e − 02 | 1e − 06 | False | | | 64 | .55 | .0001 | False | |
| stanfordcars | Adam | 1e − 04 | 1e − 06 | True | | | 8 | .40 | .0001 | False | |
| tissuemnist | Adam | 1e − 04 | 1e − 06 | True | | | 8 | .40 | .0001 | False | |
| waterbirds | Adam | 1e − 04 | 1e − 06 | False | | | 16 | .60 | .0001 | False | ResNet18-50 |

| Dataset | optim. | l. rate | w. decay | t. decay mom. | SGD nest. | λ | α | β | b_norm | Arch. |
|---|---|---|---|---|---|---|---|---|---|---|
| SVHN | Adam | 1e − 04 | 1e − 06 | True | | | 8 | .40 | .0001 | False | |
| catsdogs | Adam | 1e − 03 | 1e − 06 | False | | | 16 | .65 | .0001 | False | |
| cifar10 | SGD | 1e − 02 | 1e − 03 | True | .99 | True | 32 | .50 | .001 | True | VGG |

Table A9: Best configurations for CONFIDNET, divided by architectures.

**FTTransformer**

| Dataset | optim. | l. rate | w. decay | decay t. | mom. | SGD nest. | optim. unc. | l. rate unc. | mom. unc. | SGD nest. unc. | n_blocks | d_token | att._drop. | res._drop. | d_ffn_factor | ffn_drop. | Arch. |
|---|---|---|---|---|---|---|---|---|---|---|---|---|---|---|---|---|---|
| adult | Adam | 1e−05 | 1e−05 | True | | | AdamW | 1e−06 | | | 4 | 64 | .50 | | 2.33 | .15 | |
| compass | Adam | 1e−03 | 1e−05 | True | | | AdamW | 1e−06 | | | 2 | 64 | .40 | | 2.00 | .35 | |
| covtype | Adam | 1e−04 | 1e−05 | False | | | Adam | 1e−05 | | | 2 | 448 | .10 | .15 | 2.00 | .25 | |
| electricity | Adam | 1e−05 | 1e−06 | False | | | Adam | 1e−05 | | | 4 | 448 | .10 | .05 | 1.67 | .15 | |
| eye | SGD | 1e−04 | 1e−04 | False | .96 | True | AdamW | 1e−06 | | | 4 | 128 | .35 | | .67 | .15 | |
| heloc | AdamW | 1e−05 | 1e−05 | False | | | SGD | 1e−06 | .92 | True | 2 | 128 | .50 | | 2.33 | .45 | |
| higgs | AdamW | 1e−04 | 1e−04 | False | | | Adam | 1e−06 | | | 1 | 512 | .40 | | 2.67 | .30 | |
| house | AdamW | 1e−03 | 1e−03 | False | | | Adam | 1e−06 | | | 1 | 512 | .40 | | 2.67 | .30 | |
| jannis | AdamW | 1e−04 | 1e−03 | False | | | Adam | 1e−06 | | | 1 | 512 | .40 | | 2.67 | .30 | |
| kddipums97 | AdamW | 1e−03 | 1e−04 | False | | | Adam | 1e−05 | | | 1 | 512 | .40 | | 2.67 | .25 | |
| letter | SGD | 1e−03 | 1e−03 | False | .97 | True | AdamW | 1e−06 | | | 3 | 128 | .35 | | .67 | .15 | |
| magic | AdamW | 1e−03 | 1e−03 | False | | | Adam | 1e−06 | | | 1 | 512 | .40 | | 2.67 | .30 | |
| online | AdamW | 1e−03 | 1e−03 | False | | | Adam | 1e−06 | | | 1 | 512 | .40 | | 2.67 | .30 | |
| phoneme | AdamW | 1e−03 | 1e−03 | False | | | Adam | 1e−06 | | | 1 | 512 | .40 | | 2.67 | .30 | |
| pol | Adam | 1e−03 | 1e−05 | False | | | Adam | 1e−05 | | | 4 | 384 | .05 | .20 | 2.00 | .25 | |
| r1 | AdamW | 1e−04 | 1e−03 | False | | | Adam | 1e−06 | | | 1 | 448 | .40 | | 2.67 | .25 | |
| upselling | SGD | 1e−04 | 1e−04 | False | .96 | True | AdamW | 1e−06 | | | 3 | 128 | .35 | | .67 | .15 | |

**TabResnet**

| Dataset | optim. | l. rate | w. decay | decay t. | optim. unc. | l. rate unc. | n_blocks | d_token | d_main | d_hidden_f | d_drop_first | d_drop_sec. | Arch. |
|---|---|---|---|---|---|---|---|---|---|---|---|---|---|
| aloi | Adam | 1e−04 | 1e−05 | False | AdamW | 1e−05 | 4 | 512 | 192 | 2.00 | .15 | .15 | |
| bank | Adam | 1e−03 | 1e−03 | True | Adam | 1e−04 | 3 | 64 | 128 | 2.00 | .40 | .45 | |
| giveme | Adam | 1e−04 | 1e−03 | False | AdamW | 1e−04 | 4 | 448 | 448 | 1.00 | .30 | .40 | |
| helena | Adam | 1e−04 | 1e−06 | True | AdamW | 1e−07 | 3 | 128 | 512 | 3.00 | .25 | .40 | |
| indian | AdamW | 1e−04 | 1e−04 | True | Adam | 1e−05 | 1 | 128 | 320 | 1.00 | .35 | .25 | |
| miniboone | Adam | 1e−02 | 1e−03 | True | Adam | 1e−04 | 2 | 64 | 128 | 2.00 | .40 | .45 | |
| ucicredit | Adam | 1e−02 | 1e−03 | True | Adam | 1e−04 | 2 | 64 | 128 | 2.00 | .40 | .45 | |

**Resnet18-50**

| Dataset | optim. | l. rate | w. decay | decay t. | mom. | SGD nest. | optim. unc. | l. rate unc. | mom. unc. | SGD nest. unc. | zero_init_resid | Arch. |
|---|---|---|---|---|---|---|---|---|---|---|---|---|
| MNIST | Adam | 1e−04 | 1e−04 | False | | | AdamW | 1e−07 | | | True | |
| bloodmnist | SGD | 1e−01 | 1e−04 | True | .9 | False | Adam | 1e−06 | | | False | |
| breastmnist | Adam | 1e−05 | 1e−04 | False | | | AdamW | 1e−07 | | | True | |
| chestmnist | SGD | 1e−01 | 1e−03 | True | .98 | True | AdamW | 1e−06 | | | False | |
| dermamnist | AdamW | 1e−05 | 1e−05 | False | | | AdamW | 1e−06 | | | True | |
| food101 | SGD | 1e−03 | 1e−03 | True | .99 | True | Adam | 1e−06 | | | False | |
| octmnist | SGD | 1e−02 | 1e−04 | True | .95 | True | Adam | 1e−05 | | | False | |
| organamnist | Adam | 1e−03 | 1e−04 | False | | | SGD | 1e−04 | .99 | True | False | |
| organcmnist | Adam | 1e−04 | 1e−04 | False | | | AdamW | 1e−07 | | | False | |
| organsmnist | SGD | 1e−02 | 1e−03 | True | .96 | True | Adam | 1e−06 | | | True | |
| oxfordpets | Adam | 1e−04 | 1e−04 | False | | | SGD | 1e−05 | .97 | False | False | |
| pathmnist | Adam | 1e−04 | 1e−03 | False | | | AdamW | 1e−06 | | | True | |
| pneumoniamnist | Adam | 1e−04 | 1e−04 | False | | | SGD | 1e−06 | .95 | True | False | |
| retinamnist | AdamW | 1e−05 | 1e−04 | False | | | Adam | 1e−07 | | | False | |
| stanfordcars | Adam | 1e−04 | 1e−03 | True | | | AdamW | 1e−08 | | | True | |
| tissuemnist | Adam | 1e−03 | 1e−03 | True | | | Adam | 1e−08 | | | False | |
| waterbirds | Adam | 1e−04 | 1e−04 | False | | | AdamW | 1e−08 | | | True | |

**VGG**

| Dataset | optim. | l. rate | w. decay | decay t. | optim. unc. | l. rate unc. | b_norm | Arch. |
|---|---|---|---|---|---|---|---|---|
| SVHN | Adam | 1e−04 | 1e−03 | False | AdamW | 1e−07 | True | |
| catsdogs | AdamW | 1e−03 | 1e−05 | False | Adam | 1e−05 | False | |
| cifar10 | Adam | 1e−03 | 1e−04 | False | AdamW | 1e−07 | True | |

Table A10: Best configurations for REG, divided by architectures.

| Dataset | optim. | l. rate | w. decay | t. decay mom. | SGD nest. | optim. unc. | l. rate unc. | mom. | SGD unc. nest. | n_blocks | d_token | att._drop. | res._drop. | d_ffn_factor | ffn_drop. | Arch. |
|---|---|---|---|---|---|---|---|---|---|---|---|---|---|---|---|---|
| adult | AdamW | 1e−05 | 1e−04 | False | | SGD | 1e−05 | | True | 2 | 128 | .50 | | 2.33 | .45 | FTTransformer |
| compass | SGD | 1e−01 | 1e−06 | True | .9 False | AdamW | 1e−07 | | | 4 | 256 | .10 | | 1.67 | .35 | |
| covtype | Adam | 1e−03 | 1e−05 | True | | AdamW | 1e−06 | | | 2 | 320 | .67 | | 1.67 | .15 | |
| electricity | SGD | 1e−03 | 1e−04 | True | .99 True | AdamW | 1e−04 | | | 2 | 128 | .35 | | 1.33 | | |
| eye | SGD | 1e−04 | 1e−03 | False | .96 True | Adam | 1e−03 | | | 3 | 128 | .35 | | .67 | | |
| heloc | Adam | 1e−03 | 1e−05 | True | | SGD | 1e−05 | | | 3 | 64 | .15 | | 2.00 | .25 | |
| higgs | AdamW | 1e−04 | 1e−04 | False | .9 | SGD | 1e−04 | False | | 1 | 64 | .50 | | 2.33 | .50 | |
| house | SGD | 1e−03 | 1e−04 | False | .95 True | AdamW | 1e−06 | True | | 2 | 256 | .35 | | .67 | .15 | |
| jannis | Adam | 1e−04 | 1e−05 | True | | AdamW | 1e−04 | | | 3 | 64 | .25 | | 2.67 | .05 | |
| kddipums97 | AdamW | 1e−03 | 1e−03 | False | | AdamW | 1e−07 | | | 4 | 512 | .40 | .05 | 2.67 | .30 | |
| letter | SGD | 1e−02 | 1e−04 | False | .9 False | AdamW | 1e−06 | | | 1 | 320 | .20 | | 2.67 | .25 | |
| magic | Adam | 1e−04 | 1e−05 | True | | AdamW | 1e−04 | | | 3 | 128 | .05 | .05 | 1.33 | .25 | |
| online | Adam | 1e−04 | 1e−06 | False | | Adam | 1e−05 | | | 1 | 384 | .30 | | 1.67 | .20 | |
| phoneme | AdamW | 1e−03 | 1e−03 | False | | AdamW | 1e−04 | | | 1 | 512 | .40 | | 2.67 | .30 | |
| pol | AdamW | 1e−03 | 1e−03 | True | | Adam | 1e−05 | | | 4 | 320 | .25 | | 2.00 | .05 | |
| r1 | Adam | 1e−04 | 1e−03 | True | | SGD | 1e−06 | | | 2 | 128 | .10 | | .67 | .30 | |
| upselling | SGD | 1e−04 | 1e−03 | False | .96 True | AdamW | 1e−05 | | | 3 | 128 | .35 | | .67 | .15 | |

| Dataset | optim. | l. rate | w. decay | t. decay mom. | SGD nest. | optim. unc. | l. rate unc. | mom. | SGD unc. nest. | n_blocks | d_main | d_hidden_f | d_drop_first | d_drop_sec. | Arch. |
|---|---|---|---|---|---|---|---|---|---|---|---|---|---|---|---|
| aloi | Adam | 1e−03 | 1e−06 | True | | AdamW | 1e−06 | | | 3 | 128 | 4.00 | .50 | .45 | TabResnet |
| bank | SGD | 1e−01 | 1e−06 | True | .99 False | AdamW | 1e−07 | | | 4 | 256 | 3.00 | .45 | .30 | |
| give me | Adam | 1e−04 | 1e−05 | False | | AdamW | 1e−04 | | | 4 | 512 | 2.00 | .15 | .15 | |
| helena | Adam | 1e−04 | 1e−03 | True | | SGD | 1e−06 | False | | 3 | 192 | 3.00 | .25 | .05 | |
| indian | AdamW | 1e−04 | 1e−03 | True | .9 | AdamW | 1e−04 | | .99 | 3 | 256 | 3.00 | .25 | .05 | |
| miniboone | Adam | 1e−04 | 1e−03 | True | | SGD | 1e−05 | | | 3 | 320 | 1.00 | .10 | .25 | |
| ucicredit | AdamW | 1e−05 | 1e−04 | True | .96 True | Adam | 1e−07 | False | .97 | 3 | 128 | 1.00 | .25 | .15 | TabTransformer |

| Dataset | optim. | l. rate | w. decay | t. decay mom. | SGD nest. | optim. unc. | l. rate unc. | mom. | SGD unc. nest. | b_norm | Arch. |
|---|---|---|---|---|---|---|---|---|---|---|---|
| MNIST | Adam | 1e−04 | 1e−03 | False | | AdamW | 1e−07 | | | True | Resnet18-50 |
| bloodmnist | Adam | 1e−02 | 1e−06 | False | | SGD | 1e−06 | | | True | |
| breastmnist | Adam | 1e−04 | 1e−04 | True | | AdamW | 1e−03 | | | True | |
| chestmnist | AdamW | 1e−02 | 1e−04 | True | .99 | AdamW | 1e−07 | | .9 | False | |
| dermamnist | SGD | 1e−01 | 1e−04 | True | | Adam | 1e−04 | | | True | |
| food101 | Adam | 1e−01 | 1e−03 | True | .99 | AdamW | 1e−06 | False | | True | |
| octmnist | SGD | 1e−02 | 1e−05 | True | | Adam | 1e−03 | True | | True | |
| organamnist | Adam | 1e−03 | 1e−06 | False | | SGD | 1e−05 | | | True | |
| organcmnist | Adam | 1e−04 | 1e−03 | False | .99 | AdamW | 1e−07 | | .98 | True | |
| organsmnist | Adam | 1e−04 | 1e−03 | True | | AdamW | 1e−07 | | | True | |
| oxfordpets | Adam | 1e−04 | 1e−04 | True | .97 | AdamW | 1e−07 | | | True | |
| pathmnist | SGD | 1e−01 | 1e−05 | True | .97 | SGD | 1e−04 | True | | True | |
| pneumoniamnist | Adam | 1e−03 | 1e−06 | False | | Adam | 1e−07 | False | .9 | True | |
| retinamnist | SGD | 1e−01 | 1e−04 | False | .99 | AdamW | 1e−03 | | | False | |
| stanfordcars | Adam | 1e−01 | 1e−04 | True | | Adam | 1e−06 | | | True | |
| tissuemnist | SGD | 1e−01 | 1e−04 | True | .97 False | AdamW | 1e−03 | | | True | |
| waterbirds | Adam | 1e−03 | 1e−06 | False | | SGD | 1e−03 | | .97 | False | |

| Dataset | optim. | l. rate | w. decay | t. decay mom. | SGD nest. | optim. unc. | l. rate unc. | mom. | SGD unc. nest. | b_norm | Arch. |
|---|---|---|---|---|---|---|---|---|---|---|---|
| SVHN | Adam | 1e−04 | 1e−03 | False | | AdamW | 1e−06 | | | True | VGG |
| catsdogs | Adam | 1e−02 | 1e−05 | False | | AdamW | 1e−04 | | | True | |
| cifar10 | AdamW | 1e−03 | 1e−05 | False | | SGD | 1e−03 | False | .99 | True | |

Table A11: Best configurations for SELE, divided by architectures.

**FTTransformer**

| Dataset | optim. | l. rate | w. decay | t. decay | mom. | SGD nest. | optim. unc. | l. rate unc. | mom. | SGD unc. nest. | n_blocks | d_token | att._drop. | res._drop. | d_ffn_factor | ffn_drop. | Arch. |
|---|---|---|---|---|---|---|---|---|---|---|---|---|---|---|---|---|---|
| adult | SGD | $1e-03$ | $1e-06$ | True | .93 | True | AdamW | $1e-07$ | | | 4 | 256 | .05 | .10 | 1.67 | .40 | |
| compass | SGD | $1e-04$ | $1e-03$ | False | .96 | True | AdamW | $1e-05$ | | | 3 | 128 | .35 | | .67 | .15 | |
| covtype | SGD | $1e-01$ | $1e-06$ | True | .9 | False | AdamW | $1e-07$ | | | 4 | 256 | .05 | | 1.33 | .35 | |
| electricity | Adam | $1e-05$ | $1e-06$ | False | | | Adam | $1e-06$ | | | 4 | 448 | .10 | .15 | 1.67 | .25 | |
| eye | SGD | $1e-04$ | $1e-04$ | False | .96 | True | AdamW | $1e-05$ | | | 1 | 128 | .35 | | .67 | .15 | |
| heloc | Adam | $1e-03$ | $1e-05$ | True | | | SGD | $1e-04$ | .94 | True | 3 | 128 | .50 | | 1.33 | .15 | FTTransformer |
| higgs | AdamW | $1e-04$ | $1e-03$ | True | .96 | True | AdamW | $1e-04$ | | | 3 | 192 | .15 | .05 | 2.00 | .40 | |
| house | SGD | $1e-03$ | $1e-03$ | False | .94 | True | AdamW | $1e-05$ | | | 3 | 128 | .35 | .10 | .67 | .15 | |
| jannis | SGD | $1e-04$ | $1e-05$ | False | .96 | True | AdamW | $1e-07$ | | | 3 | 128 | .35 | | .67 | .20 | |
| kddipums97 | SGD | $1e-03$ | $1e-05$ | False | .95 | True | SGD | $1e-05$ | .99 | True | 3 | 128 | .35 | | .67 | .15 | |
| letter | SGD | $1e-02$ | $1e-04$ | False | | | AdamW | $1e-04$ | | | 2 | 320 | .25 | .05 | 2.33 | .40 | |
| magic | Adam | $1e-03$ | $1e-03$ | True | | | Adam | $1e-04$ | | | 3 | 128 | .15 | .20 | .67 | .10 | |
| online | AdamW | $1e-05$ | $1e-06$ | False | | | Adam | $1e-04$ | | | 1 | 512 | .50 | | 2.67 | .25 | |
| phoneme | Adam | $1e-03$ | $1e-05$ | True | | | SGD | $1e-04$ | .93 | True | 4 | 448 | .10 | .05 | 1.67 | .30 | |
| pol | SGD | $1e-05$ | $1e-05$ | True | | | Adam | $1e-04$ | | | 2 | 384 | .05 | .05 | 1.67 | .25 | |
| r1 | Adam | $1e-04$ | $1e-06$ | False | | | Adam | $1e-04$ | | | 4 | 448 | .10 | .05 | 1.67 | .25 | |
| upselling | Adam | $1e-02$ | $1e-05$ | True | | | AdamW | $1e-04$ | | | 3 | 128 | .35 | | 2.67 | .05 | |

**TabResnet**

| Dataset | optim. | l. rate | w. decay | t. decay | mom. | SGD nest. | optim. unc. | l. rate unc. | mom. | SGD unc. nest. | n_blocks | d_token | d_main | d_hidden_f | d_drop_first | d_drop_sec. | Arch. |
|---|---|---|---|---|---|---|---|---|---|---|---|---|---|---|---|---|---|
| aloi | AdamW | $1e-04$ | $1e-05$ | False | | | SGD | $1e-03$ | .92 | True | 1 | 64 | 512 | 3.00 | .40 | .15 | |
| bank | Adam | $1e-02$ | $1e-03$ | True | | | Adam | $1e-03$ | | | 2 | 64 | 128 | 2.00 | .40 | .45 | |
| giveme | AdamW | $1e-04$ | $1e-04$ | True | | | Adam | $1e-05$ | | | 3 | 256 | 512 | 1.00 | .20 | .05 | TabResnet |
| helena | Adam | $1e-04$ | $1e-06$ | True | | | AdamW | $1e-03$ | | | 3 | 128 | 512 | 4.00 | .50 | .25 | |
| indian | Adam | $1e-02$ | $1e-03$ | True | | | Adam | $1e-06$ | | | 2 | 64 | 128 | 2.00 | .40 | .45 | |
| miniboone | AdamW | $1e-04$ | $1e-04$ | True | | | Adam | $1e-03$ | | | 3 | 320 | 512 | 1.00 | .20 | .15 | |
| ucicredit | Adam | $1e-02$ | $1e-03$ | True | | | Adam | | | | 2 | 64 | 128 | 2.00 | .40 | .45 | |

**Resnet18-50**

| Dataset | optim. | l. rate | w. decay | t. decay | mom. | SGD nest. | optim. unc. | l. rate unc. | mom. | SGD unc. nest. | zero_init_resid | Arch. |
|---|---|---|---|---|---|---|---|---|---|---|---|---|
| MNIST | SGD | $1e-01$ | $1e-03$ | True | .95 | False | Adam | $1e-06$ | | | True | |
| bloodmnist | Adam | $1e-02$ | $1e-06$ | False | | | SGD | $1e-04$ | | | True | |
| breastmnist | Adam | $1e-04$ | $1e-06$ | True | .95 | False | AdamW | $1e-04$ | | | True | |
| chestmnist | SGD | $1e-01$ | $1e-05$ | True | .93 | False | Adam | $1e-03$ | | | True | |
| dermamnist | SGD | $1e-01$ | $1e-04$ | True | | | AdamW | $1e-04$ | | | False | |
| food101 | Adam | $1e-03$ | $1e-02$ | True | .96 | True | Adam | $1e-04$ | | | True | |
| octmnist | SGD | $1e-01$ | $1e-01$ | True | .94 | True | SGD | $1e-06$ | | | True | |
| organamnist | SGD | $1e-04$ | $1e-04$ | False | | | AdamW | $1e-07$ | | | True | Resnet18-50 |
| organcmnist | Adam | $1e-03$ | $1e-04$ | False | | | AdamW | $1e-06$ | | | True | |
| organsmnist | Adam | $1e-04$ | $1e-03$ | False | | | AdamW | $1e-07$ | | | True | |
| oxfordpets | Adam | $1e-04$ | $1e-06$ | True | | | Adam | $1e-06$ | | | True | |
| pathmnist | SGD | $1e-01$ | $1e-04$ | True | .93 | True | Adam | $1e-06$ | | | False | |
| pneumoniamnist | AdamW | $1e-01$ | $1e-04$ | True | .99 | False | Adam | $1e-06$ | .94 | True | True | |
| retinamnist | SGD | $1e-03$ | $1e-04$ | False | | | AdamW | $1e-03$ | | | False | |
| stanfordcars | Adam | $1e-02$ | $1e-04$ | True | .94 | True | Adam | $1e-03$ | | | True | |
| tissuemnist | SGD | $1e-04$ | $1e-06$ | False | | | SGD | $1e-04$ | | | True | |

**VGG**

| Dataset | optim. | l. rate | w. decay | t. decay | mom. | SGD nest. | optim. unc. | l. rate unc. | mom. | SGD unc. nest. | b_norm | Arch. |
|---|---|---|---|---|---|---|---|---|---|---|---|---|
| SVHN | AdamW | $1e-04$ | $1e-04$ | False | | | Adam | $1e-06$ | | | False | |
| catsdogs | AdamW | $1e-03$ | $1e-04$ | False | | | SGD | $1e-03$ | .95 | True | True | VGG |
| cifar10 | AdamW | $1e-03$ | $1e-05$ | False | | | SGD | $1e-03$ | .99 | True | True | |

Table A12: Best configurations for SR, divided by architectures.

| Dataset | optim. | l. rate | w. decay | t. decay | mom. | SGD nest. | n_blocks | d_token | att._drop. | res._drop. | d.ffn_factor | ffn_drop. | Arch. |
|---|---|---|---|---|---|---|---|---|---|---|---|---|---|
| adult | SGD | 1e−03 | 1e−04 | False | | | 1 | 128 | .25 | | 1.00 | .30 | |
| compass | Adam | 1e−03 | 1e−06 | False | | | 3 | 320 | .10 | | 2.00 | .25 | |
| covtype | Adam | 1e−04 | 1e−05 | False | | | 3 | 384 | .45 | | 2.33 | .35 | |
| electricity | Adam | 1e−04 | 1e−05 | False | | | 3 | 384 | .45 | | 2.33 | .35 | |
| eye | Adam | 1e−05 | 1e−06 | False | | | 1 | 384 | .15 | .05 | 2.33 | .20 | |
| heloc | SGD | 1e−04 | 1e−06 | False | .93 | True | 4 | 128 | .30 | | 1.00 | .30 | |
| higgs | Adam | 1e−05 | 1e−06 | False | | | 2 | 384 | .20 | | 2.00 | .15 | FTTransformer |
| house | Adam | 1e−05 | 1e−06 | False | | | 2 | 320 | .20 | .10 | 2.00 | .15 | |
| jannis | Adam | 1e−05 | 1e−06 | False | | | 2 | 384 | .25 | | 2.00 | .20 | |
| kddipums97 | Adam | 1e−05 | 1e−06 | False | | | 4 | 320 | .35 | .15 | 1.00 | .10 | |
| letter | Adam | 1e−05 | 1e−06 | False | | | 3 | 384 | .45 | .20 | 2.33 | .35 | |
| magic | Adam | 1e−04 | 1e−06 | False | | | 3 | 384 | .10 | | 2.33 | .30 | |
| online | Adam | 1e−04 | 1e−05 | False | | | 1 | 384 | .45 | | 2.33 | .35 | |
| phoneme | Adam | 1e−02 | 1e−04 | False | | | 1 | 128 | .05 | .05 | 1.33 | .45 | |
| pol | Adam | 1e−03 | 1e−04 | False | | | 1 | 512 | .50 | .10 | 1.67 | .40 | |
| r1 | Adam | 1e−04 | 1e−03 | True | | | 3 | 192 | .05 | | 1.67 | .30 | |
| upselling | SGD | 1e−02 | 1e−04 | False | .91 | False | 4 | 64 | .25 | | 1.00 | .20 | |

| Dataset | optim. | l. rate | w. decay | t. decay | mom. | SGD nest. | n_blocks | d_token | d_main | d_hidden_f | d_drop_first | d_drop_sec. | Arch. |
|---|---|---|---|---|---|---|---|---|---|---|---|---|---|
| aloi | AdamW | 1e−05 | 1e−06 | False | | | 4 | 192 | 320 | 4.00 | .10 | .20 | |
| bank | AdamW | 1e−03 | 1e−04 | True | | | 4 | 256 | 64 | 3.00 | .40 | .30 | |
| giveme | AdamW | 1e−04 | 1e−05 | True | | | 3 | 64 | 256 | 2.00 | .30 | .40 | TabResnet |
| helena | AdamW | 1e−05 | 1e−06 | False | | | 3 | 192 | 320 | 4.00 | .10 | .20 | |
| indian | AdamW | 1e−03 | 1e−04 | True | | | 4 | 256 | 64 | 3.00 | .40 | .30 | |
| miniboone | AdamW | 1e−03 | 1e−04 | True | | | 4 | 256 | 64 | 3.00 | .40 | .30 | |
| ucicredit | AdamW | 1e−04 | 1e−04 | True | .99 | True | 3 | 192 | 128 | 2.00 | .25 | .35 | |

| Dataset | optim. | l. rate | w. decay | t. decay | mom. | SGD nest. | zero_init_resid | Arch. |
|---|---|---|---|---|---|---|---|---|
| MNIST | Adam | 1e−04 | 1e−04 | False | | | False | |
| bloodmnist | AdamW | 1e−03 | 1e−04 | True | | | True | |
| breastmnist | AdamW | 1e−03 | 1e−04 | True | | | True | |
| chestmnist | AdamW | 1e−04 | 1e−03 | True | | | True | |
| dermamnist | AdamW | 1e−04 | 1e−04 | True | | | True | |
| food101 | Adam | 1e−03 | 1e−04 | True | | | False | |
| octmnist | AdamW | 1e−03 | 1e−04 | True | | | False | |
| organamnist | SGD | 1e−01 | 1e−04 | True | .93 | True | False | Resnet18-50 |
| organcmnist | AdamW | 1e−03 | 1e−03 | True | | | False | |
| organsmnist | Adam | 1e−04 | 1e−04 | False | | | True | |
| oxfordpets | Adam | 1e−02 | 1e−05 | True | .97 | True | True | |
| pathmnist | SGD | 1e−02 | 1e−05 | True | | | False | |
| pneumoniamnist | Adam | 1e−04 | 1e−06 | False | | | True | |
| retinamnist | Adam | 1e−05 | 1e−06 | False | | | True | |
| stanfordcars | Adam | 1e−03 | 1e−04 | False | | | False | |
| tissuemnist | AdamW | 1e−03 | 1e−04 | True | | | True | |
| waterbirds | Adam | 1e−04 | 1e−04 | False | | | False | |

| Dataset | optim. | l. rate | w. decay | t. decay | mom. | SGD nest. | b_norm | Arch. |
|---|---|---|---|---|---|---|---|---|
| SVHN | SGD | 1e−03 | 1e−06 | False | .91 | True | False | |
| catsdogs | Adam | 1e−03 | 1e−04 | False | | | False | VGG |
| cifar10 | Adam | 1e−03 | 1e−04 | False | | | True | |

## Appendix B. Additional Experimental Results

### B.1 Q1: Results by Dataset Type

Figure B1 plots the best two and the worst two baselines mean *RelErr* by data type.

For binary tabular datasets (Figure B1a), SAT+EM+SR and SR are the best two performing methods. The former's relative error rate ranges from $\approx .508$ at $c = .99$ to $\approx .405$ at $c = .70$, while the latter achieves $\approx .511$ at $c = .99$ and $\approx .393$ at $c = .70$. The worst two methods are DG, with *RelErr* of $\approx .632$ at $c = .99$ and $\approx .559$ at .70, and REG with *RelErr* of $\approx .547$ at .99 and $\approx .527$ at .70.

For multiclass tabular datasets (Figure B1c), the best two methods are ENS+SR and SAT+SR, with a mean relative error rate of $\approx .164$ and $\approx .158$ at $c = .99$ respectively, up to $\approx .094$ and $\approx .096$ at $c = .70$. The worst methods are REG, which reaches a mean relative error rate of $\approx .211$ at $c = .99$ and of $\approx .203$ at $c = .70$, and SELNET+EM+SR, with a relative error rate ranging from $\approx .195$ at $c = .99$ to $\approx .218$ at $c = .70$.

For image datasets, methods based on ensembles, i.e., ENS and ENS+SR, achieve the lowest relative error rate. For binary image datasets (Figure B1b), ENS+SR reaches $\approx .378$ at $c = .99$ and $\approx .228$ at $c = .70$, while ENS ranges from $\approx .386$ at $c = .99$ to $\approx .234$ at $c = .70$. In this setting, the worst baselines are SELE and DG, with a mean relative error rate of $\approx .529$ and $\approx .565$ at $c = .99$ respectively, up to $\approx .564$ and $\approx .582$ at $c = .70$ respectively.

For multiclass image datasets (Figure B1d), ENS+SR passes from a mean relative error rate of $\approx .189$ at $c = .99$ to $\approx .126$ at $c = .70$, while ENS achieves $\approx .191$ at $c = .99$ up to $\approx .151$ at $c = .70$. The worst methods here are REG and SELE. The former's relative error rate ranges from $\approx .276$ at $c = .99$ to $\approx .279$ at $c = .70$, while the latter achieves $\approx .272$ at $c = .99$ and $\approx .238$ at $c = .70$.

Then, we perform the Nemenyi post hoc test to check for statistically significant differences. Figures B2 and B3 provide CD plots when considering tabular and image data respectively at $c = .99$, $c = .90$, $c = .80$ and $c = .70$. As for aggregated results, all the best performing methods are not distinguishable in a statistically significant sense.

### B.2 Q5: Additional Results

Figure B4 provides the detailed results for the out-of-distribution test sets.

Moreover, we provide additional results w.r.t. distribution shifts. We perform the same experiment as for Q5, but now considering datasets in the `OpenOOD` benchmark (Yang et al., 2022), which is specific for out-of-distribution detection, rather than randomly generated pictures. For `cifar10` we use as test set a random sample from `cifar100`, for `MNIST` a random sample from `FashionMNIST` and for `SVHN` a random sample from `cifar10`. Figure B5 reports the results and the overall mean over the 3 datasets for the 16 baselines considered.

Similarly to the experiments in Section 5.2, we observe that under this milder data shift, no baseline drops all the instances at $c = .99$. We can also see that for lower target coverages, we have higher rejection rates, as expected. Moreover, there is a clear worst-performing method, namely REG.

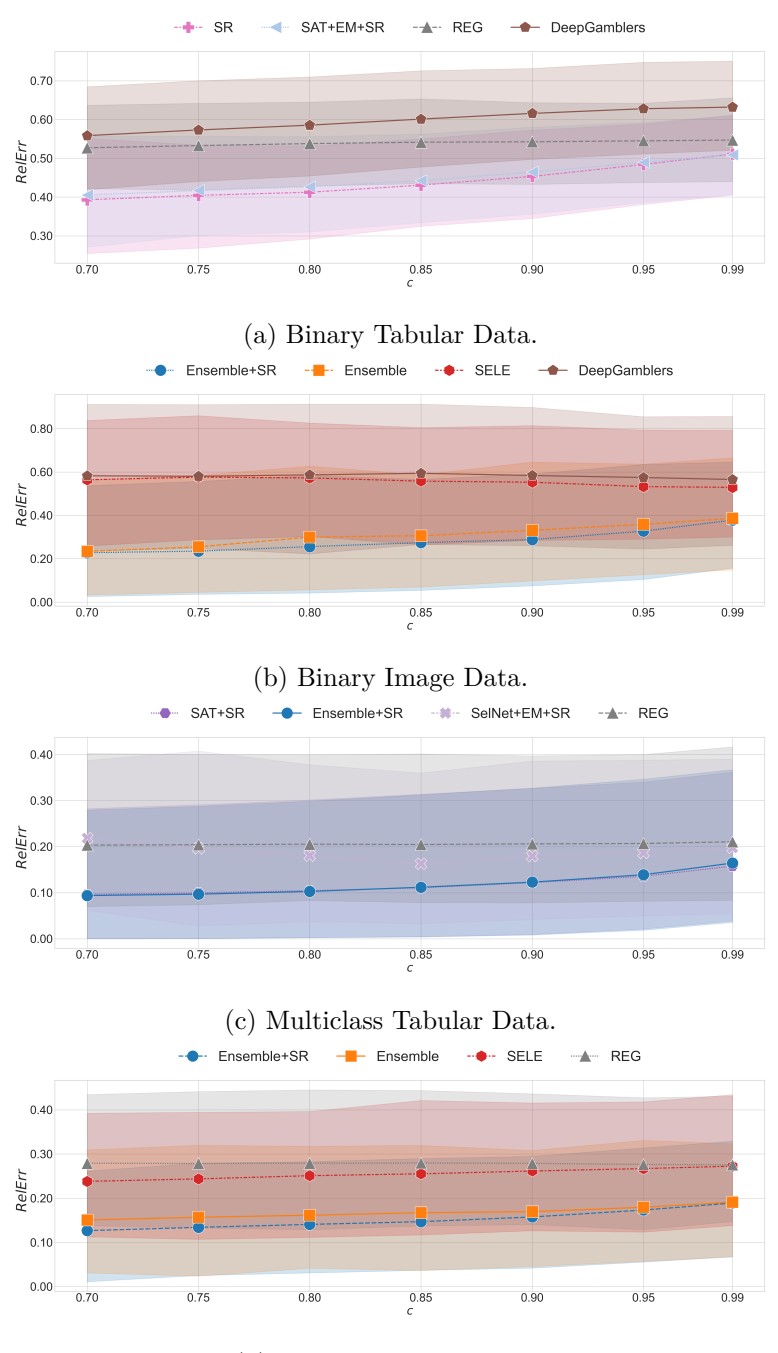

(a) Binary Tabular Data.

(b) Binary Image Data.

(c) Multiclass Tabular Data.

(d) Multiclass Image Data.

Figure B1: Q1: *RelErr* as a function of target coverage *c* for the two best and worst approaches on (a) binary tabular data, (b) binary image data, (c) multiclass tabular data and (d) multiclass image data. For readability, only the two best and two worst approaches are shown in each subplot.

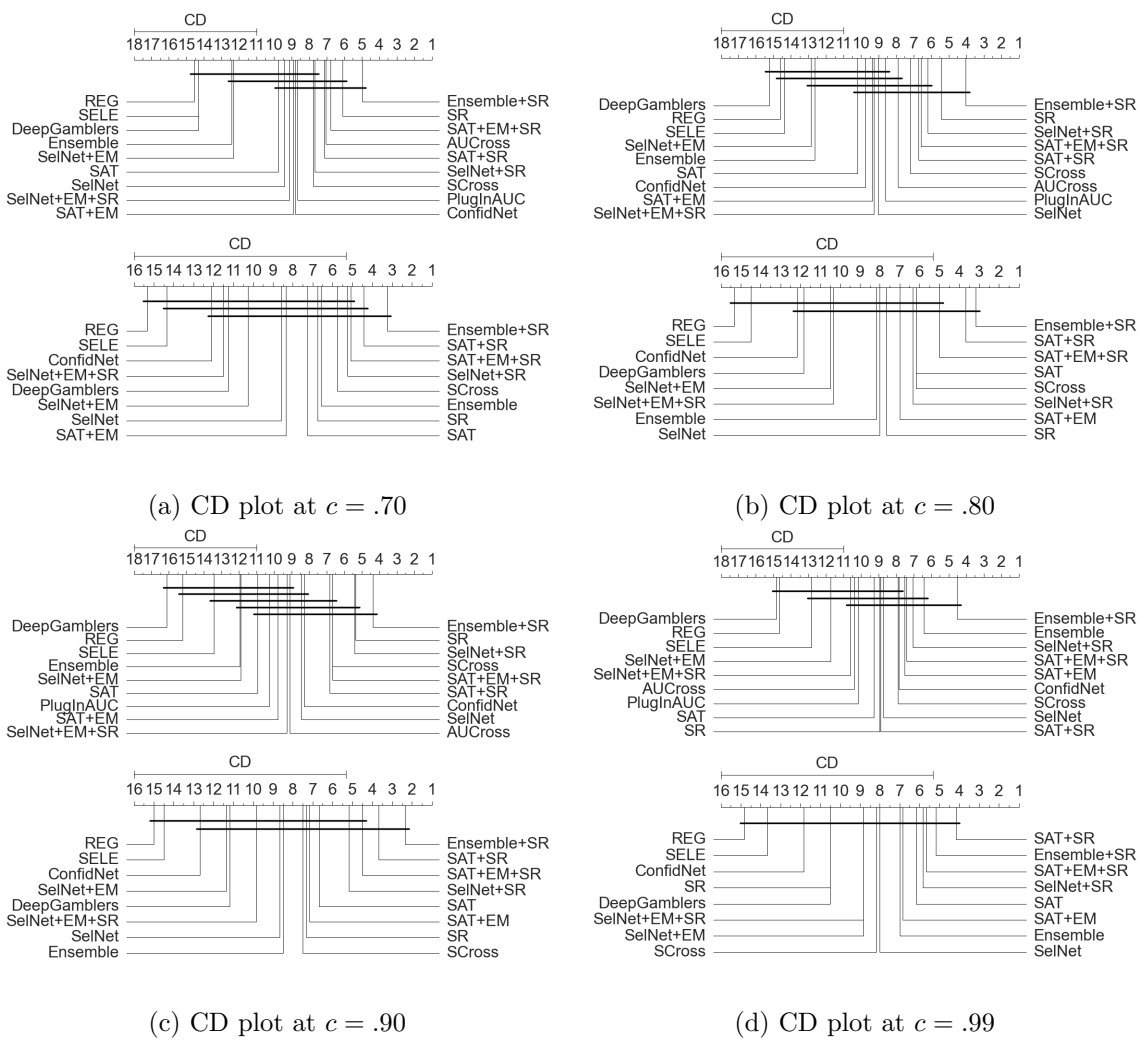

(a) CD plot at $c = .70$

(b) CD plot at $c = .80$

(c) CD plot at $c = .90$

(d) CD plot at $c = .99$

Figure B2: Q1: CD plots of relative error rate *RelErr* for different target coverages on tabular datasets. Top plots for binary datasets. Bottom plots for multiclass datasets.

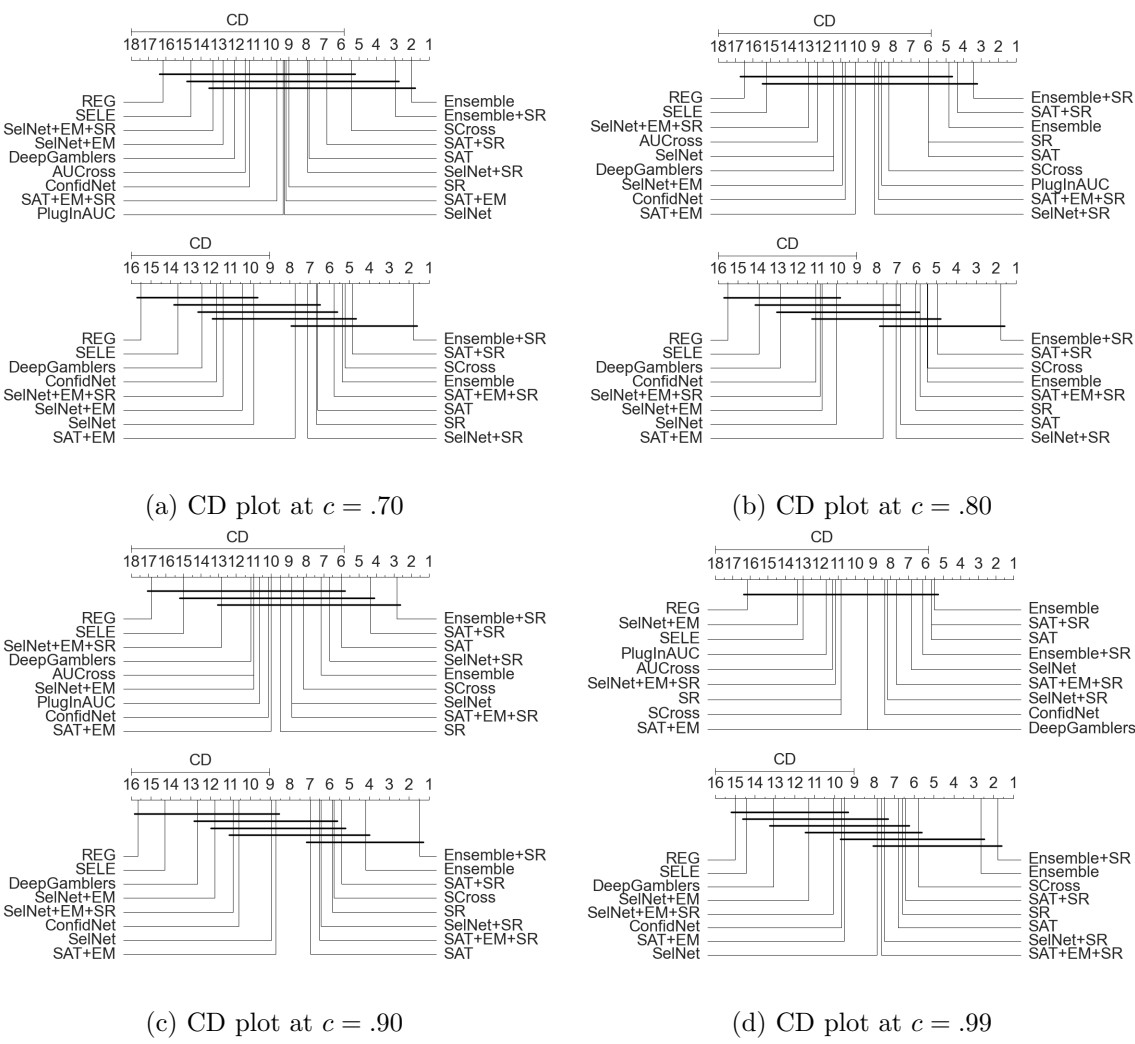

(a) CD plot at $c = .70$      (b) CD plot at $c = .80$

(c) CD plot at $c = .90$      (d) CD plot at $c = .99$

Figure B3: Q1: CD plots of relative error rate *RelErr* for different target coverages on image datasets. Top plots for binary datasets. Bottom plots for multiclass datasets.

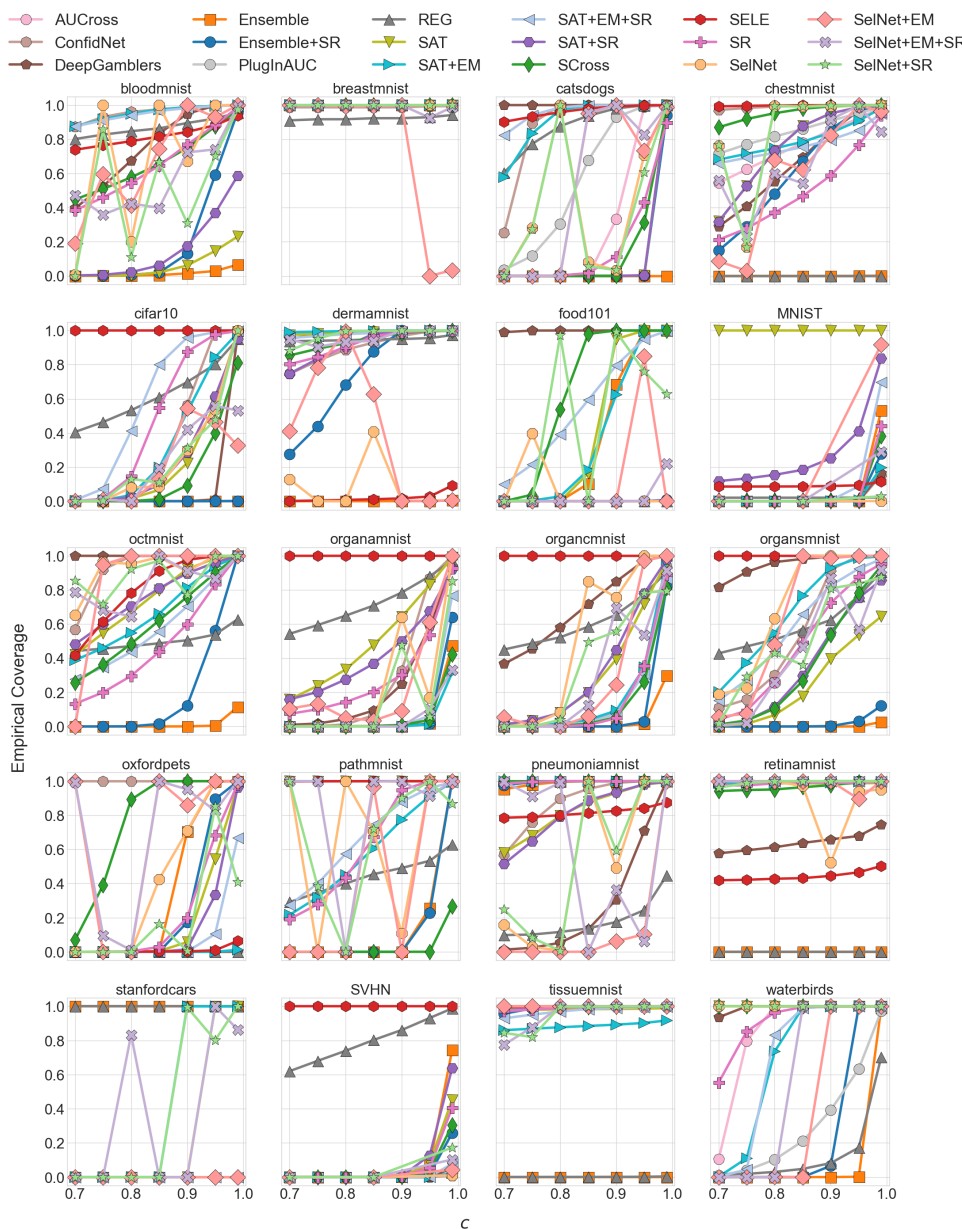

Figure B4: Q5: Empirical coverage $\hat{\phi}$ for out-of-distribution test sets on 20 image datasets for different target coverages $c$.

For `cifar10`, the method dropping more instances is ENS+SR, reaching an actual coverage of $\approx 5.7\%$ at $c = .70$. The runner-up is ENS, accepting only $\approx 6\%$ of instances. All the remaining baselines have an empirical coverage above $8\%$ at $c = .70$.

For `MNIST` and `SVHN` we observe similar patterns: eleven out of sixteen baselines reach a coverage below $1\%$ at $c = .70$. We also highlight that SELNET reaches zero coverage for $c = .75$ and $.70$ on the `SVHN` dataset.

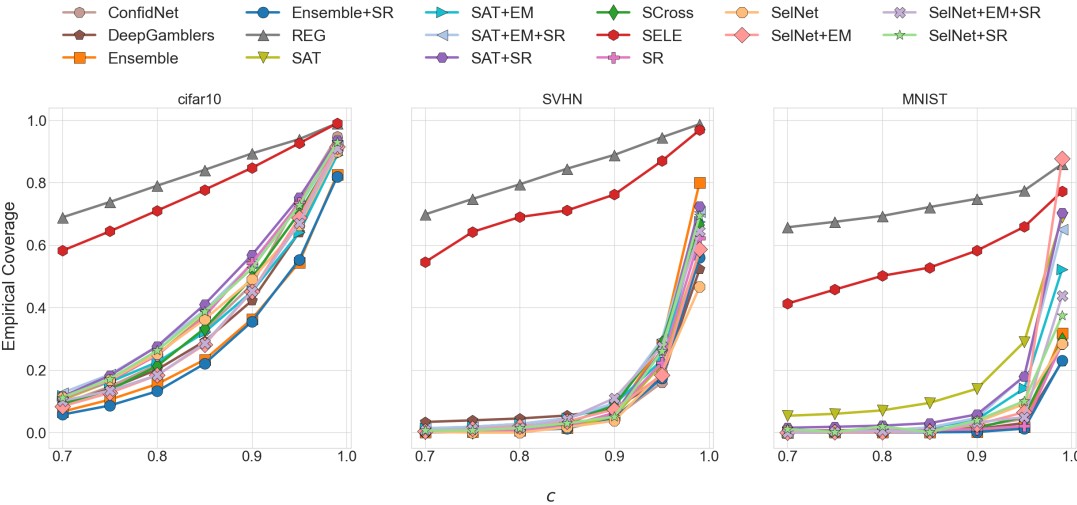

Figure B5: Q5: Empirical coverage $\hat{\phi}$ for out-of-distribution test sets on 3 image datasets for different target coverages $c$.

To conclude, the experiments show the difficulty for current SC methods to properly perform rejection under distribution shifts. For test data close to the (learned) decision boundary, the baselines correctly reject the instances, since all the methods are built to perform ambiguity rejection. For test data far from the decision boundary, the selection function become confident that the shifted instances are very likely to belong to a certain training class, ending up not rejecting the instances. We think that a potential way to mitigate these problems consists of mixing ambiguity rejection with novelty rejection methods, highlighting the need for further research in this direction.

## B.3 Dataset-level Results

We provide detailed results for all the datasets in Tables B1-B44. Each table reports $mean \pm std$ over the 100 bootstrap samples of $\hat{Err}$ and empirical coverage $\hat{\phi}$, for every baseline and every target coverage. For binary datasets, we also include the *MinCoeff* metric introduced in the main text. For error rate, we highlight the baseline column with the lowest average (and standard error in case of ties) in bold case. For *Coverage* and *MinCoeff*, we highlight in bold the values closest to target coverage $c$ and 1, respectively.

Table B1: Results for `adult`: $mean \pm std$ for $\widehat{Err}$, empirical coverage $\hat{\phi}$, and *MinCoeff*.

| Metric | c | DG | SAT | SAT+EM | SelNet | SelNet+EM | SR | SAT+SR | SAT+EM+SR | SelNet+SR | SelNet+EM+SR | ENS | ENS+SR | ConfidNet | SELE | REG | SCross | AUCross | PlugInAUC |
|---|---|---|---|---|---|---|---|---|---|---|---|---|---|---|---|---|---|---|---|
| $\hat{Err}$ | .99 | .153±.003 | .136±.004 | .125±.004 | **.123±.003** | .160±.003 | **.123±.003** | .132±.004 | .132±.004 | .132±.004 | .158±.003 | .130±.003 | .125±.003 | .138±.004 | .133±.004 | .133±.003 | .130±.004 | .131±.004 | .131±.004 |
| | .95 | .154±.003 | .126±.004 | .115±.004 | .108±.003 | .137±.004 | **.101±.003** | .118±.004 | .116±.004 | .114±.004 | .135±.004 | .130±.003 | .103±.003 | .131±.004 | .128±.004 | .123±.003 | .116±.003 | .125±.004 | .133±.004 |
| | .90 | .142±.003 | .113±.004 | .102±.004 | .096±.003 | .117±.003 | **.080±.003** | .097±.003 | .101±.003 | .096±.003 | .120±.003 | .130±.003 | **.080±.003** | .121±.004 | .119±.004 | .119±.003 | .101±.003 | .114±.003 | .134±.004 |
| | .85 | .119±.003 | .097±.003 | .088±.003 | .080±.003 | .106±.003 | **.061±.003** | .080±.003 | .083±.003 | .078±.003 | .106±.003 | .125±.003 | **.061±.003** | .106±.004 | .104±.004 | .113±.003 | .086±.003 | .103±.003 | .134±.004 |
| | .80 | .097±.003 | .079±.003 | .071±.003 | .067±.003 | .082±.003 | **.046±.002** | .067±.003 | .071±.003 | .067±.003 | .082±.003 | .126±.003 | .048±.003 | .091±.003 | .092±.004 | .107±.004 | .075±.003 | .089±.003 | .134±.004 |
| | .75 | .077±.003 | .062±.003 | .058±.003 | .056±.003 | .076±.003 | **.030±.002** | .056±.003 | .057±.003 | .055±.003 | .064±.003 | .123±.003 | .035±.002 | .074±.003 | .082±.004 | .101±.004 | .064±.003 | .073±.003 | .131±.004 |
| | .70 | .063±.003 | .055±.003 | .046±.003 | .045±.003 | .066±.003 | **.020±.002** | .044±.003 | .044±.003 | .045±.003 | .065±.003 | .118±.003 | .024±.002 | .056±.003 | .076±.003 | .099±.004 | .055±.003 | .057±.003 | .128±.004 |
| $\hat{\phi}$ | .99 | .983±.001 | .994±.001 | .967±.002 | .924±.003 | .998±.000 | .980±.001 | .985±.001 | **.991±.001** | **.991±.001** | .994±.001 | .995±.001 | .986±.001 | .980±.001 | .984±.001 | .983±.001 | .992±.001 | **.991±.001** | **.990±.001** |
| | .95 | .927±.003 | .958±.002 | .925±.002 | .913±.003 | .951±.002 | .912±.003 | .943±.002 | .945±.002 | .939±.002 | .942±.002 | .982±.001 | .917±.002 | .936±.003 | .955±.002 | .923±.003 | .955±.002 | .948±.002 | **.949±.002** |
| | .90 | .855±.004 | .914±.003 | .875±.003 | .886±.003 | .915±.003 | .842±.003 | .883±.003 | .891±.003 | .883±.003 | .924±.002 | .969±.002 | .840±.003 | .888±.003 | .906±.003 | .878±.004 | .907±.003 | **.896±.003** | **.896±.003** |
| | .85 | .793±.004 | .855±.003 | .825±.004 | .818±.003 | **.854±.003** | .771±.004 | .824±.003 | .836±.003 | .820±.003 | .960±.002 | .767±.004 | .839±.004 | .813±.004 | .825±.004 | .859±.003 | .844±.003 | .836±.004 |
| | .80 | .742±.004 | **.798±.004** | .772±.004 | .775±.004 | .796±.004 | .689±.004 | .771±.004 | .789±.004 | .772±.004 | .795±.004 | .948±.002 | .701±.004 | .790±.004 | .741±.004 | .768±.004 | .815±.004 | .790±.004 | .780±.004 |
| | .75 | .686±.004 | .732±.004 | .722±.004 | .725±.004 | .731±.004 | .616±.005 | .723±.004 | .734±.004 | .722±.004 | .675±.004 | .931±.002 | .632±.005 | **.740±.004** | .699±.005 | .723±.004 | .768±.004 | .738±.005 | .713±.005 |
| | .70 | .632±.005 | .679±.004 | .671±.004 | .672±.004 | .683±.004 | .563±.005 | .669±.004 | .679±.004 | .674±.004 | .677±.004 | .909±.002 | .579±.005 | **.687±.005** | .670±.005 | .686±.005 | .728±.004 | **.687±.005** | .659±.004 |
| *MinCoeff* | .99 | .948±.018 | .994±.019 | .949±.020 | .835±.017 | .999±.019 | .980±.019 | .983±.019 | .993±.018 | .992±.019 | .995±.019 | .999±.019 | .988±.019 | .999±.019 | **.997±.019** | .994±.019 | .988±.019 | .995±.019 | 1.007±.019 |
| | .95 | .788±.018 | .946±.019 | .872±.019 | .897±.019 | .941±.020 | .907±.019 | .935±.019 | .942±.019 | .934±.018 | .920±.019 | .977±.019 | .921±.018 | .988±.020 | .981±.019 | .960±.020 | .936±.019 | **.988±.019** | 1.035±.020 |
| | .90 | .612±.017 | .878±.019 | .770±.019 | .872±.019 | .902±.019 | .810±.019 | .870±.019 | .886±.019 | .871±.019 | .871±.020 | .964±.019 | .827±.019 | .969±.020 | .948±.019 | .944±.020 | .871±.019 | **.973±.019** | 1.068±.021 |
| | .85 | .490±.016 | .778±.017 | .690±.016 | .789±.018 | .812±.018 | .712±.018 | .794±.018 | .817±.018 | .799±.018 | .813±.018 | **.955±.019** | .726±.018 | .926±.019 | .836±.019 | .904±.020 | .798±.019 | .953±.020 | 1.104±.022 |
| | .80 | .393±.014 | .675±.016 | .604±.015 | .729±.018 | .801±.020 | .606±.016 | .730±.018 | .762±.018 | .739±.018 | .800±.019 | **.943±.019** | .637±.017 | .872±.019 | .720±.018 | .854±.020 | .731±.018 | .933±.020 | 1.143±.022 |
| | .75 | .320±.013 | .569±.015 | .533±.015 | .650±.017 | .552±.016 | .533±.017 | .659±.018 | .695±.018 | .666±.018 | .531±.016 | **.927±.020** | .550±.017 | .799±.019 | .633±.018 | .799±.020 | .655±.017 | .907±.021 | 1.184±.024 |
| | .70 | .261±.013 | .460±.015 | .479±.016 | .550±.016 | .537±.015 | .511±.018 | .582±.017 | .608±.016 | .577±.017 | .541±.015 | **.903±.019** | .504±.017 | .674±.018 | .557±.017 | .771±.020 | .597±.017 | .872±.021 | 1.222±.025 |

Table B2: Results for `aloi`: *mean* ± *std* for $\widehat{Err}$ and empirical coverage $\hat{\phi}$.

| Metric | c | DG | SAT | SAT+EM | SelNet | SelNet+EM | SR | SAT+SR | SAT+EM+SR | SelNet+SR | SelNet+EM+SR | ENS | ENS+SR | ConfidNet | SELE | REG | SCross |
|---|---|---|---|---|---|---|---|---|---|---|---|---|---|---|---|---|---|
| $\widehat{Err}$ | .99 | .035±.001 | .034±.001 | .029±.001 | .039±.001 | .031±.001 | .036±.001 | .033±.001 | **.027±.001** | .037±.001 | .030±.001 | .034±.001 | .032±.001 | .035±.001 | .063±.002 | .063±.002 | .031±.001 |
| | .95 | .026±.001 | .022±.001 | .017±.001 | .032±.001 | .047±.001 | .018±.001 | .017±.001 | **.013±.001** | .029±.001 | .035±.001 | .022±.001 | .016±.001 | .032±.001 | .056±.002 | .063±.002 | .016±.001 |
| | .90 | .022±.001 | .013±.001 | .009±.001 | .026±.001 | .044±.002 | .007±.001 | .007±.001 | .006±.001 | .018±.001 | .024±.001 | .014±.001 | **.005±.001** | .030±.001 | .049±.002 | .062±.002 | .006±.001 |
| | .85 | .019±.001 | .008±.001 | .005±.000 | .023±.001 | .018±.001 | **.002±.000** | **.002±.000** | .004±.000 | .008±.001 | .012±.001 | .008±.001 | **.002±.000** | .030±.001 | .042±.002 | .060±.002 | **.002±.000** |
| | .80 | .016±.001 | .006±.001 | .003±.000 | .022±.001 | .035±.001 | **.001±.000** | **.001±.000** | .002±.000 | .009±.001 | .011±.001 | .006±.001 | **.001±.000** | .030±.001 | .036±.001 | .059±.002 | **.001±.000** |
| | .75 | .014±.001 | .004±.000 | .002±.000 | .019±.001 | .032±.001 | .001±.000 | .001±.000 | .001±.000 | .006±.001 | .011±.001 | .002±.000 | **.000±.000** | .029±.001 | .031±.001 | .056±.002 | .001±.000 |
| | .70 | .012±.001 | .003±.000 | .002±.000 | .019±.001 | .022±.001 | **.000±.000** | **.000±.000** | .001±.000 | .005±.001 | .012±.001 | .001±.000 | **.000±.000** | .028±.001 | .027±.001 | .054±.002 | **.000±.000** |
| $\hat{\phi}$ | .99 | .991±.001 | .989±.001 | .991±.001 | **.991±.001** | .989±.001 | .991±.001 | .991±.001 | **.990±.001** | .991±.001 | .992±.001 | .991±.001 | **.990±.001** | **.991±.001** | .991±.001 | .991±.001 | .994±.001 |
| | .95 | .951±.001 | **.950±.001** | .954±.001 | **.949±.001** | **.951±.001** | **.951±.001** | .953±.001 | .956±.001 | .954±.001 | .949±.001 | **.950±.002** | .952±.002 | **.950±.001** | .952±.002 | .948±.001 | .959±.001 |
| | .90 | .904±.002 | **.900±.002** | .902±.002 | **.901±.002** | **.901±.002** | .902±.002 | .903±.002 | .905±.002 | **.899±.002** | .897±.002 | .899±.002 | **.901±.002** | **.901±.002** | .897±.002 | **.900±.002** | .916±.002 |
| | .85 | .854±.003 | .852±.003 | .850±.002 | .848±.003 | .848±.002 | .849±.003 | .847±.002 | .852±.003 | **.850±.002** | .855±.002 | .848±.002 | .851±.002 | .857±.002 | .848±.003 | **.849±.002** | .871±.002 |
| | .80 | .803±.003 | .806±.003 | .798±.003 | **.800±.003** | **.801±.003** | **.800±.003** | **.801±.003** | **.799±.003** | .803±.003 | **.800±.003** | **.800±.003** | **.800±.003** | .806±.003 | **.800±.003** | **.800±.003** | .828±.002 |
| | .75 | .755±.003 | .754±.003 | .746±.003 | **.750±.003** | .751±.003 | .755±.003 | .751±.003 | .745±.003 | .752±.003 | .752±.003 | .752±.003 | .753±.003 | .753±.003 | .751±.003 | .744±.003 | .781±.003 |
| | .70 | .706±.003 | **.702±.003** | .697±.003 | **.701±.004** | .704±.003 | .704±.003 | **.703±.003** | .694±.003 | **.702±.003** | **.702±.003** | .699±.003 | **.702±.003** | **.701±.003** | **.700±.004** | .695±.003 | .738±.003 |

Table B3: Results for `bank`: *mean* ± *std* for $\widehat{Err}$, empirical coverage $\hat{\phi}$, and *MinCoeff*.

| Metric | c | DG | SAT | SAT+EM | SelNet | SelNet+EM | SR | SAT+SR | SAT+EM+SR | SelNet+SR | SelNet+EM+SR | ENS | ENS+SR | ConfidNet | SELE | REG | SCross | AUCross | PlugInAUC |
|---|---|---|---|---|---|---|---|---|---|---|---|---|---|---|---|---|---|---|---|
| $\widehat{Err}$ | .99 | .112±.003 | .089±.002 | .087±.002 | .092±.002 | .092±.002 | .092±.002 | .087±.002 | .087±.002 | .090±.002 | .092±.002 | .092±.003 | .090±.002 | **.084±.002** | .091±.002 | .103±.003 | .090±.002 | .092±.002 | .095±.003 |
| | .95 | .090±.003 | .073±.002 | .073±.002 | .076±.002 | .075±.002 | .072±.002 | **.068±.002** | .072±.002 | .073±.002 | .073±.002 | .082±.003 | .072±.002 | .071±.002 | .081±.002 | .103±.003 | .073±.002 | .089±.003 | .093±.003 |
| | .90 | .063±.002 | .054±.002 | .055±.002 | .056±.002 | .056±.002 | .054±.002 | **.051±.002** | .053±.002 | .052±.002 | .055±.002 | .067±.002 | .054±.002 | .055±.002 | .063±.002 | .103±.003 | .055±.002 | .084±.003 | .091±.003 |
| | .85 | .041±.002 | .038±.002 | .039±.002 | .042±.002 | .041±.002 | .039±.002 | **.036±.002** | .037±.002 | .039±.002 | .042±.002 | .053±.002 | .037±.002 | .041±.002 | .050±.002 | .103±.003 | **.036±.002** | .078±.003 | .087±.003 |
| | .80 | .028±.002 | .028±.002 | .026±.002 | .027±.002 | .028±.002 | .027±.002 | .026±.002 | **.025±.001** | .027±.002 | .028±.002 | .043±.002 | .027±.002 | .026±.002 | .034±.002 | .101±.003 | .025±.002 | .069±.003 | .081±.003 |
| | .75 | .019±.001 | **.016±.001** | .018±.001 | .019±.001 | .018±.001 | .019±.001 | .017±.001 | .017±.001 | .019±.001 | .022±.001 | .033±.002 | .018±.001 | .019±.002 | .021±.002 | .099±.003 | .017±.001 | .052±.003 | .071±.003 |
| | .70 | .013±.001 | .013±.001 | **.011±.001** | .013±.001 | .014±.001 | .012±.001 | .012±.001 | **.011±.001** | .013±.001 | .015±.001 | .023±.002 | .012±.001 | .012±.001 | .015±.001 | .099±.003 | .012±.001 | .037±.002 | .056±.003 |
| $\hat{\phi}$ | .99 | **.990±.001** | **.990±.001** | .988±.001 | .992±.001 | .988±.001 | **.991±.001** | .987±.001 | .986±.001 | .988±.001 | **.991±.001** | .989±.001 | .989±.001 | .986±.001 | **.990±.001** | **.990±.001** | **.991±.001** | .988±.001 | **.991±.001** |
| | .95 | .952±.002 | .947±.002 | **.950±.002** | **.949±.002** | .947±.002 | .942±.002 | .941±.003 | .946±.002 | .946±.003 | .948±.002 | **.951±.002** | .946±.002 | **.949±.002** | **.949±.002** | **.950±.002** | **.951±.002** | .945±.002 | .946±.002 |
| | .90 | **.898±.003** | .895±.003 | .897±.003 | **.899±.003** | .897±.003 | .896±.003 | .892±.003 | .892±.003 | .890±.003 | .892±.003 | .904±.003 | .896±.003 | **.901±.003** | **.898±.003** | **.900±.003** | **.899±.003** | .897±.003 | .897±.004 |
| | .85 | .845±.004 | .846±.004 | **.850±.004** | **.851±.004** | **.849±.004** | .848±.004 | .846±.004 | .847±.004 | .848±.004 | .848±.004 | **.849±.004** | .848±.004 | .856±.003 | .854±.004 | .856±.004 | .842±.004 | .848±.004 | .855±.004 |
| | .80 | .798±.004 | .803±.004 | .799±.004 | **.801±.004** | .802±.004 | **.801±.004** | **.800±.004** | .800±.004 | .798±.004 | **.800±.004** | .799±.004 | **.799±.004** | .802±.004 | .799±.004 | .810±.004 | .790±.004 | .793±.005 | .807±.004 |
| | .75 | **.751±.004** | .753±.004 | .755±.004 | .753±.004 | .755±.004 | .754±.004 | .756±.004 | .755±.004 | **.750±.004** | .754±.004 | .749±.004 | .755±.004 | .755±.004 | .751±.004 | .755±.005 | .743±.004 | .746±.005 | .754±.005 |
| | .70 | **.700±.005** | .706±.004 | .704±.005 | .702±.004 | .704±.004 | .701±.004 | .706±.004 | .706±.004 | .705±.004 | .704±.004 | **.701±.004** | .709±.004 | .702±.005 | .702±.004 | .710±.005 | .695±.004 | .690±.005 | .699±.005 |
| *MinCoeff* | .99 | .957±.027 | .962±.028 | .959±.029 | .969±.028 | .953±.028 | .986±.029 | .959±.028 | .964±.028 | .958±.027 | .977±.028 | .974±.028 | .974±.030 | .948±.028 | .964±.028 | **.995±.029** | .980±.029 | **.997±.029** | 1.001±.029 |
| | .95 | .772±.025 | .797±.027 | .815±.028 | .798±.026 | .797±.026 | .829±.028 | .789±.026 | .854±.026 | .824±.025 | .820±.026 | .902±.028 | .855±.028 | .812±.027 | .783±.026 | .998±.030 | .865±.026 | .976±.030 | .993±.030 |
| | .90 | .539±.019 | .573±.024 | .605±.024 | .594±.022 | .603±.022 | .690±.025 | .591±.024 | .629±.024 | .585±.022 | .621±.023 | .814±.029 | .684±.024 | .612±.023 | .572±.022 | **1.000±.030** | .690±.024 | .950±.032 | .985±.032 |
| | .85 | .354±.017 | .377±.017 | .412±.019 | .355±.015 | .349±.015 | .491±.021 | .420±.019 | .452±.021 | .405±.018 | .381±.016 | .704±.029 | .511±.023 | .465±.021 | .420±.018 | **.988±.032** | .474±.022 | .913±.032 | .963±.032 |
| | .80 | .236±.014 | .240±.014 | .229±.013 | .231±.014 | .237±.014 | .315±.017 | .269±.015 | .269±.016 | .276±.016 | .277±.017 | .579±.027 | .313±.019 | .223±.014 | .280±.018 | **.986±.033** | .284±.017 | .848±.032 | .925±.031 |
| | .75 | .161±.012 | .138±.011 | .157±.010 | .161±.011 | .156±.011 | .170±.014 | .148±.011 | .162±.010 | .161±.011 | .238±.015 | .433±.023 | .165±.014 | .162±.013 | .177±.014 | **.961±.034** | .150±.012 | .759±.031 | .856±.032 |
| | .70 | .108±.011 | .108±.011 | .092±.010 | .109±.011 | .116±.011 | .106±.010 | .102±.011 | .096±.010 | .112±.011 | .131±.012 | .328±.020 | .104±.011 | .104±.011 | .124±.012 | **.964±.036** | .103±.011 | .626±.028 | .770±.033 |

Table B4: Results for `bloodmnist`: *mean* ± *std* for $\widehat{Err}$ and empirical coverage $\hat{\phi}$.

| Metric | c | DG | SAT | SAT+EM | SelNet | SelNet+EM | SR | SAT+SR | SAT+EM+SR | SelNet+SR | SelNet+EM+SR | ENS | ENS+SR | ConfidNet | SELE | REG | SCross |
|---|---|---|---|---|---|---|---|---|---|---|---|---|---|---|---|---|---|
| $\widehat{Err}$ | .99 | .066±.004 | **.033±.003** | .039±.003 | .036±.003 | .045±.003 | .043±.003 | .036±.003 | .039±.003 | .038±.003 | .043±.003 | .037±.003 | .036±.003 | .047±.004 | .064±.004 | .064±.004 | .041±.003 |
| | .95 | .056±.004 | .023±.003 | .026±.003 | .025±.003 | .034±.003 | .024±.003 | .022±.003 | .026±.003 | .021±.002 | .030±.003 | .022±.002 | **.019±.002** | .035±.003 | .063±.004 | .065±.004 | .026±.003 |
| | .90 | .043±.003 | .016±.002 | .014±.002 | .015±.002 | .021±.003 | .015±.002 | .014±.002 | .013±.002 | .013±.002 | .018±.002 | .013±.002 | **.012±.002** | .027±.004 | .062±.004 | .065±.004 | .015±.002 |
| | .85 | .037±.003 | .010±.002 | .008±.002 | .012±.002 | .009±.002 | .010±.002 | .008±.002 | **.007±.002** | .008±.002 | .008±.002 | .008±.002 | .008±.002 | .020±.004 | .063±.004 | .063±.004 | .009±.002 |
| | .80 | .029±.003 | **.005±.001** | .005±.002 | .007±.002 | .011±.002 | **.005±.001** | **.005±.001** | **.005±.001** | **.005±.001** | .007±.002 | **.005±.001** | **.005±.001** | .016±.004 | .064±.004 | .063±.004 | .005±.002 |
| | .75 | .025±.003 | .005±.001 | .004±.001 | .006±.002 | .006±.002 | .004±.001 | .005±.001 | .004±.001 | .004±.001 | .005±.001 | .004±.001 | **.002±.001** | .011±.004 | .065±.004 | .066±.004 | .003±.001 |
| | .70 | .018±.003 | .004±.001 | .002±.001 | .003±.001 | .005±.001 | .004±.001 | .003±.001 | .002±.001 | .002±.001 | .004±.001 | .002±.001 | **.001±.001** | .009±.003 | .066±.004 | .067±.004 | .002±.001 |
| $\hat{\phi}$ | .99 | **.990±.002** | .984±.002 | **.991±.002** | **.990±.002** | .989±.002 | .992±.002 | .991±.002 | .991±.002 | .992±.002 | .988±.002 | .984±.002 | **.989±.002** | .989±.004 | .991±.002 | .993±.002 | **.991±.001** |
| | .95 | .959±.003 | .946±.004 | .946±.004 | **.948±.004** | .947±.004 | .948±.004 | .946±.004 | .955±.003 | .948±.004 | **.949±.004** | .941±.004 | .942±.004 | .947±.004 | **.950±.004** | .955±.004 | **.951±.004** |
| | .90 | .909±.005 | .896±.005 | .894±.005 | **.900±.005** | .908±.004 | **.901±.006** | .891±.005 | .893±.005 | .893±.005 | **.901±.005** | .904±.005 | .894±.005 | .896±.005 | .901±.006 | .893±.006 | **.901±.005** |
| | .85 | .861±.006 | .843±.006 | **.846±.005** | .841±.006 | .842±.006 | .841±.007 | .836±.006 | .846±.005 | **.847±.006** | .840±.006 | .843±.006 | .845±.007 | .850±.007 | .844±.007 | .843±.006 | .844±.006 |
| | .80 | .814±.007 | .795±.007 | **.801±.006** | .798±.007 | .811±.006 | .795±.007 | .787±.007 | **.799±.006** | .800±.007 | .800±.007 | .802±.007 | .799±.007 | .805±.007 | .788±.008 | .795±.008 | .802±.007 |
| | .75 | .767±.007 | .751±.008 | **.747±.006** | **.747±.007** | .754±.007 | .754±.007 | .746±.008 | **.753±.006** | .747±.007 | .754±.007 | .749±.007 | **.751±.007** | .757±.007 | .737±.008 | .742±.008 | .759±.007 |
| | .70 | .705±.008 | .703±.008 | .698±.007 | **.699±.007** | .709±.008 | .705±.008 | .712±.008 | **.698±.007** | .689±.007 | **.703±.007** | .707±.008 | .702±.008 | .702±.009 | .680±.009 | .682±.009 | .712±.008 |

Table B5: Results for `breastmnist`: *mean* ± *std* for $\widehat{Err}$, empirical coverage $\hat{\phi}$, and *MinCoeff*.

| Metric | c | DG | SAT | SAT+EM | SelNet | SelNet+EM | SR | SAT+SR | SAT+EM+SR | SelNet+SR | SelNet+EM+SR | ENS | ENS+SR | ConfidNet | SELE | REG | SCross | AUCross | PlugInAUC |
|---|---|---|---|---|---|---|---|---|---|---|---|---|---|---|---|---|---|---|---|
| $\widehat{Err}$ | .99 | **.137±.029** | .161±.031 | .159±.030 | .143±.031 | .177±.033 | .176±.034 | .158±.031 | .145±.030 | .141±.031 | .163±.033 | .163±.032 | .151±.032 | .189±.035 | .189±.033 | .205±.035 | .149±.031 | .147±.031 | .188±.034 |
| | .95 | .145±.031 | .160±.031 | .158±.031 | .155±.033 | .192±.034 | .164±.033 | .154±.032 | .123±.027 | **.119±.029** | .170±.034 | .145±.031 | .122±.031 | .173±.037 | .186±.034 | .206±.036 | .138±.030 | .146±.032 | .188±.034 |
| | .90 | .150±.031 | .145±.031 | .135±.031 | .157±.035 | .155±.033 | .161±.034 | .116±.027 | .126±.028 | .108±.030 | .141±.033 | .123±.029 | **.091±.028** | .177±.038 | .182±.035 | .192±.033 | .137±.031 | .132±.030 | .165±.033 |
| | .85 | .139±.030 | .135±.031 | .129±.029 | .122±.032 | .139±.033 | .109±.032 | .107±.030 | .104±.028 | **.082±.028** | .140±.033 | .112±.026 | .092±.028 | .173±.039 | .155±.036 | .205±.035 | .131±.032 | .133±.031 | .143±.034 |
| | .80 | .133±.032 | .140±.032 | .119±.030 | .146±.033 | .078±.025 | .094±.031 | .107±.030 | .100±.029 | .095±.031 | .110±.031 | .115±.029 | **.069±.024** | .159±.039 | .162±.037 | .195±.034 | .104±.029 | .117±.030 | .107±.032 |
| | .75 | .135±.032 | .141±.032 | .108±.029 | .109±.033 | .113±.030 | .099±.032 | .101±.030 | .089±.027 | .088±.034 | .131±.035 | .052±.022 | **.051±.022** | .165±.040 | .167±.038 | .192±.035 | .076±.026 | .084±.026 | .098±.032 |
| | .70 | .144±.034 | .115±.030 | .111±.030 | .076±.029 | .092±.029 | .105±.034 | .103±.030 | .081±.026 | .064±.027 | .082±.033 | **.035±.020** | .045±.022 | .169±.041 | .142±.037 | .200±.036 | .072±.025 | .072±.025 | .103±.034 |
| $\hat{\phi}$ | .99 | 1.000±.000 | **.994±.006** | .962±.017 | .987±.009 | .973±.015 | .961±.017 | .971±.012 | .961±.015 | .969±.013 | .982±.011 | **.993±.006** | .988±.008 | .974±.014 | .981±.011 | .972±.015 | .975±.013 | .988±.008 | 1.000±.000 |
| | .95 | .949±.018 | .957±.017 | .931±.021 | .938±.019 | .976±.012 | .947±.020 | **.959±.014** | .960±.020 | .930±.020 | .918±.023 | .972±.012 | .906±.023 | .857±.028 | .961±.013 | .903±.025 | .947±.018 | **.958±.015** | 1.000±.000 |
| | .90 | .917±.024 | .925±.019 | .848±.031 | .840±.028 | .789±.035 | .923±.024 | .889±.023 | **.903±.024** | .850±.033 | .854±.032 | .879±.024 | .835±.028 | .795±.035 | .879±.025 | .840±.031 | .918±.020 | .905±.026 | .942±.021 |
| | .85 | .890±.029 | .906±.020 | .842±.031 | .806±.034 | .872±.022 | .824±.034 | .787±.029 | **.854±.028** | .766±.036 | .879±.023 | .816±.032 | .829±.029 | .747±.038 | .818±.030 | .788±.036 | .861±.029 | .898±.026 | .942±.021 |
| | .80 | .786±.034 | .876±.024 | .805±.033 | .765±.035 | .745±.034 | .749±.042 | **.787±.029** | .821±.031 | .739±.038 | .815±.032 | .792±.033 | .755±.036 | .698±.039 | .790±.033 | .714±.039 | .828±.031 | .839±.031 | .760±.040 |
| | .75 | .779±.035 | .826±.030 | .788±.035 | .652±.041 | **.748±.032** | .711±.044 | .767±.031 | .791±.032 | .613±.040 | .746±.036 | .633±.038 | .654±.042 | .672±.040 | .760±.034 | .696±.039 | .764±.035 | .769±.036 | .717±.043 |
| | .70 | .727±.038 | .727±.034 | .761±.037 | .620±.043 | .706±.039 | .673±.046 | .734±.033 | .784±.033 | .644±.043 | .536±.046 | .610±.043 | .659±.041 | **.716±.037** | .669±.039 | .745±.036 | .716±.036 | .739±.036 | .686±.044 |
| *MinCoeff* | .99 | 1.003±.047 | 1.000±.047 | .997±.048 | 1.006±.047 | **1.001±.046** | .995±.048 | 1.010±.048 | .997±.049 | 1.000±.048 | 1.013±.046 | 1.010±.046 | .998±.048 | 1.009±.047 | 1.022±.048 | .992±.049 | 1.001±.047 | 1.008±.046 | 1.003±.047 |
| | .95 | .983±.048 | .986±.049 | 1.013±.048 | 1.030±.048 | 1.002±.047 | .990±.048 | 1.005±.048 | .985±.051 | .991±.050 | .986±.049 | 1.002±.047 | 1.005±.048 | 1.058±.050 | 1.034±.048 | 1.001±.051 | 1.011±.046 | **1.004±.045** | 1.003±.047 |
| | .90 | .969±.049 | 1.010±.049 | 1.009±.049 | 1.075±.046 | 1.015±.057 | .980±.049 | .995±.051 | .973±.052 | 1.008±.048 | 1.002±.051 | 1.027±.047 | 1.017±.049 | 1.045±.052 | 1.073±.048 | 1.001±.051 | 1.017±.046 | **1.012±.045** | .988±.048 |
| | .85 | .967±.051 | **1.002±.050** | 1.016±.048 | 1.086±.048 | 1.029±.046 | .976±.049 | 1.023±.053 | .970±.052 | .996±.054 | 1.025±.048 | 1.032±.050 | 1.014±.049 | 1.066±.052 | 1.117±.052 | 1.001±.052 | 1.053±.047 | **1.019±.046** | .980±.049 |
| | .80 | .947±.053 | .990±.050 | **1.000±.050** | 1.044±.050 | 1.032±.055 | .960±.050 | 1.023±.053 | 1.041±.051 | .984±.053 | 1.043±.049 | 1.045±.050 | 1.088±.051 | 1.126±.054 | 1.030±.051 | 1.051±.047 | 1.060±.047 | 1.060±.047 | .954±.049 |
| | .75 | .943±.053 | .977±.051 | **1.011±.049** | 1.020±.053 | 1.008±.056 | .949±.051 | 1.014±.054 | .982±.055 | 1.012±.057 | .994±.051 | 1.097±.052 | 1.037±.056 | 1.078±.053 | 1.131±.052 | 1.032±.053 | 1.063±.048 | 1.065±.048 | .942±.051 |
| | .70 | .926±.056 | .999±.056 | **.999±.050** | 1.058±.052 | 1.010±.057 | .975±.051 | 1.005±.054 | .991±.055 | 1.002±.056 | .963±.060 | 1.110±.053 | 1.084±.053 | 1.071±.054 | 1.149±.053 | 1.019±.054 | 1.066±.047 | 1.076±.047 | .957±.051 |

Table B6: Results for `catsdogs`: $mean \pm std$ for $\widehat{Err}$, empirical coverage $\hat{\phi}$, and *MinCoeff*.

| Metric | c | DG | SAT | SAT+EM | SelNet | SelNet+EM | SR | SAT+SR | SAT+EM+SR | SelNet+SR | SelNet+EM+SR | ENS | ENS+SR | ConfidNet | SELE | REG | SCross | AUCross | PlugInAUC |
|---|---|---|---|---|---|---|---|---|---|---|---|---|---|---|---|---|---|---|---|
| $\widehat{Err}$ | .99 | .040±.003 | .040±.003 | .042±.003 | .066±.003 | .060±.003 | .068±.003 | .040±.003 | .042±.003 | .067±.003 | .060±.003 | .042±.003 | .045±.003 | .052±.003 | .060±.003 | .063±.004 | .082±.004 | .079±.004 | .070±.004 |
| | .95 | .026±.002 | .027±.002 | .030±.003 | .051±.003 | .047±.003 | .052±.003 | .027±.002 | .030±.002 | .050±.003 | .046±.003 | .032±.003 | .030±.002 | .044±.003 | .059±.003 | .062±.004 | .071±.004 | .056±.003 | .059±.003 |
| | .90 | .014±.002 | .017±.002 | .019±.002 | .023±.002 | .079±.004 | .039±.003 | .018±.002 | .021±.002 | .023±.002 | .083±.004 | .023±.002 | .019±.002 | .034±.003 | .060±.003 | .062±.004 | .057±.004 | .050±.003 | .049±.003 |
| | .85 | .008±.002 | .010±.001 | .013±.002 | .023±.003 | .049±.004 | .027±.003 | .010±.002 | .014±.002 | .024±.003 | .047±.004 | .015±.002 | .011±.002 | .026±.002 | .060±.003 | .061±.004 | .049±.003 | .042±.003 | .035±.003 |
| | .80 | .004±.001 | .008±.001 | .009±.002 | .029±.003 | .019±.002 | .017±.002 | .007±.001 | .008±.001 | .029±.003 | .019±.002 | .009±.001 | .008±.001 | .018±.002 | .058±.003 | .062±.004 | .044±.003 | .037±.003 | .014±.002 |
| | .75 | .002±.001 | .005±.001 | .007±.001 | .008±.001 | .017±.002 | .010±.002 | .005±.001 | .007±.001 | .092±.005 | .016±.002 | .006±.001 | .005±.001 | .014±.002 | .058±.003 | .062±.004 | .039±.003 | .033±.003 | .009±.001 |
| | .70 | .002±.001 | .005±.001 | .007±.001 | .008±.001 | .011±.002 | .005±.001 | .005±.001 | .006±.001 | .008±.001 | .011±.002 | .004±.001 | .004±.001 | .011±.002 | .058±.004 | .061±.004 | .035±.003 | .032±.003 | .007±.001 |
| $\hat{\phi}$ | .99 | .991±.001 | .986±.001 | .988±.001 | .989±.001 | .994±.001 | .992±.001 | .985±.002 | .988±.001 | .989±.001 | .993±.001 | .988±.002 | .992±.001 | .992±.001 | .990±.001 | .991±.001 | .995±.001 | .995±.001 | .989±.001 |
| | .95 | .956±.003 | .946±.003 | .952±.003 | .947±.003 | .955±.003 | .952±.003 | .946±.003 | .951±.003 | .945±.003 | .957±.003 | .944±.003 | .951±.003 | .954±.003 | .950±.003 | .955±.003 | .969±.003 | .966±.003 | .942±.003 |
| | .90 | .899±.005 | .897±.004 | .904±.004 | .891±.005 | .901±.004 | .912±.004 | .899±.004 | .908±.004 | .890±.005 | .901±.004 | .899±.004 | .898±.004 | .907±.004 | .904±.004 | .901±.005 | .936±.003 | .943±.004 | .893±.005 |
| | .85 | .848±.006 | .850±.004 | .852±.005 | .847±.004 | .853±.004 | .861±.005 | .851±.005 | .854±.005 | .850±.005 | .851±.004 | .856±.005 | .852±.005 | .850±.005 | .854±.005 | .844±.005 | .910±.005 | .917±.004 | .849±.005 |
| | .80 | .800±.006 | .797±.005 | .802±.005 | .788±.005 | .796±.006 | .804±.006 | .791±.005 | .806±.005 | .791±.005 | .796±.006 | .811±.006 | .813±.006 | .785±.006 | .801±.006 | .788±.006 | .880±.006 | .893±.005 | .800±.005 |
| | .75 | .744±.006 | .742±.006 | .752±.005 | .736±.006 | .735±.005 | .750±.006 | .735±.006 | .752±.005 | .739±.006 | .732±.005 | .763±.006 | .761±.006 | .741±.006 | .748±.007 | .733±.007 | .855±.006 | .867±.006 | .757±.006 |
| | .70 | .693±.006 | .685±.006 | .709±.006 | .712±.007 | .684±.006 | .691±.007 | .688±.006 | .702±.006 | .708±.007 | .682±.006 | .712±.007 | .711±.007 | .690±.006 | .699±.007 | .690±.007 | .820±.006 | .848±.006 | .713±.007 |
| *MinCoeff* | .99 | 1.004±.017 | .999±.017 | 1.006±.017 | .998±.017 | 1.002±.017 | .997±.017 | .999±.017 | 1.007±.017 | .999±.017 | 1.001±.017 | .994±.017 | .997±.017 | 1.005±.017 | 1.007±.017 | 1.002±.017 | .999±.017 | .999±.017 | 1.010±.017 |
| | .95 | 1.001±.017 | .989±.017 | 1.012±.017 | .982±.017 | .997±.017 | .971±.017 | .989±.017 | 1.012±.017 | .980±.017 | 1.006±.017 | .972±.017 | .968±.017 | 1.016±.017 | 1.021±.017 | 1.000±.017 | .977±.017 | .978±.017 | 1.031±.018 |
| | .90 | .969±.017 | .964±.017 | 1.020±.017 | 1.002±.017 | .929±.017 | .945±.017 | .963±.017 | 1.022±.017 | 1.004±.017 | .925±.017 | .941±.017 | .932±.018 | 1.029±.017 | 1.027±.017 | .998±.017 | .949±.017 | .959±.017 | 1.061±.017 |
| | .85 | .977±.017 | .943±.017 | 1.019±.017 | .969±.018 | .890±.018 | .913±.017 | .943±.017 | 1.026±.018 | .963±.017 | .888±.018 | .914±.018 | .901±.018 | 1.059±.017 | 1.031±.018 | .998±.018 | .926±.017 | .940±.017 | 1.082±.018 |
| | .80 | .964±.017 | .917±.018 | 1.020±.018 | .932±.017 | .913±.018 | .879±.017 | .906±.018 | 1.034±.018 | .929±.017 | .914±.018 | .886±.018 | .878±.018 | 1.103±.018 | 1.029±.018 | 1.004±.019 | .895±.017 | .920±.017 | 1.098±.018 |
| | .75 | .944±.017 | .910±.018 | 1.013±.018 | .906±.018 | .903±.019 | .864±.018 | .891±.018 | 1.035±.018 | .902±.018 | .915±.019 | .865±.018 | .851±.018 | 1.134±.018 | 1.034±.018 | 1.009±.019 | .873±.017 | .895±.017 | 1.109±.018 |
| | .70 | .929±.018 | .933±.019 | 1.008±.018 | .919±.019 | .907±.019 | .875±.019 | .902±.018 | 1.038±.019 | .922±.019 | .905±.019 | .836±.018 | .816±.018 | 1.165±.019 | 1.039±.019 | 1.014±.019 | .832±.017 | .875±.017 | 1.123±.018 |

Table B7: Results for `chestmnist`: $mean \pm std$ for $\widehat{Err}$, empirical coverage $\hat{\phi}$, and *MinCoeff*.

| Metric | c | DG | SAT | SAT+EM | SelNet | SelNet+EM | SR | SAT+SR | SAT+EM+SR | SelNet+SR | SelNet+EM+SR | ENS | ENS+SR | ConfidNet | SELE | REG | SCross | AUCross | PlugInAUC |
|---|---|---|---|---|---|---|---|---|---|---|---|---|---|---|---|---|---|---|---|
| $\widehat{Err}$ | .99 | .101±.002 | .100±.002 | .100±.002 | .099±.002 | .099±.002 | .099±.002 | .100±.002 | .100±.002 | .100±.002 | .100±.002 | .102±.002 | .099±.002 | .101±.002 | .101±.002 | .103±.002 | .100±.002 | .104±.002 | .103±.002 |
| | .95 | .093±.002 | .090±.002 | .089±.002 | .091±.002 | .090±.002 | .091±.002 | .090±.002 | .089±.002 | .090±.002 | .091±.002 | .099±.002 | .090±.002 | .091±.002 | .092±.002 | .102±.002 | .090±.002 | .104±.002 | .103±.002 |
| | .90 | .086±.002 | .080±.002 | .081±.002 | .080±.002 | .082±.002 | .081±.002 | .081±.002 | .081±.002 | .081±.002 | .084±.002 | .096±.002 | .081±.002 | .082±.002 | .084±.002 | .100±.002 | .081±.002 | .103±.002 | .104±.002 |
| | .85 | .081±.002 | .074±.002 | .073±.002 | .074±.002 | .073±.002 | .073±.002 | .074±.002 | .073±.002 | .074±.002 | .097±.002 | .090±.002 | .073±.002 | .075±.002 | .078±.002 | .099±.002 | .073±.002 | .103±.002 | .103±.002 |
| | .80 | .076±.002 | .066±.002 | .066±.002 | .066±.002 | .067±.002 | .067±.002 | .066±.002 | .066±.002 | .066±.002 | .129±.002 | .086±.002 | .066±.002 | .069±.002 | .071±.002 | .097±.002 | .066±.002 | .101±.002 | .103±.002 |
| | .75 | .072±.002 | .059±.002 | .060±.002 | .061±.002 | .061±.002 | .059±.002 | .059±.002 | .060±.002 | .060±.002 | .137±.003 | .083±.002 | .059±.002 | .061±.002 | .065±.002 | .096±.002 | .061±.002 | .100±.002 | .103±.002 |
| | .70 | .068±.002 | .055±.002 | .054±.002 | .055±.002 | .057±.002 | .055±.002 | .055±.002 | .055±.002 | .055±.002 | .204±.003 | .079±.002 | .054±.002 | .058±.002 | .059±.002 | .095±.002 | .055±.002 | .100±.003 | .100±.003 |
| $\hat{\phi}$ | .99 | .991±.001 | .989±.001 | .989±.001 | .989±.001 | .989±.001 | .989±.001 | .988±.001 | .989±.001 | .990±.001 | .990±.001 | .987±.001 | .989±.001 | .990±.001 | .990±.001 | .992±.001 | .991±.001 | .990±.001 | .990±.001 |
| | .95 | .953±.001 | .952±.001 | .949±.001 | .952±.001 | .949±.001 | .950±.001 | .952±.001 | .949±.001 | .952±.001 | .950±.002 | .945±.002 | .950±.001 | .950±.001 | .950±.002 | .953±.001 | .954±.001 | .950±.001 | .949±.002 |
| | .90 | .899±.002 | .899±.002 | .903±.002 | .901±.002 | .899±.002 | .898±.002 | .899±.002 | .902±.002 | .901±.002 | .899±.002 | .900±.002 | .902±.002 | .899±.002 | .901±.002 | .900±.002 | .906±.002 | .900±.002 | .897±.002 |
| | .85 | .846±.002 | .850±.002 | .850±.002 | .854±.002 | .849±.002 | .852±.002 | .850±.002 | .850±.002 | .853±.002 | .857±.002 | .847±.003 | .851±.002 | .850±.002 | .852±.002 | .853±.002 | .855±.002 | .853±.003 | .846±.003 |
| | .80 | .799±.003 | .798±.003 | .802±.002 | .799±.002 | .801±.003 | .802±.002 | .799±.002 | .802±.002 | .801±.003 | .801±.002 | .798±.003 | .800±.003 | .801±.002 | .805±.002 | .800±.002 | .805±.002 | .803±.003 | .798±.003 |
| | .75 | .748±.003 | .748±.003 | .754±.002 | .748±.003 | .751±.003 | .750±.003 | .749±.003 | .748±.003 | .749±.003 | .757±.003 | .748±.003 | .751±.003 | .752±.003 | .752±.003 | .748±.003 | .756±.003 | .751±.003 | .750±.003 |
| | .70 | .701±.003 | .701±.003 | .698±.003 | .700±.003 | .702±.002 | .702±.003 | .702±.003 | .701±.003 | .699±.003 | .699±.003 | .701±.003 | .698±.003 | .702±.003 | .698±.003 | .696±.003 | .707±.003 | .700±.003 | .699±.003 |
| *MinCoeff* | .99 | .978±.020 | .971±.021 | .969±.020 | .967±.020 | .965±.020 | .962±.021 | .973±.021 | .969±.020 | .970±.021 | .974±.020 | .993±.021 | .962±.021 | .979±.020 | .981±.021 | .999±.021 | .972±.021 | 1.003±.021 | 1.001±.021 |
| | .95 | .908±.019 | .875±.019 | .864±.018 | .878±.020 | .876±.019 | .879±.019 | .876±.019 | .864±.018 | .879±.019 | .880±.019 | .963±.022 | .871±.020 | .887±.020 | .892±.020 | .984±.021 | .879±.020 | 1.003±.022 | 1.001±.022 |
| | .90 | .837±.018 | .782±.018 | .785±.018 | .782±.018 | .801±.019 | .783±.019 | .782±.018 | .785±.018 | .785±.018 | .811±.019 | .928±.023 | .791±.018 | .795±.019 | .819±.019 | .961±.021 | .788±.018 | .997±.023 | 1.002±.022 |
| | .85 | .783±.018 | .716±.018 | .706±.017 | .720±.018 | .710±.017 | .711±.018 | .716±.018 | .707±.017 | .719±.018 | .803±.019 | .876±.023 | .705±.017 | .731±.018 | .753±.018 | .956±.021 | .711±.017 | .993±.023 | .997±.022 |
| | .80 | .741±.019 | .645±.018 | .642±.016 | .643±.016 | .655±.017 | .651±.017 | .647±.018 | .642±.016 | .644±.016 | .822±.021 | .839±.022 | .642±.016 | .666±.018 | .687±.018 | .936±.022 | .645±.018 | .980±.024 | .999±.023 |
| | .75 | .698±.019 | .578±.017 | .587±.017 | .589±.018 | .593±.018 | .577±.017 | .580±.016 | .586±.017 | .589±.018 | .822±.021 | .801±.023 | .577±.018 | .618±.018 | .632±.018 | .923±.022 | .597±.017 | .967±.023 | .994±.023 |
| | .70 | .665±.019 | .533±.016 | .527±.018 | .538±.017 | .550±.016 | .533±.017 | .534±.016 | .531±.018 | .537±.017 | .905±.023 | .770±.023 | .521±.017 | .567±.017 | .572±.018 | .911±.021 | .535±.017 | .967±.024 | .969±.024 |

Table B8: Results for `cifar10`: $mean \pm std$ for $\widehat{Err}$ and empirical coverage $\hat{\phi}$.

| Metric | c | DG | SAT | SAT+EM | SelNet | SelNet+EM | SR | SAT+SR | SAT+EM+SR | SelNet+SR | SelNet+EM+SR | ENS | ENS+SR | ConfidNet | SELE | REG | SCross |
|---|---|---|---|---|---|---|---|---|---|---|---|---|---|---|---|---|---|
| $\widehat{Err}$ | .99 | .093±.002 | .088±.003 | .099±.003 | .078±.002 | .064±.002 | .090±.003 | .088±.003 | .098±.003 | .079±.003 | .064±.002 | .059±.002 | .058±.002 | .091±.003 | .178±.002 | .119±.003 | .088±.002 |
| | .95 | .074±.003 | .069±.002 | .078±.002 | .073±.002 | .049±.002 | .071±.003 | .067±.002 | .075±.002 | .074±.002 | .046±.002 | .045±.002 | .039±.002 | .074±.003 | .115±.003 | .119±.003 | .069±.002 |
| | .90 | .055±.002 | .047±.002 | .056±.002 | .045±.002 | .031±.002 | .051±.002 | .047±.002 | .053±.002 | .045±.002 | .030±.002 | .031±.002 | .025±.002 | .053±.002 | .112±.003 | .119±.003 | .049±.002 |
| | .85 | .040±.002 | .033±.002 | .039±.002 | .037±.002 | .019±.001 | .037±.002 | .032±.002 | .037±.002 | .037±.002 | .019±.001 | .021±.001 | .014±.001 | .039±.002 | .108±.003 | .120±.003 | .032±.002 |
| | .80 | .029±.002 | .020±.001 | .026±.002 | .028±.002 | .011±.001 | .024±.002 | .022±.001 | .026±.002 | .026±.002 | .013±.001 | .013±.001 | .009±.001 | .025±.002 | .105±.003 | .120±.003 | .021±.001 |
| | .75 | .020±.002 | .015±.001 | .019±.001 | .019±.001 | .007±.001 | .015±.002 | .014±.001 | .017±.001 | .018±.001 | .008±.001 | .008±.001 | .006±.001 | .018±.001 | .101±.003 | .121±.003 | .015±.001 |
| | .70 | .012±.001 | .009±.001 | .014±.001 | .011±.001 | .006±.001 | .011±.001 | .010±.001 | .011±.001 | .011±.001 | .008±.001 | .005±.001 | .003±.001 | .011±.001 | .098±.003 | .121±.003 | .010±.001 |
| $\hat{\phi}$ | .99 | .992±.001 | .989±.001 | .989±.001 | .990±.001 | .990±.001 | .992±.001 | .990±.001 | .990±.001 | .991±.001 | .990±.001 | .987±.001 | .989±.001 | .992±.001 | .995±.001 | .990±.001 | .988±.001 |
| | .95 | .951±.002 | .950±.002 | .947±.002 | .945±.002 | .951±.002 | .952±.002 | .946±.002 | .942±.002 | .949±.002 | .947±.002 | .946±.002 | .948±.002 | .956±.002 | .954±.002 | .942±.002 | .948±.002 |
| | .90 | .895±.002 | .901±.003 | .893±.003 | .894±.003 | .900±.003 | .902±.002 | .901±.003 | .892±.003 | .896±.003 | .900±.003 | .896±.003 | .899±.003 | .902±.002 | .908±.003 | .894±.003 | .897±.003 |
| | .85 | .848±.003 | .854±.003 | .842±.003 | .852±.003 | .847±.003 | .853±.003 | .854±.003 | .855±.003 | .845±.003 | .846±.003 | .841±.004 | .844±.003 | .857±.003 | .848±.003 | .848±.003 | .844±.003 |
| | .80 | .799±.003 | .800±.003 | .790±.003 | .799±.003 | .801±.003 | .799±.003 | .805±.003 | .799±.003 | .796±.004 | .800±.004 | .791±.004 | .796±.004 | .803±.003 | .809±.003 | .804±.003 | .791±.004 |
| | .75 | .751±.003 | .751±.003 | .745±.003 | .756±.004 | .756±.004 | .752±.003 | .752±.003 | .746±.003 | .754±.004 | .756±.004 | .750±.004 | .751±.004 | .756±.004 | .758±.004 | .750±.004 | .742±.004 |
| | .70 | .694±.003 | .695±.004 | .701±.004 | .695±.004 | .710±.004 | .703±.004 | .697±.004 | .696±.004 | .696±.004 | .708±.004 | .701±.004 | .702±.004 | .704±.004 | .711±.004 | .698±.004 | .689±.004 |

Table B9: Results for `compass`: $mean \pm std$ for $\widehat{Err}$, empirical coverage $\hat{\phi}$, and *MinCoeff*.

| Metric | c | DG | SAT | SAT+EM | SelNet | SelNet+EM | SR | SAT+SR | SAT+EM+SR | SelNet+SR | SelNet+EM+SR | ENS | ENS+SR | ConfidNet | SELE | REG | SCross | AUCross | PlugInAUC |
|---|---|---|---|---|---|---|---|---|---|---|---|---|---|---|---|---|---|---|---|
| $\widehat{Err}$ | .99 | .310±.008 | .298±.008 | .301±.008 | .293±.007 | .287±.008 | .292±.007 | .297±.008 | .302±.008 | .293±.007 | .288±.008 | .289±.007 | .285±.007 | .297±.007 | .296±.008 | .307±.008 | .300±.008 | .300±.008 | .292±.007 |
| | .95 | .303±.009 | .293±.008 | .296±.009 | .290±.008 | .290±.007 | .284±.007 | .293±.008 | .295±.008 | .284±.007 | .287±.007 | .289±.007 | .275±.008 | .282±.008 | .292±.008 | .302±.008 | .291±.008 | .285±.008 | .284±.008 |
| | .90 | .294±.009 | .287±.008 | .288±.009 | .276±.008 | .280±.008 | .272±.007 | .282±.008 | .286±.008 | .272±.007 | .282±.008 | .282±.008 | .264±.008 | .276±.008 | .283±.008 | .300±.008 | .273±.008 | .269±.008 | .272±.007 |
| | .85 | .290±.009 | .275±.009 | .279±.009 | .273±.008 | .267±.008 | .262±.007 | .276±.008 | .276±.008 | .266±.008 | .272±.008 | .282±.008 | .257±.008 | .266±.008 | .276±.009 | .298±.008 | .260±.008 | .255±.008 | .265±.007 |
| | .80 | .282±.009 | .266±.008 | .274±.009 | .262±.008 | .257±.009 | .252±.006 | .262±.008 | .271±.008 | .255±.008 | .257±.009 | .279±.008 | .245±.008 | .256±.008 | .274±.009 | .295±.009 | .247±.008 | .245±.009 | .255±.009 |
| | .75 | .280±.009 | .255±.008 | .262±.009 | .250±.009 | .246±.008 | .244±.008 | .250±.009 | .262±.009 | .243±.008 | .247±.009 | .273±.008 | .234±.008 | .248±.008 | .269±.009 | .290±.009 | .237±.009 | .236±.009 | .248±.008 |
| | .70 | .274±.009 | .243±.009 | .253±.009 | .239±.008 | .237±.009 | .231±.008 | .234±.009 | .252±.009 | .229±.009 | .234±.009 | .271±.009 | .221±.008 | .235±.009 | .257±.010 | .283±.009 | .231±.009 | .229±.009 | .243±.008 |
| $\hat{\phi}$ | .99 | .990±.002 | .988±.002 | .987±.002 | .986±.002 | .988±.002 | .995±.001 | .991±.002 | .990±.002 | .987±.002 | .990±.002 | .989±.002 | .989±.002 | .991±.002 | .985±.002 | .993±.001 | .990±.002 | .986±.002 | .989±.002 |
| | .95 | .954±.004 | .962±.003 | .945±.004 | .945±.004 | .955±.004 | .953±.003 | .943±.003 | .951±.004 | .941±.004 | .948±.004 | .947±.004 | .951±.004 | .932±.005 | .946±.004 | .957±.004 | .949±.004 | .937±.004 | .937±.004 |
| | .90 | .902±.005 | .911±.005 | .902±.005 | .891±.005 | .894±.005 | .907±.005 | .896±.005 | .896±.006 | .893±.006 | .902±.006 | .901±.005 | .901±.005 | .888±.005 | .888±.006 | .907±.005 | .875±.005 | .841±.007 | .883±.006 |
| | .85 | .857±.005 | .846±.007 | .855±.006 | .841±.007 | .840±.006 | .845±.006 | .836±.006 | .851±.007 | .834±.006 | .839±.007 | .855±.006 | .847±.006 | .835±.007 | .854±.006 | .854±.007 | .743±.007 | .778±.008 | .837±.006 |
| | .80 | .804±.006 | .788±.007 | .814±.007 | .798±.007 | .805±.008 | .802±.007 | .794±.007 | .815±.007 | .794±.007 | .801±.007 | .809±.008 | .800±.007 | .784±.008 | .802±.007 | .801±.008 | .743±.007 | .725±.008 | .786±.006 |
| | .75 | .750±.007 | .742±.007 | .760±.008 | .748±.007 | .738±.007 | .746±.007 | .741±.007 | .772±.007 | .741±.008 | .736±.007 | .760±.008 | .743±.008 | .730±.008 | .759±.006 | .758±.008 | .689±.008 | .664±.008 | .739±.006 |
| | .70 | .699±.008 | .695±.008 | .719±.008 | .699±.008 | .698±.008 | .692±.008 | .695±.007 | .689±.008 | .718±.008 | .709±.008 | .681±.008 | .715±.008 | .697±.008 | .674±.008 | .715±.007 | .698±.008 | .629±.008 | .690±.007 |
| *MinCoeff* | .99 | 1.004±.019 | 1.000±.019 | .998±.019 | 1.000±.018 | 1.000±.019 | 1.001±.019 | 1.002±.019 | 1.002±.019 | 1.002±.019 | 1.000±.019 | 1.000±.019 | 1.008±.019 | 1.002±.018 | .999±.018 | 1.002±.019 | 1.001±.019 | 1.004±.019 | 1.012±.019 |
| | .95 | 1.003±.019 | 1.004±.019 | .985±.020 | 1.004±.019 | .999±.019 | 1.004±.018 | 1.004±.019 | .993±.019 | 1.002±.019 | .996±.019 | 1.006±.019 | 1.008±.019 | .997±.019 | .991±.019 | .997±.019 | .999±.020 | 1.002±.019 | 1.012±.019 |
| | .90 | 1.000±.019 | 1.006±.019 | .981±.020 | 1.009±.018 | .992±.019 | 1.000±.019 | 1.007±.020 | .982±.020 | .989±.019 | .998±.020 | .999±.019 | .999±.019 | .994±.019 | .967±.020 | .985±.019 | .966±.021 | 1.011±.020 | 1.015±.020 |
| | .85 | .997±.020 | 1.001±.020 | .971±.021 | 1.017±.020 | 1.005±.020 | .998±.019 | 1.015±.020 | .973±.020 | .982±.020 | 1.003±.020 | .999±.020 | .994±.020 | 1.000±.019 | .980±.020 | .969±.020 | .984±.021 | 1.021±.021 | 1.028±.020 |
| | .80 | .991±.020 | 1.010±.020 | .968±.020 | 1.019±.020 | .990±.021 | 1.002±.020 | 1.002±.020 | .965±.020 | .988±.020 | .990±.020 | .995±.020 | .993±.021 | 1.012±.020 | .978±.020 | .951±.020 | .969±.021 | 1.034±.021 | 1.036±.020 |
| | .75 | .999±.020 | 1.011±.021 | .961±.021 | 1.010±.020 | .996±.021 | .996±.021 | 1.006±.021 | .965±.021 | .988±.020 | 1.000±.021 | .994±.021 | .998±.021 | 1.003±.021 | .971±.021 | .947±.021 | .966±.022 | 1.048±.022 | 1.055±.020 |
| | .70 | 1.005±.021 | 1.015±.022 | .959±.021 | 1.010±.021 | .990±.022 | 1.003±.022 | 1.012±.022 | .956±.021 | .996±.021 | .984±.021 | .983±.022 | 1.000±.022 | .994±.023 | .963±.022 | .931±.021 | .960±.023 | 1.063±.023 | 1.071±.020 |

Table B10: Results for `covtype`: $mean \pm std$ for $\widehat{Err}$ and empirical coverage $\hat{\phi}$.

| Metric | c | DG | SAT | SAT+EM | SelNet | SelNet+EM | SR | SAT+SR | SAT+EM+SR | SelNet+SR | SelNet+EM+SR | ENS | ENS+SR | ConfidNet | SELE | REG | SCross |
|---|---|---|---|---|---|---|---|---|---|---|---|---|---|---|---|---|---|
| $\widehat{Err}$ | .99 | .036±.001 | .028±.000 | .029±.000 | .027±.000 | .031±.000 | .041±.001 | .027±.000 | .028±.000 | .025±.000 | .029±.000 | .029±.000 | .027±.000 | .035±.000 | .070±.001 | .056±.001 | .037±.000 |
| | .95 | .029±.001 | .016±.000 | .017±.000 | .015±.000 | .020±.000 | .027±.000 | .013±.000 | .014±.000 | .012±.000 | .017±.000 | .020±.000 | .015±.000 | .025±.000 | .069±.001 | .057±.001 | .024±.000 |
| | .90 | .023±.000 | .007±.000 | .009±.000 | .009±.000 | .011±.000 | .016±.000 | .006±.000 | .006±.000 | .005±.000 | .008±.000 | .011±.000 | .007±.000 | .018±.000 | .067±.001 | .058±.001 | .015±.000 |
| | .85 | .018±.000 | .004±.000 | .005±.000 | .005±.000 | .007±.000 | .010±.000 | .003±.000 | .003±.000 | .002±.000 | .005±.000 | .006±.000 | .004±.000 | .014±.000 | .065±.001 | .059±.001 | .009±.000 |
| | .80 | .014±.000 | .002±.000 | .003±.000 | .002±.000 | .004±.000 | .007±.000 | .002±.000 | .002±.000 | .001±.000 | .003±.000 | .003±.000 | .002±.000 | .011±.000 | .064±.001 | .059±.001 | .006±.000 |
| | .75 | .011±.000 | .002±.000 | .002±.000 | .001±.000 | .002±.000 | .005±.000 | .001±.000 | .001±.000 | .001±.000 | .001±.000 | .002±.000 | .001±.000 | .009±.000 | .062±.001 | .059±.001 | .004±.000 |
| | .70 | .009±.000 | .001±.000 | .001±.000 | .001±.000 | .001±.000 | .003±.000 | .001±.000 | .001±.000 | .001±.000 | .001±.000 | .030±.001 | .001±.000 | .001±.000 | .060±.001 | .059±.001 | .002±.000 |
| $\hat{\phi}$ | .99 | .989±.000 | .989±.000 | .990±.000 | .990±.000 | .990±.000 | .989±.000 | .991±.000 | .990±.000 | .990±.000 | .990±.000 | .991±.000 | .990±.000 | .990±.000 | .989±.000 | .989±.000 | .992±.000 |
| | .95 | .950±.001 | .948±.001 | .950±.001 | .950±.001 | .950±.001 | .949±.001 | .948±.001 | .951±.001 | .950±.001 | .951±.001 | .950±.001 | .950±.001 | .950±.001 | .948±.001 | .949±.001 | .955±.001 |
| | .90 | .901±.001 | .898±.001 | .899±.001 | .900±.001 | .898±.001 | .899±.001 | .899±.001 | .899±.001 | .899±.001 | .899±.001 | .898±.001 | .900±.001 | .901±.001 | .898±.001 | .899±.001 | .900±.001 |
| | .85 | .850±.001 | .849±.001 | .849±.001 | .851±.001 | .849±.001 | .849±.001 | .849±.001 | .849±.001 | .850±.001 | .849±.001 | .851±.001 | .850±.001 | .849±.001 | .847±.001 | .848±.001 | .866±.001 |
| | .80 | .797±.001 | .799±.001 | .799±.001 | .799±.001 | .798±.001 | .801±.001 | .799±.001 | .798±.001 | .800±.001 | .797±.001 | .801±.001 | .801±.001 | .798±.001 | .796±.001 | .800±.001 | .820±.001 |
| | .75 | .748±.001 | .749±.001 | .749±.001 | .749±.001 | .748±.001 | .750±.001 | .749±.001 | .750±.001 | .750±.001 | .748±.001 | .751±.001 | .750±.001 | .750±.001 | .746±.001 | .752±.002 | .775±.001 |
| | .70 | .698±.001 | .699±.001 | .700±.001 | .700±.002 | .697±.001 | .699±.001 | .700±.001 | .701±.001 | .699±.001 | .700±.001 | .701±.001 | .700±.001 | .700±.001 | .698±.001 | .699±.002 | .728±.001 |

Table B11: Results for `dermamnist`: $mean \pm std$ for $\widehat{Err}$ and empirical coverage $\hat{\phi}$.

| Metric | c | DG | SAT | SAT+EM | SelNet | SelNet+EM | SR | SAT+SR | SAT+EM+SR | SelNet+SR | SelNet+EM+SR | ENS | ENS+SR | ConfidNet | SELE | REG | SCross |
|---|---|---|---|---|---|---|---|---|---|---|---|---|---|---|---|---|---|
| $\widehat{Err}$ | .99 | .269±.010 | .229±.009 | .239±.009 | .243±.010 | .241±.010 | .228±.009 | .232±.009 | .236±.009 | .239±.011 | .240±.010 | .226±.009 | .223±.009 | .245±.010 | .270±.009 | .273±.009 | .231±.009 |
| | .95 | .250±.010 | .217±.009 | .230±.009 | .227±.010 | .227±.010 | .207±.009 | .214±.009 | .222±.009 | .224±.011 | .221±.009 | .213±.008 | .206±.009 | .238±.010 | .263±.010 | .267±.010 | .214±.008 |
| | .90 | .213±.010 | .202±.009 | .212±.010 | .207±.010 | .210±.010 | .191±.009 | .194±.009 | .196±.009 | .195±.011 | .206±.010 | .194±.009 | .182±.009 | .218±.010 | .259±.010 | .263±.011 | .193±.008 |
| | .85 | .194±.010 | .180±.009 | .201±.009 | .198±.010 | .198±.010 | .177±.009 | .173±.009 | .174±.009 | .182±.010 | .180±.010 | .175±.009 | .161±.009 | .200±.010 | .252±.010 | .263±.010 | .166±.009 |
| | .80 | .180±.010 | .159±.009 | .179±.009 | .192±.010 | .183±.011 | .152±.009 | .151±.009 | .162±.009 | .202±.010 | .199±.010 | .159±.010 | .147±.008 | .179±.010 | .242±.011 | .261±.011 | .146±.008 |
| | .75 | .163±.010 | .144±.009 | .162±.010 | .164±.011 | .165±.010 | .129±.009 | .129±.009 | .140±.009 | .182±.010 | .185±.010 | .146±.010 | .123±.008 | .161±.011 | .224±.011 | .258±.012 | .126±.008 |
| | .70 | .143±.011 | .120±.009 | .143±.009 | .146±.010 | .148±.010 | .114±.008 | .110±.009 | .119±.009 | .172±.010 | .165±.010 | .130±.010 | .108±.008 | .147±.010 | .203±.010 | .254±.011 | .109±.009 |
| $\hat{\phi}$ | .99 | .994±.002 | .983±.003 | .993±.002 | .993±.002 | .992±.002 | .991±.002 | .989±.002 | .989±.002 | .990±.002 | .992±.002 | .994±.002 | .994±.002 | .991±.002 | .989±.002 | .993±.002 | .989±.002 |
| | .95 | .962±.004 | .946±.004 | .965±.004 | .955±.005 | .964±.004 | .947±.005 | .949±.005 | .950±.005 | .954±.004 | .959±.004 | .956±.005 | .960±.004 | .967±.004 | .933±.007 | .957±.005 | .952±.004 |
| | .90 | .898±.008 | .900±.006 | .906±.007 | .912±.006 | .920±.006 | .905±.006 | .896±.007 | .892±.008 | .898±.007 | .907±.007 | .896±.006 | .910±.006 | .913±.007 | .892±.008 | .905±.006 | .905±.006 |
| | .85 | .862±.008 | .849±.008 | .868±.007 | .866±.008 | .882±.007 | .865±.007 | .851±.008 | .845±.008 | .852±.008 | .857±.007 | .837±.008 | .852±.008 | .871±.008 | .850±.008 | .870±.008 | .849±.007 |
| | .80 | .820±.009 | .785±.009 | .818±.008 | .820±.009 | .823±.009 | .797±.009 | .796±.009 | .816±.009 | .800±.010 | .807±.010 | .789±.008 | .809±.008 | .818±.009 | .803±.008 | .818±.009 | .800±.008 |
| | .75 | .760±.009 | .746±.010 | .757±.008 | .757±.010 | .764±.009 | .739±.010 | .749±.009 | .760±.011 | .760±.011 | .763±.010 | .730±.009 | .750±.009 | .766±.010 | .748±.009 | .777±.009 | .749±.009 |
| | .70 | .709±.010 | .687±.010 | .705±.010 | .711±.011 | .705±.010 | .702±.010 | .692±.010 | .698±.011 | .720±.010 | .703±.010 | .684±.009 | .702±.010 | .711±.010 | .694±.010 | .736±.010 | .681±.010 |

Table B12: Results for `electricity`: $mean \pm std$ for $\widehat{Err}$, empirical coverage $\hat{\phi}$, and $MinCoeff$.

| Metric | c | DG | SAT | SAT+EM | SelNet | SelNet+EM | SR | SAT+SR | SAT+EM+SR | SelNet+SR | SelNet+EM+SR | ENS | ENS+SR | ConfidNet | SELE | REG | SCross | AUCross | PlugInAUC |
|---|---|---|---|---|---|---|---|---|---|---|---|---|---|---|---|---|---|---|---|
| $\widehat{Err}$ | .99 | .248±.004 | .176±.005 | .163±.005 | .179±.004 | .187±.005 | .174±.004 | .176±.005 | .162±.005 | .179±.004 | .185±.005 | .162±.005 | .161±.005 | .167±.004 | .206±.005 | .200±.004 | .169±.005 | .170±.005 | .174±.004 |
| | .95 | .245±.004 | .164±.004 | .152±.004 | .172±.005 | .174±.004 | .162±.004 | .162±.004 | .149±.005 | .166±.004 | .170±.004 | .159±.005 | .150±.004 | .155±.004 | .196±.004 | .158±.004 | .161±.005 | .161±.005 | .166±.005 |
| | .90 | .238±.005 | .151±.004 | .138±.004 | .161±.004 | .162±.005 | .147±.004 | .145±.004 | .135±.004 | .155±.005 | .160±.004 | .155±.005 | .134±.004 | .139±.004 | .190±.005 | .193±.004 | .144±.004 | .148±.005 | .166±.004 |
| | .85 | .227±.005 | .135±.004 | .125±.004 | .149±.005 | .143±.005 | .135±.004 | .133±.004 | .120±.004 | .139±.005 | .139±.004 | .148±.005 | .120±.004 | .129±.004 | .182±.005 | .190±.005 | .130±.004 | .136±.004 | .143±.004 |
| | .80 | .217±.005 | .121±.004 | .112±.004 | .136±.004 | .138±.005 | .119±.004 | .117±.004 | .107±.004 | .125±.004 | .138±.005 | .140±.005 | .107±.004 | .117±.004 | .176±.005 | .187±.005 | .117±.004 | .121±.004 | .130±.005 |
| | .75 | .203±.005 | .110±.004 | .102±.004 | .125±.005 | .125±.005 | .107±.004 | .101±.004 | .096±.004 | .113±.004 | .125±.005 | .131±.004 | .093±.004 | .109±.004 | .169±.005 | .184±.005 | .104±.004 | .106±.004 | .118±.004 |
| | .70 | .186±.005 | .096±.004 | .088±.004 | .124±.004 | .116±.005 | .097±.004 | .089±.004 | .084±.004 | .112±.005 | .117±.005 | .127±.005 | .085±.004 | .098±.004 | .160±.005 | .178±.005 | .095±.004 | .092±.004 | .105±.004 |
| $\hat{\phi}$ | .99 | .990±.001 | .991±.001 | .993±.001 | .992±.001 | .995±.001 | .990±.001 | .989±.001 | .990±.001 | .990±.001 | .992±.001 | .990±.001 | .992±.001 | .988±.001 | .990±.001 | .989±.001 | .992±.001 | .990±.001 | .989±.001 |
| | .95 | .949±.003 | .949±.003 | .951±.002 | .961±.002 | .958±.002 | .950±.003 | .949±.003 | .954±.002 | .951±.002 | .953±.003 | .955±.002 | .951±.003 | .950±.003 | .953±.002 | .948±.003 | .954±.002 | .949±.002 | .951±.002 |
| | .90 | .898±.004 | .901±.004 | .902±.004 | .911±.003 | .907±.004 | .899±.003 | .896±.004 | .905±.004 | .908±.003 | .902±.004 | .909±.003 | .900±.004 | .901±.004 | .908±.004 | .900±.003 | .909±.004 | .900±.003 | .903±.004 |
| | .85 | .849±.004 | .850±.004 | .859±.004 | .859±.004 | .852±.004 | .856±.004 | .851±.004 | .852±.004 | .854±.004 | .846±.004 | .860±.004 | .852±.004 | .853±.004 | .853±.005 | .844±.004 | .863±.004 | .856±.004 | .856±.005 |
| | .80 | .801±.004 | .803±.005 | .817±.005 | .817±.004 | .807±.005 | .807±.005 | .798±.005 | .805±.005 | .806±.004 | .801±.005 | .807±.004 | .803±.005 | .802±.005 | .796±.005 | .795±.005 | .813±.005 | .808±.005 | .802±.005 |
| | .75 | .752±.005 | .763±.005 | .769±.005 | .765±.005 | .754±.005 | .760±.005 | .751±.005 | .761±.005 | .749±.005 | .750±.005 | .756±.004 | .752±.005 | .757±.005 | .749±.005 | .750±.005 | .767±.005 | .761±.006 | .758±.004 |
| | .70 | .703±.005 | .716±.005 | .719±.006 | .710±.005 | .721±.005 | .717±.005 | .706±.005 | .722±.006 | .712±.005 | .711±.005 | .712±.006 | .712±.006 | .715±.005 | .700±.005 | .692±.005 | .719±.006 | .709±.006 | .712±.005 |
| $MinCoeff$ | .99 | 1.003±.011 | 1.000±.011 | 1.001±.011 | 1.000±.011 | 1.000±.011 | .998±.011 | 1.002±.011 | 1.002±.011 | .998±.011 | 1.001±.011 | 1.003±.011 | 1.001±.011 | .998±.011 | 1.000±.011 | 1.004±.011 | .999±.011 | 1.002±.011 | 1.002±.011 |
| | .95 | 1.021±.011 | 1.003±.011 | 1.004±.011 | .999±.011 | 1.002±.011 | 1.000±.011 | 1.010±.011 | 1.005±.011 | .997±.011 | 1.004±.011 | 1.008±.011 | 1.002±.011 | .999±.011 | 1.001±.011 | 1.006±.011 | .997±.011 | 1.013±.011 | 1.012±.011 |
| | .90 | 1.037±.011 | 1.006±.012 | 1.007±.012 | .991±.011 | 1.009±.012 | 1.006±.011 | 1.009±.012 | 1.014±.011 | .993±.011 | 1.009±.012 | 1.017±.011 | 1.005±.012 | 1.003±.012 | 1.000±.011 | 1.007±.012 | 1.000±.011 | 1.024±.011 | 1.031±.012 |
| | .85 | 1.049±.012 | 1.011±.012 | 1.013±.012 | .999±.012 | 1.009±.012 | 1.008±.012 | 1.018±.012 | 1.019±.012 | 1.005±.012 | 1.009±.013 | 1.027±.011 | 1.003±.012 | 1.006±.012 | 1.012±.012 | 1.004±.012 | 1.004±.011 | 1.031±.011 | 1.042±.012 |
| | .80 | 1.060±.012 | 1.013±.013 | 1.017±.012 | .997±.012 | 1.026±.012 | 1.012±.012 | 1.021±.012 | 1.024±.012 | 1.006±.011 | 1.030±.012 | 1.035±.011 | 1.008±.013 | 1.007±.013 | 1.025±.013 | 1.007±.013 | 1.006±.012 | 1.045±.012 | 1.057±.012 |
| | .75 | 1.066±.013 | 1.025±.013 | 1.024±.012 | 1.002±.012 | 1.036±.012 | 1.016±.012 | 1.029±.013 | 1.030±.012 | 1.013±.012 | 1.035±.013 | 1.036±.012 | 1.011±.013 | 1.010±.013 | 1.039±.014 | 1.005±.013 | 1.003±.013 | 1.062±.012 | 1.074±.012 |
| | .70 | 1.067±.014 | 1.033±.014 | 1.028±.013 | 1.002±.014 | 1.050±.013 | 1.020±.013 | 1.034±.013 | 1.033±.012 | 1.008±.013 | 1.052±.013 | 1.042±.013 | 1.014±.013 | 1.013±.013 | 1.057±.014 | .988±.014 | 1.002±.013 | 1.073±.013 | 1.097±.012 |

Table B13: Results for `eye`: $mean \pm std$ for $\widehat{Err}$, empirical coverage $\hat{\phi}$, and $MinCoeff$.

| Metric | c | DG | SAT | SAT+EM | SelNet | SelNet+EM | SR | SAT+SR | SAT+EM+SR | SelNet+SR | SelNet+EM+SR | ENS | ENS+SR | ConfidNet | SELE | REG | SCross | AUCross | PlugInAUC |
|---|---|---|---|---|---|---|---|---|---|---|---|---|---|---|---|---|---|---|---|
| $\widehat{Err}$ | .99 | .433±.013 | .441±.014 | .415±.012 | .419±.012 | .413±.013 | .425±.013 | .444±.014 | .414±.012 | .415±.012 | .414±.013 | .427±.013 | .421±.013 | .414±.012 | .411±.013 | .412±.013 | .426±.013 | .425±.013 | .427±.013 |
| | .95 | .435±.013 | .436±.015 | .417±.012 | .414±.012 | .418±.012 | .420±.014 | .436±.015 | .412±.012 | .415±.012 | .411±.013 | .439±.014 | .419±.013 | .414±.012 | .405±.013 | .413±.013 | .421±.013 | .421±.013 | .421±.014 |
| | .90 | .435±.014 | .433±.016 | .415±.012 | .422±.012 | .413±.014 | .413±.014 | .439±.015 | .411±.013 | .412±.012 | .406±.013 | .451±.014 | .412±.013 | .408±.012 | .398±.013 | .412±.013 | .417±.013 | .418±.014 | .412±.013 |
| | .85 | .435±.014 | .429±.016 | .410±.013 | .413±.013 | .416±.014 | .407±.014 | .431±.015 | .397±.013 | .393±.012 | .415±.013 | .457±.015 | .407±.013 | .405±.012 | .400±.013 | .415±.013 | .410±.014 | .411±.015 | .409±.014 |
| | .80 | .437±.014 | .429±.017 | .412±.013 | .406±.013 | .421±.013 | .401±.014 | .424±.016 | .394±.013 | .398±.013 | .390±.013 | .452±.015 | .403±.014 | .400±.013 | .399±.014 | .412±.014 | .402±.014 | .407±.015 | .402±.014 |
| | .75 | .437±.014 | .431±.017 | .401±.014 | .403±.013 | .419±.013 | .394±.014 | .421±.016 | .385±.013 | .391±.013 | .387±.014 | .443±.016 | .396±.014 | .391±.014 | .395±.015 | .410±.015 | .394±.014 | .400±.015 | .399±.014 |
| | .70 | .433±.014 | .424±.018 | .399±.014 | .399±.014 | .416±.014 | .389±.015 | .412±.017 | .377±.014 | .384±.014 | .389±.014 | .443±.017 | .393±.014 | .382±.014 | .386±.016 | .407±.015 | .392±.014 | .397±.015 | .387±.014 |
| $\hat{\phi}$ | .99 | .994±.002 | .983±.004 | .993±.002 | .985±.003 | .989±.002 | .993±.002 | .992±.002 | .987±.003 | .990±.003 | .990±.003 | .985±.003 | .991±.002 | .992±.002 | .989±.003 | .995±.002 | .989±.002 | .991±.002 | .985±.003 |
| | .95 | .947±.006 | .943±.006 | .947±.005 | .950±.006 | .947±.005 | .968±.005 | .959±.005 | .938±.006 | .954±.006 | .949±.006 | .928±.007 | .951±.006 | .953±.006 | .941±.007 | .945±.006 | .954±.006 | .950±.005 | .941±.006 |
| | .90 | .889±.008 | .889±.008 | .886±.008 | .904±.008 | .902±.007 | .898±.008 | .915±.007 | .895±.009 | .896±.008 | .897±.007 | .872±.008 | .893±.008 | .895±.007 | .883±.009 | .908±.007 | .907±.008 | .911±.007 | .894±.008 |
| | .85 | .847±.009 | .846±.010 | .833±.008 | .855±.009 | .862±.009 | .860±.009 | .845±.009 | .833±.010 | .855±.009 | .856±.009 | .837±.009 | .855±.009 | .861±.009 | .853±.010 | .851±.008 | .862±.009 | .855±.008 | .858±.009 |
| | .80 | .811±.010 | .787±.011 | .796±.009 | .796±.010 | .815±.010 | .809±.010 | .799±.011 | .784±.010 | .795±.009 | .792±.011 | .779±.010 | .806±.010 | .790±.010 | .775±.011 | .787±.010 | .816±.010 | .808±.010 | .814±.010 |
| | .75 | .758±.012 | .731±.012 | .746±.010 | .753±.011 | .760±.012 | .762±.011 | .757±.010 | .747±.011 | .748±.011 | .748±.011 | .727±.011 | .759±.010 | .741±.011 | .738±.011 | .725±.011 | .772±.011 | .751±.011 | .762±.011 |
| | .70 | .716±.013 | .687±.012 | .707±.011 | .715±.011 | .712±.012 | .694±.011 | .706±.010 | .697±.012 | .692±.012 | .696±.013 | .679±.011 | .717±.011 | .682±.013 | .686±.013 | .680±.012 | .717±.011 | .715±.011 | .702±.011 |
| $MinCoeff$ | .99 | .999±.029 | 1.007±.030 | .999±.029 | .997±.029 | 1.001±.029 | 1.005±.029 | 1.003±.029 | 1.001±.030 | .996±.030 | 1.003±.030 | 1.012±.030 | 1.007±.029 | 1.006±.029 | 1.003±.030 | 1.001±.029 | 1.004±.030 | 1.001±.030 | 1.000±.030 |
| | .95 | .989±.030 | .996±.029 | .996±.029 | .993±.031 | .985±.030 | 1.004±.030 | .991±.030 | .996±.030 | .999±.030 | .997±.030 | 1.045±.031 | 1.009±.029 | 1.007±.030 | .997±.030 | .995±.029 | 1.009±.030 | .999±.030 | 1.006±.030 |
| | .90 | .981±.031 | .999±.031 | .979±.031 | .977±.032 | .981±.029 | .998±.031 | .995±.030 | .991±.032 | 1.009±.030 | 1.001±.031 | 1.080±.031 | 1.011±.030 | .996±.031 | .993±.031 | .995±.030 | 1.010±.030 | .990±.031 | 1.015±.030 |
| | .85 | .972±.031 | 1.001±.031 | .980±.031 | .979±.031 | .970±.029 | 1.009±.031 | .993±.033 | .999±.032 | 1.001±.030 | 1.007±.031 | 1.096±.031 | 1.019±.031 | .986±.031 | .993±.031 | 1.008±.033 | 1.005±.031 | 1.006±.030 | 1.008±.030 |
| | .80 | .962±.031 | 1.004±.031 | .968±.031 | .971±.034 | .946±.030 | 1.011±.031 | .989±.034 | .987±.033 | .979±.032 | 1.003±.030 | 1.101±.034 | 1.008±.032 | .967±.034 | 1.018±.034 | 1.017±.032 | 1.002±.031 | .998±.032 | 1.009±.031 |
| | .75 | .949±.031 | 1.020±.032 | .969±.033 | .972±.035 | .938±.032 | 1.007±.031 | .981±.035 | .989±.033 | .972±.033 | .999±.032 | 1.098±.035 | 1.005±.033 | .970±.036 | 1.008±.034 | 1.022±.033 | 1.001±.032 | 1.011±.033 | 1.006±.033 |
| | .70 | .947±.033 | 1.026±.034 | .962±.034 | .945±.037 | .929±.033 | 1.001±.034 | .976±.037 | .977±.033 | .950±.034 | 1.005±.032 | 1.100±.037 | .996±.033 | .969±.038 | 1.024±.035 | 1.031±.032 | .997±.034 | 1.021±.035 | 1.004±.034 |

Table B14: Results for `food101`: $mean \pm std$ for $\widehat{Err}$ and empirical coverage $\hat{\phi}$.

| Metric | c | DG | SAT | SAT+EM | SelNet | SelNet+EM | SR | SAT+SR | SAT+EM+SR | SelNet+SR | SelNet+EM+SR | ENS | ENS+SR | ConfidNet | SELE | REG | SCross |
|---|---|---|---|---|---|---|---|---|---|---|---|---|---|---|---|---|---|
| $\widehat{Err}$ | .99 | .464±.004 | .269±.003 | .306±.004 | .322±.003 | .341±.004 | .256±.003 | .268±.003 | .304±.004 | .318±.003 | .339±.004 | .215±.003 | .211±.003 | .271±.003 | .388±.003 | .384±.004 | .242±.003 |
| | .95 | .442±.004 | .250±.003 | .286±.004 | .287±.004 | .328±.003 | .233±.003 | .244±.003 | .280±.004 | .266±.004 | .317±.003 | .196±.003 | .190±.003 | .256±.003 | .380±.004 | .383±.004 | .220±.003 |
| | .90 | .416±.004 | .226±.003 | .260±.004 | .264±.003 | .315±.004 | .205±.003 | .219±.003 | .250±.004 | .232±.003 | .292±.003 | .172±.003 | .165±.003 | .237±.003 | .370±.004 | .382±.004 | .198±.003 |
| | .85 | .392±.004 | .203±.003 | .237±.003 | .283±.003 | .299±.004 | .174±.003 | .191±.003 | .225±.004 | .243±.003 | .264±.003 | .150±.003 | .134±.002 | .219±.003 | .361±.003 | .383±.004 | .174±.003 |
| | .80 | .366±.004 | .179±.003 | .212±.003 | .224±.004 | .278±.004 | .149±.003 | .162±.003 | .199±.004 | .175±.003 | .237±.003 | .128±.003 | .108±.002 | .200±.003 | .353±.003 | .382±.004 | .151±.003 |
| | .75 | .341±.004 | .156±.003 | .190±.003 | .210±.003 | .266±.003 | .126±.003 | .141±.003 | .172±.004 | .148±.003 | .218±.003 | .106±.002 | .086±.002 | .185±.003 | .344±.004 | .383±.004 | .128±.003 |
| | .70 | .318±.004 | .137±.003 | .169±.003 | .219±.004 | .246±.003 | .106±.003 | .117±.003 | .145±.003 | .157±.003 | .189±.003 | .088±.002 | .069±.002 | .170±.003 | .334±.004 | .386±.004 | .107±.002 |
| $\hat{\phi}$ | .99 | .989±.001 | .988±.001 | .993±.001 | .991±.001 | .990±.001 | .990±.001 | .987±.001 | .991±.001 | .990±.001 | .988±.001 | .990±.001 | .987±.001 | .991±.001 | .991±.001 | .992±.001 | .991±.001 |
| | .95 | .948±.001 | .946±.001 | .951±.002 | .952±.002 | .951±.001 | .951±.002 | .946±.002 | .948±.002 | .951±.002 | .948±.002 | .951±.001 | .951±.002 | .951±.002 | .952±.001 | .955±.001 | .955±.001 |
| | .90 | .899±.002 | .897±.002 | .896±.002 | .897±.002 | .903±.002 | .899±.002 | .900±.002 | .891±.002 | .903±.002 | .902±.002 | .898±.002 | .894±.002 | .901±.002 | .902±.002 | .909±.002 | .915±.002 |
| | .85 | .851±.002 | .848±.002 | .846±.003 | .850±.003 | .852±.003 | .846±.003 | .848±.003 | .844±.003 | .850±.002 | .849±.003 | .847±.002 | .847±.003 | .849±.003 | .856±.003 | .860±.003 | .872±.003 |
| | .80 | .801±.003 | .796±.003 | .793±.003 | .792±.003 | .801±.003 | .795±.003 | .792±.003 | .797±.003 | .795±.003 | .799±.003 | .794±.003 | .795±.003 | .798±.003 | .809±.003 | .810±.003 | .826±.003 |
| | .75 | .752±.003 | .744±.004 | .745±.003 | .750±.003 | .751±.003 | .746±.003 | .747±.003 | .745±.003 | .753±.003 | .752±.003 | .744±.004 | .745±.004 | .750±.003 | .760±.003 | .758±.003 | .777±.003 |
| | .70 | .700±.003 | .696±.004 | .695±.004 | .705±.003 | .700±.003 | .700±.003 | .696±.003 | .695±.004 | .706±.004 | .701±.004 | .698±.004 | .697±.004 | .700±.004 | .709±.003 | .704±.003 | .730±.003 |

Table B15: Results for `giveme`: *mean ± std* for $\widehat{Err}$, empirical coverage $\hat{\phi}$, and *MinCoeff*.

| Metric | c | DG | SAT | SAT+EM | SelNet | SelNet+EM | SR | SAT+SR | SAT+EM+SR | SelNet+SR | SelNet+EM+SR | ENS | ENS+SR | ConfidNet | SELE | REG | SCross | AUCross | PlugInAUC |
|---|---|---|---|---|---|---|---|---|---|---|---|---|---|---|---|---|---|---|---|
| $\widehat{Err}$ | .99 | .061±.001 | .060±.001 | .060±.001 | .060±.001 | .060±.001 | .061±.001 | .061±.001 | .059±.001 | .059±.001 | .062±.001 | .061±.001 | .060±.001 | .059±.001 | .060±.001 | .065±.001 | .060±.001 | .064±.001 | .065±.001 |
| | .95 | .047±.001 | .046±.001 | .046±.001 | .046±.001 | .046±.001 | .046±.001 | .046±.001 | .046±.001 | .046±.001 | .046±.001 | .050±.001 | .047±.001 | .046±.001 | .047±.001 | .065±.001 | .047±.001 | .064±.001 | .066±.001 |
| | .90 | .036±.001 | .035±.001 | .035±.001 | .035±.001 | .035±.001 | .035±.001 | .035±.001 | .035±.001 | .035±.001 | .037±.001 | .046±.001 | .035±.001 | .035±.001 | .036±.001 | .065±.001 | .036±.001 | .065±.001 | .068±.001 |
| | .85 | .031±.001 | .029±.001 | .029±.001 | .029±.001 | .029±.001 | .030±.001 | .029±.001 | .029±.001 | .029±.001 | .040±.001 | .045±.001 | .029±.001 | .029±.001 | .036±.001 | .065±.001 | .036±.001 | .065±.002 | .069±.001 |
| | .80 | .028±.001 | .026±.001 | .025±.001 | .025±.001 | .025±.001 | .026±.001 | .025±.001 | .025±.001 | .025±.001 | .034±.001 | .043±.001 | .026±.001 | .025±.001 | .026±.001 | .057±.001 | .026±.001 | .065±.002 | .070±.002 |
| | .75 | .025±.001 | .023±.001 | .023±.001 | .023±.001 | .023±.001 | .023±.001 | .023±.001 | .023±.001 | .023±.001 | .028±.001 | .041±.001 | .023±.001 | .023±.001 | .023±.001 | .053±.001 | .024±.001 | .065±.002 | .071±.002 |
| | .70 | .023±.001 | .022±.001 | .021±.001 | .021±.001 | .022±.001 | .022±.001 | .022±.001 | .022±.001 | .021±.001 | .026±.001 | .038±.001 | .021±.001 | .021±.001 | .021±.001 | .051±.001 | .023±.001 | .063±.002 | .072±.002 |
| $\hat{\phi}$ | .99 | .990±.001 | .989±.001 | .989±.001 | .989±.001 | .989±.001 | .991±.001 | .991±.001 | .988±.001 | .988±.001 | .990±.001 | .990±.001 | .990±.001 | .989±.001 | .990±.001 | .991±.000 | .991±.000 | .994±.000 | .989±.001 |
| | .95 | .948±.001 | .947±.001 | .949±.001 | .950±.001 | .949±.001 | .949±.001 | .949±.001 | .949±.001 | .949±.001 | .950±.001 | .948±.001 | .950±.001 | .950±.001 | .949±.001 | .967±.001 | .952±.001 | .964±.001 | .947±.001 |
| | .90 | .896±.002 | .897±.002 | .896±.002 | .898±.002 | .899±.002 | .898±.002 | .898±.002 | .897±.002 | .899±.002 | .894±.002 | .896±.002 | .898±.002 | .898±.002 | .897±.002 | .967±.001 | .904±.002 | .927±.002 | .895±.002 |
| | .85 | .848±.002 | .847±.002 | .847±.002 | .846±.002 | .846±.002 | .849±.002 | .847±.002 | .847±.002 | .843±.002 | .845±.002 | .847±.002 | .847±.002 | .847±.002 | .854±.002 | .854±.002 | .891±.002 | .850±.002 | |
| | .80 | .799±.002 | .799±.002 | .794±.003 | .797±.002 | .796±.002 | .796±.002 | .799±.002 | .795±.003 | .798±.002 | .794±.002 | .798±.002 | .797±.002 | .797±.002 | .798±.002 | .807±.002 | .803±.002 | .846±.002 | .803±.002 |
| | .75 | .748±.002 | .750±.003 | .747±.003 | .750±.003 | .748±.003 | .747±.003 | .749±.003 | .748±.003 | .747±.003 | .748±.002 | .748±.003 | .748±.003 | .749±.003 | .759±.003 | .767±.003 | .806±.002 | .754±.003 | |
| | .70 | .696±.002 | .701±.003 | .697±.003 | .697±.003 | .699±.003 | .698±.003 | .701±.003 | .698±.003 | .698±.003 | .696±.003 | .697±.003 | .701±.003 | .696±.003 | .707±.003 | .730±.003 | .756±.003 | .706±.003 | |
| *MinCoeff* | .99 | .936±.020 | .936±.021 | .925±.021 | .901±.020 | .905±.020 | .942±.021 | .939±.020 | .918±.020 | .908±.021 | .926±.020 | .945±.020 | .933±.020 | .932±.021 | .933±.020 | .994±.022 | .941±.020 | 1.000±.021 | 1.001±.021 |
| | .95 | .718±.019 | .684±.017 | .691±.017 | .691±.017 | .685±.017 | .705±.018 | .722±.018 | .697±.017 | .689±.017 | .685±.017 | .781±.019 | .706±.018 | .690±.017 | .715±.017 | .990±.022 | .702±.018 | 1.003±.022 | 1.018±.022 |
| | .90 | .545±.017 | .520±.015 | .519±.016 | .524±.015 | .523±.016 | .526±.016 | .528±.016 | .523±.016 | .525±.015 | .548±.016 | .716±.019 | .530±.016 | .524±.015 | .536±.016 | .990±.022 | .537±.016 | 1.009±.022 | 1.044±.023 |
| | .85 | .463±.016 | .437±.015 | .433±.015 | .440±.015 | .437±.016 | .443±.015 | .440±.015 | .437±.015 | .437±.016 | .600±.019 | .694±.019 | .439±.015 | .436±.016 | .437±.015 | .890±.020 | .449±.015 | 1.012±.024 | 1.061±.024 |
| | .80 | .419±.016 | .384±.015 | .380±.015 | .381±.015 | .378±.015 | .387±.015 | .382±.015 | .381±.015 | .379±.015 | .511±.019 | .672±.020 | .387±.015 | .380±.015 | .390±.015 | .859±.020 | .396±.014 | 1.014±.025 | 1.074±.025 |
| | .75 | .375±.016 | .347±.015 | .345±.015 | .347±.015 | .348±.016 | .344±.015 | .344±.015 | .346±.015 | .342±.015 | .414±.017 | .636±.020 | .349±.015 | .343±.015 | .345±.015 | .804±.019 | .365±.016 | 1.017±.026 | 1.094±.026 |
| | .70 | .344±.016 | .323±.015 | .319±.016 | .319±.016 | .330±.016 | .325±.016 | .324±.015 | .322±.016 | .316±.017 | .400±.017 | .592±.020 | .320±.015 | .318±.017 | .317±.016 | .768±.021 | .339±.016 | .995±.028 | 1.110±.029 |

Table B16: Results for `helena`: *mean ± std* for $\widehat{Err}$ and empirical coverage $\hat{\phi}$.

| Metric | c | DG | SAT | SAT+EM | SelNet | SelNet+EM | SR | SAT+SR | SAT+EM+SR | SelNet+SR | SelNet+EM+SR | ENS | ENS+SR | ConfidNet | SELE | REG | SCross |
|---|---|---|---|---|---|---|---|---|---|---|---|---|---|---|---|---|---|
| $\widehat{Err}$ | .99 | .618±.004 | .614±.004 | .619±.004 | .613±.004 | .613±.004 | .615±.004 | .612±.004 | .618±.004 | .611±.004 | .614±.004 | .612±.004 | .610±.004 | .615±.004 | .633±.004 | .632±.004 | .611±.004 |
| | .95 | .608±.004 | .605±.004 | .609±.004 | .600±.004 | .601±.004 | .603±.004 | .599±.004 | .605±.004 | .599±.005 | .599±.004 | .605±.004 | .599±.004 | .607±.004 | .623±.004 | .631±.004 | .597±.004 |
| | .90 | .594±.004 | .592±.005 | .597±.004 | .583±.004 | .585±.005 | .585±.004 | .585±.004 | .590±.004 | .582±.004 | .584±.005 | .594±.004 | .582±.004 | .595±.004 | .609±.004 | .631±.004 | .581±.004 |
| | .85 | .581±.004 | .581±.005 | .584±.005 | .570±.005 | .570±.005 | .569±.004 | .566±.005 | .572±.004 | .564±.005 | .566±.005 | .582±.004 | .563±.004 | .586±.005 | .597±.005 | .628±.004 | .563±.005 |
| | .80 | .567±.005 | .567±.005 | .570±.005 | .552±.005 | .557±.005 | .547±.005 | .548±.005 | .553±.005 | .547±.005 | .552±.005 | .568±.005 | .545±.005 | .577±.004 | .583±.005 | .628±.004 | .547±.005 |
| | .75 | .555±.005 | .551±.005 | .557±.005 | .540±.005 | .541±.005 | .529±.005 | .528±.005 | .536±.005 | .529±.005 | .534±.005 | .555±.005 | .526±.005 | .566±.005 | .570±.005 | .624±.004 | .529±.005 |
| | .70 | .541±.005 | .535±.006 | .540±.006 | .530±.005 | .525±.005 | .510±.005 | .510±.005 | .517±.005 | .515±.005 | .520±.005 | .540±.005 | .509±.005 | .556±.005 | .555±.005 | .621±.004 | .509±.005 |
| $\hat{\phi}$ | .99 | .989±.001 | .991±.001 | .990±.001 | .989±.001 | .989±.001 | .987±.001 | .989±.001 | .991±.001 | .991±.001 | .990±.001 | .988±.001 | .989±.001 | .991±.001 | .988±.001 | .987±.001 | .988±.001 |
| | .95 | .947±.002 | .953±.002 | .944±.002 | .945±.002 | .946±.002 | .946±.002 | .949±.002 | .948±.002 | .948±.002 | .948±.002 | .953±.002 | .950±.002 | .949±.002 | .939±.002 | .944±.002 | .946±.002 |
| | .90 | .898±.002 | .900±.003 | .892±.003 | .890±.003 | .892±.002 | .895±.003 | .900±.003 | .901±.003 | .899±.003 | .898±.003 | .898±.002 | .900±.003 | .894±.002 | .889±.003 | .894±.003 | .897±.003 |
| | .85 | .846±.003 | .852±.003 | .845±.004 | .838±.003 | .840±.003 | .849±.003 | .846±.003 | .848±.003 | .842±.003 | .846±.003 | .846±.003 | .843±.003 | .844±.003 | .838±.003 | .842±.003 | .846±.003 |
| | .80 | .792±.003 | .801±.003 | .796±.004 | .788±.003 | .790±.004 | .789±.003 | .792±.003 | .797±.003 | .798±.003 | .793±.003 | .799±.003 | .794±.004 | .798±.003 | .788±.004 | .788±.003 | .795±.004 |
| | .75 | .742±.004 | .752±.004 | .751±.005 | .745±.004 | .744±.004 | .739±.004 | .743±.004 | .748±.004 | .747±.004 | .747±.003 | .752±.004 | .743±.004 | .745±.004 | .743±.004 | .740±.003 | .746±.004 |
| | .70 | .695±.004 | .701±.004 | .699±.004 | .700±.004 | .688±.004 | .693±.004 | .697±.004 | .697±.004 | .697±.004 | .698±.004 | .702±.004 | .699±.004 | .697±.004 | .690±.004 | .691±.003 | .699±.004 |

Table B17: Results for `heloc`: *mean ± std* for $\widehat{Err}$, empirical coverage $\hat{\phi}$, and *MinCoeff*.

| Metric | c | DG | SAT | SAT+EM | SelNet | SelNet+EM | SR | SAT+SR | SAT+EM+SR | SelNet+SR | SelNet+EM+SR | ENS | ENS+SR | ConfidNet | SELE | REG | SCross | AUCross | PlugInAUC |
|---|---|---|---|---|---|---|---|---|---|---|---|---|---|---|---|---|---|---|---|
| $\widehat{Err}$ | .99 | .309±.010 | .279±.009 | .277±.010 | .279±.011 | .277±.010 | .285±.009 | .277±.009 | .280±.011 | .277±.010 | .277±.010 | .275±.010 | .273±.010 | .283±.010 | .289±.010 | .274±.010 | .276±.010 | .260±.010 | .285±.009 |
| | .95 | .307±.011 | .272±.009 | .273±.010 | .281±.011 | .279±.010 | .281±.010 | .276±.009 | .280±.010 | .275±.011 | .271±.010 | .273±.010 | .263±.010 | .283±.010 | .283±.010 | .292±.009 | .268±.010 | .260±.010 | .277±.010 |
| | .90 | .301±.011 | .259±.010 | .263±.010 | .265±.010 | .260±.011 | .261±.010 | .259±.010 | .262±.010 | .261±.011 | .260±.010 | .273±.010 | .258±.011 | .275±.010 | .275±.010 | .290±.010 | .259±.010 | .245±.010 | .270±.010 |
| | .85 | .296±.011 | .253±.010 | .248±.011 | .253±.010 | .247±.011 | .246±.011 | .245±.010 | .249±.011 | .253±.011 | .248±.010 | .266±.011 | .242±.010 | .281±.010 | .274±.011 | .291±.010 | .252±.010 | .237±.010 | .242±.011 |
| | .80 | .290±.011 | .253±.010 | .243±.011 | .242±.011 | .256±.011 | .232±.010 | .231±.010 | .233±.011 | .240±.010 | .238±.011 | .258±.011 | .225±.011 | .288±.010 | .276±.011 | .287±.010 | .236±.011 | .224±.011 | .239±.012 |
| | .75 | .283±.012 | .218±.011 | .226±.011 | .228±.011 | .321±.013 | .216±.011 | .215±.011 | .223±.011 | .225±.011 | .226±.011 | .255±.011 | .213±.011 | .291±.011 | .278±.012 | .285±.010 | .228±.011 | .207±.011 | .238±.012 |
| | .70 | .283±.012 | .202±.011 | .213±.011 | .209±.011 | .210±.011 | .202±.011 | .201±.011 | .208±.011 | .207±.011 | .212±.011 | .242±.012 | .202±.011 | .294±.011 | .285±.012 | .284±.011 | .210±.011 | .193±.011 | .231±.012 |
| $\hat{\phi}$ | .99 | .994±.001 | .994±.002 | .987±.003 | .993±.002 | .993±.002 | .997±.001 | .986±.003 | .993±.002 | .990±.002 | .989±.003 | .996±.001 | .985±.003 | .990±.002 | .999±.001 | .990±.002 | .990±.002 | .986±.003 | .989±.002 |
| | .95 | .974±.003 | .947±.005 | .968±.004 | 1.000±.000 | .999±.001 | .986±.002 | .984±.003 | .990±.002 | .974±.004 | .963±.004 | .946±.005 | .947±.005 | .988±.002 | .999±.001 | .951±.005 | .974±.003 | .907±.006 | .950±.004 |
| | .90 | .951±.005 | .893±.006 | .905±.006 | .916±.006 | .908±.006 | .916±.006 | .913±.006 | .913±.006 | .907±.006 | .913±.006 | .904±.006 | .933±.006 | .911±.006 | .905±.007 | .901±.008 | .927±.007 | .861±.007 | .908±.006 |
| | .85 | .911±.006 | .864±.008 | .864±.008 | .877±.008 | .847±.009 | .860±.007 | .866±.007 | .858±.008 | .866±.009 | .816±.009 | .853±.008 | .852±.007 | .860±.007 | .878±.008 | .862±.009 | .894±.008 | .830±.007 | .814±.008 |
| | .80 | .854±.007 | .864±.008 | .801±.008 | .823±.009 | .818±.008 | .821±.007 | .820±.008 | .798±.009 | .816±.009 | .811±.009 | .800±.008 | .806±.009 | .821±.009 | .795±.009 | .906±.009 | .867±.009 | .781±.009 | .783±.009 |
| | .75 | .777±.009 | .762±.010 | .754±.009 | .761±.009 | .747±.011 | .764±.009 | .768±.009 | .758±.009 | .759±.009 | .763±.010 | .750±.009 | .759±.010 | .769±.009 | .735±.010 | .761±.011 | .754±.009 | .731±.010 | .766±.009 |
| | .70 | .777±.009 | .711±.011 | .714±.010 | .716±.010 | .715±.010 | .716±.010 | .720±.010 | .704±.010 | .711±.011 | .704±.010 | .688±.009 | .702±.010 | .713±.009 | .692±.010 | .724±.011 | .690±.010 | .690±.010 | .722±.010 |
| *MinCoeff* | .99 | .997±.023 | 1.001±.023 | 1.003±.024 | 1.001±.023 | .999±.023 | .999±.023 | 1.002±.023 | 1.001±.023 | 1.004±.023 | 1.002±.023 | 1.001±.023 | 1.002±.024 | 1.009±.023 | 1.002±.023 | .994±.024 | 1.001±.023 | 1.008±.023 | 1.003±.023 |
| | .95 | .991±.024 | 1.005±.023 | 1.004±.024 | 1.000±.023 | .999±.023 | .999±.023 | 1.001±.023 | 1.001±.023 | 1.002±.023 | 1.001±.024 | .979±.024 | 1.011±.023 | 1.010±.023 | 1.002±.023 | .989±.024 | 1.000±.024 | 1.014±.024 | 1.005±.024 |
| | .90 | .990±.024 | 1.016±.024 | 1.017±.025 | 1.006±.025 | 1.000±.025 | 1.007±.023 | 1.005±.024 | 1.009±.024 | 1.016±.024 | 1.008±.025 | .958±.025 | 1.006±.023 | 1.033±.025 | 1.000±.025 | .986±.024 | .995±.025 | 1.011±.024 | 1.011±.025 |
| | .85 | .980±.026 | 1.009±.024 | 1.020±.024 | 1.004±.026 | 1.010±.024 | 1.013±.024 | 1.013±.024 | 1.010±.025 | 1.015±.025 | 1.012±.025 | .952±.026 | 1.014±.024 | 1.063±.026 | .977±.025 | .983±.025 | .988±.025 | 1.016±.025 | 1.026±.026 |
| | .80 | .975±.026 | 1.009±.024 | 1.019±.025 | 1.026±.025 | .965±.026 | 1.022±.024 | 1.012±.025 | 1.018±.025 | 1.019±.026 | 1.015±.025 | .934±.026 | 1.013±.025 | 1.114±.026 | .963±.024 | .983±.027 | .984±.027 | 1.022±.025 | 1.041±.026 |
| | .75 | .977±.027 | 1.001±.026 | 1.025±.025 | 1.018±.026 | .767±.023 | 1.025±.026 | 1.014±.026 | 1.024±.025 | 1.023±.027 | 1.020±.025 | .922±.028 | 1.012±.026 | 1.157±.026 | .958±.025 | .995±.029 | .975±.027 | 1.030±.026 | 1.053±.027 |
| | .70 | .977±.027 | 1.006±.027 | 1.026±.026 | 1.011±.028 | 1.028±.027 | 1.031±.027 | 1.012±.027 | 1.022±.026 | 1.024±.027 | 1.015±.026 | .894±.028 | 1.005±.028 | 1.207±.027 | .941±.027 | 1.002±.029 | .950±.028 | 1.043±.027 | 1.079±.027 |

Table B18: Results for `higgs`: *mean ± std* for $\widehat{Err}$, empirical coverage $\hat{\phi}$, and *MinCoeff*.

| Metric | c | DG | SAT | SAT+EM | SelNet | SelNet+EM | SR | SAT+SR | SAT+EM+SR | SelNet+SR | SelNet+EM+SR | ENS | ENS+SR | ConfidNet | SELE | REG | SCross | AUCross | PlugInAUC |
|---|---|---|---|---|---|---|---|---|---|---|---|---|---|---|---|---|---|---|---|
| $\widehat{Err}$ | .99 | .385±.005 | .271±.003 | .286±.004 | .271±.003 | .279±.003 | .273±.003 | .273±.003 | .286±.004 | .271±.003 | .279±.003 | .269±.003 | .268±.003 | .280±.003 | .287±.003 | .296±.003 | .278±.004 | .279±.003 | .275±.004 |
| | .95 | .388±.010 | .269±.004 | .279±.004 | .264±.003 | .269±.003 | .264±.004 | .265±.003 | .277±.004 | .263±.003 | .270±.003 | .264±.003 | .260±.003 | .273±.003 | .282±.003 | .296±.003 | .268±.004 | .273±.003 | .268±.004 |
| | .90 | .388±.016 | .261±.003 | .268±.004 | .255±.003 | .263±.003 | .253±.004 | .255±.003 | .268±.004 | .249±.003 | .262±.003 | .258±.003 | .248±.004 | .266±.003 | .275±.003 | .296±.003 | .264±.004 | .258±.004 | .264±.004 |
| | .85 | .387±.021 | .253±.004 | .256±.004 | .247±.003 | .251±.003 | .242±.004 | .243±.004 | .258±.004 | .245±.003 | .249±.003 | .252±.003 | .236±.004 | .257±.003 | .269±.003 | .296±.003 | .242±.004 | .256±.004 | .246±.004 |
| | .80 | .382±.022 | .247±.004 | .246±.004 | .230±.003 | .306±.004 | .231±.004 | .231±.004 | .245±.004 | .229±.003 | .237±.003 | .246±.004 | .225±.004 | .248±.004 | .263±.003 | .294±.004 | .231±.004 | .244±.004 | .237±.004 |
| | .75 | .377±.023 | .239±.004 | .233±.004 | .217±.004 | .301±.004 | .219±.004 | .220±.004 | .233±.004 | .215±.003 | .228±.004 | .240±.004 | .213±.004 | .240±.004 | .259±.003 | .295±.004 | .231±.004 | .231±.004 | .225±.004 |
| | .70 | .370±.025 | .230±.004 | .219±.004 | .207±.004 | .306±.004 | .207±.004 | .209±.004 | .221±.004 | .209±.004 | .217±.004 | .234±.004 | .201±.004 | .231±.004 | .253±.003 | .294±.004 | .208±.004 | .219±.004 | .213±.004 |
| $\hat{\phi}$ | .99 | .990±.001 | .991±.001 | .992±.001 | .989±.001 | .990±.001 | .988±.001 | .990±.001 | .991±.001 | .989±.001 | .990±.001 | .990±.001 | .990±.001 | .990±.001 | .990±.001 | .991±.001 | .991±.001 | .990±.001 | .991±.001 |
| | .95 | .949±.002 | .948±.002 | .953±.001 | .951±.002 | .951±.001 | .947±.002 | .952±.002 | .951±.001 | .953±.002 | .948±.002 | .945±.002 | .952±.001 | .950±.001 | .949±.002 | .948±.002 | .951±.002 | .946±.002 | .951±.002 |
| | .90 | .897±.004 | .899±.002 | .903±.002 | .903±.002 | .900±.002 | .903±.002 | .902±.002 | .901±.002 | .898±.002 | .900±.002 | .896±.002 | .898±.002 | .902±.002 | .897±.002 | .899±.002 | .893±.002 | .896±.002 | .902±.002 |
| | .85 | .847±.005 | .843±.002 | .849±.003 | .855±.002 | .851±.002 | .852±.003 | .852±.003 | .853±.002 | .853±.002 | .848±.002 | .848±.003 | .850±.002 | .849±.003 | .844±.003 | .847±.002 | .842±.003 | .849±.003 | .849±.002 |
| | .80 | .796±.004 | .792±.003 | .803±.003 | .798±.002 | .798±.003 | .804±.003 | .800±.003 | .800±.003 | .800±.003 | .803±.003 | .799±.003 | .802±.003 | .796±.003 | .792±.003 | .797±.003 | .794±.003 | .796±.003 | .800±.003 |
| | .75 | .749±.005 | .743±.003 | .752±.003 | .750±.003 | .741±.003 | .755±.003 | .748±.003 | .751±.003 | .747±.003 | .754±.003 | .753±.003 | .753±.003 | .748±.003 | .750±.003 | .749±.003 | .742±.003 | .745±.003 | .752±.003 |
| | .70 | .696±.007 | .689±.003 | .698±.003 | .703±.003 | .698±.003 | .702±.004 | .701±.003 | .699±.003 | .703±.003 | .705±.003 | .705±.003 | .707±.003 | .699±.003 | .702±.003 | .696±.003 | .692±.003 | .697±.003 | .703±.003 |
| *MinCoeff* | .99 | 1.005±.008 | 1.000±.007 | 1.002±.006 | 1.001±.006 | 1.000±.006 | 1.001±.006 | 1.000±.006 | 1.001±.006 | 1.001±.006 | 1.002±.006 | 1.000±.006 | 1.001±.006 | 1.002±.006 | 1.000±.007 | 1.000±.006 | 1.002±.006 | 1.003±.006 | 1.002±.007 |
| | .95 | 1.022±.018 | 1.002±.007 | 1.006±.006 | 1.006±.006 | 1.002±.007 | 1.003±.006 | 1.001±.006 | 1.009±.006 | 1.003±.007 | 1.003±.006 | 1.004±.007 | 1.004±.007 | 1.006±.007 | 1.002±.007 | 1.000±.007 | 1.007±.007 | 1.011±.007 | 1.007±.007 |
| | .90 | 1.038±.031 | 1.006±.007 | 1.015±.007 | 1.009±.006 | .998±.007 | 1.007±.007 | 1.007±.007 | 1.019±.007 | 1.007±.007 | 1.009±.007 | 1.007±.007 | 1.009±.007 | 1.010±.007 | 1.005±.007 | .996±.007 | 1.013±.007 | 1.024±.007 | 1.018±.007 |
| | .85 | 1.052±.041 | 1.010±.007 | 1.029±.007 | 1.013±.007 | 1.166±.007 | 1.018±.007 | 1.011±.007 | 1.011±.007 | 1.008±.008 | 1.016±.008 | 1.007±.007 | 1.012±.007 | 1.033±.007 | 1.003±.007 | .988±.025 | 1.016±.013 | 1.038±.007 | 1.040±.007 |
| | .80 | 1.061±.045 | 1.014±.007 | 1.038±.007 | 1.012±.007 | 1.166±.007 | 1.018±.007 | 1.016±.007 | 1.008±.008 | 1.018±.007 | 1.019±.008 | 1.009±.007 | 1.019±.008 | 1.024±.008 | 1.016±.008 | .997±.007 | 1.031±.007 | 1.043±.008 | 1.042±.008 |
| | .75 | 1.069±.051 | 1.020±.008 | 1.053±.008 | 1.018±.007 | 1.200±.007 | 1.025±.007 | 1.020±.007 | 1.060±.007 | 1.024±.007 | 1.027±.008 | 1.011±.008 | 1.024±.008 | 1.033±.008 | 1.017±.008 | .997±.008 | 1.041±.008 | 1.075±.008 | 1.058±.008 |
| | .70 | 1.079±.058 | 1.026±.008 | 1.070±.008 | 1.029±.008 | 1.247±.007 | 1.031±.008 | 1.024±.007 | 1.075±.008 | 1.022±.008 | 1.032±.008 | 1.013±.008 | 1.028±.008 | 1.045±.008 | 1.023±.008 | .997±.008 | 1.055±.008 | 1.093±.009 | 1.072±.008 |

Table B19: Results for `house`: *mean ± std* for $\widehat{Err}$, empirical coverage $\hat{\phi}$, and *MinCoeff*.

| Metric | c | DG | SAT | SAT+EM | SelNet | SelNet+EM | SR | SAT+SR | SAT+EM+SR | SelNet+SR | SelNet+EM+SR | ENS | ENS+SR | ConfidNet | SELE | REG | SCross | AUCross | PlugInAUC |
|---|---|---|---|---|---|---|---|---|---|---|---|---|---|---|---|---|---|---|---|
| $\widehat{Err}$ | .99 | .217±.007 | .126±.007 | .130±.007 | .131±.006 | .133±.006 | .128±.006 | .124±.007 | .129±.007 | .128±.006 | .132±.006 | .117±.006 | .116±.006 | .128±.007 | .140±.006 | .143±.006 | **.120±.006** | .124±.006 | .127±.006 |
| | .95 | .208±.007 | .114±.006 | .120±.006 | .122±.006 | .123±.006 | .117±.006 | .113±.006 | .116±.006 | .114±.006 | .116±.006 | .106±.006 | .116±.006 | .131±.006 | .146±.006 | **.105±.006** | .117±.006 | .120±.006 |
| | .90 | .190±.007 | .098±.006 | .106±.006 | .111±.006 | .112±.006 | .101±.006 | .099±.006 | .103±.005 | .097±.006 | .100±.006 | .109±.006 | .092±.006 | .099±.006 | .123±.006 | .147±.007 | **.091±.006** | .110±.006 | .112±.007 |
| | .85 | .175±.007 | .081±.005 | .089±.005 | .093±.006 | .096±.006 | .083±.006 | **.078±.005** | .085±.005 | .075±.006 | .088±.005 | .099±.006 | **.078±.005** | .085±.006 | .113±.006 | .147±.007 | .079±.006 | .097±.006 | .101±.006 |
| | .80 | .158±.008 | .068±.005 | .078±.006 | .077±.006 | .080±.006 | **.064±.005** | .066±.005 | .077±.006 | .073±.006 | .077±.006 | .093±.006 | **.064±.005** | .068±.005 | .102±.006 | .149±.008 | .065±.006 | .088±.005 | .088±.006 |
| | .75 | .144±.008 | .057±.005 | .069±.006 | .062±.006 | .066±.006 | .057±.005 | .057±.005 | .071±.006 | .062±.006 | .064±.005 | .086±.006 | **.055±.005** | .060±.005 | .095±.006 | .148±.008 | .056±.006 | .075±.006 | .078±.006 |
| | .70 | .141±.008 | .050±.005 | .060±.006 | .055±.006 | .052±.005 | **.049±.005** | .050±.005 | .061±.006 | .052±.005 | .054±.005 | .072±.006 | **.049±.005** | .052±.005 | .085±.006 | .149±.008 | .053±.005 | .063±.006 | .067±.006 |
| $\hat{\phi}$ | .99 | .984±.002 | .988±.002 | .994±.001 | .990±.002 | .992±.002 | .983±.003 | .987±.002 | .989±.002 | .989±.002 | .984±.003 | .993±.002 | .994±.001 | **.990±.002** | .990±.002 | .992±.001 | .987±.002 | .990±.002 | .982±.002 |
| | .95 | .947±.004 | .946±.004 | .957±.004 | .952±.004 | .950±.005 | .947±.004 | .951±.004 | .953±.004 | .944±.004 | .948±.004 | .960±.004 | **.949±.004** | **.950±.004** | .945±.004 | .956±.004 | .945±.004 | .948±.005 | .940±.005 |
| | .90 | .892±.006 | **.900±.005** | .904±.006 | .913±.005 | .909±.006 | .904±.005 | .898±.005 | .901±.006 | .906±.005 | **.901±.005** | .904±.005 | .891±.005 | .903±.006 | .893±.006 | .905±.005 | .893±.006 | .898±.007 | .909±.006 |
| | .85 | .841±.007 | .843±.006 | .841±.007 | **.849±.007** | .853±.007 | .844±.007 | .842±.006 | .837±.008 | **.849±.006** | .844±.006 | **.850±.007** | .848±.007 | .847±.007 | .832±.006 | .857±.007 | .844±.007 | .844±.008 | .866±.007 |
| | .80 | .779±.008 | **.800±.006** | .788±.007 | .794±.008 | .795±.008 | .772±.008 | .796±.007 | .785±.008 | **.800±.007** | .791±.008 | .807±.007 | .794±.008 | .786±.007 | .771±.007 | .794±.008 | .789±.008 | .799±.008 | .802±.008 |
| | .75 | .725±.008 | .740±.008 | .739±.009 | .742±.008 | .727±.009 | .728±.008 | .742±.009 | .742±.008 | .742±.008 | .728±.008 | .760±.007 | .738±.009 | .737±.008 | .731±.007 | .758±.009 | .733±.008 | **.752±.008** | .760±.008 |
| | .70 | .678±.008 | .686±.009 | .688±.009 | .693±.008 | .677±.009 | .686±.009 | .685±.010 | .688±.009 | .683±.008 | .706±.008 | .680±.009 | **.697±.008** | .685±.007 | .709±.010 | .687±.009 | .693±.009 | .708±.009 |
| *MinCoeff* | .99 | .996±.018 | **.999±.018** | **.999±.018** | 1.001±.019 | **1.001±.018** | .991±.018 | 1.002±.018 | .998±.018 | **1.002±.018** | 1.001±.019 | **1.001±.018** | 1.000±.019 | **1.001±.018** | 1.001±.019 | 1.003±.018 | **1.000±.018** | 1.004±.019 | 1.002±.019 |
| | .95 | .994±.019 | 1.000±.019 | .999±.019 | 1.005±.018 | 1.003±.019 | .985±.020 | 1.008±.019 | 1.005±.018 | 1.004±.019 | .998±.020 | **.998±.018** | 1.001±.019 | .999±.019 | .999±.019 | 1.026±.018 | .997±.019 | 1.025±.019 | 1.023±.019 |
| | .90 | .997±.020 | .998±.020 | **1.001±.019** | 1.011±.019 | 1.006±.020 | .980±.021 | 1.000±.020 | 1.007±.019 | 1.005±.020 | .992±.021 | .991±.019 | .997±.020 | **.998±.019** | .999±.020 | 1.029±.019 | .997±.021 | 1.055±.019 | 1.032±.019 |
| | .85 | .988±.020 | .987±.020 | **1.001±.020** | 1.018±.021 | 1.014±.021 | .972±.022 | .989±.020 | 1.000±.020 | 1.008±.022 | .991±.021 | .989±.020 | .991±.021 | .998±.020 | .992±.022 | 1.007±.020 | .994±.022 | 1.084±.020 | 1.050±.020 |
| | .80 | .973±.021 | .986±.021 | .997±.021 | 1.021±.022 | 1.004±.022 | .964±.023 | .988±.021 | 1.005±.021 | 1.008±.022 | .980±.022 | .974±.020 | .988±.022 | **1.002±.020** | .984±.023 | .987±.021 | 1.002±.022 | 1.099±.021 | 1.075±.020 |
| | .75 | .951±.021 | .971±.021 | .994±.022 | 1.020±.022 | .993±.023 | .949±.024 | .980±.020 | **.998±.022** | 1.010±.023 | .967±.023 | .967±.022 | .987±.023 | .990±.021 | .972±.023 | .975±.021 | 1.005±.023 | 1.115±.021 | 1.094±.021 |
| | .70 | .939±.023 | .958±.022 | .983±.022 | 1.009±.023 | .992±.022 | .949±.025 | .961±.022 | **.994±.022** | .996±.023 | .979±.023 | .954±.023 | .997±.023 | .991±.021 | .961±.024 | .961±.021 | 1.013±.024 | 1.143±.023 | 1.114±.022 |

Table B20: Results for `indian`: *mean ± std* for $\widehat{Err}$ and empirical coverage $\hat{\phi}$.

| Metric | c | DG | SAT | SAT+EM | SelNet | SelNet+EM | SR | SAT+SR | SAT+EM+SR | SelNet+SR | SelNet+EM+SR | ENS | ENS+SR | ConfidNet | SELE | REG | SCross |
|---|---|---|---|---|---|---|---|---|---|---|---|---|---|---|---|---|---|
| $\widehat{Err}$ | .99 | .050±.005 | .038±.004 | .041±.005 | .046±.005 | .131±.008 | .038±.004 | .041±.005 | .039±.005 | .041±.005 | .146±.008 | .040±.005 | .040±.005 | .055±.005 | .065±.006 | .084±.006 | .037±.005 |
| | .95 | .038±.005 | .029±.004 | .023±.004 | .031±.005 | .107±.008 | .023±.004 | .027±.004 | .023±.004 | .029±.004 | .123±.008 | .024±.004 | **.017±.003** | .042±.005 | .062±.006 | .079±.006 | .027±.004 |
| | .90 | .032±.005 | .020±.004 | .012±.003 | .022±.004 | .099±.008 | .011±.003 | .017±.003 | .012±.003 | .016±.003 | .115±.008 | .012±.003 | **.010±.003** | .034±.005 | .056±.006 | .079±.006 | .019±.004 |
| | .85 | .028±.005 | .012±.003 | .007±.002 | .014±.003 | .113±.008 | .007±.002 | .012±.003 | .008±.002 | .013±.003 | .093±.007 | **.006±.002** | **.006±.002** | .029±.005 | .056±.006 | .074±.006 | .013±.003 |
| | .80 | .024±.004 | .012±.003 | .005±.002 | .012±.003 | .094±.007 | .007±.002 | .012±.003 | .005±.002 | .013±.003 | .139±.009 | **.003±.001** | .005±.002 | .022±.004 | .051±.006 | .072±.007 | .007±.002 |
| | .75 | .020±.004 | .011±.003 | .004±.001 | .007±.002 | .087±.008 | **.003±.001** | .010±.003 | .004±.001 | .006±.002 | .252±.011 | **.003±.001** | **.003±.001** | .018±.004 | .044±.006 | .066±.006 | .006±.002 |
| | .70 | .016±.004 | .011±.003 | **.002±.001** | .005±.002 | .068±.007 | .003±.002 | .010±.003 | .003±.001 | .005±.002 | .237±.011 | **.003±.001** | **.002±.001** | .016±.004 | .043±.006 | .061±.006 | .004±.002 |
| $\hat{\phi}$ | .99 | .992±.002 | **.990±.002** | .992±.002 | .992±.002 | .980±.003 | .982±.003 | .988±.003 | **.991±.002** | .986±.003 | .992±.002 | .985±.003 | .992±.002 | .979±.003 | .994±.002 | **.991±.002** | .992±.002 |
| | .95 | .939±.006 | .962±.004 | .944±.005 | .953±.005 | .952±.004 | .941±.005 | .950±.006 | .941±.005 | .952±.006 | .944±.006 | .935±.005 | .931±.006 | .933±.006 | **.950±.005** | **.949±.005** | .957±.005 |
| | .90 | .877±.008 | .907±.007 | .884±.008 | .896±.007 | **.899±.006** | .885±.008 | .900±.007 | .891±.008 | .890±.008 | .888±.008 | .889±.008 | .882±.008 | .892±.007 | .908±.007 | **.901±.007** | .919±.006 |
| | .85 | .828±.010 | .842±.008 | .848±.008 | .832±.008 | .858±.008 | .831±.010 | .836±.009 | .838±.009 | .838±.009 | **.849±.008** | .826±.010 | .833±.010 | .843±.009 | .852±.009 | .825±.009 | .869±.008 |
| | .80 | .782±.010 | .796±.010 | .792±.010 | .805±.009 | **.802±.009** | .784±.011 | .784±.010 | .785±.010 | .809±.010 | .791±.010 | .777±.010 | .787±.010 | .785±.010 | .789±.010 | .759±.011 | .823±.010 |
| | .75 | .735±.011 | .739±.011 | .741±.010 | .741±.010 | **.751±.010** | .730±.011 | .744±.011 | .743±.011 | .739±.010 | .747±.012 | .728±.010 | .735±.011 | .719±.011 | .736±.011 | .705±.012 | .795±.011 |
| | .70 | .674±.012 | .705±.011 | .683±.012 | .692±.011 | .683±.011 | .689±.012 | .688±.011 | .687±.012 | **.700±.011** | .693±.012 | .688±.011 | .671±.011 | .673±.011 | .689±.011 | .669±.013 | .759±.012 |

Table B21: Results for `jannis`: *mean ± std* for $\widehat{Err}$, empirical coverage $\hat{\phi}$, and *MinCoeff*.

| Metric | c | DG | SAT | SAT+EM | SelNet | SelNet+EM | SR | SAT+SR | SAT+EM+SR | SelNet+SR | SelNet+EM+SR | ENS | ENS+SR | ConfidNet | SELE | REG | SCross | AUCross | PlugInAUC |
|---|---|---|---|---|---|---|---|---|---|---|---|---|---|---|---|---|---|---|---|
| $\widehat{Err}$ | .99 | .291±.004 | .207±.004 | .204±.004 | .205±.004 | .203±.005 | .208±.004 | .207±.004 | .204±.004 | .205±.004 | .202±.004 | .199±.003 | **.188±.004** | .210±.004 | .225±.004 | .202±.003 | .203±.003 | .210±.004 |
| | .95 | .284±.004 | .198±.004 | .196±.004 | .191±.004 | .195±.003 | .197±.003 | .195±.004 | .191±.004 | **.188±.004** | .196±.004 | .189±.004 | .200±.004 | .218±.004 | .223±.004 | .189±.003 | .192±.003 | .199±.004 |
| | .90 | .277±.004 | .186±.004 | .181±.004 | .180±.003 | .185±.004 | .184±.004 | .180±.004 | .178±.004 | .178±.003 | .182±.004 | .191±.004 | .177±.004 | .189±.004 | .214±.004 | .222±.004 | **.176±.003** | **.176±.003** | .186±.003 |
| | .85 | .267±.004 | .172±.004 | .169±.004 | .166±.004 | .168±.004 | .167±.004 | .167±.004 | .165±.004 | .166±.004 | .172±.004 | .181±.004 | .165±.004 | .177±.003 | .210±.004 | .221±.004 | **.163±.003** | .164±.003 | .168±.004 |
| | .80 | .261±.004 | .159±.004 | .157±.004 | .151±.003 | .159±.004 | .153±.004 | .155±.004 | .152±.004 | **.150±.004** | .159±.003 | .174±.004 | **.150±.004** | .166±.003 | .210±.004 | .221±.004 | .151±.003 | .153±.003 | .155±.004 |
| | .75 | .253±.004 | .145±.004 | .141±.004 | .142±.003 | .151±.004 | .142±.004 | .141±.004 | .138±.004 | .139±.003 | .150±.004 | .164±.004 | **.134±.004** | .154±.004 | .207±.004 | .220±.004 | .138±.004 | .142±.004 | .144±.004 |
| | .70 | .244±.004 | .134±.004 | .128±.003 | .128±.004 | .144±.004 | .130±.004 | .133±.004 | .127±.004 | .127±.004 | .143±.004 | .156±.004 | **.124±.004** | .143±.004 | .203±.004 | .218±.004 | .128±.004 | .134±.004 | .136±.004 |
| $\hat{\phi}$ | .99 | .986±.001 | .993±.001 | **.991±.001** | .989±.001 | .988±.001 | .988±.001 | .988±.001 | **.990±.001** | .993±.001 | .988±.001 | .988±.001 | **.991±.001** | .986±.001 | .992±.001 | **.989±.001** | **.989±.001** | **.990±.001** | **.990±.001** | .992±.001 |
| | .95 | .943±.002 | .953±.002 | .954±.002 | .948±.002 | .946±.002 | .952±.002 | .952±.002 | .952±.002 | .948±.002 | .948±.002 | .947±.002 | .947±.002 | **.950±.002** | .947±.002 | .945±.002 | .948±.002 | .950±.002 | .955±.002 |
| | .90 | .893±.003 | .908±.003 | .903±.003 | .903±.003 | .907±.003 | .907±.003 | .907±.003 | **.900±.003** | .908±.003 | .905±.003 | .903±.003 | **.901±.003** | **.900±.003** | .899±.003 | .893±.003 | **.899±.003** | **.900±.003** | .907±.003 |
| | .85 | .845±.003 | .857±.003 | .855±.004 | .857±.003 | .859±.004 | .858±.003 | .851±.004 | .853±.003 | .860±.004 | .853±.004 | .853±.004 | .839±.003 | .856±.005 | .851±.003 | .848±.004 | .846±.003 | .851±.004 | .854±.003 |
| | .80 | .792±.004 | .802±.004 | .807±.004 | .807±.003 | .808±.004 | .800±.004 | .803±.004 | .805±.004 | .805±.003 | .803±.004 | .795±.004 | .803±.004 | .807±.003 | .800±.004 | .797±.004 | **.799±.004** | .803±.004 | .797±.004 |
| | .75 | .744±.005 | .752±.005 | .754±.004 | .755±.004 | .747±.004 | .753±.004 | .751±.004 | **.751±.004** | **.751±.004** | .748±.005 | .738±.004 | .748±.004 | .755±.004 | .751±.004 | .748±.004 | .748±.004 | .750±.004 | .748±.004 |
| | .70 | .691±.005 | .701±.005 | .701±.004 | .702±.004 | .702±.005 | .702±.004 | .703±.005 | **.700±.004** | .701±.005 | .701±.005 | .700±.004 | .698±.004 | .703±.004 | .703±.004 | .696±.004 | .695±.004 | .701±.004 | .703±.004 |
| *MinCoeff* | .99 | .997±.010 | **.999±.010** | .997±.010 | 1.000±.010 | 1.001±.010 | 1.001±.010 | .999±.010 | .998±.010 | .999±.010 | .999±.010 | .999±.010 | .998±.010 | .997±.010 | .999±.010 | 1.002±.010 | 1.001±.010 | 1.001±.010 | 1.000±.010 |
| | .95 | .991±.010 | .996±.010 | .996±.010 | **1.001±.010** | 1.002±.010 | 1.001±.010 | .996±.010 | .996±.010 | **.999±.010** | 1.002±.011 | .998±.011 | 1.000±.010 | .994±.010 | .996±.010 | 1.005±.010 | **1.001±.010** | **1.003±.010** | 1.005±.010 |
| | .90 | .983±.010 | .996±.011 | .995±.011 | 1.002±.010 | .991±.011 | 1.007±.010 | .993±.010 | .996±.011 | 1.004±.011 | 1.006±.010 | 1.007±.011 | 1.005±.011 | .994±.010 | .986±.010 | 1.018±.011 | **.999±.010** | 1.011±.010 | 1.009±.011 |
| | .85 | .955±.011 | .987±.012 | .985±.011 | .979±.011 | **.998±.010** | 1.010±.011 | .987±.011 | .980±.011 | .986±.011 | .992±.010 | 1.008±.011 | 1.002±.011 | .987±.011 | .957±.011 | 1.040±.011 | .984±.011 | 1.025±.011 | 1.029±.011 |
| | .75 | .937±.011 | .984±.011 | .972±.011 | **.999±.011** | .972±.011 | 1.012±.011 | .982±.011 | .968±.011 | 1.002±.011 | .987±.011 | 1.007±.012 | .997±.011 | .984±.011 | .948±.011 | 1.057±.011 | .973±.011 | 1.034±.011 | 1.040±.011 |
| | .70 | .914±.011 | .977±.012 | .959±.011 | .979±.011 | 1.011±.012 | **1.005±.011** | .974±.012 | .953±.011 | .988±.011 | 1.010±.012 | 1.007±.012 | .987±.011 | .975±.012 | .937±.011 | 1.073±.012 | .963±.011 | 1.044±.011 | 1.054±.012 |

Table B22: Results for `kddipums97`: *mean ± std* for $\widehat{Err}$, empirical coverage $\hat{\phi}$, and *MinCoeff*.

| Metric | c | DG | SAT | SAT+EM | SelNet | SelNet+EM | SR | SAT+SR | SAT+EM+SR | SelNet+SR | SelNet+EM+SR | ENS | ENS+SR | ConfidNet | SELE | REG | SCross | AUCross | PlugInAUC |
|---|---|---|---|---|---|---|---|---|---|---|---|---|---|---|---|---|---|---|---|
| $\widehat{Err}$ | .99 | .123±.010 | .154±.011 | .130±.011 | .136±.011 | .148±.011 | .137±.010 | .150±.011 | .130±.011 | .115±.010 | .145±.011 | .137±.010 | .138±.010 | **.119±.010** | .151±.011 | .143±.011 | .145±.011 | .136±.010 |
| | .95 | .121±.010 | .138±.011 | .120±.011 | .121±.010 | .137±.010 | .119±.010 | .129±.010 | .111±.010 | .115±.010 | .136±.011 | .129±.010 | .130±.010 | **.109±.010** | .146±.011 | .147±.012 | .115±.011 | .118±.011 | .124±.010 |
| | .90 | .110±.009 | .113±.011 | .101±.010 | .093±.010 | .114±.011 | .104±.009 | .111±.010 | .091±.010 | .091±.010 | .117±.011 | .120±.010 | .117±.010 | .103±.010 | .136±.010 | .147±.012 | **.090±.010** | **.089±.010** | .106±.010 |
| | .85 | .085±.009 | .096±.010 | .086±.010 | .080±.010 | .102±.011 | .087±.010 | .097±.010 | .090±.010 | .075±.009 | .092±.010 | .117±.011 | .098±.010 | .083±.009 | .131±.010 | .118±.012 | .063±.009 | **.061±.009** | .093±.009 |
| | .80 | .058±.008 | .081±.009 | .058±.008 | .067±.010 | .090±.009 | .078±.009 | .085±.010 | .060±.008 | .068±.010 | .095±.010 | .105±.010 | .082±.010 | .061±.008 | .118±.011 | .149±.012 | .054±.009 | **.040±.007** | .075±.009 |
| | .75 | .048±.007 | .073±.009 | .051±.008 | .077±.009 | .077±.009 | .061±.009 | .078±.010 | .052±.008 | .067±.009 | .077±.009 | .098±.010 | .068±.010 | .050±.008 | .111±.011 | .144±.012 | .045±.008 | **.040±.007** | .067±.009 |
| | .70 | .041±.007 | .056±.008 | .043±.007 | .059±.009 | .071±.010 | .045±.008 | .064±.009 | .043±.007 | .046±.008 | .074±.010 | .088±.010 | .052±.009 | .041±.008 | .108±.011 | .140±.012 | **.031±.007** | .038±.007 | .057±.008 |
| $\hat{\phi}$ | .99 | .992±.003 | .993±.002 | .997±.002 | .992±.003 | .994±.002 | .992±.003 | **.991±.003** | .989±.003 | .983±.004 | .988±.003 | .991±.003 | .991±.003 | .984±.004 | .988±.004 | .967±.005 | .978±.004 | **.989±.003** |
| | .95 | .961±.006 | .946±.007 | .955±.006 | .958±.007 | .947±.006 | .938±.007 | **.949±.006** | .934±.008 | .942±.007 | .952±.007 | .955±.007 | .959±.006 | .941±.007 | .958±.006 | **.948±.007** | .934±.010 | .895±.010 | .961±.006 |
| | .90 | .926±.008 | .881±.010 | .901±.010 | .881±.011 | .895±.009 | .893±.008 | .885±.010 | .895±.009 | .895±.010 | .912±.008 | .902±.010 | .912±.008 | .914±.009 | .907±.010 | .894±.009 | .817±.012 | .816±.012 | **.898±.008** |
| | .85 | .852±.011 | .833±.012 | .850±.011 | .830±.011 | .849±.011 | .849±.010 | .847±.011 | .863±.011 | .822±.012 | .849±.011 | .867±.012 | .858±.009 | .847±.012 | .834±.012 | .750±.012 | .748±.012 | .858±.009 |
| | .80 | .788±.013 | .798±.013 | .782±.013 | .778±.013 | .793±.012 | .812±.012 | .800±.012 | .786±.013 | .776±.012 | .807±.012 | .832±.013 | .805±.011 | **.801±.013** | .812±.014 | .793±.013 | .708±.013 | .694±.013 | .813±.011 |
| | .75 | .740±.013 | .766±.013 | .737±.013 | .773±.013 | .773±.013 | **.750±.012** | .757±.013 | .773±.013 | .744±.013 | .754±.014 | .743±.013 | .777±.014 | .769±.013 | .746±.014 | .762±.014 | .675±.013 | .675±.013 | .767±.013 |
| | .70 | .691±.014 | .717±.013 | .701±.013 | **.700±.013** | .692±.014 | .701±.013 | .733±.013 | .699±.013 | .706±.013 | .715±.014 | .731±.014 | .725±.013 | .693±.014 | .713±.015 | .707±.016 | .631±.014 | .657±.013 | .705±.014 |
| *MinCoeff* | .99 | 1.012±.030 | 1.010±.030 | 1.007±.030 | 1.004±.030 | 1.010±.030 | 1.006±.030 | 1.010±.029 | 1.007±.030 | 1.004±.030 | 1.002±.030 | 1.006±.030 | **1.007±.030** | 1.015±.029 | **1.000±.030** | .992±.030 | .997±.031 | 1.007±.031 | 1.005±.030 |
| | .95 | 1.030±.030 | 1.007±.030 | .992±.030 | **1.002±.029** | 1.003±.030 | .997±.031 | 1.017±.030 | 1.004±.030 | 1.001±.030 | 1.001±.031 | 1.018±.031 | 1.028±.030 | .997±.030 | .957±.031 | .983±.032 | .973±.031 | 1.004±.030 |
| | .90 | 1.032±.031 | 1.006±.032 | 1.011±.031 | 1.010±.031 | 1.003±.032 | **.998±.031** | .996±.031 | 1.019±.031 | 1.007±.031 | 1.011±.031 | .965±.032 | 1.016±.031 | 1.045±.031 | .980±.031 | .912±.031 | .947±.032 | .953±.032 | 1.005±.030 |
| | .85 | 1.029±.031 | .990±.033 | 1.015±.032 | .978±.033 | 1.001±.032 | .988±.033 | 1.026±.031 | .991±.031 | .962±.033 | **1.006±.031** | 1.028±.031 | .967±.032 | .868±.032 | .923±.034 | .945±.034 | 1.007±.031 |
| | .80 | 1.019±.032 | .983±.034 | 1.013±.032 | .991±.033 | .952±.034 | .987±.032 | .970±.033 | 1.013±.032 | .970±.034 | .953±.034 | .935±.035 | .978±.033 | 1.028±.031 | .927±.034 | .839±.034 | .865±.036 | .938±.035 | **1.011±.032** |
| | .75 | .982±.033 | .963±.034 | .974±.034 | .964±.034 | .937±.035 | .977±.035 | .966±.034 | .957±.035 | .975±.034 | .928±.036 | .900±.036 | .746±.014 | .987±.034 | .884±.034 | .800±.034 | .696±.035 | .911±.036 | **1.009±.033** |
| | .70 | .932±.035 | .938±.035 | .942±.033 | .908±.036 | .873±.038 | .932±.035 | .933±.035 | .939±.034 | .943±.034 | .899±.035 | .869±.037 | **.944±.035** | .936±.035 | .828±.035 | .772±.033 | .845±.037 | .887±.037 | 1.065±.033 |

Table B23: Results for `letter`: $mean \pm std$ for $\widehat{Err}$ and empirical coverage $\hat{\phi}$.

| Metric | c | DG | SAT | SAT+EM | SelNet | SelNet+EM | SR | SAT+SR | SAT+EM+SR | SelNet+SR | SelNet+EM+SR | ENS | ENS+SR | ConfidNet | SELE | REG | SCross |
|---|---|---|---|---|---|---|---|---|---|---|---|---|---|---|---|---|---|
| $\widehat{Err}$ | .99 | .122±.005 | .016±.002 | .014±.002 | .017±.002 | .017±.002 | .026±.002 | .015±.002 | **.012±.002** | .014±.002 | .014±.002 | **.012±.002** | **.012±.002** | .033±.003 | .061±.004 | .048±.003 | .020±.002 |
| | .95 | .105±.005 | .006±.001 | **.005±.001** | .013±.002 | .012±.002 | .014±.002 | .006±.001 | **.005±.001** | .006±.001 | .006±.001 | **.005±.001** | **.005±.001** | .028±.003 | .058±.004 | .047±.003 | .009±.002 |
| | .90 | .084±.005 | **.002±.001** | **.002±.001** | .007±.001 | .009±.002 | .006±.001 | **.002±.001** | **.002±.001** | **.002±.001** | **.002±.001** | **.002±.001** | **.002±.001** | .022±.003 | .048±.004 | .047±.004 | .005±.001 |
| | .85 | .070±.005 | .001±.000 | .002±.001 | .007±.001 | .007±.001 | .003±.001 | **.000±.000** | .001±.001 | **.000±.000** | .001±.001 | .001±.001 | .001±.001 | .019±.002 | .041±.003 | .046±.004 | .002±.001 |
| | .80 | .053±.004 | **.000±.000** | .002±.001 | .004±.001 | .006±.001 | .002±.001 | **.000±.000** | **.000±.000** | .001±.001 | .001±.001 | .001±.001 | **.000±.000** | .015±.002 | .039±.003 | .044±.004 | .001±.001 |
| | .75 | .042±.004 | **.000±.000** | .001±.001 | .003±.001 | .003±.001 | .001±.001 | **.000±.000** | .001±.001 | **.000±.000** | .001±.001 | **.000±.000** | **.000±.000** | .014±.002 | .039±.003 | .044±.004 | .001±.001 |
| | .70 | .037±.004 | **.000±.000** | .001±.001 | .003±.001 | .003±.001 | .001±.001 | **.000±.000** | **.000±.000** | .001±.001 | **.000±.000** | **.000±.000** | **.000±.000** | .012±.002 | .038±.003 | .042±.004 | .001±.000 |
| $\hat{\phi}$ | .99 | .991±.001 | .994±.001 | .995±.001 | **.990±.002** | .993±.001 | .994±.001 | .992±.002 | .993±.001 | .986±.002 | **.990±.002** | .993±.001 | .993±.001 | .993±.001 | **.990±.002** | .992±.002 | .991±.001 |
| | .95 | .955±.003 | .957±.003 | .953±.004 | .956±.003 | **.950±.003** | .957±.003 | .954±.003 | .954±.003 | .949±.004 | .949±.003 | .951±.003 | .953±.003 | .949±.004 | .954±.003 | .939±.004 | .959±.003 |
| | .90 | .904±.004 | **.903±.005** | **.903±.005** | .911±.004 | .908±.004 | .907±.005 | .905±.005 | .905±.005 | **.901±.005** | **.902±.005** | **.902±.005** | **.903±.005** | **.901±.005** | .907±.005 | .890±.005 | .923±.005 |
| | .85 | .860±.006 | .857±.006 | .848±.006 | .861±.005 | .853±.006 | .855±.006 | .860±.006 | .854±.006 | .851±.007 | .855±.006 | .854±.006 | .863±.005 | **.849±.005** | .842±.005 | .882±.005 | |
| | .80 | .807±.006 | .811±.006 | **.801±.006** | .806±.006 | .810±.006 | **.799±.007** | .810±.006 | .805±.006 | .798±.007 | .797±.007 | .811±.006 | .808±.006 | **.803±.006** | .807±.006 | .789±.006 | .838±.006 |
| | .75 | .756±.007 | .757±.007 | .751±.007 | .745±.007 | .751±.007 | .755±.007 | .764±.007 | **.749±.007** | .759±.007 | .753±.007 | .763±.007 | .759±.007 | **.751±.007** | .766±.007 | .736±.006 | .793±.006 |
| | .70 | .707±.008 | .698±.007 | .703±.008 | .701±.007 | .711±.008 | .698±.008 | .702±.007 | .697±.007 | .708±.008 | .712±.008 | .716±.008 | .715±.007 | **.701±.007** | .710±.007 | .690±.004 | .747±.007 |

Table B24: Results for `magic`: $mean \pm std$ for $\widehat{Err}$, empirical coverage $\hat{\phi}$, and $MinCoeff$.

| Metric | c | DG | SAT | SAT+EM | SelNet | SelNet+EM | SR | SAT+SR | SAT+EM+SR | SelNet+SR | SelNet+EM+SR | ENS | ENS+SR | ConfidNet | SELE | REG | SCross | AUCross | PlugInAUC |
|---|---|---|---|---|---|---|---|---|---|---|---|---|---|---|---|---|---|---|---|
| $\widehat{Err}$ | .99 | .225±.008 | .142±.007 | .155±.007 | .143±.007 | .148±.007 | .155±.008 | .143±.007 | .153±.007 | **.141±.007** | .147±.008 | .153±.008 | .149±.007 | .147±.008 | .157±.007 | .158±.008 | .159±.008 | .155±.008 |
| | .95 | .217±.008 | **.127±.007** | .141±.007 | .129±.007 | .144±.007 | .136±.007 | .128±.007 | .142±.007 | **.127±.007** | .142±.007 | .147±.007 | .134±.007 | .132±.007 | .147±.008 | .153±.007 | .141±.007 | .153±.008 |
| | .90 | .203±.008 | .109±.007 | .123±.007 | .112±.007 | .125±.007 | .118±.007 | .110±.007 | .123±.007 | **.106±.007** | .125±.007 | .140±.007 | .118±.007 | .120±.007 | .142±.008 | .144±.007 | .125±.007 | .147±.008 |
| | .85 | .186±.008 | .099±.007 | .108±.006 | .099±.007 | .121±.007 | .112±.007 | **.097±.007** | .106±.006 | .098±.007 | .121±.007 | .134±.007 | .105±.007 | .107±.006 | .134±.008 | .139±.007 | .114±.007 | .137±.008 |
| | .80 | .170±.008 | .087±.006 | .092±.006 | **.085±.006** | .100±.006 | .098±.006 | .086±.006 | .094±.006 | .086±.006 | .099±.006 | .129±.007 | .095±.006 | .099±.006 | .120±.008 | .130±.007 | .101±.007 | .133±.008 |
| | .75 | .156±.007 | .076±.006 | .084±.006 | **.071±.006** | .090±.006 | .088±.006 | .075±.006 | .083±.006 | .090±.006 | .119±.007 | .086±.006 | .086±.006 | .112±.007 | .124±.007 | .089±.006 | .108±.008 | .116±.008 |
| | .70 | .144±.008 | .072±.006 | .072±.006 | **.069±.006** | .086±.005 | .075±.006 | **.069±.006** | .070±.006 | .073±.006 | .087±.006 | .105±.007 | .072±.005 | .072±.006 | .105±.008 | .119±.007 | .078±.006 | .103±.008 |
| $\hat{\phi}$ | .99 | .986±.003 | .981±.002 | .995±.001 | **.990±.002** | .988±.002 | .997±.001 | .987±.002 | .992±.002 | .989±.002 | .990±.002 | .992±.002 | .984±.002 | .985±.002 | .990±.002 | .986±.002 | .992±.002 | .991±.002 |
| | .95 | .954±.004 | .936±.005 | .955±.005 | .945±.004 | .959±.004 | .945±.004 | .942±.005 | .959±.004 | .951±.005 | .952±.004 | .953±.005 | .941±.005 | .936±.005 | .957±.004 | .956±.004 | **.951±.004** | .958±.004 |
| | .90 | .904±.006 | .860±.006 | .891±.007 | **.896±.006** | **.897±.006** | .883±.006 | .890±.006 | .892±.006 | .887±.007 | **.898±.006** | .915±.006 | .885±.006 | .889±.006 | .909±.006 | .887±.005 | .904±.006 | .907±.006 |
| | .85 | **.849±.006** | .843±.007 | .841±.008 | .845±.007 | .855±.007 | .839±.007 | .839±.007 | .836±.007 | .846±.007 | .851±.007 | .870±.007 | .837±.007 | .841±.007 | .857±.007 | .845±.007 | .860±.007 | .870±.007 |
| | .80 | .798±.007 | .783±.008 | .791±.008 | .788±.008 | .808±.007 | .793±.008 | .778±.008 | .797±.008 | .790±.009 | .809±.008 | .819±.009 | .793±.008 | **.799±.007** | **.800±.007** | .805±.007 | .805±.007 | .817±.008 | .822±.007 |
| | .75 | **.749±.008** | .724±.009 | .744±.008 | .731±.008 | .762±.009 | .736±.009 | .734±.009 | .744±.008 | .734±.009 | .760±.009 | .758±.009 | .745±.009 | .747±.008 | .752±.008 | .760±.008 | .762±.008 | .764±.009 | .768±.008 |
| | .70 | **.698±.008** | .687±.010 | **.698±.009** | .688±.009 | .700±.009 | .693±.010 | .686±.009 | .697±.009 | .708±.008 | .705±.009 | .700±.009 | .694±.009 | .704±.009 | .701±.009 | .719±.009 | .718±.009 | .714±.010 | .719±.009 |
| $MinCoeff$ | .99 | 1.001±.019 | .996±.019 | **.997±.019** | 1.001±.019 | .996±.019 | .997±.019 | **.999±.019** | .999±.019 | .999±.019 | .998±.019 | .999±.019 | .994±.019 | 1.001±.019 | .999±.019 | .995±.019 | **.998±.019** | 1.002±.019 | 1.003±.019 |
| | .95 | 1.007±.020 | .989±.019 | **1.000±.019** | 1.002±.019 | .996±.019 | .999±.019 | .999±.019 | **.998±.019** | .999±.019 | .997±.019 | .990±.020 | .998±.019 | .995±.019 | 1.013±.020 | 1.000±.019 | .992±.020 | 1.014±.020 | 1.029±.020 |
| | .90 | 1.017±.020 | .994±.020 | **.998±.019** | .999±.019 | .998±.020 | 1.009±.020 | .990±.020 | **.998±.019** | .994±.020 | 1.001±.019 | .984±.020 | **1.003±.020** | .994±.020 | 1.034±.021 | 1.017±.020 | .989±.020 | 1.042±.021 | 1.046±.021 |
| | .85 | 1.023±.021 | .997±.022 | **.995±.021** | 1.010±.021 | 1.003±.021 | 1.027±.021 | .999±.021 | .995±.021 | 1.009±.020 | 1.005±.021 | .982±.021 | 1.013±.021 | 1.047±.022 | 1.055±.021 | 1.024±.020 | .985±.021 | 1.060±.022 | 1.073±.021 |
| | .80 | 1.034±.022 | 1.010±.023 | **.999±.022** | 1.017±.022 | 1.017±.021 | 1.034±.022 | 1.009±.023 | 1.005±.022 | 1.016±.021 | .972±.022 | 1.031±.022 | 1.024±.022 | 1.066±.021 | 1.033±.021 | **1.002±.022** | 1.080±.022 | 1.101±.022 |
| | .75 | 1.044±.022 | 1.023±.024 | 1.014±.024 | 1.034±.024 | 1.027±.023 | 1.057±.023 | 1.018±.023 | 1.024±.023 | 1.035±.023 | 1.024±.023 | .966±.023 | 1.040±.023 | 1.049±.023 | 1.077±.021 | 1.040±.021 | **1.002±.023** | 1.104±.022 | 1.126±.023 |
| | .70 | 1.046±.022 | 1.044±.024 | 1.038±.025 | 1.037±.024 | 1.047±.024 | 1.065±.023 | 1.048±.024 | 1.040±.024 | 1.026±.023 | 1.040±.024 | .959±.023 | 1.047±.024 | 1.060±.024 | 1.080±.022 | 1.036±.022 | **1.007±.024** | 1.132±.023 | 1.161±.023 |

Table B25: Results for `miniboone`: $mean \pm std$ for $\widehat{Err}$, empirical coverage $\hat{\phi}$, and $MinCoeff$.

| Metric | c | DG | SAT | SAT+EM | SelNet | SelNet+EM | SR | SAT+SR | SAT+EM+SR | SelNet+SR | SelNet+EM+SR | ENS | ENS+SR | ConfidNet | SELE | REG | SCross | AUCross | PlugInAUC |
|---|---|---|---|---|---|---|---|---|---|---|---|---|---|---|---|---|---|---|---|
| $\widehat{Err}$ | .99 | .065±.002 | .061±.002 | **.057±.002** | .059±.002 | .066±.002 | .061±.002 | .060±.002 | **.044±.002** | .044±.002 | .046±.002 | .067±.002 | .062±.002 | .060±.002 | .067±.002 | .075±.002 | .073±.002 | .064±.002 | .068±.002 | .062±.002 |
| | .95 | .055±.002 | .049±.002 | .047±.002 | .047±.002 | .052±.002 | .046±.002 | **.044±.002** | **.044±.002** | .046±.002 | .050±.002 | .051±.002 | .046±.002 | .043±.002 | .052±.002 | .071±.002 | .074±.002 | .062±.002 | .066±.002 | .048±.002 |
| | .90 | .047±.002 | .036±.002 | .033±.002 | .033±.002 | .038±.002 | .033±.002 | **.032±.002** | **.032±.002** | .033±.002 | .035±.002 | .040±.002 | .034±.002 | .038±.002 | .067±.002 | .073±.002 | .036±.002 | .064±.002 | .036±.002 |
| | .85 | .041±.002 | .024±.001 | .021±.001 | .024±.001 | .027±.001 | .022±.001 | .023±.001 | **.021±.001** | .024±.001 | .029±.001 | .031±.002 | .023±.001 | .027±.001 | .065±.002 | .071±.002 | .025±.001 | .058±.002 | .029±.002 |
| | .80 | .036±.002 | .017±.001 | .017±.001 | .018±.001 | .020±.001 | .018±.001 | **.016±.001** | **.016±.001** | .018±.001 | .020±.001 | .023±.001 | .017±.001 | .021±.001 | .062±.002 | .070±.002 | .019±.001 | .049±.002 | .022±.001 |
| | .75 | .030±.002 | .013±.001 | **.012±.001** | .014±.001 | .017±.001 | .012±.001 | **.012±.001** | **.012±.001** | .013±.001 | .016±.001 | .018±.001 | .013±.001 | .015±.001 | .060±.002 | .069±.002 | .013±.001 | .037±.002 | .017±.001 |
| | .70 | .027±.001 | .010±.001 | **.009±.001** | .010±.001 | .012±.001 | .010±.001 | **.009±.001** | **.009±.001** | .010±.001 | .013±.001 | .014±.001 | .010±.001 | .012±.001 | .059±.002 | .069±.003 | .010±.001 | .029±.002 | .015±.001 |
| $\hat{\phi}$ | .99 | .989±.001 | .987±.001 | .988±.001 | .988±.001 | .988±.001 | .991±.001 | .989±.001 | .990±.001 | .991±.001 | .992±.001 | .990±.001 | .990±.001 | .986±.001 | .986±.001 | .991±.001 | .989±.001 | .994±.001 | **.990±.001** |
| | .95 | .946±.002 | .947±.002 | .952±.002 | **.950±.002** | .946±.002 | .947±.002 | .947±.002 | **.950±.002** | .951±.002 | .947±.002 | .946±.002 | .950±.002 | .947±.002 | .945±.002 | .949±.002 | .946±.002 | .971±.001 | .946±.002 |
| | .90 | .895±.003 | .897±.003 | .895±.003 | .894±.003 | .895±.003 | .896±.003 | .897±.003 | .896±.003 | .897±.003 | .894±.003 | .897±.003 | **.900±.003** | **.899±.003** | .898±.003 | **.899±.003** | .892±.003 | .939±.002 | .894±.003 |
| | .85 | .847±.003 | .843±.003 | .845±.003 | .843±.003 | .839±.003 | .840±.003 | .845±.003 | .845±.003 | .845±.003 | .840±.003 | .845±.003 | **.850±.003** | .845±.003 | .850±.003 | .849±.003 | .837±.003 | .904±.002 | .843±.003 |
| | .80 | **.798±.003** | .790±.003 | **.798±.003** | .790±.003 | .790±.003 | .792±.004 | .790±.004 | .795±.003 | .791±.003 | .791±.003 | .793±.003 | **.800±.003** | .794±.004 | **.800±.003** | .798±.003 | .786±.003 | .864±.003 | .790±.004 |
| | .75 | .740±.004 | .741±.004 | .743±.004 | .735±.004 | .740±.004 | .741±.004 | .738±.004 | .746±.004 | .734±.004 | .738±.004 | .742±.004 | **.750±.004** | .740±.004 | **.750±.004** | .729±.004 | .820±.003 | .734±.004 |
| | .70 | .692±.004 | .692±.004 | .693±.004 | .686±.004 | .689±.004 | .692±.004 | .689±.004 | .691±.004 | .688±.004 | .686±.004 | .693±.004 | **.700±.004** | .691±.004 | .697±.004 | .704±.003 | .670±.004 | .771±.004 | .685±.004 |
| $MinCoeff$ | .99 | 1.000±.008 | 1.000±.008 | .999±.008 | 1.000±.008 | .999±.008 | .999±.008 | .999±.008 | .999±.008 | .999±.008 | 1.000±.008 | 1.001±.008 | .999±.008 | 1.000±.008 | .996±.008 | .995±.008 | **.999±.008** | 1.005±.008 | 1.004±.008 |
| | .95 | .990±.008 | .993±.008 | .990±.008 | .994±.008 | .996±.008 | .990±.009 | .993±.008 | .991±.008 | .996±.008 | .994±.008 | 1.005±.008 | .996±.009 | **.997±.008** | .992±.009 | .964±.008 | .988±.009 | 1.022±.009 | 1.016±.008 |
| | .90 | .978±.009 | .982±.009 | .972±.008 | .983±.008 | .984±.009 | .981±.009 | .982±.009 | .980±.009 | .986±.008 | .984±.009 | 1.006±.009 | .985±.009 | .991±.009 | **.997±.009** | .956±.009 | .972±.009 | 1.048±.009 | 1.032±.008 |
| | .85 | .954±.009 | .968±.009 | .955±.009 | .973±.009 | .976±.008 | .946±.009 | .969±.009 | .961±.009 | .978±.009 | .987±.009 | 1.002±.009 | .973±.009 | .989±.009 | .967±.009 | .946±.009 | 1.072±.009 | 1.044±.008 |
| | .80 | 1.034±.022 | 1.010±.023 | **.999±.022** | .936±.010 | .952±.009 | .943±.009 | .948±.009 | .944±.009 | .938±.009 | .952±.009 | .954±.009 | **.995±.009** | .957±.009 | .961±.009 | .991±.010 | .983±.009 | .916±.009 | 1.097±.009 | 1.060±.009 |
| | .75 | .879±.009 | .925±.009 | .904±.010 | .917±.009 | .916±.009 | .922±.009 | .912±.009 | .907±.010 | .918±.009 | .924±.009 | .988±.010 | .931±.009 | .935±.009 | .993±.010 | **1.000±.009** | .877±.010 | 1.118±.009 | 1.077±.009 |
| | .70 | .830±.009 | .892±.009 | .869±.010 | .879±.010 | .891±.010 | .890±.009 | .883±.009 | .873±.010 | .874±.010 | .894±.010 | **.989±.010** | .905±.010 | .904±.010 | .985±.011 | 1.014±.009 | .824±.010 | 1.133±.008 | 1.089±.010 |

Table B26: Results for `MNIST`: $mean \pm std$ for $\widehat{Err}$ and empirical coverage $\hat{\phi}$.

| Metric | c | DG | SAT | SAT+EM | SelNet | SelNet+EM | SR | SAT+SR | SAT+EM+SR | SelNet+SR | SelNet+EM+SR | ENS | ENS+SR | ConfidNet | SELE | REG | SCross |
|---|---|---|---|---|---|---|---|---|---|---|---|---|---|---|---|---|---|
| $\widehat{Err}$ | .99 | .005±.001 | .009±.001 | .010±.001 | **.002±.000** | .006±.001 | .016±.001 | .003±.000 | .007±.001 | .008±.001 | .008±.001 | .012±.001 | .003±.000 | **.002±.000** | .007±.001 | .010±.001 | .013±.001 | .004±.000 |
| | .95 | .001±.000 | .004±.001 | .003±.000 | **.001±.000** | .002±.000 | .001±.000 | .002±.000 | .002±.000 | **.001±.000** | .001±.000 | **.001±.000** | **.001±.000** | .002±.000 | .010±.001 | .013±.001 | **.001±.000** |
| | .90 | **.000±.000** | .003±.000 | .001±.000 | **.000±.000** | .001±.000 | **.000±.000** | .001±.000 | .001±.000 | **.000±.000** | **.000±.000** | .001±.000 | **.000±.000** | **.000±.000** | .001±.000 | .009±.001 | .013±.001 | **.000±.000** |
| | .85 | **.000±.000** | .002±.000 | .001±.000 | **.000±.000** | .003±.001 | **.000±.000** | **.000±.000** | **.000±.000** | **.000±.000** | .001±.000 | **.000±.000** | **.000±.000** | **.000±.000** | **.000±.000** | .009±.001 | .013±.001 | **.000±.000** |
| | .80 | **.000±.000** | .001±.000 | **.000±.000** | **.000±.000** | .001±.000 | **.000±.000** | **.000±.000** | **.000±.000** | **.000±.000** | **.000±.000** | **.000±.000** | **.000±.000** | **.000±.000** | .008±.001 | .013±.001 | **.000±.000** |
| | .75 | **.000±.000** | .001±.000 | **.000±.000** | **.000±.000** | **.000±.000** | **.000±.000** | **.000±.000** | **.000±.000** | **.000±.000** | **.000±.000** | **.000±.000** | **.000±.000** | **.000±.000** | .008±.001 | .012±.001 | **.000±.000** |
| | .70 | **.000±.000** | **.000±.000** | **.000±.000** | **.000±.000** | **.000±.000** | .001±.000 | **.000±.000** | **.000±.000** | **.000±.000** | **.000±.000** | **.000±.000** | **.000±.000** | **.000±.000** | .001±.000 | .008±.001 | .012±.001 | **.000±.000** |
| $\hat{\phi}$ | .99 | .989±.001 | .991±.001 | .988±.001 | .976±.001 | .991±.001 | .988±.001 | .991±.001 | .991±.001 | .989±.001 | .991±.001 | .989±.001 | .989±.001 | .988±.001 | .992±.001 | .991±.001 | **.990±.001** | .990±.001 |
| | .95 | .949±.002 | .952±.002 | .948±.002 | .933±.002 | .948±.002 | .948±.002 | .948±.002 | .949±.002 | .950±.002 | **.948±.002** | **.950±.002** | .950±.002 | .949±.002 | .948±.002 | .949±.002 | .949±.002 | .954±.002 |
| | .90 | **.900±.003** | .898±.003 | .898±.003 | .882±.003 | .895±.003 | **.901±.003** | .893±.003 | .899±.003 | .903±.003 | **.900±.003** | .894±.002 | .895±.002 | .895±.003 | .894±.003 | .906±.002 |
| | .85 | .848±.003 | .842±.003 | .845±.003 | .842±.003 | .853±.003 | .847±.003 | .839±.003 | .848±.003 | .848±.003 | .852±.003 | .846±.003 | .844±.003 | .840±.003 | **.849±.003** | .845±.003 | .862±.003 |
| | .80 | .792±.003 | .788±.004 | **.802±.003** | .792±.004 | **.802±.003** | **.799±.003** | .792±.004 | .797±.003 | .798±.004 | .811±.003 | .795±.003 | .794±.003 | .791±.003 | **.800±.003** | .789±.004 | .818±.003 |
| | .75 | .745±.004 | .741±.004 | .754±.003 | .738±.004 | .747±.004 | .754±.004 | .743±.004 | .753±.004 | .745±.004 | .745±.003 | .745±.003 | **.750±.004** | .740±.004 | .747±.004 | .744±.004 | .772±.004 |
| | .70 | .695±.004 | .694±.004 | **.701±.004** | .699±.004 | .700±.004 | .708±.004 | .693±.004 | .709±.004 | .699±.004 | .694±.004 | .694±.004 | .706±.004 | .690±.004 | .697±.004 | .690±.004 | .721±.004 |

Table B27: Results for `octmnist`: $mean \pm std$ for $\widehat{Err}$ and empirical coverage $\hat{\phi}$.

| Metric | c | DG | SAT | SAT+EM | SelNet | SelNet+EM | SR | SAT+SR | SAT+EM+SR | SelNet+SR | SelNet+EM+SR | ENS | ENS+SR | ConfidNet | SELE | REG | SCross |
|---|---|---|---|---|---|---|---|---|---|---|---|---|---|---|---|---|---|
| $\widehat{Err}$ | .99 | .099±.002 | .083±.002 | .079±.002 | .077±.002 | .088±.002 | .082±.002 | .083±.002 | .079±.002 | .077±.002 | .088±.002 | .075±.002 | **.072±.002** | .089±.002 | .111±.002 | .108±.002 | .079±.002 |
| | .95 | .088±.002 | .067±.002 | .063±.002 | .063±.002 | .076±.002 | .067±.002 | .066±.002 | .063±.002 | .061±.002 | .073±.002 | .065±.002 | **.056±.002** | .073±.002 | .103±.002 | .108±.002 | .065±.002 |
| | .90 | .076±.002 | .052±.002 | .046±.001 | .051±.002 | .059±.002 | .049±.002 | .050±.001 | .045±.001 | .047±.002 | .058±.002 | .052±.002 | **.042±.001** | .055±.002 | .093±.002 | .108±.002 | .049±.001 |
| | .85 | .066±.002 | .039±.001 | .035±.001 | .039±.002 | .046±.002 | .040±.001 | .039±.001 | .034±.001 | .037±.001 | .044±.001 | .042±.002 | **.031±.001** | .043±.002 | .083±.002 | .108±.002 | .037±.001 |
| | .80 | .059±.002 | .030±.001 | .028±.001 | .030±.001 | .054±.002 | .030±.001 | .030±.001 | .028±.001 | .029±.001 | .053±.002 | .033±.001 | **.023±.001** | .030±.001 | .073±.002 | .108±.002 | .029±.001 |
| | .75 | .053±.002 | .024±.001 | .022±.001 | .024±.001 | .037±.001 | .024±.001 | .024±.001 | .022±.001 | .023±.001 | .038±.002 | .026±.001 | **.018±.001** | .026±.001 | .066±.002 | .108±.002 | .024±.001 |
| | .70 | .048±.001 | .019±.001 | .018±.001 | .018±.001 | .024±.001 | .019±.001 | .018±.001 | .018±.001 | .018±.001 | .023±.001 | .021±.001 | **.014±.001** | .021±.001 | .057±.002 | .108±.003 | .019±.001 |
| $\hat{\phi}$ | .99 | **.990±.001** | **.990±.001** | .991±.001 | **.990±.001** | .992±.001 | .989±.001 | **.990±.001** | .991±.001 | **.990±.001** | .992±.001 | **.990±.001** | .989±.001 | **.990±.001** | .989±.001 | .989±.001 | .991±.001 |
| | .95 | .952±.002 | .949±.001 | **.950±.002** | .947±.002 | .952±.002 | .947±.001 | .952±.002 | .952±.002 | .946±.002 | **.951±.002** | **.950±.001** | .946±.001 | **.950±.002** | .951±.002 | .951±.002 | .954±.002 |
| | .90 | .902±.002 | .898±.002 | **.899±.002** | **.899±.002** | .900±.002 | **.899±.002** | **.899±.002** | .898±.002 | .897±.002 | **.899±.002** | .896±.002 | .898±.002 | .898±.002 | **.900±.002** | .900±.002 | .900±.002 |
| | .85 | **.849±.002** | .848±.002 | .849±.002 | .848±.002 | **.850±.002** | .852±.002 | .851±.002 | .849±.002 | .851±.002 | .847±.002 | .843±.002 | .846±.002 | **.850±.003** | .844±.002 | **.849±.003** | .857±.002 |
| | .80 | .804±.003 | .798±.003 | **.800±.003** | .798±.002 | .796±.003 | .799±.003 | .799±.003 | .800±.002 | .798±.002 | .794±.002 | .796±.002 | .795±.003 | .793±.003 | .794±.003 | .795±.003 | .809±.003 |
| | .75 | .754±.003 | **.751±.003** | **.750±.003** | .747±.003 | .754±.003 | .747±.003 | **.749±.003** | .751±.003 | .748±.003 | .749±.003 | .748±.003 | .746±.003 | .747±.003 | .748±.003 | .745±.003 | .762±.003 |
| | .70 | .705±.003 | .695±.003 | .699±.003 | .697±.003 | .697±.003 | .696±.003 | .697±.003 | **.699±.003** | **.699±.003** | .698±.003 | .703±.003 | .697±.003 | .692±.003 | .699±.003 | .692±.003 | .718±.003 |

Table B28: Results for `online`: $mean \pm std$ for $\widehat{Err}$, empirical coverage $\hat{\phi}$, and $MinCoeff$.

| Metric | c | DG | SAT | SAT+EM | SelNet | SelNet+EM | SR | SAT+SR | SAT+EM+SR | SelNet+SR | SelNet+EM+SR | ENS | ENS+SR | ConfidNet | SELE | REG | SCross | AUCross | PlugInAUC |
|---|---|---|---|---|---|---|---|---|---|---|---|---|---|---|---|---|---|---|---|
| $\widehat{Err}$ | .99 | **.091±.006** | .095±.006 | .093±.006 | .100±.006 | .099±.006 | .094±.006 | .094±.006 | .092±.006 | .102±.006 | .101±.006 | .093±.006 | .093±.006 | .092±.006 | .094±.006 | .103±.006 | .092±.006 | .096±.006 | .099±.006 |
| | .95 | .079±.006 | .087±.006 | **.078±.006** | .083±.006 | .089±.006 | .081±.006 | .082±.006 | .084±.006 | .088±.006 | .088±.006 | .091±.006 | **.078±.006** | .081±.006 | .093±.006 | .106±.007 | .085±.006 | .095±.006 | .099±.006 |
| | .90 | .066±.005 | .073±.005 | .067±.005 | .071±.005 | .068±.006 | .064±.005 | .067±.005 | .065±.005 | .068±.005 | .069±.006 | .084±.006 | **.063±.005** | .069±.005 | .087±.006 | .108±.007 | .070±.005 | .095±.006 | .098±.006 |
| | .85 | .056±.005 | .061±.005 | .054±.005 | .059±.006 | .062±.006 | .051±.005 | .053±.004 | .055±.005 | .062±.006 | .062±.006 | .075±.006 | .049±.005 | .051±.005 | .082±.006 | .111±.008 | **.047±.005** | .088±.006 | .095±.007 |
| | .80 | .048±.005 | .047±.005 | .040±.004 | .046±.005 | .051±.005 | .040±.005 | .042±.004 | .042±.004 | .047±.005 | .049±.005 | .068±.005 | **.038±.004** | .043±.005 | .079±.006 | .113±.008 | .039±.004 | .081±.006 | .090±.007 |
| | .75 | .039±.005 | .038±.004 | **.031±.004** | .041±.005 | .041±.005 | .032±.004 | .035±.004 | **.031±.004** | .041±.005 | .042±.004 | .060±.005 | .033±.004 | .036±.004 | .071±.006 | .115±.008 | .032±.004 | .067±.006 | .084±.007 |
| | .70 | .032±.004 | .031±.004 | **.026±.004** | .030±.004 | .039±.005 | .028±.004 | .027±.004 | .028±.004 | .033±.004 | .035±.004 | .052±.005 | .028±.004 | .033±.005 | .057±.005 | .120±.008 | .028±.004 | .065±.006 | .071±.007 |
| $\hat{\phi}$ | .99 | .988±.002 | .988±.002 | **.991±.002** | **.991±.002** | .988±.002 | .985±.002 | **.990±.002** | .987±.002 | **.992±.002** | .990±.002 | .990±.002 | .994±.002 | .987±.002 | .986±.002 | **.989±.002** | .993±.002 | .993±.002 | .988±.002 |
| | .95 | .948±.005 | .957±.004 | .955±.004 | .952±.004 | .954±.004 | .958±.004 | .955±.004 | .958±.004 | .960±.004 | .960±.003 | .954±.004 | .957±.004 | .954±.004 | **.951±.004** | .944±.005 | .971±.003 | .962±.004 | .953±.004 |
| | .90 | .902±.006 | .907±.006 | .925±.006 | .908±.006 | .920±.006 | .914±.006 | .915±.005 | .920±.006 | .901±.007 | .923±.006 | .896±.006 | .913±.006 | .916±.006 | .895±.006 | .887±.007 | .934±.005 | .916±.006 | **.902±.005** |
| | .85 | .869±.008 | .868±.007 | .869±.008 | .869±.007 | .868±.007 | .862±.008 | .868±.007 | .868±.007 | .870±.007 | .871±.007 | **.851±.007** | .864±.008 | .857±.008 | .834±.007 | .829±.008 | .854±.008 | .858±.007 | .844±.007 |
| | .80 | .829±.008 | .821±.009 | .820±.009 | .820±.008 | .822±.009 | .819±.009 | .819±.008 | .822±.009 | .826±.008 | .824±.009 | .816±.008 | .820±.009 | .821±.008 | .790±.008 | .795±.009 | .806±.008 | **.799±.008** | .790±.008 |
| | .75 | .779±.009 | .777±.009 | .767±.009 | .775±.010 | .776±.009 | .770±.009 | .776±.009 | .769±.009 | .780±.009 | .779±.009 | .768±.009 | .777±.009 | .778±.008 | .733±.009 | **.742±.009** | .766±.010 | .749±.009 | .731±.008 |
| | .70 | .708±.009 | .732±.009 | .712±.009 | .721±.011 | .729±.010 | .732±.009 | .716±.010 | .719±.009 | .718±.010 | .718±.010 | .707±.011 | .729±.009 | .710±.009 | .680±.009 | .690±.009 | .720±.010 | **.698±.010** | .675±.009 |
| $MinCoeff$ | .99 | .983±.046 | **.991±.047** | .990±.049 | .978±.047 | .976±.048 | .972±.047 | .982±.048 | .968±.048 | .972±.047 | .983±.047 | **.994±.047** | .994±.048 | .981±.048 | .991±.048 | 1.015±.048 | .993±.048 | 1.006±.048 | 1.008±.047 |
| | .95 | .913±.047 | .893±.045 | .887±.048 | .866±.046 | .890±.048 | .911±.050 | .907±.047 | .901±.047 | .911±.046 | .900±.046 | .933±.046 | .901±.047 | .900±.048 | .986±.050 | 1.060±.049 | .938±.047 | **1.010±.046** | 1.018±.047 |
| | .90 | .811±.046 | .759±.042 | .822±.047 | .751±.047 | .782±.046 | .819±.048 | .818±.046 | .809±.046 | .734±.044 | .803±.045 | .821±.048 | .780±.046 | .793±.044 | .972±.051 | 1.045±.051 | .867±.046 | **1.031±.048** | 1.034±.049 |
| | .85 | .735±.045 | .638±.039 | .675±.045 | .727±.042 | .699±.043 | .630±.042 | .686±.042 | .667±.044 | .721±.041 | .700±.043 | .750±.045 | .635±.042 | .657±.042 | .945±.052 | 1.046±.052 | .630±.044 | **1.012±.052** | 1.046±.052 |
| | .80 | .684±.045 | .502±.038 | .565±.045 | .472±.038 | .459±.038 | .554±.043 | .567±.041 | .590±.044 | .536±.038 | .489±.039 | .685±.043 | .533±.040 | .548±.041 | .900±.053 | 1.063±.051 | .539±.041 | **.997±.055** | 1.042±.056 |
| | .75 | .614±.047 | .374±.033 | .484±.040 | .264±.031 | .267±.030 | .467±.040 | .492±.039 | .479±.040 | .304±.031 | .270±.028 | .605±.043 | .462±.039 | .411±.039 | .806±.050 | 1.068±.055 | .475±.039 | **.965±.058** | 1.040±.057 |
| | .70 | .580±.046 | .205±.027 | .416±.039 | .193±.028 | .250±.030 | .418±.040 | .430±.038 | .426±.038 | .213±.028 | .229±.029 | .489±.043 | .372±.039 | .363±.040 | .730±.053 | 1.094±.055 | .433±.037 | .964±.060 | **.994±.058** |

Table B29: Results for `organamnist`: $mean \pm std$ for $\widehat{Err}$ and empirical coverage $\hat{\phi}$.

| Metric | c | DG | SAT | SAT+EM | SelNet | SelNet+EM | SR | SAT+SR | SAT+EM+SR | SelNet+SR | SelNet+EM+SR | ENS | ENS+SR | ConfidNet | SELE | REG | SCross |
|---|---|---|---|---|---|---|---|---|---|---|---|---|---|---|---|---|---|
| $\widehat{Err}$ | .99 | .005±.001 | .002±.001 | .003±.000 | .003±.000 | **.001±.000** | .002±.000 | .002±.000 | .002±.000 | **.001±.000** | **.001±.000** | **.001±.000** | **.001±.000** | .007±.001 | .009±.001 | .009±.001 | .002±.000 |
| | .95 | .002±.000 | **.000±.000** | .001±.000 | .001±.000 | **.000±.000** | **.000±.000** | **.000±.000** | **.000±.000** | **.000±.000** | **.000±.000** | **.000±.000** | **.000±.000** | .002±.001 | .008±.001 | .009±.001 | **.000±.000** |
| | .90 | .001±.000 | **.000±.000** | **.000±.000** | **.000±.000** | **.000±.000** | **.000±.000** | **.000±.000** | **.000±.000** | **.000±.000** | **.000±.000** | **.000±.000** | **.000±.000** | .001±.000 | .008±.001 | .009±.001 | **.000±.000** |
| | .85 | **.000±.000** | **.000±.000** | **.000±.000** | **.000±.000** | **.000±.000** | **.000±.000** | **.000±.000** | **.000±.000** | **.000±.000** | .045±.002 | **.000±.000** | **.000±.000** | .001±.000 | .008±.001 | .009±.001 | **.000±.000** |
| | .80 | **.000±.000** | **.000±.000** | **.000±.000** | **.000±.000** | **.000±.000** | **.000±.000** | **.000±.000** | **.000±.000** | **.000±.000** | .048±.002 | **.000±.000** | **.000±.000** | **.000±.000** | .008±.001 | .009±.001 | **.000±.000** |
| | .75 | **.000±.000** | **.000±.000** | **.000±.000** | **.000±.000** | **.000±.000** | **.000±.000** | **.000±.000** | **.000±.000** | **.000±.000** | .002±.000 | **.000±.000** | **.000±.000** | **.000±.000** | .007±.001 | .010±.001 | **.000±.000** |
| | .70 | .001±.000 | **.000±.000** | **.000±.000** | **.000±.000** | **.000±.000** | **.000±.000** | **.000±.000** | **.000±.000** | **.000±.000** | .006±.001 | **.000±.000** | **.000±.000** | **.000±.000** | .007±.001 | .010±.001 | **.000±.000** |
| $\hat{\phi}$ | .99 | .987±.001 | **.990±.001** | .989±.001 | .987±.001 | **.990±.001** | **.990±.001** | **.990±.001** | .987±.001 | **.992±.001** | **.990±.001** | **.990±.001** | **.990±.001** | .991±.001 | .991±.001 | **.991±.001** | **.990±.001** |
| | .95 | .952±.002 | .954±.002 | **.950±.002** | **.951±.002** | .955±.002 | .954±.002 | **.950±.002** | .949±.002 | .954±.002 | .954±.002 | .956±.002 | .955±.002 | **.952±.002** | .952±.002 | **.952±.002** | .956±.002 |
| | .90 | .905±.002 | .904±.003 | **.900±.003** | .906±.003 | .904±.003 | .909±.002 | **.901±.003** | **.900±.003** | .902±.003 | .910±.003 | .904±.003 | .903±.003 | .905±.003 | **.900±.003** | .906±.002 | .910±.003 |
| | .85 | **.848±.003** | .853±.003 | **.849±.003** | .852±.003 | .853±.003 | .859±.003 | .852±.003 | .846±.003 | .858±.003 | .922±.002 | .853±.003 | .855±.003 | .858±.003 | **.851±.003** | .855±.003 | .869±.003 |
| | .80 | **.800±.004** | .805±.003 | **.802±.003** | .807±.004 | .806±.004 | .806±.004 | .808±.003 | .802±.004 | .808±.004 | .953±.002 | .805±.003 | .805±.003 | .807±.003 | **.803±.004** | .805±.004 | .829±.003 |
| | .75 | **.750±.003** | .755±.004 | .755±.004 | .768±.004 | .760±.004 | .758±.004 | .757±.004 | .758±.004 | .758±.004 | .961±.002 | .753±.004 | .756±.004 | .762±.003 | **.753±.004** | .753±.004 | .787±.003 |
| | .70 | .705±.004 | .698±.004 | **.700±.004** | .710±.004 | .709±.004 | .706±.004 | .720±.004 | .703±.004 | .715±.004 | .951±.002 | .712±.004 | .710±.004 | .709±.004 | .705±.004 | **.700±.004** | .745±.003 |

Table B30: Results for `organcmnist`: $mean \pm std$ for $\widehat{Err}$ and empirical coverage $\hat{\phi}$.

| Metric | c | DG | SAT | SAT+EM | SelNet | SelNet+EM | SR | SAT+SR | SAT+EM+SR | SelNet+SR | SelNet+EM+SR | ENS | ENS+SR | ConfidNet | SELE | REG | SCross |
|---|---|---|---|---|---|---|---|---|---|---|---|---|---|---|---|---|---|
| $\widehat{Err}$ | .99 | .034±.003 | .023±.002 | .023±.002 | .021±.002 | .025±.002 | .020±.002 | .023±.002 | .021±.002 | .020±.002 | .024±.002 | .019±.002 | .018±.002 | .033±.002 | .043±.003 | .046±.003 | **.017±.002** |
| | .95 | .023±.002 | .012±.002 | .014±.002 | .013±.002 | .016±.002 | .010±.001 | .010±.001 | .012±.002 | .009±.001 | .012±.002 | .008±.001 | **.005±.001** | .027±.002 | .039±.003 | .045±.003 | .007±.001 |
| | .90 | .015±.002 | .004±.001 | .007±.001 | .007±.001 | .005±.001 | .005±.001 | .003±.001 | .007±.001 | .003±.001 | .004±.001 | .004±.001 | **.001±.001** | .020±.002 | .036±.003 | .046±.003 | .003±.001 |
| | .85 | .011±.002 | .002±.001 | .004±.001 | .006±.001 | .003±.001 | .002±.001 | **.001±.000** | .004±.001 | .003±.001 | .001±.001 | .003±.001 | **.000±.000** | .016±.002 | .032±.003 | .045±.003 | .001±.001 |
| | .80 | .008±.001 | **.001±.000** | **.001±.000** | .002±.001 | .004±.001 | .010±.001 | **.001±.000** | .002±.001 | .001±.001 | .001±.001 | .001±.001 | **.000±.000** | .012±.002 | .030±.003 | .044±.003 | **.001±.000** |
| | .75 | .006±.001 | .001±.000 | .001±.000 | .001±.001 | .001±.001 | .002±.001 | **.000±.000** | .001±.000 | .001±.000 | .001±.000 | .040±.003 | **.000±.000** | **.000±.000** | .028±.003 | .040±.003 | **.000±.000** |
| | .70 | .005±.001 | .001±.000 | **.000±.000** | .001±.002 | .001±.001 | .001±.001 | **.000±.000** | .001±.000 | .001±.000 | **.000±.000** | .111±.006 | **.000±.000** | **.000±.000** | .026±.003 | .041±.004 | **.000±.000** |
| $\hat{\phi}$ | .99 | .992±.001 | .991±.001 | .992±.001 | .990±.001 | **.991±.001** | .990±.001 | **.991±.001** | .989±.001 | **.991±.001** | .990±.001 | .994±.001 | .993±.001 | **.990±.001** | **.990±.001** | **.990±.001** | **.990±.001** |
| | .95 | .958±.003 | .952±.003 | .953±.003 | .958±.003 | .950±.003 | .958±.003 | .954±.003 | .955±.003 | .960±.003 | **.950±.003** | **.950±.003** | .953±.003 | **.950±.003** | .954±.003 | .949±.003 | .961±.003 |
| | .90 | .911±.004 | .897±.005 | **.902±.004** | .891±.005 | .897±.005 | .903±.005 | **.900±.005** | .902±.005 | .903±.004 | .897±.004 | .905±.005 | .904±.005 | .897±.005 | .910±.004 | .900±.005 | .919±.004 |
| | .85 | .862±.005 | **.850±.005** | .842±.006 | .853±.005 | .855±.005 | .848±.006 | **.850±.005** | **.851±.005** | .856±.005 | .854±.005 | **.850±.005** | .853±.005 | .854±.006 | .859±.005 | .859±.005 | .869±.005 |
| | .80 | .809±.006 | .796±.006 | .794±.006 | .809±.006 | .792±.005 | .801±.006 | .800±.006 | .796±.006 | .801±.006 | .804±.005 | .799±.006 | .798±.006 | .801±.006 | **.811±.005** | **.801±.005** | .825±.006 |
| | .75 | .754±.006 | .742±.006 | .752±.007 | .761±.006 | .753±.006 | .745±.006 | .750±.007 | .750±.007 | .766±.006 | .757±.006 | .757±.007 | .754±.007 | .756±.006 | .767±.006 | **.748±.006** | .783±.006 |
| | .70 | .710±.006 | .698±.007 | .705±.007 | .704±.007 | .694±.007 | .703±.007 | **.701±.007** | .706±.007 | .708±.007 | .715±.007 | **.701±.007** | .703±.007 | .706±.006 | .718±.006 | **.702±.006** | .725±.007 |

Table B31: Results for `organsmnist`: $mean \pm std$ for $\widehat{Err}$ and empirical coverage $\hat{\phi}$.

| Metric | c | DG | SAT | SAT+EM | SelNet | SelNet+EM | SR | SAT+SR | SAT+EM+SR | SelNet+SR | SelNet+EM+SR | ENS | ENS+SR | ConfidNet | SELE | REG | SCross |
|---|---|---|---|---|---|---|---|---|---|---|---|---|---|---|---|---|---|
| $\widehat{Err}$ | .99 | .162±.005 | .072±.004 | .066±.004 | .077±.004 | .079±.004 | .066±.003 | .071±.004 | .066±.004 | .077±.004 | .076±.004 | .055±.003 | **.053±.003** | .065±.004 | .114±.005 | .116±.005 | .064±.004 |
| | .95 | .146±.004 | .057±.004 | .052±.003 | .063±.004 | .063±.003 | .051±.003 | .054±.003 | .050±.003 | .055±.004 | .066±.004 | .043±.003 | **.036±.003** | .051±.004 | .107±.005 | .116±.005 | .049±.003 |
| | .90 | .134±.004 | .039±.003 | .032±.002 | .041±.003 | .038±.003 | .031±.003 | .035±.003 | .029±.002 | .038±.003 | .044±.003 | .037±.003 | **.016±.002** | .036±.003 | .098±.005 | .117±.005 | .032±.002 |
| | .85 | .121±.004 | .022±.002 | .017±.002 | .026±.002 | .033±.003 | .017±.002 | .018±.002 | .015±.002 | .018±.002 | .041±.003 | .033±.002 | **.008±.001** | .031±.003 | .089±.005 | .116±.005 | .017±.002 |
| | .80 | .106±.004 | .013±.002 | .010±.002 | .017±.002 | .015±.002 | .008±.001 | .010±.002 | .006±.001 | .011±.002 | .037±.003 | .026±.003 | **.004±.001** | .028±.003 | .082±.005 | .114±.005 | .008±.001 |
| | .75 | .098±.004 | .006±.002 | .006±.001 | .010±.002 | .007±.001 | .004±.001 | .004±.001 | **.002±.001** | .006±.001 | .042±.004 | .017±.002 | **.002±.001** | .025±.003 | .077±.005 | .110±.005 | .005±.001 |
| | .70 | .093±.004 | .004±.001 | .004±.001 | .007±.002 | .003±.001 | .003±.001 | .003±.001 | **.001±.000** | .003±.001 | .028±.003 | .008±.002 | **.001±.000** | .023±.003 | .072±.005 | .105±.005 | .002±.001 |
| $\hat{\phi}$ | .99 | .989±.002 | .993±.001 | .989±.002 | **.990±.001** | **.990±.001** | .989±.002 | .988±.002 | .988±.002 | .994±.001 | .986±.002 | .994±.001 | **.990±.002** | **.990±.001** | .988±.002 | **.991±.001** | .987±.002 |
| | .95 | .941±.003 | .955±.003 | .952±.003 | .959±.003 | .956±.003 | .955±.003 | **.950±.003** | .952±.003 | .946±.003 | .948±.003 | .942±.003 | **.950±.003** | .941±.003 | .953±.003 | .957±.002 | .957±.002 |
| | .90 | .890±.004 | .903±.004 | **.902±.004** | .903±.004 | .897±.004 | .902±.004 | .902±.004 | .906±.004 | .899±.004 | .894±.004 | .892±.004 | .899±.004 | .907±.004 | **.901±.004** | .902±.004 | .914±.004 |
| | .85 | .845±.005 | **.852±.005** | **.852±.005** | .856±.005 | .845±.005 | .845±.005 | **.849±.005** | .854±.005 | .858±.005 | .847±.005 | .857±.005 | .860±.005 | .847±.005 | .857±.005 | .866±.005 | .817±.006 |
| | .80 | .798±.006 | .805±.006 | .810±.005 | .803±.006 | .803±.006 | .800±.006 | **.799±.006** | .809±.005 | .798±.006 | .793±.006 | .809±.006 | .808±.006 | .812±.006 | .792±.006 | .809±.006 | .817±.006 |
| | .75 | .754±.006 | .743±.007 | .774±.006 | .761±.006 | .753±.006 | .738±.006 | .744±.007 | .762±.007 | .750±.006 | .757±.007 | .753±.007 | .759±.007 | .756±.007 | .747±.007 | .756±.006 | .769±.006 |
| | .70 | .708±.007 | .702±.007 | .719±.006 | .709±.006 | .701±.007 | .695±.007 | **.699±.007** | .720±.006 | .705±.007 | .699±.007 | .703±.007 | .705±.007 | .710±.007 | **.701±.007** | .704±.006 | .724±.007 |

Table B32: Results for `oxfordpets`: $mean \pm std$ for $\widehat{Err}$, empirical coverage $\hat{\phi}$, and $MinCoeff$.

| Metric | c | DG | SAT | SAT+EM | SelNet | SelNet+EM | SR | SAT+SR | SAT+EM+SR | SelNet+SR | SelNet+EM+SR | ENS | ENS+SR | ConfidNet | SELE | REG | SCross | AUCross | PlugInAUC |
|---|---|---|---|---|---|---|---|---|---|---|---|---|---|---|---|---|---|---|---|
| $\widehat{Err}$ | .99 | .317±.012 | .053±.006 | .087±.007 | **.039±.005** | .166±.009 | .059±.006 | .039±.005 | .041±.005 | .166±.008 | .166±.010 | .039±.005 | .041±.005 | .050±.006 | .169±.009 | .170±.009 | .041±.005 | .040±.005 | .057±.006 |
| | .95 | .315±.012 | .035±.006 | .072±.007 | .055±.006 | .120±.008 | .039±.006 | .036±.006 | .075±.006 | .047±.005 | .123±.009 | .029±.005 | .029±.005 | .050±.006 | .165±.009 | .172±.009 | **.029±.004** | .030±.005 | .058±.005 |
| | .90 | .315±.013 | .022±.004 | .058±.006 | .024±.005 | .131±.009 | .025±.005 | .025±.004 | .057±.006 | .021±.004 | .125±.008 | .018±.004 | **.014±.003** | .046±.005 | .163±.009 | .171±.004 | .017±.004 | .023±.004 | .027±.005 |
| | .85 | .314±.013 | .016±.003 | .044±.005 | **.006±.002** | .180±.010 | .020±.004 | .016±.003 | .042±.005 | .007±.002 | .178±.010 | .004±.002 | .006±.003 | .030±.005 | .161±.009 | .168±.010 | .013±.003 | .013±.003 | .016±.004 |
| | .80 | .315±.013 | .013±.003 | .032±.005 | .018±.004 | .107±.007 | .012±.003 | .012±.003 | .033±.005 | .020±.004 | .090±.007 | .004±.002 | **.002±.002** | .030±.005 | .156±.009 | .165±.010 | .007±.002 | .006±.002 | .013±.003 |
| | .75 | .316±.014 | .005±.002 | .029±.005 | .004±.002 | .038±.006 | .007±.002 | **.002±.002** | .024±.004 | .005±.002 | .035±.005 | .003±.002 | .003±.002 | .021±.005 | .155±.010 | .167±.011 | .005±.002 | .005±.002 | .010±.003 |
| | .70 | .313±.014 | .003±.001 | .016±.004 | .006±.002 | .150±.010 | .006±.002 | .003±.001 | .020±.004 | .005±.002 | .159±.011 | **.001±.001** | **.001±.001** | .015±.004 | .155±.010 | .172±.011 | **.001±.001** | .002±.002 | .007±.003 |
| $\hat{\phi}$ | .99 | **.992±.002** | .992±.002 | .980±.003 | .985±.003 | **.989±.003** | .992±.002 | **.989±.003** | .982±.004 | **.990±.003** | .982±.004 | **.989±.003** | **.989±.003** | .985±.003 | .987±.003 | .985±.003 | **.991±.002** | **.992±.002** | **.990±.003** |
| | .95 | .950±.006 | **.949±.006** | .936±.006 | .951±.006 | .929±.007 | .943±.006 | .953±.005 | .947±.006 | .958±.005 | .949±.006 | .952±.005 | .951±.006 | .943±.006 | .935±.006 | .945±.006 | .953±.006 | .957±.005 | .943±.006 |
| | .90 | .912±.008 | .904±.007 | .890±.007 | .901±.008 | .898±.008 | .896±.008 | .911±.007 | .888±.008 | .892±.008 | **.901±.007** | .907±.007 | .903±.007 | .903±.008 | .888±.008 | .883±.008 | .902±.008 | .908±.008 | .898±.008 |
| | .85 | .860±.010 | .849±.009 | .835±.009 | .827±.009 | .819±.010 | .855±.010 | **.852±.005** | **.852±.005** | .835±.009 | .892±.008 | .864±.008 | .827±.009 | .828±.009 | .843±.009 | .854±.009 | .831±.009 | .858±.010 | .844±.010 |
| | .80 | .819±.011 | .803±.010 | .770±.010 | **.798±.009** | .831±.008 | .800±.012 | .802±.010 | .776±.010 | .780±.010 | .794±.011 | .771±.010 | .771±.010 | .788±.010 | .811±.009 | .768±.011 | .807±.010 | .809±.010 | .800±.011 |
| | .75 | .758±.012 | **.751±.010** | .734±.010 | .741±.012 | .737±.012 | .750±.012 | .753±.010 | .725±.011 | .751±.011 | .730±.012 | .725±.012 | .725±.012 | .739±.010 | .768±.010 | .707±.013 | .755±.011 | .755±.011 | .755±.013 |
| | .70 | .717±.013 | .693±.011 | .676±.011 | .676±.012 | .695±.013 | .691±.013 | .696±.011 | .677±.011 | .676±.012 | .703±.013 | .674±.011 | .681±.011 | **.698±.010** | .708±.011 | .669±.013 | .703±.011 | .714±.011 | .697±.013 |
| $MinCoeff$ | .99 | 1.008±.018 | 1.005±.018 | 1.011±.018 | 1.013±.018 | 1.005±.018 | 1.007±.018 | 1.008±.018 | 1.009±.018 | 1.005±.018 | 1.005±.018 | 1.007±.018 | 1.007±.018 | .996±.019 | 1.004±.018 | **1.001±.018** | 1.007±.018 | 1.005±.018 | 1.024±.018 |
| | .95 | 1.012±.018 | 1.014±.019 | 1.023±.019 | 1.019±.018 | **1.004±.019** | 1.021±.018 | 1.012±.019 | 1.016±.019 | 1.038±.018 | 1.012±.019 | 1.022±.018 | 1.025±.018 | .977±.019 | 1.008±.020 | .997±.019 | 1.014±.019 | 1.016±.018 | 1.024±.018 |
| | .90 | 1.011±.019 | 1.021±.019 | 1.036±.019 | 1.034±.019 | 1.015±.019 | 1.044±.018 | 1.021±.019 | 1.033±.020 | 1.041±.019 | 1.020±.020 | 1.041±.018 | 1.043±.018 | .965±.019 | 1.021±.020 | **.994±.019** | 1.032±.019 | 1.036±.019 | 1.014±.019 |
| | .85 | 1.013±.019 | 1.026±.019 | 1.061±.020 | 1.063±.019 | 1.035±.019 | 1.054±.019 | 1.028±.019 | 1.051±.020 | 1.069±.019 | 1.028±.019 | 1.066±.018 | 1.067±.019 | .951±.019 | 1.035±.020 | **.992±.020** | 1.044±.019 | 1.053±.019 | 1.078±.018 |
| | .80 | **1.011±.019** | 1.028±.020 | 1.093±.021 | 1.043±.019 | 1.046±.019 | 1.077±.019 | 1.028±.020 | 1.086±.020 | 1.015±.019 | 1.033±.020 | 1.082±.020 | 1.085±.020 | .936±.021 | 1.030±.020 | .997±.022 | 1.052±.020 | 1.067±.020 | 1.100±.019 |
| | .75 | 1.010±.020 | 1.042±.019 | 1.119±.021 | 1.035±.020 | 1.070±.021 | 1.100±.020 | 1.043±.020 | 1.115±.021 | **1.011±.019** | 1.035±.022 | 1.100±.020 | 1.102±.020 | .921±.022 | 1.031±.021 | .987±.022 | 1.072±.020 | 1.097±.019 | 1.133±.020 |
| | .70 | **1.014±.021** | 1.047±.021 | 1.159±.020 | 1.173±.018 | 1.086±.020 | 1.109±.021 | 1.048±.021 | 1.147±.020 | 1.134±.019 | 1.026±.022 | 1.120±.019 | 1.118±.020 | .899±.022 | 1.033±.022 | .980±.024 | 1.089±.021 | 1.116±.020 | 1.167±.020 |

Table B33: Results for `pathmnist`: $mean \pm std$ for $\widehat{Err}$ and empirical coverage $\hat{\phi}$.

| Metric | c | DG | SAT | SAT+EM | SelNet | SelNet+EM | SR | SAT+SR | SAT+EM+SR | SelNet+SR | SelNet+EM+SR | ENS | ENS+SR | ConfidNet | SELE | REG | SCross |
|---|---|---|---|---|---|---|---|---|---|---|---|---|---|---|---|---|---|
| $\widehat{Err}$ | .99 | .027±.001 | .011±.001 | .020±.001 | .014±.001 | .022±.001 | .010±.001 | .011±.001 | .019±.001 | .012±.001 | .019±.001 | .007±.001 | **.006±.001** | .020±.001 | .028±.001 | .033±.001 | .008±.001 |
| | .95 | .019±.001 | .004±.000 | .010±.001 | .007±.001 | .016±.001 | .003±.000 | .003±.000 | .008±.001 | .005±.001 | .013±.001 | .002±.000 | .001±.000 | .006±.001 | .028±.001 | .034±.001 | .003±.000 |
| | .90 | .013±.001 | .002±.000 | .004±.001 | .003±.000 | .023±.001 | .002±.000 | .002±.000 | .004±.000 | .002±.000 | .022±.001 | .001±.000 | .000±.000 | .002±.000 | .027±.001 | .035±.001 | .001±.000 |
| | .85 | .010±.001 | .001±.000 | .002±.000 | .001±.000 | .004±.000 | .001±.000 | .001±.000 | .002±.000 | .001±.000 | .004±.000 | .000±.000 | .000±.000 | .001±.000 | .025±.001 | .036±.001 | .001±.000 |
| | .80 | .007±.001 | **.000±.000** | **.000±.000** | .001±.000 | .002±.000 | .001±.000 | **.000±.000** | .001±.000 | .001±.000 | .001±.000 | **.000±.000** | **.000±.000** | **.000±.000** | .024±.001 | .037±.001 | .001±.000 |
| | .75 | .006±.001 | **.000±.000** | .001±.000 | **.000±.000** | .012±.001 | .001±.000 | **.000±.000** | .001±.000 | **.000±.000** | .010±.001 | **.000±.000** | **.000±.000** | **.000±.000** | .023±.001 | .038±.001 | **.000±.000** |
| | .70 | .005±.001 | **.000±.000** | .001±.000 | .001±.000 | .002±.000 | .001±.000 | **.000±.000** | .001±.000 | **.000±.000** | .002±.000 | **.000±.000** | **.000±.000** | .001±.000 | .022±.001 | .038±.001 | **.000±.000** |
| $\hat{\phi}$ | .99 | .989±.001 | .989±.001 | .989±.001 | .990±.001 | .989±.001 | .991±.001 | **.990±.001** | .989±.001 | **.990±.001** | .988±.001 | .989±.001 | .989±.001 | .989±.001 | .992±.001 | **.990±.001** | .992±.001 |
| | .95 | .948±.002 | **.951±.002** | .949±.001 | .951±.002 | .951±.002 | **.950±.001** | **.950±.001** | **.951±.001** | .948±.002 | .951±.002 | **.951±.001** | **.950±.001** | .947±.001 | .955±.002 | .944±.002 | .958±.001 |
| | .90 | .898±.002 | **.901±.002** | .899±.002 | **.899±.002** | .899±.002 | **.900±.002** | .902±.002 | **.901±.002** | **.901±.002** | .900±.002 | .903±.002 | **.902±.002** | .895±.002 | .902±.002 | .895±.002 | .914±.002 |
| | .85 | .849±.002 | .848±.002 | .852±.003 | .852±.002 | .853±.002 | **.850±.002** | .848±.002 | .852±.002 | .854±.002 | .856±.002 | **.851±.002** | .852±.002 | .845±.002 | .851±.002 | **.849±.002** | .863±.002 |
| | .80 | .797±.003 | .798±.002 | .801±.003 | .807±.002 | .805±.003 | **.800±.003** | .798±.003 | .804±.003 | .803±.002 | .803±.003 | .799±.002 | .796±.002 | .793±.003 | .799±.003 | **.800±.002** | .807±.003 |
| | .75 | .749±.003 | **.751±.003** | .754±.003 | .754±.003 | .745±.003 | .744±.003 | .752±.003 | .752±.003 | **.750±.003** | .752±.003 | .751±.003 | .744±.003 | .742±.003 | **.749±.003** | .755±.003 | .748±.003 |
| | .70 | .697±.003 | .703±.003 | .702±.003 | .706±.003 | **.700±.003** | .693±.003 | .704±.003 | .706±.003 | .701±.003 | .705±.003 | .706±.003 | .695±.003 | .690±.003 | **.701±.003** | .701±.003 | .695±.003 |

Table B34: Results for `phoneme`: $mean \pm std$ for $\widehat{Err}$, empirical coverage $\hat{\phi}$, and $MinCoeff$.

| Metric | c | DG | SAT | SAT+EM | SelNet | SelNet+EM | SR | SAT+SR | SAT+EM+SR | SelNet+SR | SelNet+EM+SR | ENS | ENS+SR | ConfidNet | SELE | REG | SCross | AUCross | PlugInAUC |
|---|---|---|---|---|---|---|---|---|---|---|---|---|---|---|---|---|---|---|---|
| $\widehat{Err}$ | .99 | .255±.017 | .133±.013 | .141±.015 | .134±.014 | .161±.015 | .137±.015 | .136±.014 | .142±.015 | **.127±.013** | .159±.015 | .145±.014 | .140±.015 | .141±.015 | .165±.014 | .166±.014 | .180±.015 | .189±.015 | .136±.014 |
| | .95 | .228±.017 | .124±.014 | .138±.015 | .125±.015 | .142±.015 | .117±.014 | .125±.014 | **.116±.014** | .141±.015 | .141±.015 | .134±.015 | .129±.014 | .165±.015 | .168±.014 | .171±.016 | .176±.015 | .184±.015 | .128±.014 |
| | .90 | .220±.017 | .118±.014 | .124±.014 | .114±.013 | .123±.015 | **.103±.014** | .121±.014 | .108±.014 | .104±.013 | .115±.014 | .156±.016 | .112±.014 | .110±.013 | .163±.015 | .171±.014 | .157±.016 | .176±.015 | .124±.015 |
| | .85 | .217±.017 | .101±.013 | .121±.014 | **.081±.014** | .119±.014 | .092±.014 | .109±.014 | .095±.013 | .085±.013 | .107±.014 | .165±.017 | .084±.013 | .108±.013 | .161±.015 | .170±.015 | .148±.015 | .164±.016 | .107±.015 |
| | .80 | .214±.018 | .093±.013 | .111±.014 | .094±.014 | .109±.014 | .089±.014 | .094±.013 | .084±.013 | .094±.013 | .069±.013 | .169±.018 | .083±.013 | .098±.013 | .161±.015 | .168±.015 | .139±.015 | .158±.016 | .096±.014 |
| | .75 | .224±.018 | .079±.013 | .099±.014 | .080±.012 | .101±.014 | .078±.013 | .085±.014 | .079±.012 | .084±.013 | .080±.012 | .174±.019 | **.067±.013** | .082±.012 | .158±.016 | .161±.015 | .136±.015 | .151±.016 | .095±.014 |
| | .70 | .218±.018 | .073±.012 | .097±.015 | .081±.014 | .094±.013 | .070±.012 | .070±.012 | .072±.012 | .074±.014 | .070±.012 | .178±.019 | .063±.012 | **.063±.011** | .158±.016 | .156±.016 | .133±.016 | .147±.016 | .093±.015 |
| $\hat{\phi}$ | .99 | .978±.005 | .984±.005 | .992±.004 | .989±.004 | 1.000±.000 | .997±.002 | .984±.005 | **.990±.003** | .970±.007 | .990±.004 | .988±.004 | .994±.003 | .989±.004 | .995±.003 | .988±.004 | .980±.005 | .994±.003 | **.992±.003** |
| | .95 | .921±.011 | .948±.010 | .961±.008 | .931±.009 | .960±.008 | .938±.009 | .945±.009 | .929±.010 | .949±.009 | .936±.010 | .947±.009 | .975±.007 | .954±.008 | .946±.009 | .966±.007 | **.950±.008** | .961±.008 | .939±.009 |
| | .90 | .862±.012 | .907±.012 | .927±.009 | .894±.012 | .903±.011 | .895±.011 | .909±.012 | .893±.012 | .892±.012 | .858±.012 | .898±.012 | .916±.011 | .899±.011 | .922±.010 | **.902±.010** | .914±.011 | .912±.011 | .907±.012 |
| | .85 | .837±.014 | .857±.013 | .874±.012 | .832±.016 | .866±.014 | .840±.014 | .865±.014 | .827±.014 | .870±.014 | .832±.014 | .851±.016 | .851±.014 | .861±.012 | .890±.012 | **.847±.012** | .882±.013 | .867±.014 | .839±.015 |
| | .80 | .791±.015 | .819±.014 | .813±.016 | .786±.015 | .799±.017 | .812±.016 | .820±.015 | .785±.016 | .787±.015 | .797±.016 | .787±.017 | .804±.016 | .828±.014 | .841±.015 | **.791±.013** | .844±.014 | .828±.015 | .811±.016 |
| | .75 | .726±.016 | .760±.017 | .752±.016 | .767±.017 | .730±.018 | .764±.018 | .762±.017 | .734±.016 | .767±.016 | .740±.018 | .764±.018 | .754±.017 | .769±.015 | .804±.014 | **.755±.014** | .805±.017 | .812±.016 | .755±.018 |
| | .70 | .693±.018 | .704±.018 | .691±.019 | .735±.019 | .706±.018 | .729±.019 | .685±.018 | .694±.018 | .733±.018 | .678±.019 | .739±.019 | .704±.019 | **.705±.016** | .765±.017 | **.707±.015** | .765±.018 | .778±.018 | .712±.019 |
| $MinCoeff$ | .99 | 1.011±.038 | 1.006±.037 | 1.008±.037 | .993±.037 | **1.000±.037** | 1.000±.037 | 1.005±.037 | 1.007±.037 | 1.006±.039 | .993±.037 | **1.001±.037** | 1.003±.038 | 1.002±.037 | 1.005±.037 | 1.008±.037 | 1.008±.037 | 1.003±.037 | **1.001±.037** |
| | .95 | 1.037±.039 | 1.008±.038 | 1.009±.038 | .984±.039 | .988±.037 | 1.017±.038 | 1.003±.037 | 1.019±.038 | .992±.039 | .999±.038 | 1.029±.038 | **1.004±.037** | 1.005±.037 | 1.050±.040 | .974±.037 | 1.026±.038 | 1.015±.039 | 1.026±.037 |
| | .90 | 1.032±.039 | .998±.039 | 1.018±.038 | 1.012±.038 | **.993±.037** | 1.023±.039 | .998±.038 | 1.011±.038 | 1.029±.038 | .975±.037 | 1.086±.038 | 1.000±.038 | 1.016±.037 | 1.063±.040 | .975±.037 | 1.029±.039 | 1.052±.039 | 1.045±.037 |
| | .85 | 1.010±.040 | .987±.040 | 1.017±.039 | **.995±.038** | .974±.039 | 1.013±.039 | .998±.039 | 1.016±.039 | 1.019±.038 | .970±.038 | 1.084±.040 | 1.020±.038 | .999±.039 | 1.090±.042 | .967±.036 | 1.049±.040 | 1.085±.040 | 1.075±.039 |
| | .80 | 1.002±.041 | .992±.041 | 1.019±.041 | .989±.040 | .967±.039 | 1.007±.040 | .964±.041 | 1.022±.042 | .988±.040 | .971±.039 | 1.110±.042 | **1.006±.040** | .998±.041 | 1.128±.045 | .950±.037 | 1.065±.041 | 1.113±.042 | 1.104±.039 |
| | .75 | .961±.042 | .995±.042 | .987±.042 | **.995±.038** | .996±.040 | 1.005±.042 | .967±.042 | 1.012±.042 | .990±.038 | .967±.039 | 1.124±.042 | 1.008±.040 | .988±.042 | 1.155±.044 | .939±.039 | 1.073±.041 | 1.119±.042 | 1.147±.042 |
| | .70 | .938±.043 | .995±.044 | .965±.045 | 1.006±.042 | **1.004±.040** | 1.007±.042 | .958±.043 | .993±.041 | 1.035±.042 | .951±.041 | 1.118±.044 | 1.019±.043 | .984±.044 | 1.259±.047 | .910±.041 | 1.069±.043 | 1.143±.042 | 1.165±.043 |

Table B35: Results for `pneumoniamnist`: $mean \pm std$ for $\widehat{Err}$, empirical coverage $\hat{\phi}$, and $MinCoeff$.

| Metric | c | DG | SAT | SAT+EM | SelNet | SelNet+EM | SR | SAT+SR | SAT+EM+SR | SelNet+SR | SelNet+EM+SR | ENS | ENS+SR | ConfidNet | SELE | REG | SCross | AUCross | PlugInAUC |
|---|---|---|---|---|---|---|---|---|---|---|---|---|---|---|---|---|---|---|---|
| $\widehat{Err}$ | .99 | .038±.006 | .033±.005 | .041±.006 | .036±.006 | .042±.006 | .033±.005 | .032±.005 | .041±.006 | .037±.006 | .045±.006 | .031±.005 | .030±.005 | **.028±.005** | .040±.006 | .047±.006 | .045±.006 | .048±.006 | .033±.005 |
| | .95 | .035±.006 | .020±.005 | .026±.006 | .025±.005 | .015±.004 | .017±.004 | .021±.005 | .024±.005 | .022±.005 | .015±.004 | .022±.005 | **.012±.003** | .019±.004 | .040±.006 | .047±.006 | .032±.006 | .031±.005 | .019±.004 |
| | .90 | .025±.005 | .013±.004 | .018±.005 | **.007±.002** | **.007±.002** | .010±.003 | .013±.004 | .015±.004 | .008±.003 | .011±.003 | .013±.003 | .010±.003 | .014±.004 | .040±.006 | .045±.007 | .019±.005 | .018±.004 | .009±.003 |
| | .85 | .019±.005 | .008±.003 | .011±.004 | .006±.003 | .009±.003 | .006±.002 | .008±.003 | .012±.004 | .006±.003 | .010±.004 | .007±.003 | **.004±.002** | .010±.004 | .037±.007 | .045±.007 | .013±.004 | .008±.003 | .005±.002 |
| | .80 | .015±.004 | .005±.002 | .009±.004 | .006±.002 | .007±.003 | .005±.002 | .005±.002 | .009±.004 | .004±.002 | .062±.008 | .004±.002 | **.003±.002** | .010±.004 | .037±.007 | .045±.007 | .007±.003 | .008±.003 | .003±.002 |
| | .75 | .009±.003 | .002±.001 | .006±.003 | .002±.001 | .008±.003 | .004±.002 | .003±.002 | .006±.003 | .002±.001 | .054±.008 | **.001±.001** | .002±.002 | .010±.004 | .035±.007 | .046±.008 | **.001±.001** | .006±.003 | .002±.001 |
| | .70 | .008±.003 | .002±.002 | .005±.002 | .002±.002 | .005±.002 | .002±.002 | .001±.001 | .004±.002 | .003±.002 | .083±.009 | **.000±.000** | **.000±.000** | .008±.004 | .033±.007 | .047±.008 | **.000±.000** | .004±.002 | .002±.001 |
| $\hat{\phi}$ | .99 | .985±.003 | .986±.003 | .993±.002 | .994±.002 | .989±.003 | .984±.003 | .985±.003 | .994±.002 | .996±.002 | .993±.003 | **.991±.003** | .989±.003 | .985±.003 | .987±.003 | .992±.002 | .986±.003 | .985±.004 | .989±.003 |
| | .95 | **.947±.006** | .949±.007 | .934±.007 | .964±.005 | .957±.006 | .945±.006 | .952±.007 | .943±.006 | .963±.007 | .954±.006 | .961±.006 | .945±.007 | .939±.008 | .954±.006 | .946±.007 | .927±.007 | .943±.007 | .954±.006 |
| | .90 | .902±.009 | .906±.008 | .892±.008 | .895±.008 | .902±.008 | .906±.008 | .904±.009 | .889±.009 | .889±.009 | .900±.008 | .917±.007 | .912±.008 | .924±.008 | .875±.010 | .895±.009 | .904±.008 | .869±.009 | .845±.009 |
| | .85 | .863±.010 | .859±.010 | **.848±.009** | .867±.009 | .840±.010 | .857±.009 | .862±.011 | .843±.010 | .864±.009 | .848±.010 | .847±.010 | .860±.009 | .822±.011 | .852±.011 | .837±.009 | .784±.011 | .778±.012 | .864±.010 |
| | .80 | .806±.011 | **.805±.011** | .793±.011 | .807±.011 | .790±.011 | .805±.011 | .806±.011 | .789±.011 | .795±.011 | .801±.012 | .810±.010 | **.800±.011** | .779±.011 | .795±.013 | .784±.013 | .706±.012 | .702±.013 | .809±.011 |
| | .75 | .745±.013 | .741±.011 | .761±.013 | .758±.012 | .731±.013 | .746±.012 | .744±.011 | .751±.012 | .749±.013 | .721±.013 | .742±.012 | .739±.013 | .732±.012 | .728±.015 | .727±.014 | .633±.012 | .638±.013 | **.749±.012** |
| | .70 | .711±.013 | **.700±.013** | .716±.014 | .695±.014 | .699±.014 | .686±.013 | .687±.012 | .701±.013 | .686±.015 | .688±.013 | .682±.013 | .672±.013 | .682±.015 | .694±.015 | .753±.018 | .557±.014 | .571±.014 | .722±.012 |
| $MinCoeff$ | .99 | .995±.017 | 1.000±.017 | **1.000±.017** | 1.004±.016 | .999±.016 | 1.006±.017 | 1.001±.016 | 1.002±.017 | 1.002±.016 | 1.000±.016 | .999±.017 | **1.002±.017** | .996±.017 | 1.006±.017 | **1.001±.017** | **1.002±.017** | 1.007±.017 | 1.007±.017 |
| | .95 | .990±.017 | 1.005±.017 | 1.023±.017 | 1.005±.017 | 1.021±.017 | 1.027±.018 | 1.008±.017 | 1.016±.017 | 1.005±.017 | 1.023±.017 | 1.016±.017 | 1.015±.018 | .992±.018 | 1.014±.016 | **.998±.017** | 1.007±.017 | 1.048±.017 | 1.019±.017 |
| | .90 | .991±.018 | 1.018±.018 | 1.038±.018 | 1.017±.018 | 1.052±.018 | 1.051±.018 | 1.016±.018 | 1.037±.018 | 1.015±.017 | 1.055±.017 | 1.017±.018 | 1.019±.018 | .978±.019 | 1.017±.016 | **.994±.017** | 1.012±.019 | 1.108±.017 | 1.033±.018 |
| | .85 | .988±.019 | 1.026±.018 | 1.061±.019 | 1.030±.017 | .999±.019 | 1.081±.017 | 1.020±.018 | 1.063±.018 | 1.032±.018 | .991±.020 | 1.039±.018 | 1.039±.017 | .963±.020 | 1.029±.017 | **.983±.018** | 1.029±.020 | 1.153±.017 | 1.048±.018 |
| | .80 | .986±.019 | 1.032±.019 | 1.091±.018 | 1.048±.019 | 1.058±.019 | 1.116±.017 | 1.009±.019 | 1.081±.018 | 1.046±.019 | 1.086±.019 | 1.056±.019 | 1.052±.019 | .948±.021 | 1.033±.017 | **.973±.019** | 1.058±.021 | 1.200±.015 | 1.068±.017 |
| | .75 | .993±.020 | 1.053±.019 | 1.107±.017 | 1.054±.020 | 1.224±.014 | 1.159±.016 | 1.042±.019 | 1.108±.018 | 1.024±.021 | 1.137±.017 | 1.071±.019 | 1.078±.019 | .929±.022 | 1.026±.019 | **.982±.020** | 1.076±.023 | 1.250±.012 | 1.103±.018 |
| | .70 | .994±.020 | 1.055±.020 | 1.126±.017 | 1.090±.017 | 1.146±.017 | 1.205±.017 | 1.058±.019 | 1.145±.017 | 1.049±.018 | 1.106±.017 | 1.088±.019 | 1.100±.019 | .900±.022 | 1.032±.020 | **.986±.020** | 1.085±.023 | 1.314±.010 | 1.127±.018 |

Table B36: Results for `pol`: $mean \pm std$ for $\widehat{Err}$ and empirical coverage $\hat{\phi}$.

| Metric | c | DG | SAT | SAT+EM | SelNet | SelNet+EM | SR | SAT+SR | SAT+EM+SR | SelNet+SR | SelNet+EM+SR | ENS | ENS+SR | ConfidNet | SELE | REG | SCross |
|---|---|---|---|---|---|---|---|---|---|---|---|---|---|---|---|---|---|
| $\widehat{Err}$ | .99 | .047±.004 | .049±.004 | .060±.004 | .052±.004 | .061±.005 | .070±.005 | **.046±.004** | .061±.005 | .051±.004 | .061±.005 | .061±.005 | .062±.005 | .075±.005 | .063±.005 | .081±.005 | .105±.006 |
| | .95 | .040±.004 | .038±.004 | .050±.004 | .043±.004 | .058±.004 | .045±.004 | **.026±.004** | .047±.004 | .038±.004 | .063±.004 | .061±.005 | .059±.004 | .063±.005 | .062±.005 | .075±.006 | .064±.005 |
| | .90 | .028±.003 | .023±.003 | .035±.004 | .034±.003 | .042±.004 | .030±.004 | **.017±.003** | .029±.004 | .025±.003 | .063±.005 | .037±.004 | .023±.003 | .045±.005 | .055±.005 | .070±.006 | .059±.005 |
| | .85 | .014±.002 | .010±.002 | .017±.003 | .017±.003 | .035±.003 | .014±.003 | **.006±.002** | .017±.003 | .018±.003 | .049±.005 | .023±.003 | .012±.002 | .029±.003 | .063±.005 | .067±.005 | .032±.004 |
| | .80 | .005±.002 | **.002±.001** | .003±.001 | .003±.001 | .003±.001 | .007±.002 | **.002±.001** | .004±.002 | .004±.001 | .047±.004 | .007±.002 | .003±.001 | .006±.001 | .064±.006 | .065±.005 | .008±.002 |
| | .75 | **.000±.000** | **.000±.000** | .001±.001 | **.000±.000** | **.000±.000** | **.000±.000** | **.000±.000** | .001±.001 | **.000±.000** | .027±.003 | **.000±.000** | **.000±.000** | **.000±.000** | .066±.006 | .064±.006 | .001±.001 |
| | .70 | **.000±.000** | **.000±.000** | .000±.001 | .000±.001 | **.000±.000** | **.000±.000** | **.000±.000** | **.000±.000** | **.000±.000** | .063±.006 | **.000±.000** | **.000±.000** | **.000±.000** | .063±.006 | .061±.005 | **.000±.000** |
| $\hat{\phi}$ | .99 | .994±.002 | .987±.002 | .987±.002 | .988±.002 | .985±.002 | .992±.002 | .981±.002 | .989±.002 | .986±.002 | .986±.002 | .987±.002 | .991±.002 | **.990±.002** | .981±.003 | .987±.002 | .988±.002 |
| | .95 | .949±.004 | **.950±.004** | .958±.003 | .953±.003 | .948±.004 | .938±.004 | .940±.004 | .957±.004 | .943±.004 | .948±.004 | .941±.004 | .957±.004 | .934±.004 | .938±.004 | .938±.004 | .945±.004 |
| | .90 | .898±.006 | .903±.006 | .907±.005 | .899±.006 | .893±.006 | .898±.006 | .898±.006 | .908±.006 | .891±.006 | **.899±.005** | .908±.005 | .898±.006 | .912±.006 | .892±.005 | .893±.006 | .900±.006 |
| | .85 | .846±.007 | .857±.007 | .856±.007 | .850±.007 | .850±.007 | .850±.007 | .848±.008 | .862±.007 | .852±.007 | .857±.007 | .858±.007 | .850±.008 | .864±.007 | **.847±.005** | .845±.007 | .844±.007 |
| | .80 | .802±.008 | .804±.008 | .808±.008 | .802±.008 | .805±.008 | .814±.008 | .802±.008 | .809±.008 | .804±.008 | .798±.008 | .809±.008 | .803±.008 | .802±.008 | **.798±.006** | .795±.008 | .796±.008 |
| | .75 | .767±.008 | .757±.008 | **.752±.008** | .757±.009 | .760±.008 | .756±.008 | .756±.008 | .754±.008 | .761±.009 | .756±.008 | .755±.009 | .753±.009 | .753±.009 | .762±.007 | **.749±.008** | .747±.009 |
| | .70 | .713±.009 | .708±.009 | .707±.009 | .710±.009 | .708±.009 | .712±.009 | .706±.009 | .709±.009 | .717±.009 | .830±.004 | .708±.009 | .712±.009 | .701±.009 | .714±.009 | **.700±.009** | .682±.009 |

Table B37: Results for `retinamnist`: $mean \pm std$ for $\widehat{Err}$ and empirical coverage $\hat{\phi}$.

| Metric | c | DG | SAT | SAT+EM | SelNet | SelNet+EM | SR | SAT+SR | SAT+EM+SR | SelNet+SR | SelNet+EM+SR | ENS | ENS+SR | ConfidNet | SELE | REG | SCross |
|---|---|---|---|---|---|---|---|---|---|---|---|---|---|---|---|---|---|
| $\widehat{Err}$ | .99 | .502±.029 | .463±.028 | .492±.030 | .498±.026 | .453±.030 | .451±.027 | .467±.028 | .488±.030 | .507±.027 | .459±.031 | .485±.028 | **.448±.027** | .485±.026 | .499±.030 | .497±.027 | .471±.027 |
| | .95 | .494±.028 | .445±.029 | .483±.030 | .498±.026 | .527±.029 | .444±.027 | .468±.028 | .475±.027 | .498±.025 | .532±.029 | .443±.027 | **.440±.028** | .476±.027 | .478±.028 | .492±.027 | .459±.027 |
| | .90 | .474±.028 | .435±.031 | .461±.031 | .479±.026 | .503±.029 | .431±.028 | .443±.030 | .456±.031 | .447±.028 | .500±.029 | .453±.029 | **.419±.029** | .459±.028 | .440±.031 | .495±.028 | .455±.027 |
| | .85 | .463±.029 | .424±.032 | .458±.031 | .457±.028 | .489±.032 | .427±.029 | .427±.030 | .438±.030 | .436±.028 | .491±.030 | .445±.029 | **.414±.029** | .440±.031 | .459±.029 | .495±.028 | .435±.028 |
| | .80 | .442±.031 | .425±.032 | .444±.032 | .456±.029 | .456±.029 | .415±.031 | .415±.030 | .430±.031 | .444±.033 | .429±.032 | .452±.029 | **.394±.030** | .408±.031 | .435±.029 | .494±.030 | .419±.028 |
| | .75 | .416±.032 | .404±.032 | .427±.033 | .467±.032 | .466±.033 | **.379±.033** | .404±.030 | .415±.031 | .412±.031 | .503±.031 | .448±.030 | .385±.031 | .405±.031 | .445±.030 | .494±.032 | .412±.029 |
| | .70 | .388±.034 | .380±.034 | .406±.035 | .449±.031 | .416±.032 | **.358±.034** | .385±.031 | .385±.031 | .414±.035 | .414±.035 | .415±.029 | .361±.032 | .401±.033 | .439±.034 | .490±.035 | .393±.031 |
| $\hat{\phi}$ | .99 | .981±.008 | .974±.009 | .992±.005 | .959±.012 | .959±.011 | .988±.006 | .993±.005 | .994±.004 | .984±.008 | .975±.009 | .978±.009 | .987±.007 | .994±.005 | .986±.007 | .975±.010 | 1.000±.000 |
| | .95 | .921±.015 | .921±.014 | .953±.011 | .950±.013 | .966±.010 | .953±.013 | .983±.007 | **.951±.011** | .955±.013 | .977±.009 | .944±.014 | .972±.009 | .972±.009 | .973±.019 | .921±.015 | .972±.010 |
| | .90 | .862±.020 | .877±.019 | **.898±.018** | .934±.014 | .931±.017 | .919±.018 | .910±.015 | .912±.015 | .849±.018 | .889±.016 | .913±.016 | .937±.013 | .903±.018 | .936±.014 | .860±.019 | .940±.015 |
| | .85 | .835±.022 | .821±.022 | .888±.019 | .871±.020 | .818±.022 | **.857±.020** | .857±.020 | .832±.022 | .859±.019 | .844±.022 | .833±.022 | .904±.016 | .819±.023 | .860±.019 | .825±.020 | .907±.018 |
| | .80 | .780±.023 | .800±.022 | .856±.021 | .833±.021 | .779±.024 | .836±.021 | .822±.020 | **.794±.021** | .773±.023 | .753±.027 | .787±.025 | .817±.021 | .745±.026 | .840±.020 | **.804±.020** | .866±.021 |
| | .75 | .724±.027 | .773±.023 | .815±.022 | .775±.024 | .732±.024 | .762±.024 | .767±.022 | **.749±.023** | .682±.025 | .755±.025 | .758±.027 | .780±.024 | .726±.026 | .803±.023 | .736±.023 | .812±.023 |
| | .70 | .685±.028 | .706±.027 | .717±.022 | .718±.024 | .730±.026 | .702±.026 | .742±.022 | .669±.025 | .669±.025 | .683±.023 | .717±.027 | .751±.025 | .696±.027 | .720±.024 | **.690±.024** | .731±.029 |

Table B38: Results for `rl`: *mean ± std* for $\widehat{Err}$, empirical coverage $\hat{\phi}$, and *MinCoeff*.

| Metric | c | DG | SAT | SAT+EM | SelNet | SelNet+EM | SR | SAT+SR | SAT+EM+SR | SelNet+SR | SelNet+EM+SR | ENS | ENS+SR | ConfidNet | SELE | REG | SCross | AUCross | PlugInAUC |
|---|---|---|---|---|---|---|---|---|---|---|---|---|---|---|---|---|---|---|---|
| $\widehat{Err}$ | .99 | .499±.016 | .289±.015 | .271±.014 | .260±.014 | .295±.015 | .261±.015 | .292±.015 | .273±.014 | .256±.014 | .294±.015 | .244±.016 | **.236±.015** | .266±.014 | .331±.016 | .334±.015 | .276±.016 | .274±.016 | .263±.015 |
| | .95 | .497±.016 | .287±.015 | .264±.015 | .236±.015 | .283±.015 | .254±.015 | .285±.016 | .266±.015 | .245±.016 | .285±.016 | .241±.016 | **.225±.016** | .261±.014 | .329±.015 | .335±.015 | .263±.016 | .266±.016 | .256±.015 |
| | .90 | .499±.018 | .274±.015 | .259±.015 | .264±.016 | .276±.016 | .246±.016 | .276±.017 | .260±.015 | .257±.017 | .272±.017 | .229±.016 | **.216±.016** | .259±.015 | .325±.015 | .330±.015 | .245±.016 | .252±.017 | .244±.016 |
| | .85 | .499±.019 | .270±.016 | .248±.016 | .269±.015 | .280±.018 | .229±.016 | .267±.017 | .248±.015 | .250±.015 | .265±.016 | .234±.016 | **.209±.016** | .246±.015 | .325±.016 | .324±.016 | .236±.016 | .229±.017 | .233±.016 |
| | .80 | .506±.020 | .266±.016 | .237±.017 | .231±.016 | .253±.017 | .217±.016 | .258±.017 | .235±.016 | .229±.017 | .249±.017 | .238±.017 | **.204±.017** | .233±.015 | .323±.016 | .321±.016 | .215±.016 | .221±.018 | .230±.017 |
| | .75 | .513±.020 | .262±.016 | .213±.016 | .245±.016 | .252±.018 | .198±.016 | .249±.018 | .217±.016 | .252±.017 | .236±.017 | **.192±.016** | .222±.016 | .319±.016 | .302±.016 | .200±.017 | .211±.018 | .223±.017 |
| | .70 | .510±.021 | .256±.017 | .207±.017 | .224±.018 | .241±.018 | .195±.017 | .231±.019 | .202±.017 | .219±.017 | .233±.017 | .227±.017 | **.175±.016** | .215±.015 | .319±.017 | .293±.016 | .188±.018 | .193±.017 | .205±.017 |
| $\hat{\phi}$ | .99 | .992±.003 | .991±.003 | .986±.003 | .992±.003 | .982±.004 | .984±.003 | .998±.002 | .992±.003 | .994±.003 | .981±.004 | **.990±.003** | .971±.005 | .960±.005 | .986±.004 | .989±.004 | .979±.005 | .987±.003 | .994±.002 |
| | .95 | .974±.005 | .942±.007 | .939±.008 | .942±.008 | .950±.007 | .943±.007 | .967±.006 | **.950±.007** | .973±.005 | .943±.007 | .967±.005 | .927±.008 | .941±.008 | .955±.007 | .968±.005 | .905±.008 | .919±.009 | .943±.008 |
| | .90 | .909±.009 | **.901±.009** | .896±.010 | .898±.010 | .902±.010 | .902±.008 | .897±.010 | .897±.010 | .879±.010 | .881±.010 | .914±.009 | .885±.009 | .878±.010 | .926±.009 | .928±.007 | .826±.012 | .841±.012 | .878±.011 |
| | .85 | .871±.010 | .843±.011 | .842±.012 | .858±.014 | .845±.011 | .846±.011 | .853±.012 | .839±.011 | .831±.013 | **.847±.010** | .854±.012 | .848±.011 | .840±.012 | .880±.011 | .869±.010 | .765±.013 | .764±.013 | .821±.012 |
| | .80 | .822±.012 | .812±.012 | .787±.014 | .814±.011 | .802±.014 | .794±.013 | **.803±.012** | .777±.014 | .815±.012 | .801±.013 | .804±.014 | .786±.013 | .788±.014 | .834±.011 | .830±.011 | .703±.014 | .704±.014 | **.800±.013** |
| | .75 | .785±.013 | .779±.014 | .716±.015 | .781±.012 | .753±.014 | .739±.014 | **.752±.013** | .730±.015 | .744±.015 | .739±.014 | .770±.013 | .739±.014 | .746±.013 | .753±.013 | .757±.013 | .647±.015 | .667±.016 | .759±.013 |
| | .70 | .728±.015 | .723±.015 | .670±.016 | .678±.016 | .718±.015 | **.701±.014** | .704±.014 | .687±.016 | .700±.015 | .705±.015 | .724±.014 | .691±.015 | .696±.015 | .727±.013 | .713±.014 | .590±.017 | .608±.017 | .713±.015 |
| *MinCoeff* | .99 | **.999±.032** | 1.004±.032 | .999±.033 | .997±.033 | .999±.033 | 1.002±.033 | 1.003±.032 | 1.001±.033 | 1.001±.032 | .998±.032 | 1.009±.033 | 1.005±.034 | 1.007±.033 | .999±.033 | 1.004±.032 | **.999±.032** | 1.004±.033 | 1.005±.033 |
| | .95 | .994±.033 | .998±.033 | .996±.033 | 1.008±.034 | .992±.034 | 1.003±.032 | 1.003±.033 | 1.007±.034 | 1.013±.033 | 1.002±.033 | 1.018±.034 | 1.008±.034 | .999±.034 | .997±.034 | **.995±.032** | .980±.035 | 1.022±.034 | 1.017±.034 |
| | .90 | .998±.036 | .991±.033 | .992±.033 | 1.014±.033 | .992±.035 | **1.000±.033** | 1.008±.035 | 1.000±.035 | 1.019±.034 | 1.021±.034 | 1.030±.036 | 1.020±.034 | .995±.036 | .998±.035 | **.999±.033** | .982±.037 | 1.040±.036 | 1.031±.035 |
| | .85 | .998±.038 | **.993±.033** | .994±.034 | 1.020±.036 | .987±.036 | 1.009±.036 | .992±.036 | .994±.036 | 1.044±.034 | 1.021±.035 | 1.031±.035 | 1.019±.035 | .993±.037 | 1.006±.035 | .994±.035 | 1.004±.038 | 1.048±.039 | 1.040±.035 |
| | .80 | 1.012±.039 | .999±.033 | .984±.038 | 1.014±.036 | .990±.038 | 1.016±.037 | .982±.037 | 1.002±.038 | .993±.035 | .989±.035 | 1.053±.036 | 1.040±.035 | .975±.038 | 1.021±.035 | .983±.037 | 1.001±.040 | 1.023±.040 | 1.038±.035 |
| | .75 | 1.026±.040 | **1.003±.034** | .952±.038 | **1.005±.035** | .999±.038 | 1.028±.039 | 1.000±.038 | .983±.038 | 1.002±.036 | 1.016±.037 | 1.062±.037 | 1.045±.035 | .981±.038 | 1.012±.037 | .974±.038 | .997±.040 | 1.036±.040 | 1.040±.036 |
| | .70 | 1.019±.042 | **1.008±.036** | .979±.039 | 1.033±.039 | .990±.039 | 1.027±.039 | .986±.038 | .985±.037 | .985±.038 | .971±.036 | 1.073±.038 | 1.045±.037 | .992±.040 | 1.007±.037 | .971±.040 | .985±.042 | 1.062±.040 | 1.062±.038 |

Table B39: Results for `stanfordcars`: *mean ± std* for $\widehat{Err}$ and empirical coverage $\hat{\phi}$.

| Metric | c | DG | SAT | SAT+EM | SelNet | SelNet+EM | SR | SAT+SR | SAT+EM+SR | SelNet+SR | SelNet+EM+SR | ENS | ENS+SR | ConfidNet | SELE | REG | SCross |
|---|---|---|---|---|---|---|---|---|---|---|---|---|---|---|---|---|---|
| $\widehat{Err}$ | .99 | .236±.008 | .178±.007 | .170±.006 | .318±.009 | .469±.008 | .173±.007 | .165±.006 | .317±.008 | .469±.008 | **.107±.005** | .109±.006 | .230±.008 | .429±.009 | .431±.009 | .162±.007 |
| | .95 | .223±.008 | .165±.007 | .150±.006 | .285±.007 | .442±.009 | .145±.006 | .160±.007 | .143±.006 | .263±.008 | .430±.009 | .091±.005 | **.086±.005** | .224±.008 | .416±.009 | .433±.009 | .141±.007 |
| | .90 | .204±.008 | .146±.007 | .123±.006 | .281±.008 | .433±.008 | .116±.005 | .131±.006 | .115±.006 | .245±.007 | .411±.008 | .075±.005 | **.064±.004** | .211±.008 | .405±.009 | .437±.010 | .123±.006 |
| | .85 | .190±.008 | .127±.006 | .109±.006 | .276±.009 | .378±.010 | .096±.005 | .114±.006 | .097±.005 | .224±.008 | .349±.010 | .067±.004 | **.048±.004** | .199±.008 | .395±.010 | .433±.010 | .100±.006 |
| | .80 | .176±.008 | .111±.006 | .098±.006 | .274±.008 | .393±.010 | .075±.005 | .099±.006 | .078±.005 | .208±.008 | .372±.010 | .060±.005 | **.039±.004** | .182±.008 | .382±.010 | .437±.010 | .085±.006 |
| | .75 | .157±.008 | .099±.006 | .089±.006 | .268±.009 | .377±.008 | .058±.004 | .081±.006 | .061±.004 | .174±.008 | .340±.009 | .056±.005 | **.028±.004** | .173±.008 | .373±.010 | .435±.010 | .067±.005 |
| | .70 | .148±.008 | .081±.006 | .072±.005 | .256±.009 | .316±.009 | .046±.004 | .068±.005 | .046±.004 | .166±.008 | .277±.010 | .050±.004 | **.022±.003** | .164±.008 | .362±.011 | .438±.011 | .056±.005 |
| $\hat{\phi}$ | .99 | .988±.002 | .994±.001 | .994±.001 | .984±.002 | .989±.002 | **.989±.002** | .987±.002 | .988±.002 | .993±.001 | .991±.002 | **.990±.002** | .991±.002 | .986±.002 | .992±.002 | .987±.002 | .993±.001 |
| | .95 | .958±.003 | .966±.003 | .955±.004 | .958±.004 | **.948±.003** | .954±.003 | .966±.003 | **.952±.003** | .954±.003 | **.949±.003** | .958±.004 | .954±.004 | .954±.004 | .954±.003 | .952±.004 | .960±.003 |
| | .90 | .910±.005 | .925±.004 | .907±.005 | .886±.006 | **.900±.005** | .896±.005 | .917±.004 | .904±.004 | .911±.004 | **.899±.005** | .903±.005 | **.901±.005** | .919±.005 | .911±.005 | .906±.005 | .922±.004 |
| | .85 | .868±.006 | .875±.005 | .861±.006 | .840±.007 | **.802±.007** | .792±.006 | .866±.006 | .857±.005 | .855±.006 | .843±.006 | .855±.006 | **.852±.006** | .866±.005 | .862±.006 | .867±.006 | .875±.005 |
| | .80 | .815±.006 | .833±.006 | .814±.006 | .804±.007 | **.802±.007** | .792±.006 | .817±.007 | **.804±.006** | .815±.006 | .805±.007 | **.804±.006** | .806±.007 | .813±.006 | .818±.006 | .810±.007 | .833±.006 |
| | .75 | .761±.007 | .789±.007 | .775±.007 | .761±.008 | .757±.007 | **.748±.007** | .776±.008 | .755±.007 | .764±.008 | .761±.008 | .756±.007 | **.751±.007** | .768±.007 | .766±.006 | .756±.008 | .790±.007 |
| | .70 | .715±.007 | .733±.007 | .716±.007 | .712±.007 | .713±.007 | .708±.007 | .730±.008 | .707±.007 | .716±.008 | .715±.008 | **.706±.007** | **.706±.007** | .727±.007 | .719±.007 | .715±.008 | .748±.007 |

Table B40: Results for `SVHN`: *mean ± std* for $\widehat{Err}$ and empirical coverage $\hat{\phi}$.

| Metric | c | DG | SAT | SAT+EM | SelNet | SelNet+EM | SR | SAT+SR | SAT+EM+SR | SelNet+SR | SelNet+EM+SR | ENS | ENS+SR | ConfidNet | SELE | REG | SCross |
|---|---|---|---|---|---|---|---|---|---|---|---|---|---|---|---|---|---|
| $\widehat{Err}$ | .99 | .038±.002 | **.030±.001** | .037±.001 | **.030±.001** | .043±.002 | .039±.001 | **.030±.001** | .036±.001 | .031±.001 | .043±.002 | .036±.001 | .034±.001 | .036±.001 | .050±.002 | .050±.002 | .038±.002 |
| | .95 | .022±.001 | .017±.001 | .020±.001 | **.016±.001** | .026±.001 | .021±.001 | .017±.001 | .020±.001 | **.016±.001** | .024±.001 | .021±.001 | .018±.001 | .019±.001 | .049±.002 | .050±.002 | .021±.001 |
| | .90 | .013±.001 | **.008±.001** | .011±.001 | .009±.001 | .014±.001 | .011±.001 | **.008±.001** | .010±.001 | .009±.001 | .013±.001 | .011±.001 | .010±.001 | .009±.001 | .048±.002 | .050±.002 | .011±.001 |
| | .85 | .012±.001 | **.006±.001** | .007±.001 | **.006±.001** | .009±.001 | .007±.001 | **.006±.001** | .008±.001 | **.006±.001** | .009±.001 | .007±.001 | .007±.001 | .007±.001 | .045±.001 | .050±.002 | .008±.001 |
| | .80 | .011±.001 | **.005±.001** | .006±.001 | **.005±.001** | .008±.001 | .006±.001 | .006±.001 | **.005±.001** | **.005±.001** | .007±.001 | .006±.001 | **.005±.001** | .006±.001 | .042±.002 | .050±.002 | .006±.001 |
| | .75 | .011±.001 | **.004±.001** | .005±.001 | .005±.001 | .005±.001 | .005±.001 | .005±.001 | .005±.001 | .005±.001 | .005±.001 | .005±.001 | .005±.001 | .006±.001 | .037±.002 | .051±.002 | .005±.001 |
| | .70 | .010±.001 | **.004±.001** | .005±.001 | .005±.001 | .005±.001 | **.004±.001** | **.004±.001** | .005±.001 | .005±.001 | **.004±.001** | .005±.001 | **.004±.001** | .005±.001 | .035±.002 | .051±.002 | **.004±.001** |
| $\hat{\phi}$ | .99 | .991±.001 | **.990±.001** | .991±.001 | **.990±.001** | .992±.001 | .991±.001 | .991±.001 | .992±.001 | .991±.001 | .992±.001 | .990±.001 | .991±.001 | .993±.001 | **.990±.001** | .991±.001 | .991±.001 |
| | .95 | .953±.002 | .959±.001 | .952±.002 | .954±.002 | .955±.001 | .952±.002 | .960±.001 | .953±.002 | .954±.001 | .952±.002 | .953±.002 | .953±.002 | .955±.001 | .947±.002 | **.951±.002** | .952±.001 |
| | .90 | .903±.002 | .905±.002 | .906±.002 | .904±.002 | .905±.002 | .905±.002 | .906±.002 | .905±.002 | .906±.002 | .906±.002 | .903±.002 | .904±.002 | .904±.002 | **.900±.002** | .897±.002 | .909±.002 |
| | .85 | .855±.003 | .855±.003 | .855±.002 | .858±.003 | .858±.002 | .855±.003 | .858±.003 | **.853±.002** | .855±.003 | .857±.003 | .856±.002 | .856±.003 | .856±.002 | .847±.003 | **.852±.003** | .865±.003 |
| | .80 | **.799±.003** | .807±.003 | .805±.003 | .832±.003 | .808±.003 | .807±.003 | .809±.003 | .802±.003 | .808±.003 | .814±.003 | .808±.003 | .807±.003 | .802±.003 | **.800±.003** | .804±.003 | .821±.003 |
| | .75 | **.753±.003** | .758±.003 | .755±.003 | .807±.003 | **.753±.003** | .758±.003 | .755±.003 | .756±.003 | .759±.003 | .755±.003 | .760±.003 | .761±.003 | **.753±.003** | .746±.003 | .754±.003 | .776±.003 |
| | .70 | .703±.003 | .705±.003 | .709±.003 | .712±.003 | .713±.003 | .709±.003 | .707±.003 | .711±.003 | .712±.003 | .710±.003 | .707±.003 | .708±.003 | .706±.003 | .695±.003 | **.702±.003** | .738±.003 |

Table B41: Results for `tissuemnist`: *mean ± std* for $\widehat{Err}$ and empirical coverage $\hat{\phi}$.

| Metric | c | DG | SAT | SAT+EM | SelNet | SelNet+EM | SR | SAT+SR | SAT+EM+SR | SelNet+SR | SelNet+EM+SR | ENS | ENS+SR | ConfidNet | SELE | REG | SCross |
|---|---|---|---|---|---|---|---|---|---|---|---|---|---|---|---|---|---|
| $\widehat{Err}$ | .99 | .331±.002 | .308±.001 | .310±.002 | .310±.002 | .357±.002 | .309±.002 | .309±.002 | .309±.002 | .309±.002 | .356±.002 | .297±.002 | **.295±.002** | .303±.002 | .370±.002 | .386±.002 | .314±.002 |
| | .95 | .319±.002 | .295±.002 | .296±.002 | .296±.002 | .347±.002 | .302±.002 | .293±.002 | .294±.002 | .295±.002 | .344±.002 | .286±.002 | **.278±.002** | .296±.002 | .359±.002 | .386±.002 | .298±.002 |
| | .90 | .305±.002 | .278±.002 | .278±.002 | .278±.002 | .334±.002 | .283±.002 | .273±.002 | .274±.002 | .273±.002 | .340±.002 | .274±.002 | **.258±.002** | .287±.002 | .347±.002 | .387±.002 | .280±.002 |
| | .85 | .292±.002 | .262±.002 | .261±.002 | .261±.002 | .321±.002 | .265±.002 | .256±.002 | .257±.002 | .255±.002 | .343±.002 | .260±.002 | **.239±.002** | .277±.002 | .333±.002 | .388±.002 | .262±.002 |
| | .80 | .279±.003 | .245±.002 | .243±.002 | .239±.002 | .302±.002 | .249±.002 | .238±.002 | .239±.002 | .235±.002 | .335±.003 | .247±.002 | **.221±.002** | .265±.002 | .320±.002 | .388±.002 | .246±.002 |
| | .75 | .268±.003 | .228±.002 | .226±.002 | .222±.002 | .288±.002 | .230±.002 | .222±.002 | .223±.002 | .217±.002 | .327±.003 | .232±.002 | **.204±.002** | .256±.002 | .306±.002 | .391±.002 | .227±.002 |
| | .70 | .256±.003 | .211±.002 | .208±.002 | .210±.002 | .270±.003 | .212±.002 | .203±.002 | .204±.002 | .202±.002 | .317±.003 | .218±.002 | **.187±.002** | .244±.002 | .291±.002 | .391±.002 | .208±.002 |
| $\hat{\phi}$ | .99 | .991±.000 | .988±.000 | .991±.000 | .990±.000 | .990±.000 | .989±.000 | **.990±.000** | .991±.000 | .991±.000 | .989±.000 | **.990±.000** | .988±.000 | .990±.000 | **.990±.000** | .990±.000 | **.990±.000** |
| | .95 | .952±.001 | .950±.001 | **.951±.001** | **.950±.001** | **.950±.001** | .946±.001 | **.950±.001** | **.951±.001** | **.951±.001** | **.949±.001** | **.949±.001** | .945±.001 | .952±.001 | .948±.001 | .952±.001 | **.949±.001** |
| | .90 | **.899±.001** | **.899±.001** | **.899±.001** | **.900±.001** | **.901±.001** | .895±.002 | **.899±.001** | .898±.001 | .898±.002 | **.901±.001** | .894±.002 | .893±.001 | **.900±.002** | .897±.001 | **.901±.002** | **.899±.001** |
| | .85 | .848±.002 | .849±.002 | **.850±.002** | .853±.002 | .848±.002 | .846±.002 | .848±.002 | **.850±.002** | **.850±.002** | .847±.002 | .843±.002 | .842±.002 | .851±.002 | .845±.002 | .855±.002 | **.850±.002** |
| | .80 | .795±.002 | .797±.002 | **.800±.002** | .798±.002 | .796±.002 | .797±.002 | **.801±.002** | .802±.002 | .798±.002 | **.800±.002** | .796±.002 | .793±.002 | .802±.002 | .797±.002 | .805±.002 | .801±.002 |
| | .75 | .747±.002 | .748±.002 | **.750±.002** | **.751±.002** | .749±.002 | .745±.002 | .753±.002 | .753±.002 | .754±.002 | **.750±.002** | .746±.002 | .747±.002 | .752±.002 | .748±.002 | .753±.002 | .753±.002 |
| | .70 | .698±.002 | **.698±.002** | **.700±.002** | **.702±.002** | .698±.002 | .699±.002 | **.700±.002** | .705±.002 | .704±.002 | .697±.002 | .698±.002 | **.700±.002** | .702±.002 | .695±.002 | .703±.002 | .702±.002 |

Table B42: Results for `ucicredit`: *mean±std* for $\widehat{Err}$, empirical coverage $\hat{\phi}$, and *MinCoeff*.

| Metric | c | DG | SAT | SAT+EM | SelNet | SelNet+EM | SR | SAT+SR | SAT+EM+SR | SelNet+SR | SelNet+EM+SR | ENS | ENS+SR | ConfidNet | SELE | REG | SCross | AUCross | PlugInAUC |
|---|---|---|---|---|---|---|---|---|---|---|---|---|---|---|---|---|---|---|---|
| $\widehat{Err}$ | .99 | .181±.004 | .182±.005 | .179±.004 | .177±.004 | .179±.004 | .179±.005 | .181±.004 | .177±.004 | **.176±.004** | .179±.004 | .181±.004 | .178±.005 | .180±.004 | .181±.005 | .188±.004 | .180±.005 | .184±.005 | .184±.005 |
| | .95 | .170±.004 | .172±.005 | **.166±.004** | **.166±.004** | .170±.004 | .168±.005 | .168±.004 | .167±.004 | .169±.004 | .177±.004 | .164±.005 | .171±.005 | .186±.005 | .170±.004 | .184±.005 | .185±.005 |
| | .90 | .160±.004 | .159±.004 | .154±.004 | **.153±.004** | .156±.004 | **.153±.004** | .156±.004 | .155±.004 | **.153±.004** | .158±.004 | .174±.005 | .154±.005 | .155±.005 | .161±.004 | .184±.005 | .155±.004 | .186±.005 | .186±.005 |
| | .85 | .151±.004 | .148±.005 | .146±.004 | .144±.004 | **.143±.004** | .144±.004 | .146±.004 | .144±.004 | .143±.005 | .151±.004 | .171±.005 | .146±.004 | .146±.004 | .151±.005 | .180±.005 | .146±.004 | .187±.005 | .189±.005 |
| | .80 | .144±.004 | .140±.005 | .139±.004 | .137±.004 | .139±.004 | .137±.004 | .139±.004 | .140±.004 | .143±.005 | .143±.005 | .167±.005 | .138±.004 | **.134±.004** | .143±.005 | .177±.005 | .138±.004 | .186±.005 | .190±.005 |
| | .75 | .139±.005 | **.128±.005** | .132±.005 | .131±.004 | .130±.005 | .131±.004 | .134±.004 | .132±.005 | .139±.005 | .133±.004 | .163±.005 | .130±.005 | .129±.005 | .135±.005 | .175±.006 | .130±.004 | .188±.005 | .190±.005 |
| | .70 | .136±.005 | .123±.004 | .126±.005 | .126±.005 | .125±.005 | .126±.005 | **.122±.004** | .126±.005 | .130±.005 | .146±.005 | .159±.005 | .125±.005 | .125±.005 | .128±.005 | .171±.005 | .124±.005 | .190±.006 | .191±.006 |
| $\hat{\phi}$ | .99 | .989±.001 | **.991±.001** | .990±.001 | .989±.001 | .987±.001 | .990±.001 | **.991±.001** | .987±.001 | .988±.001 | **.989±.001** | **.990±.001** | .990±.001 | .992±.001 | **.989±.001** | .992±.001 | **.990±.001** | .989±.001 |
| | .95 | **.951±.003** | .957±.002 | .947±.003 | **.951±.003** | .946±.003 | .952±.003 | .952±.003 | .945±.003 | **.950±.003** | .952±.003 | .942±.003 | .950±.003 | .953±.003 | .948±.003 | .944±.003 | .958±.003 | .953±.003 | .948±.003 |
| | .90 | .902±.004 | **.898±.004** | .899±.004 | .896±.004 | .896±.004 | .897±.004 | **.899±.004** | .900±.004 | .898±.004 | .904±.004 | **.901±.004** | .901±.004 | .903±.004 | .892±.004 | .897±.004 | .908±.004 | .902±.004 | .897±.004 |
| | .85 | .851±.005 | .843±.005 | .848±.005 | .846±.005 | .845±.004 | .849±.005 | .860±.005 | .849±.005 | .858±.005 | .857±.005 | **.849±.004** | .857±.005 | .848±.005 | .840±.005 | .844±.005 | .863±.005 | .852±.004 | .849±.005 |
| | .80 | .803±.005 | .798±.005 | .794±.006 | **.800±.005** | **.799±.005** | .804±.005 | .805±.005 | .801±.005 | .811±.005 | .801±.005 | .803±.005 | .806±.005 | .795±.006 | .796±.005 | .790±.005 | .813±.005 | .806±.005 | .794±.006 |
| | .75 | .743±.006 | .741±.006 | .745±.006 | .756±.006 | .752±.006 | .756±.006 | .758±.006 | .760±.006 | .766±.006 | **.750±.006** | .760±.005 | .751±.006 | .750±.006 | .746±.006 | .742±.006 | .758±.006 | .750±.006 | .744±.006 |
| | .70 | **.701±.006** | .691±.007 | .699±.007 | .710±.006 | .698±.006 | **.701±.006** | .705±.006 | .709±.007 | .714±.006 | **.701±.006** | .715±.006 | .706±.007 | .710±.007 | .705±.006 | .690±.006 | **.701±.006** | .698±.007 | .693±.007 |
| *MinCoeff* | .99 | .991±.021 | .987±.021 | .994±.021 | .986±.021 | .979±.021 | .990±.021 | .988±.021 | .993±.021 | .983±.022 | .987±.021 | **1.000±.021** | .993±.021 | .987±.021 | .979±.021 | .989±.021 | .993±.021 | 1.006±.021 | 1.006±.021 |
| | .95 | .943±.022 | .948±.022 | .941±.021 | .940±.022 | .922±.021 | .953±.022 | .951±.022 | .937±.022 | .942±.021 | .942±.021 | **.988±.022** | .953±.023 | .947±.021 | .916±.020 | .980±.022 | .958±.022 | 1.015±.021 | 1.021±.022 |
| | .90 | .893±.022 | .881±.022 | .871±.022 | .871±.023 | .843±.022 | .891±.021 | .899±.022 | .893±.021 | .872±.022 | .891±.022 | **.980±.022** | .897±.022 | .898±.022 | .816±.020 | .966±.023 | .908±.021 | 1.030±.022 | 1.034±.022 |
| | .85 | .813±.022 | .805±.023 | .773±.022 | .763±.020 | .754±.021 | .835±.022 | .856±.023 | .834±.022 | .819±.022 | .847±.023 | .950±.023 | .859±.022 | .811±.022 | .748±.021 | .943±.023 | .859±.022 | **1.046±.023** | 1.058±.022 |
| | .80 | .748±.020 | .743±.023 | .693±.022 | .620±.019 | .624±.019 | .771±.021 | .746±.021 | .774±.022 | .824±.026 | .661±.022 | .909±.022 | .781±.021 | .706±.021 | .702±.021 | .926±.025 | .796±.023 | **1.057±.024** | 1.079±.024 |
| | .75 | .701±.021 | .667±.023 | .631±.021 | .592±.020 | .589±.021 | .692±.022 | .736±.023 | .709±.023 | .824±.026 | .661±.022 | .909±.023 | .688±.022 | .618±.021 | .641±.021 | **.924±.026** | .729±.022 | 1.081±.025 | 1.090±.025 |
| | .70 | .682±.022 | .617±.022 | .588±.021 | .568±.021 | .564±.021 | .629±.022 | .664±.024 | .637±.021 | .787±.025 | .843±.027 | .873±.024 | .636±.021 | .581±.022 | .600±.021 | **.904±.027** | .645±.021 | 1.109±.026 | 1.113±.025 |

Table B43: Results for `upselling`: $mean \pm std$ for $\widehat{Err}$, empirical coverage $\hat{\phi}$, and $MinCoeff$.

| Metric | c | DG | SAT | SAT+EM | SelNet | SelNet+EM | SR | SAT+SR | SAT+EM+SR | SelNet+SR | SelNet+EM+SR | ENS | ENS+SR | ConfidNet | SELE | REG | SCross | AUCross | PlugInAUC |
|---|---|---|---|---|---|---|---|---|---|---|---|---|---|---|---|---|---|---|---|
| $\widehat{Err}$ | .99 | .277 ± .014 | .168 ± .013 | .171 ± .014 | .164 ± .013 | .178 ± .014 | .186 ± .015 | .169 ± .014 | .171 ± .014 | **.161 ± .013** | .179 ± .013 | .189 ± .013 | .186 ± .013 | .177 ± .012 | .185 ± .014 | .185 ± .013 | .177 ± .012 | .185 ± .012 | .191 ± .015 |
| | .95 | .278 ± .014 | .160 ± .013 | .163 ± .013 | .162 ± .013 | .171 ± .014 | .174 ± .015 | .163 ± .013 | .159 ± .013 | **.152 ± .013** | .171 ± .013 | .195 ± .014 | .168 ± .013 | .166 ± .012 | .179 ± .014 | .178 ± .013 | .177 ± .012 | .185 ± .012 | .183 ± .015 |
| | .90 | .276 ± .015 | .150 ± .014 | .150 ± .013 | .145 ± .012 | .155 ± .013 | .163 ± .014 | **.142 ± .013** | **.142 ± .013** | **.142 ± .013** | .156 ± .013 | .208 ± .014 | .156 ± .013 | .148 ± .012 | .164 ± .014 | .172 ± .014 | .177 ± .012 | .185 ± .012 | .164 ± .015 |
| | .85 | .275 ± .015 | .136 ± .013 | .134 ± .013 | **.122 ± .012** | .139 ± .013 | .145 ± .014 | .126 ± .013 | .129 ± .012 | .128 ± .013 | .137 ± .013 | .221 ± .015 | .142 ± .013 | .138 ± .012 | .162 ± .014 | .168 ± .014 | .177 ± .012 | .185 ± .012 | .157 ± .015 |
| | .80 | .291 ± .016 | **.104 ± .012** | .121 ± .013 | .115 ± .012 | .121 ± .012 | .130 ± .013 | .107 ± .012 | .116 ± .012 | .110 ± .013 | .124 ± .012 | .230 ± .016 | .118 ± .011 | .114 ± .012 | .144 ± .014 | .168 ± .015 | .176 ± .012 | .185 ± .012 | .147 ± .014 |
| | .75 | .297 ± .016 | .094 ± .011 | .110 ± .012 | .104 ± .012 | .102 ± .011 | .106 ± .012 | **.092 ± .012** | .103 ± .012 | .101 ± .012 | .105 ± .012 | .244 ± .016 | .095 ± .011 | .098 ± .012 | .128 ± .014 | .164 ± .014 | .171 ± .012 | .185 ± .012 | .123 ± .013 |
| | .70 | .304 ± .017 | .078 ± .011 | .081 ± .010 | .075 ± .010 | .080 ± .010 | .080 ± .011 | .080 ± .011 | .086 ± .011 | **.077 ± .011** | .077 ± .010 | .257 ± .017 | .076 ± .010 | **.074 ± .010** | .120 ± .013 | .158 ± .014 | .171 ± .012 | .185 ± .012 | .108 ± .013 |
| $\hat{\phi}$ | .99 | .994 ± .002 | .988 ± .004 | .985 ± .004 | .993 ± .003 | .984 ± .004 | .989 ± .004 | .988 ± .004 | .995 ± .002 | .984 ± .004 | .994 ± .002 | .992 ± .003 | **.990 ± .003** | .993 ± .003 | .990 ± .003 | **.991 ± .003** | .978 ± .005 | .998 ± .001 | .996 ± .002 |
| | .95 | .948 ± .007 | .950 ± .007 | **.951 ± .007** | .964 ± .006 | .950 ± .008 | .951 ± .008 | .965 ± .006 | .953 ± .007 | .940 ± .009 | .944 ± .007 | .961 ± .006 | .941 ± .007 | .953 ± .006 | .962 ± .006 | .939 ± .007 | .978 ± .005 | .998 ± .001 | .955 ± .006 |
| | .90 | .888 ± .010 | .906 ± .010 | .883 ± .011 | .915 ± .010 | .918 ± .009 | .923 ± .010 | .884 ± .012 | .887 ± .011 | .902 ± .011 | .912 ± .010 | **.900 ± .010** | .895 ± .010 | .913 ± .010 | .907 ± .008 | .881 ± .009 | .978 ± .005 | .998 ± .001 | .890 ± .010 |
| | .85 | .833 ± .012 | .860 ± .012 | .842 ± .012 | .841 ± .013 | .863 ± .013 | .879 ± .011 | .841 ± .013 | .849 ± .013 | .851 ± .013 | .848 ± .012 | .847 ± .011 | .858 ± .012 | .867 ± .012 | .881 ± .009 | **.852 ± .010** | .978 ± .005 | .998 ± .001 | .872 ± .011 |
| | .80 | .759 ± .014 | .808 ± .014 | .800 ± .013 | .805 ± .013 | .801 ± .014 | .833 ± .012 | .803 ± .014 | .807 ± .014 | .795 ± .015 | .816 ± .013 | .810 ± .012 | .809 ± .013 | .822 ± .014 | .823 ± .011 | **.798 ± .012** | .977 ± .005 | .998 ± .001 | .825 ± .012 |
| | .75 | .718 ± .015 | .764 ± .014 | .758 ± .014 | .772 ± .014 | .758 ± .014 | .786 ± .014 | .758 ± .015 | .768 ± .014 | .769 ± .015 | .760 ± .014 | .760 ± .012 | .768 ± .015 | .772 ± .015 | .762 ± .014 | **.754 ± .013** | .964 ± .006 | .998 ± .001 | .772 ± .013 |
| | .70 | .677 ± .015 | .724 ± .015 | **.702 ± .015** | .722 ± .015 | .706 ± .016 | .725 ± .015 | .721 ± .016 | .739 ± .015 | .722 ± .016 | .724 ± .015 | .722 ± .014 | .709 ± .015 | .723 ± .015 | .730 ± .014 | .717 ± .012 | .964 ± .006 | .998 ± .001 | .721 ± .014 |
| $MinCoeff$ | .99 | 1.000 ± .031 | 1.003 ± .031 | 1.004 ± .031 | 1.002 ± .031 | 1.001 ± .031 | 1.001 ± .030 | 1.005 ± .030 | 1.002 ± .030 | **1.000 ± .031** | .999 ± .030 | 1.009 ± .031 | 1.001 ± .031 | 1.003 ± .030 | 1.001 ± .031 | .999 ± .031 | 1.003 ± .031 | 1.001 ± .031 | 1.001 ± .031 |
| | .95 | .977 ± .032 | **1.000 ± .031** | 1.004 ± .033 | **1.001 ± .031** | **1.001 ± .031** | .995 ± .031 | 1.005 ± .031 | 1.016 ± .030 | 1.004 ± .031 | 1.000 ± .031 | .985 ± .032 | 1.011 ± .031 | **.997 ± .031** | **1.005 ± .031** | 1.011 ± .032 | 1.003 ± .031 | **1.001 ± .031** | 1.013 ± .031 |
| | .90 | .950 ± .033 | .998 ± .031 | 1.017 ± .034 | .994 ± .032 | 1.004 ± .031 | .997 ± .032 | .993 ± .032 | 1.020 ± .031 | **1.001 ± .032** | 1.007 ± .031 | .937 ± .034 | 1.012 ± .031 | .996 ± .031 | 1.027 ± .032 | 1.007 ± .032 | 1.003 ± .031 | 1.001 ± .031 | 1.021 ± .032 |
| | .85 | .921 ± .034 | .999 ± .031 | 1.018 ± .035 | .991 ± .033 | 1.005 ± .033 | 1.004 ± .033 | **.996 ± .032** | 1.024 ± .031 | .992 ± .032 | 1.018 ± .032 | .885 ± .035 | 1.006 ± .031 | 1.005 ± .030 | 1.035 ± .032 | .998 ± .033 | 1.003 ± .031 | 1.001 ± .031 | 1.022 ± .032 |
| | .80 | .840 ± .034 | 1.008 ± .032 | 1.022 ± .034 | 1.007 ± .032 | 1.011 ± .032 | 1.007 ± .033 | .997 ± .032 | 1.026 ± .032 | 1.009 ± .033 | 1.004 ± .033 | .839 ± .035 | 1.012 ± .032 | .997 ± .032 | 1.048 ± .034 | .984 ± .033 | 1.002 ± .031 | **1.001 ± .031** | 1.041 ± .033 |
| | .75 | .799 ± .034 | 1.018 ± .034 | 1.021 ± .036 | 1.019 ± .033 | 1.013 ± .033 | 1.019 ± .034 | 1.013 ± .033 | 1.015 ± .033 | 1.016 ± .034 | 1.016 ± .033 | .781 ± .035 | 1.012 ± .033 | 1.015 ± .033 | 1.077 ± .034 | .976 ± .034 | 1.000 ± .032 | **1.001 ± .031** | 1.050 ± .035 |
| | .70 | .746 ± .036 | 1.031 ± .035 | 1.039 ± .037 | 1.026 ± .034 | 1.042 ± .035 | 1.029 ± .035 | 1.020 ± .035 | 1.025 ± .034 | 1.017 ± .035 | 1.022 ± .033 | .781 ± .036 | 1.023 ± .036 | 1.024 ± .034 | 1.095 ± .036 | .965 ± .035 | 1.000 ± .032 | **1.001 ± .031** | 1.066 ± .037 |

Table B44: Results for `waterbirds`: $mean \pm std$ for $\widehat{Err}$, empirical coverage $\hat{\phi}$, and $MinCoeff$.

| Metric | c | DG | SAT | SAT+EM | SelNet | SelNet+EM | SR | SAT+SR | SAT+EM+SR | SelNet+SR | SelNet+EM+SR | ENS | ENS+SR | ConfidNet | SELE | REG | SCross | AUCross | PlugInAUC |
|---|---|---|---|---|---|---|---|---|---|---|---|---|---|---|---|---|---|---|---|
| $\widehat{Err}$ | .99 | .143 ± .008 | .093 ± .006 | .109 ± .007 | .114 ± .007 | .120 ± .007 | .094 ± .006 | .094 ± .006 | .110 ± .007 | .115 ± .007 | .117 ± .007 | **.083 ± .006** | **.083 ± .006** | .101 ± .006 | .139 ± .007 | .144 ± .007 | .102 ± .006 | .108 ± .006 | .099 ± .007 |
| | .95 | .142 ± .008 | .078 ± .006 | .094 ± .007 | .109 ± .006 | .113 ± .007 | .078 ± .006 | .080 ± .006 | .090 ± .007 | .102 ± .007 | .119 ± .007 | .075 ± .006 | **.064 ± .006** | .093 ± .006 | .137 ± .007 | .141 ± .007 | .084 ± .005 | .110 ± .006 | .097 ± .007 |
| | .90 | .134 ± .008 | .064 ± .005 | .085 ± .007 | .101 ± .007 | .087 ± .007 | .068 ± .006 | .062 ± .005 | .073 ± .006 | .095 ± .007 | .091 ± .007 | .063 ± .005 | **.049 ± .005** | .081 ± .006 | .133 ± .007 | .141 ± .007 | .065 ± .005 | .115 ± .006 | .098 ± .007 |
| | .85 | .112 ± .008 | .051 ± .005 | .073 ± .006 | .086 ± .006 | .083 ± .006 | .056 ± .005 | .049 ± .005 | .064 ± .006 | .090 ± .006 | .076 ± .006 | .052 ± .005 | **.039 ± .005** | .064 ± .005 | .127 ± .007 | .139 ± .007 | .048 ± .005 | .119 ± .007 | .096 ± .007 |
| | .80 | .089 ± .007 | .043 ± .005 | .056 ± .005 | .098 ± .007 | .095 ± .007 | .047 ± .006 | .040 ± .005 | .053 ± .006 | .069 ± .006 | .063 ± .006 | .041 ± .005 | **.031 ± .004** | .055 ± .005 | .126 ± .008 | .139 ± .008 | .035 ± .004 | .127 ± .007 | .094 ± .007 |
| | .75 | .072 ± .007 | .033 ± .004 | .044 ± .005 | .082 ± .006 | .080 ± .007 | .040 ± .005 | .033 ± .004 | .041 ± .005 | .065 ± .005 | .055 ± .006 | .033 ± .005 | **.023 ± .004** | .049 ± .005 | .124 ± .008 | .141 ± .008 | .029 ± .004 | .130 ± .008 | .088 ± .007 |
| | .70 | .058 ± .006 | .027 ± .004 | .042 ± .005 | .079 ± .007 | .063 ± .006 | .034 ± .005 | .027 ± .004 | .040 ± .005 | .075 ± .006 | .049 ± .005 | .028 ± .004 | **.018 ± .003** | .038 ± .004 | .119 ± .008 | .141 ± .008 | **.017 ± .004** | .135 ± .008 | .083 ± .006 |
| $\hat{\phi}$ | .99 | **.989 ± .002** | **.991 ± .002** | .986 ± .003 | .981 ± .003 | **.990 ± .002** | **.991 ± .002** | .994 ± .001 | .987 ± .003 | **.989 ± .002** | .985 ± .002 | **.992 ± .002** | **.990 ± .002** | **.990 ± .002** | .973 ± .004 | .995 ± .001 | .988 ± .002 | .987 ± .002 | .994 ± .002 |
| | .95 | **.951 ± .005** | .946 ± .005 | .936 ± .006 | .938 ± .005 | .943 ± .005 | .943 ± .005 | .959 ± .005 | .936 ± .006 | .927 ± .006 | .943 ± .005 | **.949 ± .005** | **.950 ± .004** | .945 ± .005 | .937 ± .005 | .956 ± .005 | .932 ± .005 | .939 ± .005 | .940 ± .006 |
| | .90 | .904 ± .006 | .897 ± .007 | .884 ± .008 | .899 ± .007 | .913 ± .006 | .906 ± .006 | .904 ± .007 | .879 ± .007 | .889 ± .006 | .912 ± .007 | **.901 ± .006** | **.900 ± .007** | .895 ± .007 | .878 ± .007 | .896 ± .006 | .857 ± .007 | .886 ± .007 | .894 ± .007 |
| | .85 | **.849 ± .007** | .841 ± .009 | .845 ± .009 | .842 ± .009 | .835 ± .009 | .855 ± .008 | .844 ± .008 | .826 ± .008 | .842 ± .009 | .831 ± .009 | .856 ± .007 | .849 ± .009 | .845 ± .008 | .812 ± .008 | .854 ± .007 | .775 ± .008 | .821 ± .007 | .847 ± .008 |
| | .80 | **.801 ± .009** | .785 ± .010 | .776 ± .009 | **.799 ± .009** | .803 ± .009 | .804 ± .009 | .775 ± .010 | .779 ± .009 | .790 ± .009 | **.801 ± .009** | .796 ± .009 | **.800 ± .009** | .808 ± .008 | .761 ± .009 | .807 ± .008 | .695 ± .009 | .752 ± .009 | .807 ± .008 |
| | .75 | **.749 ± .010** | .716 ± .010 | .723 ± .009 | .756 ± .010 | .757 ± .009 | .754 ± .010 | .717 ± .010 | .729 ± .010 | .745 ± .010 | .733 ± .010 | **.746 ± .009** | .750 ± .010 | .756 ± .010 | .710 ± .010 | .759 ± .009 | .640 ± .009 | .685 ± .009 | .760 ± .009 |
| | .70 | .705 ± .010 | .677 ± .011 | .684 ± .010 | .693 ± .011 | .710 ± .010 | .699 ± .011 | .671 ± .011 | .689 ± .010 | .697 ± .009 | .717 ± .010 | **.698 ± .009** | .699 ± .011 | .690 ± .010 | .664 ± .010 | .707 ± .009 | .573 ± .009 | .626 ± .010 | .701 ± .010 |
| $MinCoeff$ | .99 | .968 ± .037 | .987 ± .039 | .987 ± .039 | .988 ± .037 | .980 ± .037 | .991 ± .038 | .993 ± .038 | .982 ± .037 | **.997 ± .038** | .980 ± .037 | **.995 ± .037** | .988 ± .038 | 1.003 ± .038 | .978 ± .038 | 1.009 ± .038 | 1.004 ± .038 | 1.017 ± .038 | 1.012 ± .038 |
| | .95 | .840 ± .037 | .919 ± .039 | .932 ± .039 | .918 ± .037 | .913 ± .037 | .925 ± .039 | .938 ± .039 | .913 ± .039 | .901 ± .038 | .886 ± .036 | .926 ± .037 | .925 ± .039 | .952 ± .039 | .944 ± .038 | 1.001 ± .039 | **.994 ± .038** | 1.055 ± .040 | 1.034 ± .040 |
| | .90 | .683 ± .036 | .835 ± .036 | .874 ± .040 | .900 ± .038 | .871 ± .035 | .878 ± .039 | .854 ± .040 | .832 ± .038 | .873 ± .037 | .871 ± .037 | .846 ± .038 | .853 ± .037 | .876 ± .038 | .870 ± .037 | **1.006 ± .040** | 1.009 ± .041 | 1.108 ± .041 | 1.068 ± .042 |
| | .85 | .496 ± .034 | .763 ± .036 | .829 ± .039 | .664 ± .039 | .820 ± .038 | .763 ± .037 | .779 ± .039 | .648 ± .037 | .683 ± .036 | .755 ± .036 | .785 ± .039 | .755 ± .036 | .771 ± .037 | .799 ± .039 | **.992 ± .041** | 1.035 ± .042 | 1.174 ± .043 | 1.098 ± .043 |
| | .80 | .386 ± .032 | .694 ± .037 | .722 ± .037 | .435 ± .031 | .418 ± .031 | .760 ± .037 | .691 ± .039 | .717 ± .038 | .634 ± .037 | .785 ± .038 | .704 ± .039 | .704 ± .039 | .697 ± .038 | .757 ± .039 | **1.007 ± .041** | 1.032 ± .047 | 1.268 ± .046 | 1.120 ± .044 |
| | .75 | .318 ± .030 | .614 ± .037 | .638 ± .038 | .365 ± .029 | .354 ± .030 | .723 ± .038 | .621 ± .039 | .645 ± .037 | .556 ± .034 | .691 ± .040 | .656 ± .038 | .627 ± .035 | .619 ± .038 | .748 ± .040 | **1.015 ± .042** | 1.040 ± .048 | 1.349 ± .049 | 1.138 ± .044 |
| | .70 | .252 ± .026 | .578 ± .037 | .585 ± .036 | .351 ± .030 | .279 ± .028 | .690 ± .038 | .574 ± .038 | .608 ± .036 | .687 ± .042 | .652 ± .038 | .600 ± .038 | .581 ± .033 | .500 ± .034 | .722 ± .042 | **1.003 ± .044** | 1.069 ± .051 | 1.448 ± .053 | 1.180 ± .046 |

