# OpenReview forum: "Deep Neural Network Benchmarks for Selective Classification"
_DMLR — Accepted by DMLR_

### Review · Reviewer_asWe · 2024-03-28

**Recommendation:** 3
**Confidence:** 1

**Summary Of Contributions:**

This research explores Selective Classification (SC) to enhance the trustworthiness of artificial intelligence systems, particularly in sensitive areas. It uniquely benchmarks 18 SC methods across 44 datasets, assessing their performance on criteria such as error rate, coverage, and out-of-distribution data. The study reveals that no single method universally outperforms others, underscoring the need for contextual method selection. The authors contribute a survey of SC methods, an extensive evaluation framework, and a public repository to facilitate future research. This work provides certain insights into improving AI reliability through selective prediction.

**Strengths:**

Have a thorough introduction for selective classification, including learn-to-abstain, learn-to-select, and score-based ones, which is friendly for new readers.

The proposed benchmarks includes 44 datasets (20 images, 24 tabular data) along with five carefully designed metrics. Multiple common baselines are examined. For each metric, they included detailed description, and the results are plotted with helpful visualizations along with extensive analysis.

Detailed hyper-parameters are included in Appendix for replication, which is really important.

**Audience:**

Yes

**Broader Impact Concerns:**

I have no concerns here.

**Claims And Evidence:**

Yes, I believe the claims made by the submission supported by results on multiple heterogeneous datasets and multiple metrics. Also, as they mainly examined existed methods, I feel their conclusion are leaning towards objective.

**Datasets And Benchmarks:**

Table A1 listed the detailed datasets they used. They are all published datasets, and thus, this paper mainly describe the details and source.

**Extended Submissions:**

N/A

**Limitations:**

Based on the Table A2, all the training size of the used datasets is small. The biggest one is 348,605. I am unsure if the proposed benchmark is suitable to examine nowadays web-scale models. Regarding the number of datasets, the paper includes a lot, while the size of dataset is not big in my mind.

My biggest concern is how Selective Classification significance to our community. I am unsure if this technology benefits broader audience and kindly ask AE to take seriously considerations here.

**Requested Changes:**

Please address the concerns raised in the above weakness part.

Also, I feel it is a bit hard to tell the performance difference in Figure 9.

---

### Review · Reviewer_sEum · 2024-04-25

**Recommendation:** 3
**Confidence:** 1

**Summary Of Contributions:**

This paper studies the trustworthiness of machine learning models in decision-making. The paper focuses on the selective classification framework, which allows a model to not always offer a prediction. Intuitively, this framework equips a model with a mechanism that selects whether a prediction is made on a per-example basis. The goal is to find the tradeoff between the proportion of examples for which a prediction is made (i.e., the model’s coverage) and the performance improvement on the selected examples (i.e., the ones for which a prediction is made) that arises from focusing only on those cases where the model has a small chance of making a misprediction.

The paper goes beyond existing studies in three ways, including 18 selective classification methods, evaluation of several methods on 44 datasets including both image and tabular data, five different aspects of selective classification models' performance.

**Strengths:**

Strengths: The paper provides great detail about their proposed evaluation framework, including detailed background description. Then, it discusses the baselines in detail, including learn-to-abstain, learn-to-select, and score-based.

The paper also provides great detail about their experimental evaluation setting, including the datasets and baselines. It provides a detailed discussion of their results.

**Audience:**

Yes

**Claims And Evidence:**

The paper provides great details about their proposed framework. So I think the claims made are supported by evidence in the paper.

**Datasets And Benchmarks:**

The supplementary includes a zipped file that discusses the use of their data set.

**Extended Submissions:**

n/a

**Limitations:**

Weaknesses:
- While the algorithm shows impressive results on the datasets tested, the paper could benefit from a broader evaluation across a more diverse set of domains. This would help ascertain the algorithm's generalizability and effectiveness in various practical applications beyond the ones explored.

- The practical implications of their study are not very clear. I feel like this type of study could benefit from providing a more clear narrative on the implications of their study to the society. In its current form, the paper involves lots of technical materials, which makes the paper less interesting to read, in the referee's opinion.

- The paper appears to be technically solid (as evident by a large number of figures and models, methods, etc). But these materials and findings are not very interesting in the referee's opinion.

**Requested Changes:**

I think the paper could benefit from adding a discussion about the Broader Societal Impact of studying their specific proposed framework.

---

### Review · Reviewer_3eaN · 2024-07-03

**Recommendation:** 4
**Confidence:** 2

**Summary Of Contributions:**

The authors contribute an extensive benchmark study on selective classification methods for neural networks.
The study includes more baseline methods, datasets (both image- and tabular) and evaluation metrics than previous works. Conclusions are drawn about the strengths and weaknesses of baselines on empirical evidence regarding error rate, empirical coverage, rejection rates among classes, error rate when switching from bounded-abstention model to bounded-improvement model and ood performance

**Strengths:**

- The paper is very well written in my opinion
- Reproducibility is very good, all data and code is available and there is sufficient information about the used settings to obtain the results in the paper.
- The contributions are important for researchers and practitioners in the field of selective classification. In general I think this work is a valuable contribution to this field that is executed well following scientific principles.

**Audience:**

Yes

**Broader Impact Concerns:**

No clear ethical implications as far as I can see.

**Claims And Evidence:**

Yes, for the performed experiments, the evidence is extensive.

**Datasets And Benchmarks:**

In my opinion, yes.

**Extended Submissions:**

-

**Limitations:**

- No insights on the impact of NN architecture on metrics. It is unclear whether the conclusions hold for neural network models in general, or only for the chosen model architectures for each learning task. This limits generalization of the conclusions drawn.

**Requested Changes:**

- In the RQs, the terms ‘selective error rate’, ‘expected target coverage’ and ‘rejection rates’ are not defined beforehand in the background section. This should be done to improve readability and reduce confusion.
- Some more explanation is needed for the CD plots and their relation to statistical significant differences. Readers that are not familiar might get confused.
- Figure 9 should be made more readable. Either move to Appendix to use more space or make the formatting of the lines more readable.
- Maybe add a real-world use case example to the introduction as a motivation for more application-minded readers. I liked the ‘bank example’ later in the paper.